

# From caves to seamounts: the hidden diversity of tetractinellid sponges from the Balearic Islands, with the description of eight new species

Julio A. Díaz[1,2], Francesc Ordines[1], Enric Massutí[1] and Paco Cárdenas[3,4]

[1] Centre Oceanogràfic de Balears, Instituto Español de Oceanografía, Palma, Illes Balears, Spain
[2] Laboratori de Genètica, Biology Department, University of the Balearic Islands, Palma, Balearic Islands, Spain
[3] Department of Pharmaceutical Biosciences, Uppsala University, Uppsala, Sweden
[4] Museum of Evolution, Uppsala University, Uppsala, Sweden

Corresponding authors
Julio A. Díaz, julio.diaz@ieo.csic.es
Paco Cárdenas,
paco.cardenas@em.uu.se

## ABSTRACT

The sponge fauna of the Western Mediterranean stands as one of the most studied in the world. Yet sampling new habitats and a poorly studied region like the Balearic Islands highlights once again our limited knowledge of this group of animals. This work focused on demosponges of the order Tetractinellida collected in several research surveys (2016–2021) on a variety of ecosystems of the Balearic Islands, including shallow caves, seamounts and trawl fishing grounds, in a broad depth range (0–725 m). Tetractinellid material from the North Atlantic and more than twenty type specimens were also examined and, for some, re-described in this work. All species were barcoded with the traditional molecular markers COI (Folmer fragment) and 28S (C1-C2 or C1-D2 fragment). A total of 36 species were identified, mostly belonging to the family Geodiidae (15 species), thereby bringing the number of tetractinellids recorded in the Balearic Islands from 15 to 39. Eight species from this study are new: *Stelletta mortarium* sp. nov., *Penares cavernensis* sp. nov., *Penares isabellae* sp. nov., *Geodia bibilonae* sp. nov., *Geodia microsphaera* sp. nov. and *Geodia matrix* sp. nov. from the Balearic Islands; *Geodia phlegraeioides* sp. nov. and *Caminus xavierae* sp. nov. from the North East Atlantic. *Stelletta dichoclada* and *Erylus corsicus* are reported for the first time since their description in Corsica in 1983. *Pachastrella ovisternata* is documented for the first time in the Mediterranean Sea. Finally, after comparisons of type material, we propose new synonymies: *Geodia anceps* as a junior synonym of *Geodia geodina, Erylus cantabricus* as a junior synonym of *Erylus discophorus* and *Spongosorites maximus* as a junior synonym of *Characella pachastrelloides.*

## INTRODUCTION

Tetractinellida is the second most diverse demosponge order, with currently ∼1,180 described species belonging to 98 genera and 23 families (*de Voogd et al., 2023*). They are found in all oceans and latitudes, but usually more present in the deep sea and cryptic

habitats such as caves, and less frequently in light-exposed areas (*e.g.*, *Maldonado & Young, 1996*; *Grenier et al., 2018*). The astrophorin tetractinellids are known to constitute boreo-arctic North Atlantic and Western Mediterranean sponge grounds, structural habitats that increase the biodiversity and provide refuge for many demersal species of commercial interest (*Klitgaard, 1995*; *Klitgaard & Tendal, 2004*; *Maldonado et al., 2015*).

The spicular set of tetractinellids is characterized by four-branched megascleres, called triaenes, in combination with either (i) star-shaped microscleres, called asters, in the suborder Astrophorina or (ii) c/s-shaped microscleres, called sigmaspires, in the suborder Spirophorina. Some genera have developed hypersilicified spicules called desmas and have been traditionally grouped in the lithistids. These genera are now realocated to the tetractinellids (*Cárdenas, Pérez & Boury-Esnault, 2012*; *Schuster et al., 2015*). Both triaenes and microscleres can be secondary lost in some groups or species (*Cárdenas et al., 2011*; *Schuster et al., 2015*). The triaenes and asters diversified in a wide range of sizes and morphologies are occasionally found together with other microscleres such as microxeas, microrhabds, amphisanidasters, spherules or raphides. This spicular richness and heterogeneity makes the identification of tetractinellids based on spicules easier than in other demosponge groups and has attracted the attention of systematists to investigate the evolution of demosponge spicules (*Chombard, Boury-Esnault & Tillier, 1998*; *Cárdenas et al., 2010*; *Cárdenas et al., 2011*; *Cárdenas & Rapp, 2013*; *Schuster et al., 2015*). However, despite being a well-studied sponge order, several pending systematic questions remain. For instance, some groups such as Pachastrellidae, Ancorinidae or the genera *Erylus* and *Penares* are clearly polyphyletic (*Cárdenas, Pérez & Boury-Esnault, 2012*), which is often linked to the unresolved phylogenetic position of other taxa such as Calthropellidae, *Characella*, *Jaspis* and *Ecionemia*.

The Mediterranean Sea currently holds 83 species of Tetractinellida: 62 Astrophorina (including six lithistids), eight Spirophorina (including three lithistids) and nine Thoosina (*de Voogd et al., 2023*). Only 16 are currently recorded from the Balearic Islands, in contrast with the 26 species reported from the Alboran island, also in the Western Mediterranean (*Sitja & Maldonado, 2014*). Indeed, taxonomic studies on sponges in the Balearic Islands are few and fragmentary compared to other areas of the Western Mediterranean (*Bibiloni & Gili, 1982*; *Bibiloni, 1990*; *Bibiloni, 1993*; *Uriz, Rosell & Martín, 1992*; *Díaz et al., 2020*; *Díaz, Ramírez-Amaro & Ordines, 2021*). The first tetractinellid sponges reported from the Balearic Islands are *Penares helleri* (*Schmidt, 1864*) and *Penares euastrum* (*Schmidt, 1868*), found in *Bibiloni & Gili (1982)*, a faunistic work on an infralittoral cave in the island of Mallorca. This work was followed by the publication of a thesis on sponge taxonomy encompassing samples from different depths, areas and biocenosis of the Islands (*Bibiloni, 1990*) which enriched the list of tetractinellids with seven new additions: *Geodia cydonium* (Linnaeus, 1767), *Stryphnus mucronatus* (*Schmidt, 1868*), *Pachastrella monilifera* (*Schmidt, 1868*), *Poecillastra compressa* (*Bowerbank, 1866*), *Calthropella (Calthropella) pathologica* (*Schmidt, 1868*), *Dercitus (Stoeba) plicatus* (*Schmidt, 1868*) and *Jaspis johnstoni* (*Schmidt, 1862*). Later, *Uriz, Rosell & Martín (1992)* found the species *Erylus discophorus* (*Schmidt, 1862*) and *Stryphnus ponderosus* (*Bowerbank, 1866*) at the National Park of Cabrera; *Massutí & Reñones (2005)* reported the species *Thenea muricata* (Bowerbank, 1858) on fishing

grounds; *Maldonado et al. (2015)* reported *Nethea amygdaloides* (*Carter, 1876*) and the lithistid *Leiodermatium pfeifferae* (Carter, 1873) near the Emile Baudot seamount; *Santín et al. (2018)* reported *Craniella cranium* (Müller, 1776) and *Neophrissospongia nolitangere* (*Schmidt, 1870*) from the Menorca Channel.

The Balearic Islands are a Western Mediterranean archipelago of four main islands and several islets. Its marine habitats are very heterogeneous and harbor rich and diverse biocenosis, developing in habitats like karstic caves, *Posidonia oceanica* (Linnaeus) Delile, 1813 meadows, rhodoliths and soft red algae beds, coralligenous outcrops, mud and detrital bottoms, slopes, canyons and seamounts (*Canals & Ballesteros, 1997*; *Acosta et al., 2003*). These ecosystems are in a context of elevated oligotrophy, as a consequence of the scarcity of rain and the absence of rivers, which reduces the terrigenous inputs and nutrient supply. These facts contribute to the singularity of some communities, and, for instance, photosynthetic biocenosis tend to develop deeper than in adjacent areas of the Iberian Peninsula (*Ballesteros, 1994*), a fact that dilates the biological range of the species found on these habitats, enhancing mesophotic zones where suspension feeders dominate (*Zabala & Ballesteros, 1989*). The diversity of the habitats coupled with the well-preserved seamounts of the Mallorca Channel, which show a high sponge diversity (*Díaz, Ramírez-Amaro & Ordines, 2021*; *Massutí et al., 2022*) suggest that a higher number of tetractinellid species should be present. The aim of this study was to improve our knowledge on the tetractinellid fauna of the Balearic Islands using an integrative approach on newly collected samples, combining morphology and molecular markers. This study also included the revision of poorly-known type material, as well as the study of some comparable species from the North Atlantic, some of which turned out to be new.

## MATERIAL AND METHODS

### Study area

The Balearic Promontory (Fig. 1) is a seafloor elevation in the Western Mediterranean, of approximately 400 km length and 105 km wide, containing the Mallorca-Menorca shelf to the east and the Ibiza-Formentera shelf to the west. The continental shelf is narrow and shallow, with a mean depth of 87 m, characterized by the presence of calcareous sediment and by the scarcity of terrigenous input. It harbors rich photophilic habitats of soft and calcareous red algae that develop until depths of 100–150 m, leading to detrital muds of the shelf border. The slopes are very steep and descend until the surrounding abyssal plains of the Valencia Trough and the Algerian Basin (*Acosta et al., 2003*). As other areas of the Western Mediterranean, most sedimentary bottoms of the continental shelf and the upper and middle slope around the Balearic Islands, between 50 and 800 m depth, have been exploited by the trawling fleet for several decades (*Farriols et al., 2017*).

Two channels are present in the Balearic Promontory, the Menorca Channel (MeC), between Menorca and Mallorca, and the Mallorca Channel (MaC), between Mallorca and Ibiza. The first channel is narrow and shallow, and it is influenced by the strong northern winds originating in the Gulf of Lion and the hydrodynamic conditions of the Balearic sub-basin, mainly shaped by Mediterranean waters and under the influence of the Balearic

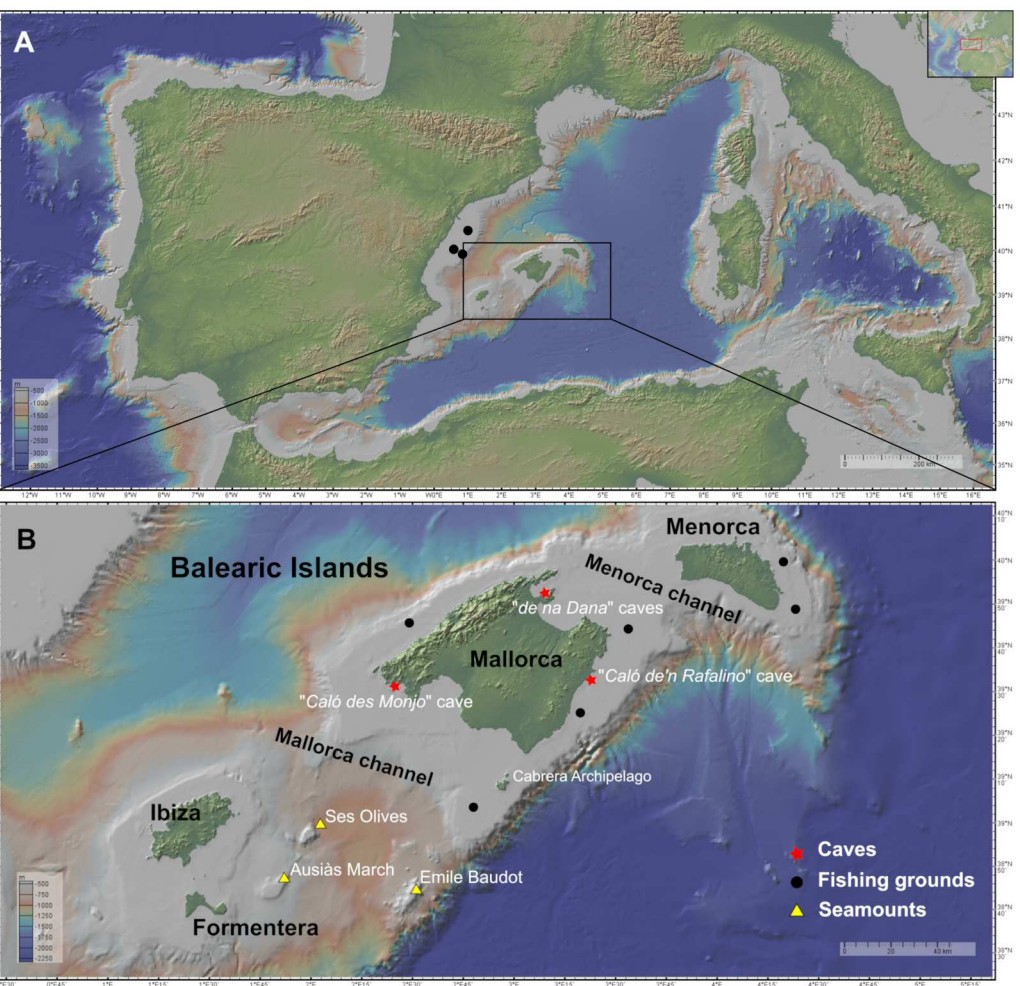

**Figure 1** **Maps of the studied area showing the location of the sampling stations of caves (red star), fishing grounds (black circle), and the seamounts SO, AM and Emile Baudot (yellow triangle).** The characteristics of the sampling stations are shown in Table 1. (A) Map of the Western Mediterranean. Black dots show the fishing grounds sampled on the Catalan shelf (next to Columbretes islands). (B) Map of the Balearic Islands. Maps made with GeoMapApp v.3.6.15 (http://www.geomapapp.org).

Current, flowing along the northern shelf edge of the Balearic Promontory (*Massutí et al., 2014*). In 2014, the MeC was included in the Natura 2000 Network, in the light of the singularity of its habitats and its high diversity of benthos (*Barbera et al., 2012*). Conversely, the MaC separating the two shelves (Mallorca-Menorca and Ibiza-Formentera) is wider, deeper and more heterogeneous than the MeC, containing not only continental shelf and slope bottoms but also abyssal plain. The MaC, being located in the Algerian sub-basin, is more influenced by the Atlantic waters (*Massutí et al., 2014*); the MaC also contains several seamounts, among which stand out Ses Olives (SO), Ausias March (AM), and Emile Baudot (EB).

SO rises from 650–900 to 250 m depth at its shallowest part; it has a flat summit composed of fine sediments. AM has a minimum depth of 86 m and a height 264 m,

with a summit in the mesophotic zone, where sediments are coarser, mainly composed of gravel and sand. Finally, EB represents a strongly irregular and uneven elevation, that rises from 900 to 94 m, with numerous mounds, depressions and rocky outcrops. Both SO and AM are of tectonic origin, while EB is of volcanic origin (*Acosta et al., 2004*). Patches of bio-constructions have been found in the summits of AM and EB, where rhodolith beds and coralligenous outcrops predominate, while rocky bottoms predominate in the flanks of SO, AM and EB, mainly colonized by filtering species, such as sponges and corals. According to these authors, in the less steep flanks and bathyal terraces of the upper and middle slope of these seamounts, muddy soft sediments are found, accumulating facies of the brachiopod *Gryphus vitreus* (Born, 1778), burrowing megafauna, small sponges and/or dead coral debris. The deepest areas of the middle slope at the base of seamounts are dominated by the finest muddy sediments and the presence of pockmarks fields (*Massutí et al., 2022*).

## Sampling

Specimens were collected during (i) seven MEDITS research surveys on board the R/V *Miguel Oliver*, carried out annually from 2016 to 2021 on fishing grounds of the Balearic Islands shelf and slopes, between 50 and 800 m depth; (ii) one MEDITS survey carried out in 2020 on board the R/V *Miguel Oliver* along the northeastern Iberian Peninsula within the same bathymetric range; and (iii) four research surveys carried out on board R/V *Ángeles Alvariño* and R/V *Sarmiento de Gamboa*, within the framework of the LIFE IP INTEMARES project at the SO, AM and EB seamounts of the MaC, in August 2018, October 2019 and July–August 2020 (Figs. 1A–1B). The MEDITS program is carried out in most of the northern coast of the Mediterranean and aims to assess the state of the demersal resources and nekton-benthic ecosystems (*Spedicato et al., 2019*). The objective of the LIFE IP INTEMARES project at the MaC is to improve the scientific knowledge on biodiversity, benthic habitats and human activities, to include SO, AM and EB seamounts in the Natura 2000 network (*Massutí et al., 2022*). Several sampling devices were used in mesophotic and bathyal bottoms for both MEDITS and INTEMARES surveys: the experimental bottom trawl gear GOC-73 (*Bertrand et al., 2002*; *Spedicato et al., 2019*), a Beam Trawl (BT), Rock Dredges (RD) and the Remote Operated Vehicle (ROV) *Liropus 2000* which was also used to film underwater. Screenshots of the film were used to study the *in situ* morphology of the specimens (Fig. 2).

Shallow water caves were explored by scuba diving or free apnea in May and November 2020 and in January, May and August 2021. Most of them can be classified as littoral marine caves created by sea erosion. They have salty water and marine fauna, being shallow (0–10 m depth) and located eastern ("Cova de sa Figuera", "Cova de ca'n Rafalino", "Cova de Cala Sa Nau"), northern ("Coves de Na Dana"), and western ("Cova Caló des Monjo") off Mallorca island (Fig. 1B). Their sizes are quite variable, "Cova de Cala Sa Nau" being the largest and the most important in terms of benthic organisms, with a spacious entrance and a main chamber having 76/36/8 m in maximum length/width/depth (*Gràcia, Clamor & Watkinson, 1998*). This cave is commonly frequented by scuba divers, especially in summertime, potentially having a negative impact on the sponge community.

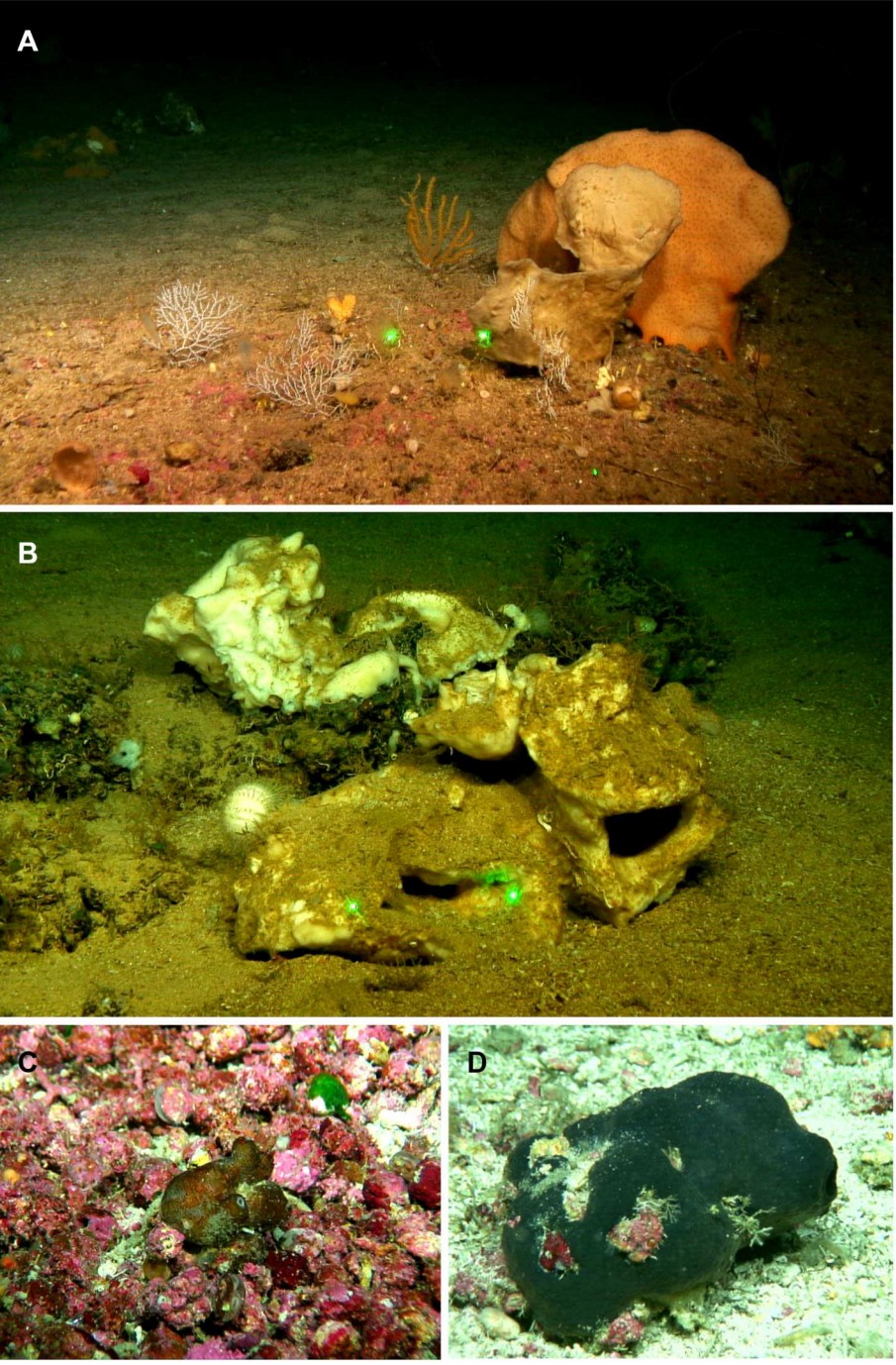

**Figure 2** **Remote Operated Vehicle (ROV) images of the tetractinellid fauna from the seamounts of the Mallorca Channel, Ses Olives (SO), Ausias March (AM) and Emile Baudot (EB).** (A) *Poecillastra compressa* (orange) specimen at 149 m depth in EB. (B) *Pachastrella ovisternata* specimen field #i808 collected at 263 m depth in the AM. (C) *Penares euastrum* (dark gray) at 90 m depth in the AM summit. (D) *Stryphnus mucronatus* specimen i827_1 collected at 100 m depth at the EB summit.

In contrast, the "Cova de ca'n Rafalino" is the smallest, being a short tunnel only several meters long, with a depth of 1–2 m and between 0.5 and 3 m wide. Inland freshwater infiltration has been observed in "Cova de ca'n Rafalino" and "Coves de na Dana". Due to the cave architecture, benthic organisms inhabiting the caves are relatively well protected from the action of waves. Details of sampling stations are summarized in Table 1. *In situ* images of the specimens were taken with an Olympus Tg5 digital camera (Fig. 3).

For oceanographic surveys, once the sampling gear was on board, sponges were separated from the rest of the catch and photographed, then macroscopic characters like morphology, color and texture were annotated prior to sample fixation. Samples for both morphological and molecular analysis were preserved in absolute ethanol (EtOH).

Specimens from this study were all deposited in the zoological collection at the Museum of Evolution, Uppsala University (Uppsala, Sweden) with UPSZMC# for non-type specimens and UPSZTY# for type material (Table S1), under Material Transfer Agreement 2023:074. Two exceptions: the holotype of *Geodia phlegraeioides* sp. nov. was deposited at the MNCN in Madrid (Spain), and the holotype of *Caminus xavierae* sp. nov. was already deposited at Naturalis in Leiden (The Netherlands). DNA extractions from the Balearic islands new species were deposited at the Museum of Evolution (holotypes) and at the Balearic Biodiversity Center (https://centrebaleardebiodiversitat.uib.eu/; paratypes), with same deposit numbers as the UPSZTY museum numbers.

The electronic version of this article in Portable Document Format (PDF) will represent a published work according to the International Commission on Zoological Nomenclature (ICZN), and hence the new names contained in the electronic version are effectively published under that Code from the electronic edition alone. This published work and the nomenclatural acts it contains have been registered in ZooBank, the online registration system for the ICZN. The ZooBank LSIDs (Life Science Identifiers) can be resolved and the associated information viewed through any standard web browser by appending the LSID to the prefix http://zoobank.org/. The LSID for this publication is: [urn:lsid:zoobank.org:pub:A88AE49E-B422-4F9A-A5E0-BB6C6B8FC185]. The online version of this work is archived and available from the following digital repositories: PeerJ, PubMed Central SCIE and CLOCKSS.

## Morphological descriptions

To obtain dissociated spicules preparations, a fragment of tissue was digested with bleach, the remaining spicules washed with pure water first, then with 50% EtOH and finally with 96% EtOH. Spicules were observed and measured with an optical microscope. For each sample, unless otherwise indicated, 25 spicules per spicule category were counted. Spicule measurements given in the text are always the range observed from all measured specimens, unless otherwise stated. Handmade thick sections with a scalpel were made to study the skeleton oganization of every species. For precious type material, such as the *Schmidt (1868)* collection, regular thick sections (100–800 μm) were made by embedding small pieces of the specimens using an Agar Low Viscosity Resin kit (Agar Scientific, Essex, UK). Embedded pieces were sectioned with a diamond wafering blade on a Buehler IsoMet™ Low Speed cutting machine. For SEM images, aliquots of suspended spicules were

Díaz et al. (2024), *PeerJ*, DOI 10.7717/peerj.16584

**Table 1  Details of the sampling stations.**

| Survey | Station | Date | Depth range | Sampling device | Latitude start | Longitude start | Latitude end | Longitude end | Area |
|---|---|---|---|---|---|---|---|---|---|
| INTEMARES_A22B_0820 | 11 | 26/08/2020 | 200–307 | ROV | 38°46′57.6″N | 1°46′40.8″E | 38°46′51″N | 1°47′0″E | Ausias March (Mallorca Channel) |
| INTEMARES_A22B_0820 | 20 | 28/08/2020 | 523–912 | ROV | 38°42′44.4″N | 2°37′8.4″E | 38°42′39.6″N | 2°36′30″E | Emile Baudot (Mallorca Channel) |
| INTEMARES_A22B_0820 | 21 | 28/08/2020 | 425–733 | ROV | 38°47′36.6″N | 2°32′49.8″E | 38°47′15″N | 2°32′56.4″E | Emile Baudot (Mallorca Channel) |
| INTEMARES_A22B_0820 | 24 | 29/08/2020 | 134–150 | ROV | 38°44′27.6″N | 2°29′16.8″E | 38°44′34.2″N | 2°29′32.4″E | Emile Baudot (Mallorca Channel) |
| INTEMARES_A22B_0820 | 25 | 29/08/2020 | 100–124 | ROV | 38°43′54.6″N | 2°30′9.6″E | 38°44′8.4″N | 2°30′36″E | Emile Baudot (Mallorca Channel) |
| INTEMARES_A22_0718 | 68 | 30/07/2018 | 135 | Rock Dredge | 38°41′54.6″N | 2°28′45.6″E | 38°4′0.06″N | 2°28′35.4″E | Emile Baudot (Mallorca Channel) |
| INTEMARES_A22_0718 | 51 | 03/08/2018 | 127 | Beam trawl | 38°44′53.9″N | 2°30′41.4″E | 38°44′58.9″N | 2°30′54.7″E | Emile Baudot (Mallorca Channel) |
| INTEMARES_A22_0718 | 52 | 03/08/2018 | 108 | Rock Dredge | 38°44′13.2″N | 2°30′3.6″E | 38°44′12.5″N | 2°30′12″E | Emile Baudot (Mallorca Channel) |
| INTEMARES_A22_0718 | 60 | 03/08/2018 | 137 | Beam trawl | 38°43′13.1″N | 2°29′29.4″E | 38°43′5.5″N | 2°29′20.4″E | Emile Baudot (Mallorca Channel) |
| INTEMARES_A22B_0720 | 8 | 21/07/2020 | 315–295 | Rock Dredge | 38°58′11.3″N | 2°0′30.6″E | 38°58′12″N | 2°0′25.2″E | Ses Olives (Mallorca Channel) |
| INTEMARES_A22B_0720 | 18 | 23/07/2020 | 112 | Beam trawl | 38°45′15.5″N | 1°46′53.4″E | 38°45′16.2″N | 1°46′54.1″E | Ausias March (Mallorca Channel) |
| INTEMARES_A22B_0720 | 19 | 23/07/2020 | 111–94 | Rock Dredge | 38°43′49.8″N | 1°45′34.2″E | 38°43′46.2″N | 1°45′43.2″E | Ausias March (Mallorca Channel) |

Díaz et al. (2024), *PeerJ*, DOI 10.7717/peerj.16584

**Table 1** (*continued*)

| Survey | Station | Date | Depth range | Sampling device | Latitude start | Longitude start | Latitude end | Longitude end | Area |
|---|---|---|---|---|---|---|---|---|---|
| INTEMARES_A22B_0720 | 21 | 23/07/2020 | 105 | Beam trawl | 38°44′55.2″N | 1°50′9.6″E | 38°45′19.2″N | 1°50′29.4″E | Ausias March (Mallorca Channel) |
| INTEMARES_A22B_0720 | 26 | 24/07/2020 | 127 | Beam trawl | 38°26′0.72″N | 1°26′20.52″E | 38°26′0.36″N | 1°26′26.28″E | Ausias March (Mallorca Channel) |
| INTEMARES_A22B_0720 | 30 | 24/07/2020 | 265–204 | Rock Dredge | 38°47′18.6″N | 1°47′0.6″E | 38°46′58.2″N | 1°47′7.8″E | Ausias March (Mallorca Channel) |
| INTEMARES_A22B_0720 | 34 | 25/07/2020 | 111–105 | Rock Dredge | 38°46′1.8″N | 1°49′5.4″E | 38°45′55.2″N | 1°49′14.4″E | Ausias March (Mallorca Channel) |
| INTEMARES_A22B_0720 | 42 | 26/07/2020 | 143–139 | Rock Dredge | 38°43′32.4″N | 2°29′16.8″E | 38°43′37.8″N | 2°29′6″E | Emile Baudot (Mallorca Channel) |
| INTEMARES_A22B_0720 | 43 | 26/07/2020 | 118–116 | Rock Dredge | 38°44′25.1″N | 2°30′40.3″E | 38°44′26.9″N | 2°30′33.5″E | Emile Baudot (Mallorca Channel) |
| INTEMARES_A22B_0720 | 45 | 26/07/2020 | 149–151 | Beam trawl | 38°42′51.8″N | 2°30′13.7″E | 38°42′28.1″N | 2°29′24″E | Emile Baudot (Mallorca Channel) |
| INTEMARES_A22B_0720 | 52 | 27/07/2020 | 297 | Beam trawl | 38°45′47.5″N | 2°31′0.5″E | 38°45′56.9″N | 2°30′37.1″E | Emile Baudot (Mallorca Channel) |
| INTEMARES_A22B_0720 | 53 | 27/07/2020 | 108–102 | Rock Dredge | 38°44′0.6″N | 2°30′43.2″E | 38°44′8.4″N | 2°30′24.6″E | Emile Baudot (Mallorca Channel) |
| INTEMARES_A22B_0720 | 54 | 27/07/2020 | 207–124 | Rock Dredge | 38°43′19.8″N | 2°30′54″E | 38°43′31.2″N | 2°30′43.8″E | Emile Baudot (Mallorca Channel) |
| INTEMARES_A22B_0720 | 59 | 28/07/2020 | 526–550 | Rock Dredge | 38°26′3.96″N | 2°26′25.56″E | 38°26′3.12″N | 2°26′29.16″E | Emile Baudot (Mallorca Channel) |
| INTEMARES_A22B_1019 | 3 | 11/10/2019 | 293–255 | Rock Dredge | 38°58′41.4″N | 1°59′13.2″E | 38°58′33″N | 1°59′13.2″E | Ses Olives (Mallorca Channel) |
| INTEMARES_A22B_1019 | 8 | 11/10/2019 | 241 | Rock Dredge | 38°57′35.4″N | 2°79′54.6″E | 38°57′42″N | 2°97′44.4″E | Ses Olives (Mallorca Channel) |
| INTEMARES_A22B_1019 | 36 | 13/10/2019 | 609 | Beam trawl | 38°57′51″N | 1°56′34.2″E | 38°57′59.4″N | 1°56′40.2″E | Ses Olives (Mallorca Channel) |
| INTEMARES_A22B_1019 | 48 | 15/10/2019 | 124 | Beam trawl | 38°43′30.6″N | 1°49′41.4″E | 38°43′39″N | 1°49′51″E | Ausias March (Mallorca Channel) |
| INTEMARES_A22B_1019 | 50 | 15/10/2019 | 98 | Beam trawl | 38°43′33.6″N | 1°48′12.6″E | 38°43′34.7″N | 1°48′23.4″E | Ausias March (Mallorca Channel) |
| INTEMARES_A22B_1019 | 58 | 15/10/2019 | 135 | Beam trawl | 38°46′55.2″N | 1°52′16.8″E | 38°47′5.4″N | 1°52′19.8″E | Ausias March (Mallorca Channel) |
| INTEMARES_A22B_1019 | 103 | 21/10/2019 | 231–302 | Rock Dredge | 38°47.4′0″N | 1°47.2′0″E | 38°47.3′0″N | 1°47.2′0″E | Ausias March (Mallorca Channel) |

**Table 1** (*continued*)

| Survey | Station | Date | Depth range | Sampling device | Latitude start | Longitude start | Latitude end | Longitude end | Area |
|---|---|---|---|---|---|---|---|---|---|
| INTEMARES_A22B_1019 | 124 | 24/10/2019 | 145–147 | Beam trawl | 38°45′19.1″N | 2°31′0.5″E | 38°45′20.9″N | 2°31′8.4″E | Emile Baudot (Mallorca Channel) |
| INTEMARES_A22B_1019 | 136 | 25/10/2019 | 141–145 | Beam trawl | 38°44′42.7″N | 2°29′25.8″E | 38°43′13.1″N | 2°29′21.5″E | Emile Baudot (Mallorca Channel) |
| INTEMARES_A22B_1019 | 158 | 27/10/2019 | 141–145 | Beam trawl | 38°42′57.6″N | 2°29′17.4″E | 38°42′55.8″N | 2°29′6″E | Emile Baudot (Mallorca Channel) |
| INTEMARES_A22B_1019 | 167 | 28/10/2019 | 147 | Beam trawl | 38°42′21.6″N | 2°29′37.3″E | 38°42′12.6″N | 2°29′29.4″E | Emile Baudot (Mallorca Channel) |
| INTEMARES_A22B_1019 | 165 | 28/10/2019 | 312 | Rock Dredge | 38°46′58.2″N | 2°31′6″E | 38°46′52.8″N | 2°31′7.8″E | Emile Baudot (Mallorca Channel) |
| INTEMARES_A22B_1019 | 177 | 29/10/2019 | 150 | Beam trawl | 38°43′57.7″N | 2°28′54.1″E | 38°43′47″N | 2°28′53.4″E | Emile Baudot (Mallorca Channel) |
| MEDITS_ES05_16 | 181 | 08/06/2016 | 142 | GOC73 | 39°1′9.48″N | 2°51′1.8″E | 39°2′15.72″N | 2°49′44.4″E | Fishing ground (Cabrera Archipelago) |
| MEDITS_ES05_17 | 194 | 12/06/2017 | 148 | GOC73 | 39°46′25.68″N | 2°27′59.22″E | 39°46′25.41″N | 2°27′59.33″E | Fishing ground (Sóller) |
| MEDITS_ES05_17 | 206 | 15/06/2017 | 134 | GOC73 | 39°47′37.2″N | 4°26′15.4″E | 39°47′37.2″N | 4°26′15.4″E | Fishing grounds (Maó) |
| MEDITS_ES05_19 | 184 | 14/06/2019 | 50 | GOC73 | 39°27′0″N | 3°20′15.6″E | 39°27′0.42″N | 3°21′6.6″E | Fishing ground (Porto-colom) |
| MEDITS_ES05_20 | 74 | 16/06/2020 | 72 | GOC73 | 40°0′30.6″N | 4°18′54.6″E | 40°0′0.6″N | 4°18′13.8″E | Fishing ground (Maó) |
| MEDITS_ES05_20 | 76 | 16/06/2020 | 132 | GOC73 | 39°47′52.2″N | 4°26′22.8″E | 39°46′36.6″N | 4°25′22.8″E | Fishing ground (Maó) |
| MEDITS_ES05_21 | 212 | 17/06/2021 | 63 | GOC73 | 39°44′52.2″N | 3°35′20.4″E | 39°4′0″N | 3°34′33.6″E | Fishing ground (Menorca channel) |
| MEDITS0521_PITIUSSES | 2 | 18/08/2021 | 54 | GOC73 | 38°35′15.72″N | 1°26′35.52″E | 38°35′45.6″N | 1°27′40.32″E | Fishing ground (South of Formentera) |
| MEDITS_ES06N_20 | 3 | 30/05/2020 | 74 | GOC73 | 40°01′57.6″N | 0°34′9.6″E | 40°0′40.8″N | 0°33′18.6″E | Fishing ground (Columbrets) |
| MEDITS_ES06N_20 | 6 | 31/05/2020 | 144 | GOC73 | 39°53′31.2″N | 0°53′9″E | 39°54′38.4″N | 0°54′34.8″E | Fishing ground (Columbrets) |
| MEDITS_ES06N_20 | 14 | 01/06/2020 | 96 | GOC73 | 40°18′39″N | 1°7′28.2″E | 40°1′0.12″N | 1°6′37.8″E | Fishing ground (Sant Carles de la Ràpita) |

**Table 1** (*continued*)

| Survey | Station | Date | Depth range | Sampling device | Latitude start | Longitude start | Latitude end | Longitude end | Area |
|---|---|---|---|---|---|---|---|---|---|
| MEDITS_ES05_17 | 219 | 18/06/2017 | 65 | GOC73 | 39°51′4.2″N | 4°05′37.8″E | 39°50′24″N | 4°06′45″E | Fishing ground (Son Bou) |
| LITORAL CAVES | – | 06/05/2020 | 0–0.5 | Free apnea | 39°33′23.49″N | 3°22′7.35″E | 39°33′23.49″N | 3°22′7.35″E | Cova de Sa Figuera (Manacor) |
| LITORAL CAVES | – | 23/05/2020 | 0–0.5 | Free apnea | 39°33′20.9″N | 3°22′2.39″E | 39°33′20.9″N | 3°22′2.39″E | Cova Caló den Rafalino (Manacor) |
| LITORAL CAVES | – | 17/01/2021 | 3–4 | Scuba diving | 39°23′31.12″N | 3°14′58.07″E | 39°23′31.12″N | 3°14′58.07″E | Cova cala Sa Nau (Felanitx) |
| LITORAL CAVES | – | 06/05/2021 | 6 | Scuba diving | 39°31′39.38″N | 2°25′50.63″E | 39°31′39.38″N | 2°25′50.63″E | Cova Caló des Monjo (Calvià) |
| LITORAL CAVES | – | 14/08/2021 | 0–0.5 | Scuba diving | 39°52′19.86″N | 3°9′8.50″E | 39°52′19.86″N | 3°9′8.50″E | Coves De Na Dana (Alcúdia) |

**Notes.**

BT, beam trawl; DR, rock dredge; SO, Ses Olives.
Ausias March: AM; Emile Baudot: EM.

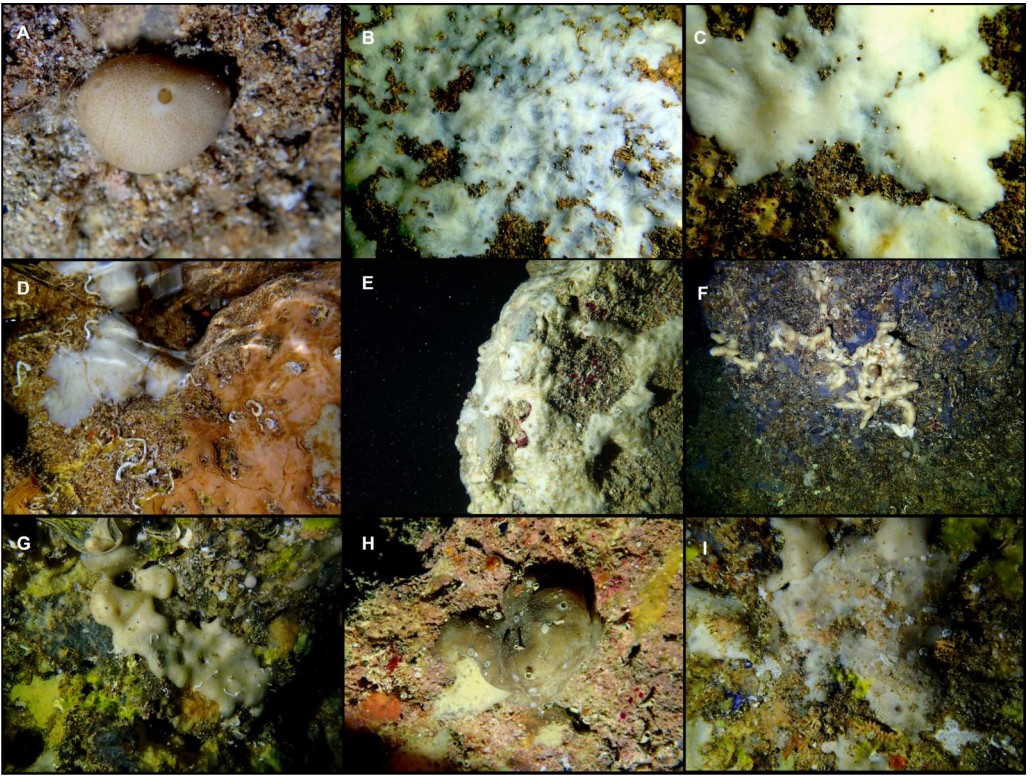

**Figure 3  Tetractinellids from Mallorca caves.** (A) *Caminella intuta* (specimen LIT05) in ''Sa cova de sa Figuera'' cave, 0–0.5 m. (B–C) *Erylus discophorus* specimens LIT72 and LIT71 collected at 0–1 m at ''Coves de na Dana'' caves. (D) *Erylus* cf. *deficiens* (white), specimen LIT10 collected at 0–0.5 m at ''Cova des Caló den Rafalino'' cave. (E) Community dominated by *Penares bibilonae* sp. nov. and *Penares cavernensis* sp. nov. (uncollected specimens) at 4–5 m at ''Cala sa Nau'' cave. (F) *Penares cavernensis* sp. nov. (uncollected specimen) at 4–5 m depth, at ''Cala sa Nau'' cave. (G) *Penares cavernensis* sp. nov. (paratype) LIT65, collected at 6 m depth at ''es Caló des Monjo'' cave. (H) *Penares cavernensis* sp. nov. (paratype) LIT45, collected at 3–4 m ''Cala sa Nau'' cave. (I) *Penares isabellae* sp. nov. (paratype) LIT66 collected at 6 m depth at ''es Caló des Monjo'' cave.

transferred onto aluminum foil, air dried, sputter coated with gold and observed under a HITACHI S-3400N scanning electron microscope (SEM) at the *Serveis Cientifíco-tècnics* of the University of the Balearic Islands (UIB). The terminology applied for the morphological description of the spicules follows *Boury-Esnault & Rützler (1997)* and *Hooper & Van Soest (2002)*.

## Molecular analysis

DNA was extracted from a piece of choanosomal tissue ($\sim$2 cm$^3$) using the DNeasy Blood and Tissue Extraction kit (Qiagen, Hilden, Germany). Polymerase chain reaction (PCR) was used to amplify the Folmer fragment (658 bp) of the mitochondrial cytochrome c oxidase subunit I (COI) and the C1-C2 ($\sim$369 bp) or C1-D2 ($\sim$800 bp.) fragments of the nuclear rDNA 28S gene.

For COI, the universal Folmer primers LCO1490/HCO2198 were used (*Folmer et al., 1994*), except for the *Craniella* species for which we used primers LCO1490/COX1R1

(*Rot et al., 2006*); this primer set amplifies a longer fragment ca. 1180 bp (Folmer + Erpenbeck fragments). When LCO1490/HCO2198 failed to amplify COI (especially for some *Erylus* and *Penares* species), the primers LCO/TetractminibarR1 were used to amplify the first 130 bp of the Folmer marker, also called the Folmer COI minibarcode (*Cárdenas & Moore, 2019*). The primers jgHCO (*Geller et al., 2013*) and ErylusCOIF2 (5′-CTCCYGGATCAATGTTGGG-3′) were then used to amplify the rest of the Folmer fragment (*Cárdenas et al., 2018*). For 28S, the primer set C1'ASTR/D2 (*Vân Le, Lecointre & Perasso, 1993*; *Cárdenas et al., 2011*) was used to get the C1-D2 domains. When the C1'ASTR/D2 primers failed to amplify 28S, we used the primers C1'/Ep3 to get the shorter C1-C2 fragment. PCR was performed in 50 μl volume reaction (34.4 μl ddH20, 5 μl Mangobuffer, 2 μl DNTPs, 3.5 MgCl$_2$, 1 μl of each primer, 1 μl BSA, 0.1 μl TAQ and 2 μl DNA). PCR thermal profile used for amplification was [94 °C/5 min; 37 cycles (94 °C/15 s, 46 °C/15 s, 72 °C/15 s); 72 °C/7 min]. PCR products were visualized with 1% agarose gel and purified using the QIAquickR PCR Purification Kit (Qiagen, Hilden, Germany) and sequenced at Macrogen Inc. (Seoul, South Korea).

Sequences were imported into BioEdit 7.0.5.2. (*Hall, 1999*) and checked for quality and accuracy with nucleotide base assignment. Sequences were aligned using Mafft (*Katoh et al., 2002*). The resulting sequences were deposited in GenBank (http://www.ncbi.nlm.nih.gov/genbank/) with the following accession numbers: ON130519–ON130569, OR045842–OR045844 and OR045913–OR045914 for COI and ON133879–ON133850 and OR044718 for 28S (Table S1). Eight COI minibarcodes (111–130 bp), too small to be submitted to GenBank, were deposited on the Sponge Barcoding Project instead (https://www.spongebarcoding.org) with sequence numbers 2683 to 2690. The final COI and 28S alignment fasta files were deposited as Data S1.

Phylogenetic analysis were conducted using two different approaches: Bayesian Inference (BI) and maximum likelihood (ML), performed with the CIPRES science gateway platform (http://www.phylo.org; *Miller, Pfeiffer & Schwartz, 2010*) using MrBayes version 3.6.2 (*Ronquist et al., 2012*) and RAxML (*Stamatakis, 2014*). For MrBayes, we conducted four independent Markov chain Monte Carlo runs of four chains each, with 5 million generations, sampling every 1,000th tree and discarding the first 25% as burn-in, while RAxML was performed under the GTRCAT model with 1,000 bootstrap iterations. Convergence was assessed by effective sample size (ESS) calculation and was visualized using TRACER version 1.5. Genetic distance (*p*-distance) and number of base differences between pairs of DNA sequences were estimated with MEGA version 10.0.5 software (*Kumar et al., 2018*).

## Comparative material and abbreviations

To help with our specimen identifications and descriptions, comparative material was used from the following institutions, for which we provide their abbreviations: BELUM Mc, Ulster Museum Belfast (Northern Ireland, UK); CEAB.POR.BIO, Porifera Collection at the 'Centro de Estudios Avanzados de Blanes' (Blanes, Spain); COLETA, 'Coleção de Referência Biológica Marinha dos Açores', reference collection of the Department of Oceanography and Fisheries, University of the Azores (Portugal); CPORCANT,

Colección PORíferos del CANTábrico, IEO-CSIC (Gijón, Spain); HBOI, Harbor Branch Oceanographic Institute, Florida Atlantic University (Fort Pierce, FL, USA); MNCN, Museo Nacional de Ciencias Naturales (Madrid, Spain); MNHN, Muséum National d'Histoire Naturelle (Paris, France); MSNG, Museo Civico di Storia Naturale ''G. Doria'' (Genoa, Italy); NHM, Natural History Museum (London, UK); PC, personal collection of P. Cárdenas, Uppsala University (Sweden); RMNH, Rijksmuseum van Natuurlijke Historie, Naturalis Biodiversity Center (Leiden, The Netherlands); SME, Station Marine d'Endoume (Marseille, France); UPSZMC/UPSZTY, zoological collection at the Museum of Evolution (Uppsala, Sweden); ZMBN, zoological collection at the Bergen Museum (Bergen, Norway); ZMUC, Zoological Museum, University of Copenhagen (Denmark).

Type material from different museums were revised or re-examined for comparison with our specimens, especially from the natural history museums collections in London (UK), Paris (France) and Genoa (Italy). Notably, tetractinellids described by *Schmidt (1868)* from Algeria, currently stored at the MNHN Paris, were all examined. This historical collection gathers samples from the French 'Exploration Scientifique de l'Algérie' in 1842 and those collected by French zoologist Henri Lacaze-Duthiers in La Calle (El Kala) in 1860–1862, while he was studying the red coral.

## RESULTS

In total, we analyzed 174 samples, belonging to nine families, 17 genera and 36 species of tetractinellids. For a given specimen, different field codes were provided depending on the collection survey. Author field collection numbers follows the nomenclature ''Lit###'' for cave samples collected with free apnea or scuba diving, ''POR###'' for samples collected during the MEDITS surveys and ''i###'' for samples collected during INTEMARES surveys. Spicule measurements given in the text are always the range observed from several specimens, unless otherwise stated. Spicule measurements for specific specimens can be found in the Tables dedicated to the different species. Two large phylogenetic trees have been obtained with COI and 28S markers (Figs. S1 and S2) and subparts of these trees will be presented next to the descriptions of the species. Taxonomic authority of new species is restricted to Díaz & Cárdenas.

### Systematics

Class Demospongiae Sollas, 1885
Subclass Heteroscleromorpha *Cárdenas, Pérez & Boury-Esnault, 2012*
Order Tetractinellida Marshall, 1876
Suborder Astrophorina Sollas, 1887
Family Ancorinidae *Schmidt, 1870*
Genus *Stelletta Schmidt, 1862*
*Stelletta dichoclada Pulitzer-Finali, 1983*
(Figs. 4–6; Table 2)

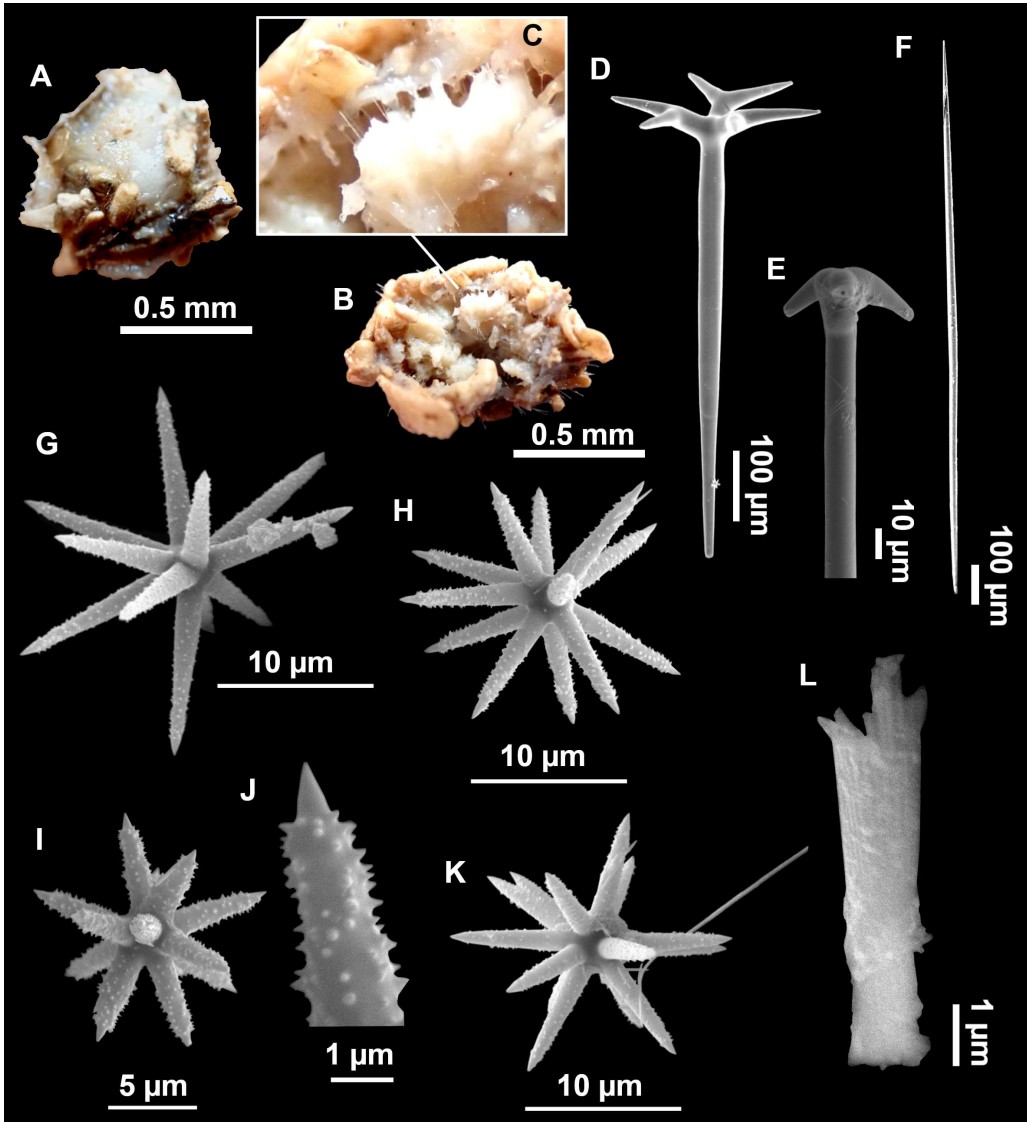

**Figure 4** *Stelletta dichoclada* (*Pulitzer-Finali, 1983*)**, specimens from the Balearic Islands.** (A) Habitus of field i715_2 on deck. (B) Habitus of field i589_1 after fixation, with (C) detail of the cortex. (D–L) SEM images of spicules from field i589_1. (D) Dichotriaene. (E) Detail of the head of an anatriaene. (F) Oxea. (G–K) Oxyasters. (L) Raphides in trichodragmata.

## Material examined

UPSZMC 190946, field#i416_A, MaC (EB), St. 177 (INTEMARES1019), 151 m, beam trawl, coll. J. A. Díaz; UPSZMC 190944, field#i589_1, MaC (AM), St. 21, (INTEMARES0720), 112 m, beam trawl, coll. J. A. Díaz; UPSZMC 190945, field#i715_2, MaC (EB), St. 45, (INTEMARES0720), 147 m, beam trawl, coll. J. A. Díaz.

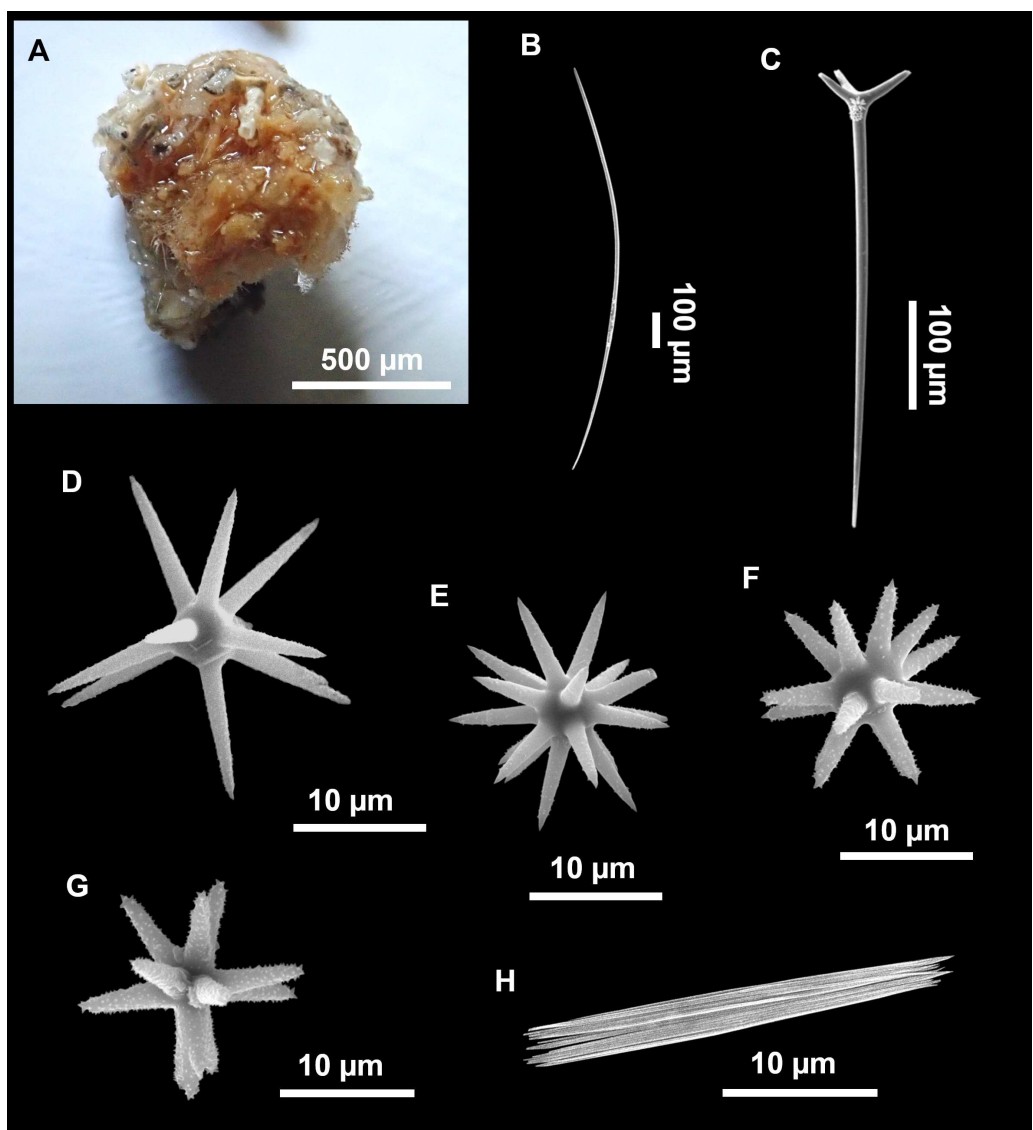

**Figure 5** **Holotype of *Stelletta dichoclada* MSNG 47152 (*Pulitzer-Finali, 1983*), from Corsica.** (B) Oxea. (C) Plagiotriaene. (D–G) Oxyasters to strongylasters. (H) Raphides in trichodragmata.

## Comparative material

*Stelletta dichoclada*, holotype, MSNG 47152, NIS.83.34a, off Calvi (Corsica) 123–147 m, detrital bottom, July 1969, dredge (Fig. 5).

*Stelletta lactea* Carter, 1871, UPSZMC 190949, Strangford Lough (Northern Ireland), 0 m, October 2021, collected by hand at low tide, coll. C. Morrow, id. C. Morrow.

## Outer morphology (Figs. 4A–4C and 5A)

Small subspherical, up to one cm in diameter, completely encrusted by calcareous sediment (Figs. 4A–4C), cortex and choanosome grayish in life and in EtOH. Sponges are slightly compressible, hispid to the naked eye. Cortex patent (Fig. 4C), about 1 mm in width.

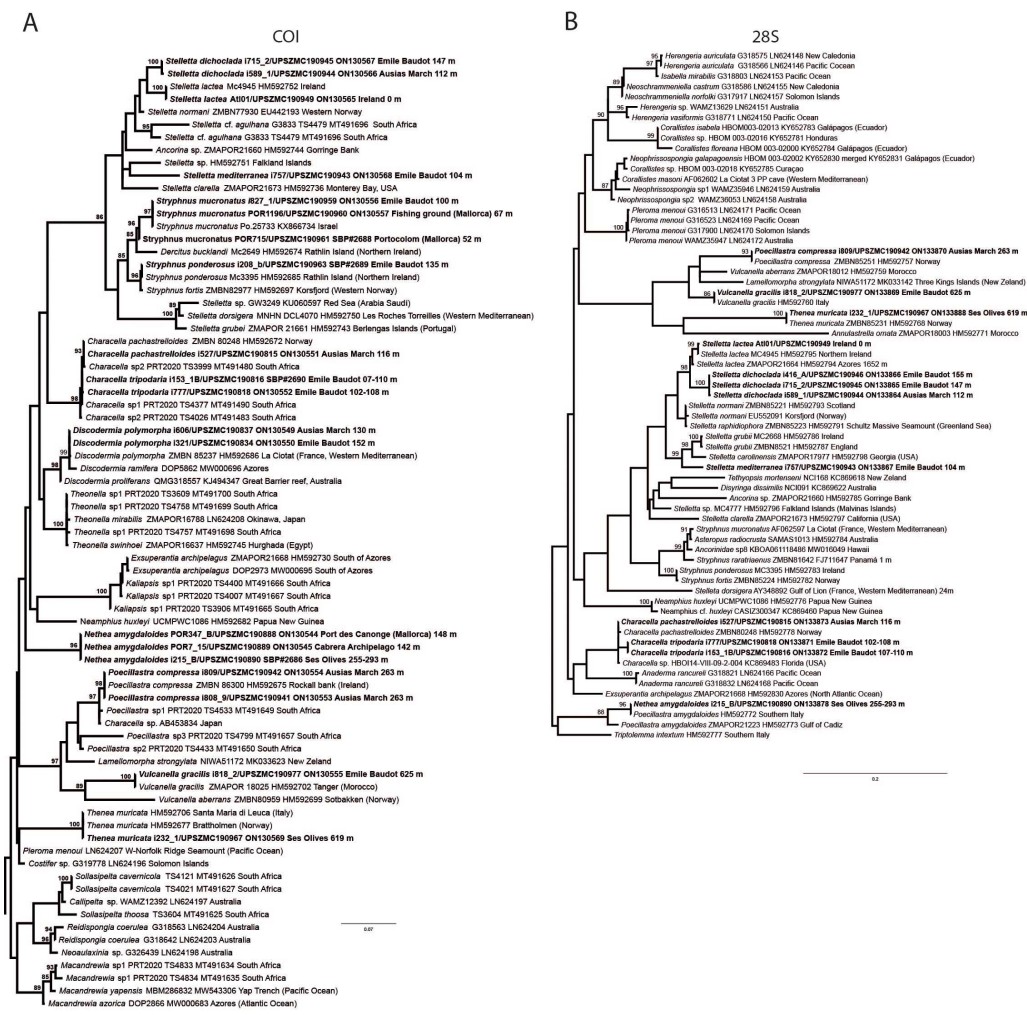

**Figure 6** Detail of COI (A) and 28S (B) phylogenetic trees showing the Ancorinidae family. In bold are new sequences from this study. Specimen codes are written as "field number/museum number" followed by Genbank accession number. The original trees can be seen as Figs. S1–S2.

## Spicules (Figs. 4D–4L Table 2)

Plagiotriaenes (as the ones observed in the holotype: Fig. 5C) small, fusiform, slightly curved rhabdome, with a slight swelling just below the cladome. Clads are pointed upwards. Rhabdome: 167–650 × 6–19 µm, clads: 26–96 × 6–16 µm. Plagiotriaenes are very scarce, and may represent immature stages of the dichotriaenes because it is common to find small and incipient bifurcated clads in the plagiotriaenes; also, the dichotriaenes are always larger.

Dichotriaenes (Fig. 4D), rhabdome robust, straight or slightly curved, fusiform. The short-sized dichotriaenes have protoclads longer or the same length as deuteroclads while in large-sized dichotriaenes, protoclads are shorter than deuteroclads. Occasionally, 1–2 clads may not be bifurcated. Rhabdome: 305-1,515/12-52 µm, protoclad 33-102/12-43 µm and deuteroclad 14-181/6-33 µm.

Peerj

**Table 2  Spicule measurements of *Stelletta dichoclada* and *Stelletta lactea*.** Measurements are given as minimum-mean-maximum for total length/minimum-mean-maximum for total width. All measurements are expressed in μm. Specimens here measured are in bold. Balearic specimen codes are the field#.

| Material | Depth (m) | Oxeas (length/width) | Anatriaene Rhabdome (length/width) Clad (length/width) | Plagiotriaenes Rhabdome (length/width) Clad (length/width) | Dichotriaenes Rhabdome (length/width) Protoclad (length/width) Deuteroclad (length/width) | Oxyasters (length) | Trichodragma (length/width) |
|---|---|---|---|---|---|---|---|
| **S. dichoclada holotype MSNG 47152 Corsica** | 123–147 | 1,134-1,463-2,555/13-19-32 | – | Rh: 167-391/6-15 (N = 2) Cl: 34-56/6-14 (N = 2) | Rh: 305-854-1,130/12-40-52 (N = 23) Pt: 33-55-71/12-33-43 Dt: 17-114-181/6-24-33 | 10-16-28 | 21-28-38/6-8-13 (N = 13) |
| **S. dichoclada i416_A  EB** | 151 | 829-1,863-3100/9-16-25 (N = 19) | – | Rh: 297-410-529/10-13-19 (N = 8) Cld: 26-59-85/9-11-15 (N = 8) | Rh: 585-1,059-1,398/20-32-45 (N = 25) Pt: 40-62-102/20-27-33 (N = 25) Dt: 27-88-123/15-20-31 (N = 25) | 7-13-25 (N = 100) | 22-27-34/4-7-12 (N = 36) |
| **S. dichoclada i589_1  AM** | 110 | 839-1,602-2,520/7-16-24 (N = 21) | Rh: 1,659-1,841/8-11 (N = 2) Cld: 15-23/6-9 (N = 2) | Rh: 412/17 Cld: 69/16 (N = 1) | Rh: 383-741-982/16-27-36 (N = 20) Pt: 41-57-77/15-23-33 Dt: 14-70-109/6-16-24 (N = 20) | 7-13-24 (N = 63) | 19-29-34/6-9-10 (N = 18) |
| **S. dichoclada i715_2  EB** | 150 | 1,130-1,762-2,067/8-20-30 (N = 14) | Rh: 2,799 (N = 1)/9-14 (N = 3) Cld: 33-38-41/9-10 (N = 4) | Rh: 276-650/10-18 (N = 4) Cld: 45-96/8-16 (N = 4) | Rh: 540-1,170-1,515/18-34-44 (N = 7) Pt: 39-74-101/16-30-36 (N = 7) Dt: 32-84-117/11-21-28 (N = 7) | 9-16-31 (N = 58) | 18-25-32/6-9-18 (N = 20) |
| **S. lactea UPSZMC 190949 N. Ireland** | 0 | 480-817-1,361/6-19-42 (N = 16) | – | Rh: 286-421-686/10-16-29 Cl: 58-94-181/10-15-26 (N = 19) | Rh: 262-477-725/11-19-26 (N = 5) Pt: 30-61-109/12-17-25 Dt: 22-35-62/8-13-18 (N = 6) | 6-10-13 | 28-36-45/6-12-19 (N = 8) |
| S. lactea Holotype Devon (North Atlantic) (*Sollas, 1888*) | Littoral | 1,250/- | – | Rh: 825/- Cl: - | Rh: 825/- Cl:-/- | 12.5 | 25/- |

*(continued on next page)*

Díaz et al. (2024), *PeerJ*, DOI 10.7717/peerj.16584

**Table 2** (*continued*)

| Material | Depth (m) | Oxeas (length/width) | Anatriaene Rhabdome (length/width) Clad (length/width) | Plagiotriaenes Rhabdome (length/width) Clad (length/width) | Dichotriaenes Rhabdome (length/width) Protoclad (length/width) Deuteroclad (length/width) | Oxyasters (length) | Trichodragma (length/width) |
|---|---|---|---|---|---|---|---|
| *S. lactea* Ionian Sea (*Pulitzer-Finali, 1983*) | 2-3 | 630-850/- | – | Rh: 350-550/- Cl: 350/- (reduced clads and rhabdome) | Very rare | 5-11 (spherasters to oxyasters) | 25/- |

**Notes.**

Rh, rhabdome; pc, protoclad; dc, deuteroclad; -, not found/not reported.

EB, Emile Baudot; AM, Ausias March; SO, Ses Olives.

Anatriaenes (Fig. 4E), very scarce, most are broken, with slightly curved rhabds. Rhabdome: 1,659-2,799/8-14 µm, cladi 14-41/6-10 µm.

Oxea (Fig. 4F), thin, slightly curved, and fusiform, 829-3,100/7-32 µm.

Oxyasters (Figs. 4G–4K), spherical, with short centrum and large actines, spiny along the whole actine with a clear pointy end. Only one size category, 7–31 µm in diameter.

Raphides in trichodragma (Fig. 4L), trichodragma length/width: 18-38/4-18 µm.

## Ecology notes

Always found on sedimentary bottoms with sand and gravels, at mesophotic depths between 112 m and 151 m.

## Genetics

We have sequenced the Folmer fragment in two pieces for specimens i589_1 and i715_2 (ON130566 and ON130567) and 28S (C1-D2) for all three specimens (ON133865, ON133864 and ON133866). The Folmer fragments of i589_1 (AM Seamount) and i715_2 (EB Seamount) have 1 bp difference. The 28S (C1-D2) of i589_1 (AM) and i416_A/#i715_2 (EB) have 1 bp difference.

## Remarks

Specimens were found on the EB and AM seamounts at 112–151 m. This is the second record of *S. dichoclada* in the literature since its original description by *Pulitzer-Finali (1983)* from a specimen collected off Corsica at similar depths (123–147 m). The holotype is a small hemispherical sponge (Fig. 5A), 0.7 cm in diameter × 0.7 cm in height; openings not visible. Cortex is conspicuous, about 1 mm in width, beige in color, crusty to the touch, resilient, and incorporates sediment. Choanosome dirty orange, softer than the cortex (fleshier). Spicules of the holotype have been re-measured (Table 2) and examined with SEM (Figs. 5B–5H). Our material differed from the holotype by the presence of a few anatriaenes in specimens i589_1 and i715_2, but not in the holotype, nor i416_A. However, anatriaenes were very uncommon, which may explain their absence in the holotype and i416_A slides. Importantly, trichodragmas were not mentioned in the original description but they are definitely present in the holotype and our specimens (Figs. 4 and 5). To ensure that those are not foreign material, we have made digestions from two different parts of the holotype body: both contained trichodragmas. The holotype contains several foreign spicules, notably microtriods to microcalthrops with annulated rugose surface, identical to those described in the same work for *Annulastrella verrucolosa* (*Pulitzer-Finali, 1983*), collected at the same station. Also, there are foreign spirasters similar to those from the order Clionaida Grant, 1826.

We have not tried to amplify the DNA from the holotype because it had been conserved in formalin. Surprisingly, in the Balearic specimens we detected two haplotypes for both markers, each time with 1 bp difference. One haplotype corresponded to i589_1, collected at the AM, while the other haplotype was shared with i416_A and i715_2, from the EB. This may suggest that each seamount harbors isolated populations, or perhaps that *S. dichoclada* represents a species complex with two cryptic species. This should be assessed in further studies, by using more variable markers and sequencing more individuals.

*Stelletta dichoclada* appears to belong to a clade with North Atlantic species (*S. lactea*, *Stelletta normani* Sollas, 1880 and *Stelletta rhaphidiophora* Hentschel, 1929) (Fig. 6 and Figs. S1–S2). In this clade all species share dichotriaenes, trichodragmas and one category of oxyasters. For both markers, the closest sister-species is the shallow-water to intertidal North Atlantic *S. lactea*. We sequenced a specimen of *S. lactea* from the intertidal area off Northern Ireland (UPSZMC 190949, ON130565 (COI), OR044718 (28S)). *Stelletta dichochlada* and *S. lactea* UPSZMC 190949 have respectively 17–18 pb difference in COI and 19–20 bp difference in 28S so they are significantly different genetically, despite their morphological strong similarities. A SEM plate for *S. lactea* has also been made to compare the microscleres (Fig. S3). In fact, both *S. lactea* and *S. dichoclada* share similar spicular types, with similar morphologies: there is no clear spicule or external morphological difference between these species. Our *S. lactea* comparative specimen did have shorter/thicker oxeas, dichotriaenes with shorter rhabds, smaller oxyasters and longer trichodragmas (Table 2) but this would need to be confirmed with the measurements of several more *S. lactea* specimens.

However, there are also several Mediterranean records of *S. lactea*: Gulf of Lion (*Boury-Esnault, 1971*; *Pouliquen, 1972*), the Tyrrhenian Sea (*Sarà, 1958*; *Sarà & Siribelli, 1960*) and the Ionian Sea (*Pulitzer-Finali, 1983*), all from shallow waters, in agreement with most North Atlantic specimens but unlike *S. dichoclada*, which seems to be a mesophotic species. There are no sequences of Mediterranean *S. lactea* specimens, so its presence in the Mediterranean Sea cannot be confirmed, and relatedness with *S. dichoclada* cannot be currently assessed.

*Stelletta mediterranea* (*Topsent, 1893*)
(Figs. 6–7, Table 3)

## Material examined
UPSZMC 190943, field#i757, MaC (EB), St. 53, (INTEMARES0720), 97–102 m, rock dredge, coll. J. A. Díaz.

## Comparative material
*Stelletta mediterranea*, holotype, MNHN DT2305 (two spicule slides), Cap l'Abeille, Banyuls, France, 30–40 m.

## Outer morphology
Small, about 1.3 cm in diameter (Fig. 7A). Hemispherical body polarized in upper (rounded) and basal (flattened) parts; hispid surface. Externally pink when alive, grayish after preservation. Choanosome color not recorded on deck, grayish after preservation. Free of agglutinated sand. Cortex 1 mm thick. Hard consistency, barely compressible. Openings inconspicuous.

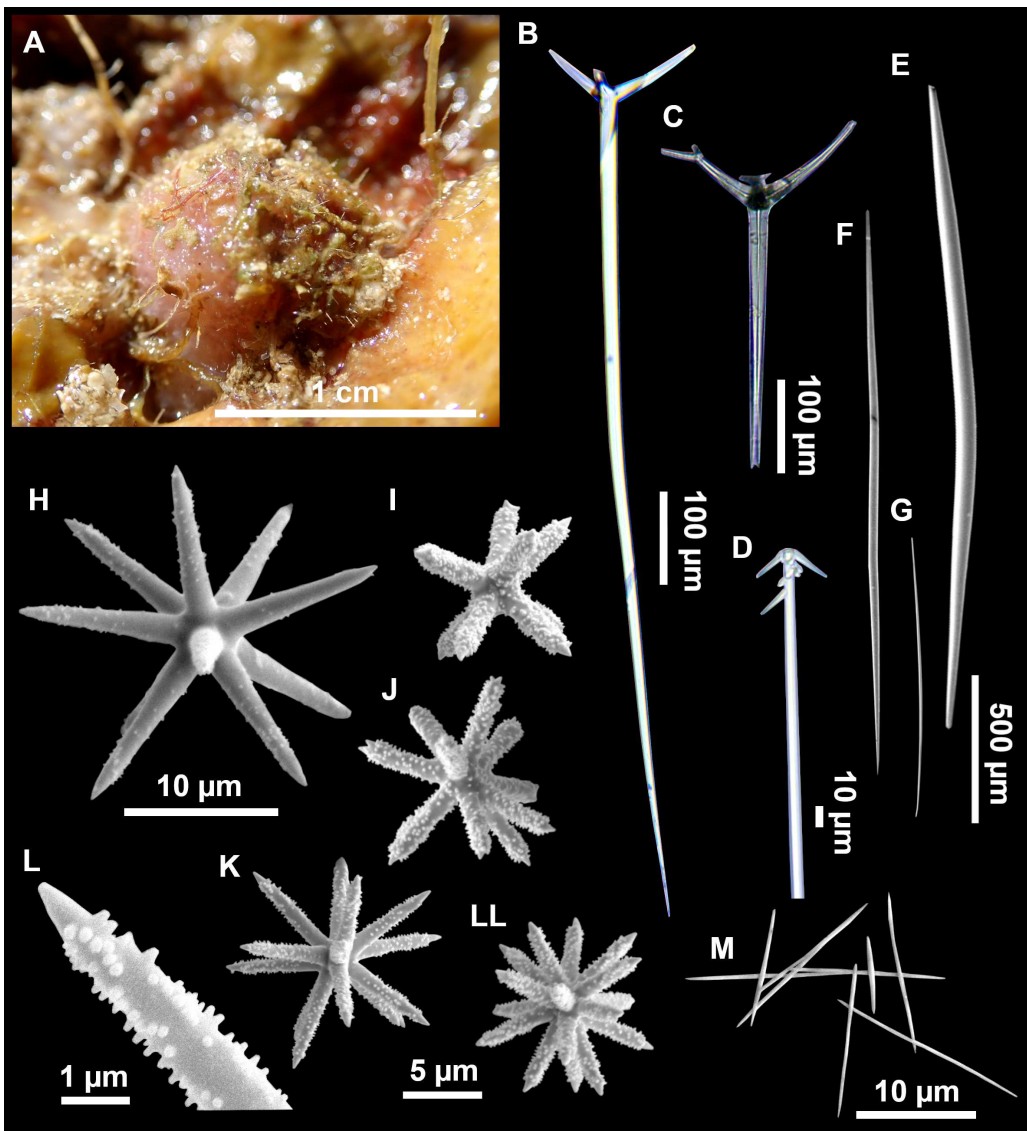

**Figure 7** *Stelletta mediterranea* (*Topsent, 1893*), **specimen #i757.** (A) Habitus on deck, before fixation. (B–D) Optical microscope images. (B) Plagiotriaene. (C) Plagiotriaene with bifurcated clads. (D) Anatriaene. (E–M) SEM images (E–F) Oxea I. (G) Oxea II. (H–LL) Oxyasters. (M) Raphides.

## Spicules

Plagiotriaenes (Fig. 7B), very scarce, rhabdome slightly curved, fusiform, measuring 240–1070 ($N = 3$)/13-27 ($N = 6$) μm. Cladome, with clads measuring 73-132/13-24 ($N = 6$) μm. A single plagiotriaene with two bifurcated clads was observed (Fig. 7C), of the same length as the regular plagiotriaenes.

Anatriaene (Fig. 7D), rare, with teratogenic clads in form of aborted hooks just beneath the cladome.

Oxea I (Figs. 7E–7F), large, robust, fusiform, most are bent at the middle, 1,528-2,189-2,699/24-47-74 μm

Díaz et al. (2024), *PeerJ*, DOI 10.7717/peerj.16584

**Table 3** **Spicule measurements of *Stelletta mediterranea*.** Measurements are given as minimum-mean-maximum for total length/minimum-mean-maximum for total width. All measurements are expressed in μm. Specimens here measured are in bold. The Balearic specimen code is the field#.

| Material | Depth | Macroscopic features | Oxeas (length/width) | Anatriaene Rhabdome (length/width) Clad (length/width) | Plagiotriaenes Rhabdome (length/width) Clad (length/width) | Oxyasters (length) | Trichodragmas (length/width) |
|---|---|---|---|---|---|---|---|
| **Holotype, Banyuls, France MNHN DT 2305** )* (*Topsent, 1894*) | 30–40 | Encrusting, hispid, 4–8 mm thick | I. 866-1,618-2,048/ 10-45-57 II.)* 650-1,300/3-4 | Rh: -/8 Cl: 37/- (N = 1) | Rh: 765-1,030-1,244/ 13-39-50 Cl: 38-122-175/- | 8-12-16 (N = 3) (abundant) | 17)*/- |
| **i757 EB** | 105 | Hemispherical, hispid, 1.3 cm | I. 1,528-2,189-2,699/ 24-47-74 (N = 45) II. 694-1,151-1,418/ 5-11-17 (N = 23) | Rh: 1,652-1,761/4-7 (N = 2) Cl: 12-18/3-4 (N = 2) | Rh: 240-1,070 (N = 3)/ 13-27 (N = 6) Cl: 73-132/ 13-24 (N = 6) | 9-14-24 (N = 48) | 13-16-20/ 3-6-10 (N = 14) |
| Alboran Sea (*Pansini, 1987*) | 70-80 | Massive, cylindrical, hispid, 8–2.5 cm | I. 2,750/66 II. 700-800/4-6 | Rh: 1,600/14 Cl: -/- | Rh: 950/30 Cl: 115/35 | 8-17 | – |

**Notes.**

Rh, rhabdome; cl, cladome; -, not found/not reported.

EB, Emile Baudot.

*In cases where measurements originate from various authors, asterisks are employed to indicate the respective source.

Oxea II (Fig. 7G), small, thin, fusiform, bent at the middle, some slightly flexuous, 694-1,151-1,418/5-11-17 ($N = 23$) μm.

Oxyasters (Figs. 7H–7LL), abundant, only one size category (9-14-24 μm) but small ones are strongyluster-like, while larger ones are more like oxyasters; 5–18 actines, less actines in larger oxyasters. Spines are distributed all along the actine in small oxyasters, and absent near the centrum in large ones.

Raphides in trichodragma (Fig. 7M), length/width measuring 13-16-20/3-6-10 ($N = 14$).

### Ecology notes

Found at the shallowest part of the EB summit (104 m). The area was rich in sponges and in coralligenous red algae. Epibiont on a large Irciniidae.

### Genetics

Only the second part of the Folmer COI fragment (ON130568) was obtained; 28S (C1-D2) was also sequenced (ON133867).

### Taxonomic remarks

This specimen is assigned to *S. mediterranea*, a poorly known species described by *Topsent (1893)* in Banyuls (France), and later recorded in the Alboran and Aegean seas (*Pansini, 1987*; *Vamvakas, 1971*). The spicules from the type slides were re-measured for the present study (Table 3). Similarities between our material and *S. mediterranea* are: (i) presence of two categories of oxeas, (ii) presence of plagiotriaenes (although Topsent called those spicules orthotriaenes, they are clearly pointing forward, (see Topsent, 1984, Plate XIV, Fig. 3)) and especially (iii) characteristic anatriaenes with teratogenic clads (Fig. 7D). Also, spicular sizes of both megascleres and microscleres fit with those of the holotype and *Pansini (1987)* (Table 3): the plagiotriaenes in the type are slightly more robust and its oxeas II are slightly thinner. Our specimen and the one from *Pansini (1987)* share a similar pink color when alive. Trichodragmas were not reported by *Pansini (1987)*, but they could have been missed since they are not abundant in our specimen. The 28S tree (Fig. 6B) clearly suggests that this species groups with *Stelletta grubii Schmidt, 1862* and *Stelletta carolinensis* (Wells, Wells & Gray, 1960) (well-supported) while its position with COI (Fig. 6A) is more ambiguous (not supported) and could be explained by the fact that we only have a small sequence.

*Stelletta mortarium* **sp. nov.** Díaz & Cárdenas
(Figs. 8–10, Table 4)

### Etymology

Due to its resemblance to a ''morter'', a type of ancient pottery kitchen bowl commonly used in Mallorcan cuisine.

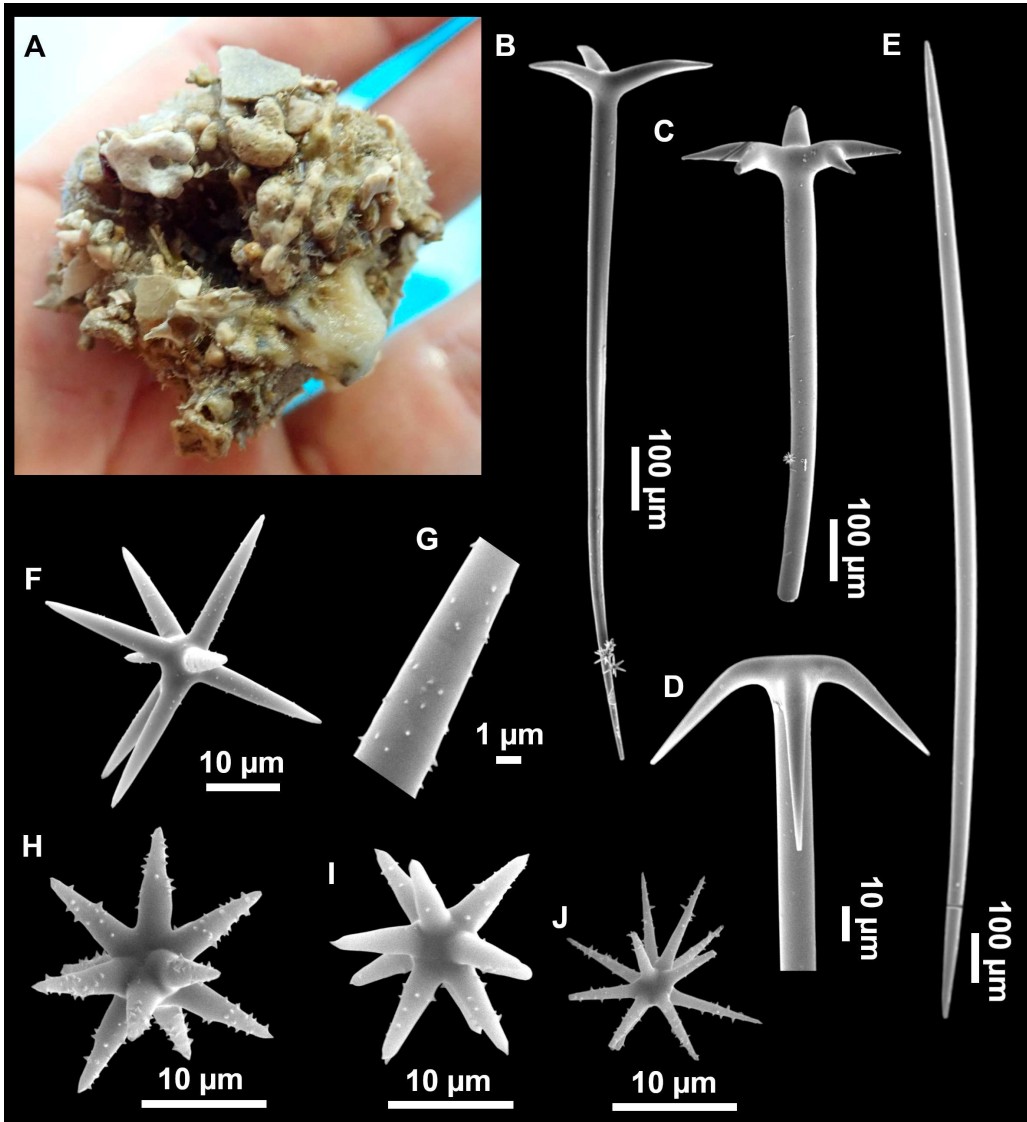

**Figure 8** **Holotype (UPSZTY 190957) of *Stelletta mortarium* sp. nov.** (A) Habitus on deck before fix-
ation. (B–J) SEM images of the holotype spicules. (B) Orthotriaenes. (C) Dichotriaenes. (D) Detail of the
cladome of an anatriaene. (E) Oxea I. (F) Oxyaster I, with detail of the spines (G). (H–J) Oxyasters II at
different development stages.

## Material examined

Holotype: UPSZTY 190957, field#i714_1, St. 45 (INTEMARES0720), MaC (EB), beam
trawl, 150 m, coll. J. A. Díaz (Fig. 8).

Paratypes: UPSZTY 190950-51, field#i352_1 and field#i352_2, St. 136, MaC (EB), beam
trawl, 146 m, coll. J. A. Díaz; UPSZTY 190952, field#i401_2, St. 167, MaC (EB), beam
trawl, 151 m, coll. J. A. Díaz; UPSZTY 190953-54, field#i406-A and field#i406-B, St. 167,
MaC (EB), beam trawl, 151 m, coll. J. A. Díaz.

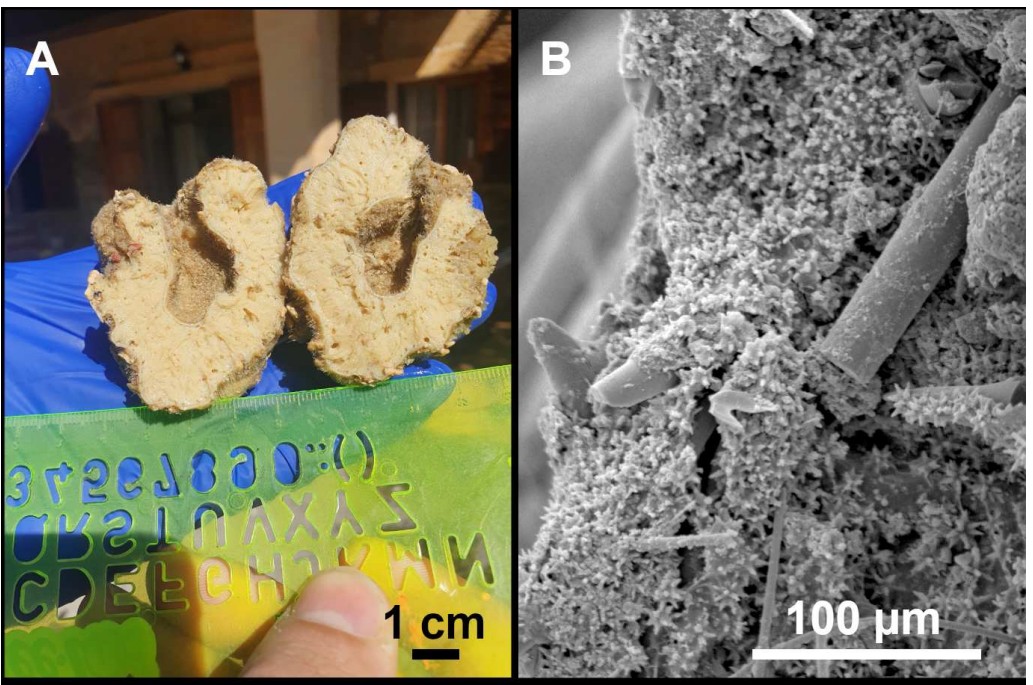

**Figure 9** ***Stelletta mortarium* sp. nov., paratype #i594.** (A) Transversal view after ethanol. (B) Detail of the cortex made up by oxyasters II.

Other specimens: UPSZMC 190955-190956, field#i582 and field#i594 (Fig. 9), St. 21 (INTEMARES0720), MaC (AM), beam trawl, 109 m, coll. J. A. Díaz.

## Comparative material

*Stelletta defensa* Pulitzer-Finali, 1983, holotype, MSNG 47153, NIS.83.36, Calvi, Corsica, Ligurian Sea, July 1969, dredge, 121–149 m, detrital bottom (Fig. S4); paratype, MSNG 47154, NIS.85.3, July 1969, dredge, 121–149 m, detrital bottom.

*Stelletta dorsigera* Schmidt, 1864, MNHN DCL4070, Roches Toreilles, France, 25 m, Oct. 1994, id. J. Vacelet and N. Boury-Esnault, COI: HM592750; 28S: AY348892.

*Stelletta grubii* Schmidt, 1862, BELUM Mc2668, Rathlin Ireland, Northern Ireland, summer 2005, id. B. Picton, 28S: HM592786 (Cárdenas et al., 2011).

*Stelletta hispida* (Bucchich, 1886), ZMBN 25636, Gulf of Cadiz, 1215 m, id. (Arnensen, 1920 (1932)).

*Stelletta tuberosa* (Topsent, 1892), MNHN DCL4066, Bay of Biscay, 4400 m, BIOGAS V expedition (Centob), id. P. Cárdenas.

*Stelletta simplicissima* (Schmidt, 1868), holotype, MNHN Schmidt collection#62, Algiers.

*Stelletta stellata* Topsent, 1893, UPSZMC 190958, South of Porto Cesareo lagoon, Apulia, SE Italy, 0.5 m, 27 July 2017. coll. P. Cárdenas and F. Cardone, id. P. Cárdenas and F. Cardone.

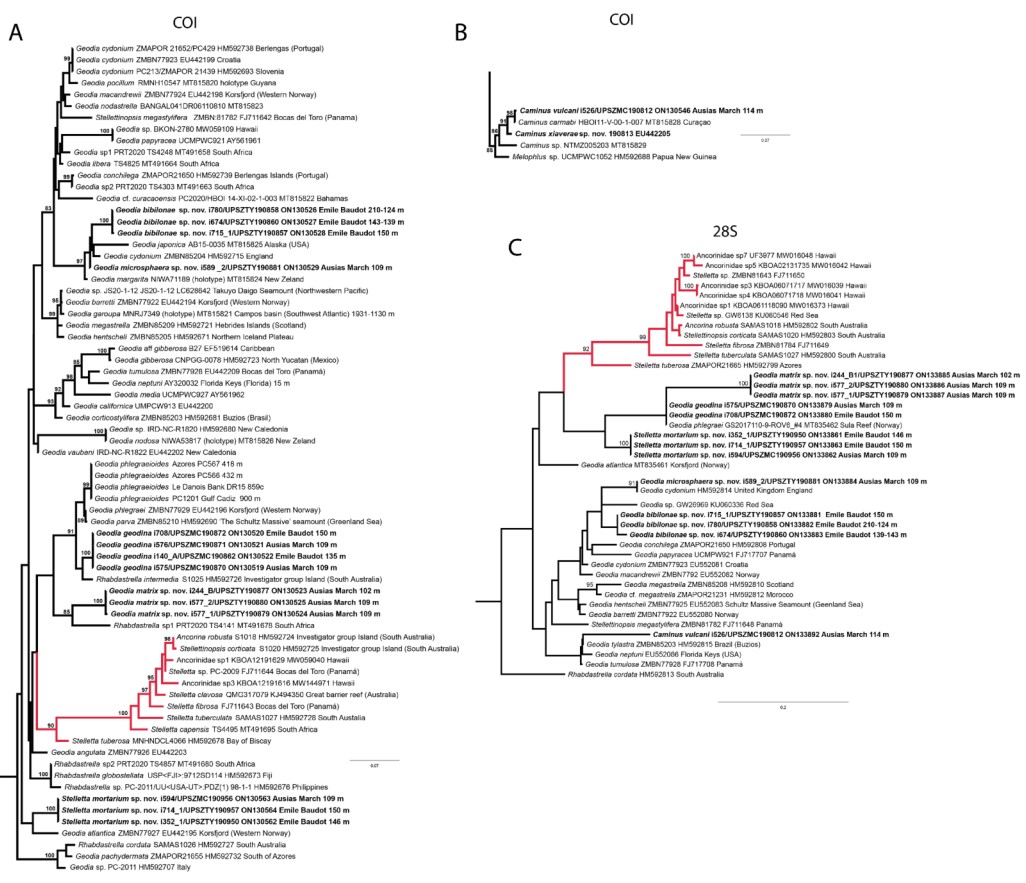

**Figure 10** Details of the COI (A–B) and 28S (C) trees of Geodiidae including the 'Geostelletta' clade (in red). Specimen codes are written as "field number/museum number" followed by Genbank accession number. The original trees can be seen as Figs. S1–S2.

## Outer morphology

Massive, circular to ellipsoid sponges, 3–6.5 cm in diameter, 2.5–6.5 cm in height with an atrium on its upper side (Figs. 8A; 9A). The atrium also has an ellipsoid shape, the opening 1.5–3 cm in diameter, and subsequent hole 1.5–3 cm deep (Fig. 9A). In specimen i582, the atrium does not generate a hole, but a concave depression at the surface. Color alive grayish (Fig. 8A). In EtOH, surface color dark gray and choanosome cream (Fig. 9A). Hard consistency, slightly compressible. Hispidation visible to the naked eye, present all over the surface, including the atrium. The atrium contains many small uniporal oscules, each with its own sphincter. Minute cribriporal pores are distributed on the sides of the specimens. Cortex ~0.5 mm thick. Abundant sediments or pebbles are incorporated into the surface, but not in the choanosome, which is fleshy.

## Spicules

Orthotriaenes (Fig. 8B), stout rhabdome, slightly curved, fusiform, with a sharp tip, measuring 482-1865/12-65 µm. Cladome also stout and with a sharp tip, 43-287/11-50

μm. The smallest triaene showed a marked swelling beneath the cladome, and its clads were more triangular.

Dichotriaenes (Fig. 8C), rare, only in specimens i714_1 and i401_2. Same size and morphology as orthotriaenes, rhabdome measuring 1,159-1,498/39-52 μm, while cladome measuring 29-71/33-44 μm (protoclad) and 86-133/24-36 μm (deuteroclad).

Anatriaenes (Fig. 8D), uncommon, rhabdome straight and stout, 1,626-3,055/5-20 μm. Cladome with tips of the cladi sharp, curved inwards, 18-87/3-18 μm. Some with underdeveloped cladome, resembling oxeas.

Protriaenes, very rare, rhabdome thin and slightly curved, measuring 1,156-1,267/5-9 μm. Cladi measuring 38-82/3-7 μm. Not found in i402_1 and i582.

Oxea I (Fig. 8E), robust, slightly curved, fusiform, 791-2,762/8-58 μm.

Oxea II, slender, slightly curved or flexuous, 753-1,627/5-11 μm. Common in specimen i352_1, very rare in i401_2, i582 and i714_1, and not observed in i594. Some have at their tip structures that remind of an aborted cladome so they may be anatriaenes with underdeveloped clads: this was pointed out by *Topsent (1893)* when describing similar oxeas II in *S. mediterranea.* which has similar oxeas II.

Oxyasters I (Fig. 8F), choanosomal, having 6–11 long actines, faintly spined (Fig. 8G), small centrum, 11–47 μm in diameter.

Oxyasters II (Figs. 8H–8J and 9B), ectosomal, having 9–15 short actines with more robust spines than oxyasters I and a centrum that is about 1/3 of the total diameter. The centrum is devoid of spines; overall measuring 11–24 μm.

### Ecology notes

The species was found in the AM and the EB, on detrital bottoms with gross sand and gravels, from 111–152 m depth.

### Genetics

COI (ON130562, ON130563, ON130564) and 28S (C1-C2) (ON133861, ON133862, ON133863) markers were obtained from i352_1 (paratype), i594 and i714_1 (holotype).

### Taxonomic remarks

There are 10 species of *Stelletta* without raphides in the Northeast Atlantic/Mediterranean region: *S. dorsigera, S. grubii, S. hispida, S. addita* (*Topsent, 1938*), *S. simplicissima, Stelletta pumex* (*Nardo, 1847*), *S. defensa, S. stellata, S. tuberosa* and *Stelletta ventricosa* (*Topsent, 1904*). *S. dorsigera* has a conspicuous dark cortex with characteristic conules, unlike the cortex of *S. mortarium* sp. nov. Also, *S. dorsigera* is subspherical and not bowl shaped like our specimens. *S. dorsigera* has smaller oxyasters than *S. mortarium* sp. nov., measuring 8–12 μm (*Uriz, 1981*) *versus* 11–47 μm. *S. grubii* lacks anatriaenes and protriaenes and its orthotriaenes have much shorter and downwards curved clads. Besides, COI/28S of *S. dorsigera* and *S. grubii* are far apart from our COI/28S sequences in our phylogenetic analyses (Fig. 10).

*S. hispida* has large plagiotriaenes instead of ortho/dichotriaenes, anatriaenes and protriaenes. Besides, it has styles instead of oxeas and a small (2, 5 cm) spherical body shape (*Bucchich, 1886*). The size of the ortho/dichotriaenes in *S. mortarium* sp. nov. are

Díaz et al. (2024), *PeerJ*, DOI 10.7717/peerj.16584

**Table 4 Spicule measurements of *Stelletta mortarium* sp. nov., given as minimum-mean-maximum for total length/minimum-mean-maximum for total width; all measurements are expressed in μm.** Specimen codes are the field#.

| Material | Depth (m) | Oxeas (length/width) | Anatriaenes Rhabdome (length/width) Clad (length/width) | Protriaenes Rhabdome (length/width) Clad (length/width) | Orthotriaenes Rhabdome (length/width) Clad (length/width) | Dichotriaenes Rhabdome (length/width) Protoclad (length/width) Deuteroclad (length/width) | Oxyasters (length) |
|---|---|---|---|---|---|---|---|
| **i714_1 (UPSZTY 190957) holotype EB** | 152 | I. 1,300-1,913-2,548/13-33-58 (*N* = 25) II. 1531/8 (*N* = 1) | Rb: -/5-13 (*N* = 3) Cl: 18-55/3-12 (*N* = 3) | Rb: 1,156/6 (*N* = 1) Cl: 69/5 (*N* = 1) | Rh:720-1,093-1,499/26-48-65 | Rh: -/52 (*N* = 1) Pt: 42/44 (*N* = 1) Dt: 107/32 (*N* = 1) | I. 16-34-47 II. 16-19-24 (*N* = 15) |
| **i352_1 paratype EB** | 146 | I. 791-1,480-2,262/8-19-38 (*N* = 13) II. 753-967-1,611/5-8-10 (*N* = 12) | Rh: 2,007-2,295/10-12 (*N* = 5) Cl: 26-53/8-11 (*N* = 5) | Rh: 1,194-1,216/5-8 (*N* = 2) Cl: 38-82/3-6 (*N* = 2) | Rh: 705-1,213-1,579/14-44-55 (*N* = 15) Cl: 48-147-221/13-36-46 (*N* = 15) | – | I. 14-27-37 II. 11-15-23 |
| **i401_2 paratype EB** | 150 | I 2,025-2,374-2,762/16-27-36 (*N* = 16) II. 985-1134/8-11 (*N* = 3) | Rb: 1,760-2,605-3,055/10-15-20 (*N* = 8) Cl: 32-61-87/10-13-18 (*N* = 8) | – | Rh: 800-1,396-1,865/24-39-51 (*N* = 12) Cl: 76-138-201/19-33-50 (*N* = 12) | Rh: 1,159-1,498/39-48 (*N* = 2) Pt: 29-71/33-42-33 (*N* = 2) Dt: 86-133/24-36 (*N* = 2) | I. 11-25-38 II. 8-13-17 |
| **i582 AM** | 112 | I. 1,599-2,469/15-51 (*N* = 7) II. 1,627/7 (*N* = 1) (flexuous) | Rb: 1,626-2,214 (*N* = 2)/7-13 (*N* = 4) Cl: 22-74 (*N* = 4)/5-11 (*N* = 4) | – | Rb: 482-1,030-1,713/12-30-53 Cl: 43-132-287/12-26-49 | – | I. 12-26-38 II. 14-15-19 (*N* = 18) |
| **i594 AM** | 112 | I.1,437-2,019-2,592/15-28-43 II. - | Rb: 2,516/14 Cl: 71/10 | Rb: 1,267 (*N* = 1)/9 (*N* = 2) Cl: 69-84/7 (*N* = 2) | Rb: 513-1,129-1,763/14-36-64 Cl: 45-141-270/11-29-47 | – | I. 15-26-38 II. 8-15-21 |

**Notes.**

Rh, rhabdome; Cl, clad; pc, protoclad; dc, deuteroclad; -, not found/not reported.

EB, Emile Baudot; AM, Ausias March.

four times longer and nearly two times thicker than in *S. addita*: dichotriaenes of *S. addita* have a rhabdome 225-350/25 µm (*vs.* 1,159-1,498/39-48 µm in *S. mortarium* sp. nov.). Also, the triaenes of *S. addita* are mostly dichotriaenes, while *S. mortarium* sp. nov., has mostly orthotriaenes. Moreover, no anatriaenes nor protriaenes were described for *S. addita*. Descriptions of the type of *S. pumex* are very poor (*Nardo, 1847*; *Schmidt, 1864*) but if we follow the short redescription made by *Sollas (1888)* it appears *S. pumex* has only one type of aster which can be quite variable (*vs.* two types of aster in *S. mortarium* sp. nov.) and only plagiotriaenes (*versus* essentially orthotriaenes in our species, along with some dicho-, ana- and protriaenes).

We found that in the MNHN *Schmidt (1868)* collection the holotypes of *S. mucronatus* (found in jar#63 labeled '*Myriastra simplicissima*' and '*Myriastra addita*') and *Stelletta simplicissima* (found in jar#62 labeled '*Stelletta mucronata*') had been exchanged. Furthermore, according to the labels of jar#63 (MNHN DT 758), the holotypes of *S. simplicissima and S. addita* should have been stored together, since originally, they were both identified as *S. simplicissima* by *Schmidt (1868)*. However, the holotype of *S. addita* was missing from either jar (#62 or #63). This may have happened when *Topsent (1938)* revised the Schmidt collection and described *S. addita*, the types were not placed back properly and the holotype of *S. addita* was misplaced and is presumably lost. The holotype of *S. simplicissima* is a brown subglobular specimen 2 × 3.5 cm. It has only stout plagiotriaenes with short clads (70–153 µm, our measurements) and very robust oxeas (2,300-2,700/80 µm, our measurements), while the largest oxeas of *S. mortarium* sp. nov. are 8–58 µm thick. Also, newly made thick sections of the holotype showed that *Schmidt (1868)* and *Sollas (1888)* overlooked short trichodragmas ~12–15/5 µm long, which are not very abundant but clearly present close to the cortex. As for *S. addita*, it has essentially dichotriaenes (and some rare orthotriaenes) and two sizes of strongylasters (*Topsent, 1938*) while *S. mortarium* **sp. nov.** has essentially orthotriaenes and two sizes of oxyasters. *S. tuberosa* is a deep-sea species from the North Atlantic found deeper (454–4,400 m) than our specimens, they have a subspherical shape, much bigger megascleres and oxyasters (*Cárdenas & Rapp, 2015*). Their COI/28S is also quite different from those of our new species (Fig. 10).

The secondary loss of sterrasters in some *Geodia* (Geodiidae family) results in the same spicule repertoire as *Stelletta* species (Ancorinidae family), with triaenes, oxeas and asters (*Cárdenas et al., 2011*). *Stelletta* is therefore currently polyphyletic, with several of its representatives (*e.g.*, *S. tuberosa*) grouping in a temporarily named 'Geostelletta' clade while others are true ancorinids (*Stelletta* sensu stricto) (*Cárdenas et al., 2011*). Both COI and 28S suggest that *S. mortarium* sp. nov. groups with *Geodia*, but not in the 'Geostelletta' clade, thereby suggesting there are several *Stelletta*-like *Geodia* clades amongst the *Geodia*. Actually, the position of *S. mortarium* sp. nov. is somewhat uncertain and poorly supported even within *Geodia* (Fig. 10). We refrain from allocating this species in the genus *Geodia*, until more species of *Stelletta* are sequenced so that new genera or subgenera can be formally created and defined based on shared morphological characters. We further note that so far none of the *Stelletta*-like *Geodia* possess raphides/trichodragmas, a spicule absent in the Geodiidae in general, so the presence of this spicule could be a good character to discriminate more efficiently some of the *Stelletta* sensu stricto species.

Genus *Stryphnus Sollas, 1886*
*Stryphnus mucronatus* (*Schmidt, 1868*)
(Figs. 2D, 6A and 11, Table 5)

## Material examined

UPSZMC 190959, field#i827_1, St. 25 (INTEMARES0820), MaC (EB), ROV, 100 m; UPSZMC 190960, field#POR1196, St. 212 (MEDITSGSA521), east of Mallorca (Cala Ratjada), 63 m, GOC-73, coll. J. A. Díaz; UPSZMC 190961, field#POR715, St. 184 (MEDITSGSA519), east of Mallorca (Portocolom), 52 m, GOC-73, coll. J. A. Díaz.

## Comparative material

*Stryphnus mucronatus*, holotype, MNHN DT758, Schmidt collection#63, Algeria, 'Exploration Scientifique de l'Algérie', 1842; PC440, field#GOR 06.80, Gettysburg Peak, Gorringe Bank, scuba diving, 36–42 m, LusoExpedição 2006, coll. J. R. Xavier, specimen mentioned in *Xavier & Van Soest (2007)*.

## Outer morphology

Massive sponges, 5–12 cm in diameter (Fig. 11A). All three specimens grow on several calcareous red algae, which are included in the sponge body. Specimen i827_1 serves as a substrate for *Haliclona (Flagellia)* sp., *P. monilifera* and a calcareous sponge. Oscula cribiporal, with several orifices surrounded by a sphincter. On the video recording we observe an oscule, 1 cm in diameter (Fig. 2D). However, on deck we observe 2–4 contracted orifices, measuring 2–4 mm in diameter. Pores inconspicuous. Same color on deck and in EtOH: dark black cortex and a slightly paler choanosome (Fig. 11B). EtOH strongly colored dark by the specimens. Hard but slightly flexible consistency. Surface visually smooth but rough to the touch. Cortex patent, 3 mm in thickness (Fig. 11B, arrow).

## Spicules

Dichotriaenes (Figs. 11C–11D), scarce, only found in i827_1. Rhabdome: 333-420-498/15-22-29 ($N = 7$), the protoclad: 40-49-62/13-20-25 $\mu$m and the deuteroclad: 21-49-73/8-13-17 $\mu$m ($N = 11$).

Plagiotriaenes (Fig. 11F), only two found, one in specimen POR715 and one in i827_1. This spicule showed tuberculous processes below the cladome. Plagiotriaenes probably represent immature stages of the dichotriaenes. Rhabdomes measure 209-302/9-12 $\mu$m ($N = 2$), while clads measure 54-79/9-10 $\mu$m ($N = 2$)

Oxeas (Fig. 11E), Fusiform, large, bent at the middle, rarely modified to styles: 400-2471/6-59 $\mu$m.

Oxyasters (Fig. 11I), small centrum and long actines, spined all over its shaft: 14–38 $\mu$m with 5–11 actines.

Amphisanidasters (Figs. 11G–11H), actines radiating from both ends of the shaft, spined: 7–15 $\mu$m long. In POR715 most are underdeveloped and have extra actines on its shaft.

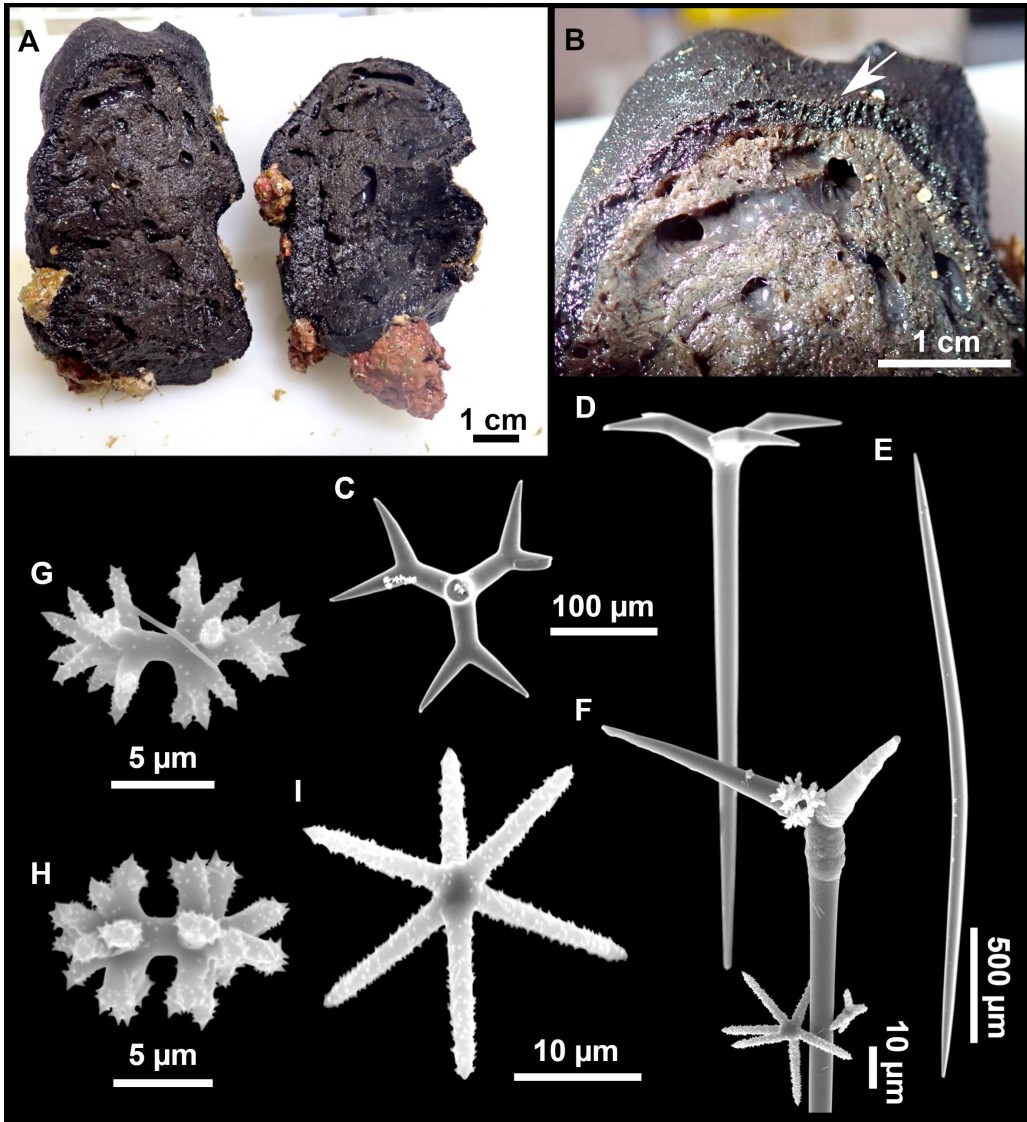

**Figure 11** *Stryphnus mucronatus* (*Schmidt, 1868*), **specimen i827_1.** (A) Habitus on deck before fixation. (B) Detail of a transversal cut, showing the cortex (arrow). (C–D) Dichotriaenes. (E) Oxea. (F) Plagiotriaene. (G–H) Amphisanidasters. (I) Oxyasters.

### Ecology notes

The species is not common in the Balearic Islands. It was found at the summit of the EB and in the fishing grounds east of Mallorca. Fishing ground stations were shallower (52 and 63 m) than the EB station (100 m). In both cases, sponges were growing on red algae bottoms.

### Genetics

The COI Folmer region from specimen i827_1 (Figs. 2D and 11A) from the EB seamount was sequenced in two parts (ON130556). The miniCOI was sequenced for POR715 and

**Table 5 Spicule measurements of *Stryphnus mucronatus* and *Stryphnus ponderosus*, given as minimum-mean-maximum for total length/minimum-mean-maximum for total width; all measurements are expressed in μm.** Specimen codes are the field#.

| Material | Depth (m) | Oxeas (length/width) | Plagiotriaenes Rhabdome (length/width) Clad (length/width) | Dichotriaenes Rhabdome (length/width) Protoclad (length/width) Deuteroclad (length/width) | Amphisanidasters (length) | Oxyasters (length) |
|---|---|---|---|---|---|---|
| *S. mucronatus* POR715 Mallorca | 52 | 400-1,247-2,036/ 6-23-42 | Rh: 302/12 Cl: 79/10 (N = 1) | – | 7-10-14 | 17-22-36 |
| *S. mucronatus* i827_1 EB | 100 | 1,137-1,889-2,356/ 15-37-59 (N = 20) | Rh: 209/9 Cl: 54/9 (N = 1) | Rh: 333-420-498/15-22-29 (N = 7) Pt: 40-49-62/13-20-25 Dt: 21-49-73/8-13-17 (N = 11) | 7-11-15 | 21-29-38 |
| *S. mucronatus* POR1196 Menorca | 63 | 962-1,857-2,471/ 9-33-49 (N = 29) | – | – | 9-11-14 (N = 23) | 14-25-36 (N = 16) |
| *S. mucronatus* PC440 Gorringes Bank | 36-42 | 807-1,224-1,571/ 7-14-18 | – | – | 9-12-16 | 19-29-44 |
| *S. ponderosus* i208 EB | 135 | 1211-1,850-2,419/ 15-31-44 (N = 11) | Rh: 650/45 Cld: 213/43 (N = 1) | Rh: 367-519-732/17-27-44 Pt: 39-71-103/14-26-42 Dt: 37-72-145/10-18-31 (N = 9) | 8-11-14 | 13-15-17 |
| *S. ponderosus* POR778_1 Benicassim Iberian Península | 76 | 1,176-1,812-2,315/ 20-41-53 | Rh: 146-337-597/9-23-31 Cld: 46-121-188/8-23-36 (N = 17) | Rh: 368-441-555/24-33-48 Pt: 63-95-119/22-30-42 Dt: 31-68-153/11-17-30 (N = 9) | 8-10-13 | 11-15-18 |
| *S. ponderosus* POR798 St Carles de la rápita Iberian Península | 95 | 1,058-1,759-2,850/ 11-31-62 | Rh: 306-514-757/16-29-51 Cld: 80-170-344/14-25-50 (N = 6) | Rh: 310-481-681/18-32-58 Pt: 62-99-158/15-29-48 Dt: 42-75-170/8-18-35 | 9-12-15 | 14-18-23 |

**Notes.**

Rh, rhabdome; Cl, clad; pc, protoclad; dc, deuteroclad; -, not found/not reported.

EB, Emile Baudot.

the second part of COI for POR1196, both from fishing grounds east of Mallorca (SBP# 2688 and ON130557). The COI of individual i827_1 and the COI fragments of POR715 and POR1196 match, and are identical to the sequence from specimen Po.25733 from the Eastern Mediterranean (Israel) (*Idan et al., 2018*).

## Taxonomic remarks

Easily recognizable species, macroscopically characterized by its massive shape, black color, and thick cortex. The spicular set is quite homogeneous between the individuals of the Balearic Islands, as well as with individuals from other localities of the Mediterranean (Table 5). Triaenes are very rare, to the point that we did not find them in one specimen. As far as we know, this rarity of triaenes has not been reported before in this species

(*Topsent, 1894*; *Topsent, 1925*; *Vacelet, 1961*; *Vacelet, 1969*; *Pansini, 1987*). The scarcity of triaenes is shared with *Stryphnus raratriaenus Cárdenas et al., 2009*, from the Caribbean. As already stated (*Cárdenas et al., 2009*), scarcity (or lack) of triaenes support the phylogenetic closeness to the genus *Asteropus Sollas, 1888*, which is essentially a *Stryphnus* without triaenes. Our findings (morphology and phylogenetic tree) suggest once again that both genera are probably synonymous, a fact that requires further revision of Atlantic species and sequencing of more species of *Asteropus* in particular.

*Stryphnus ponderosus* (*Bowerbank, 1866*)
(Figs. 6A and 12, Table 5)

### Material examined
UPSZMC 190963, field#i208_b, St. 68 (INTEMARES0718), MaC (EB), rock dredge, 135 m, coll. F. Ordines & H. Marco; UPSZMC 190964, field#POR778_1, (MEDITSGSA06N), St. 3 (2020), GOC-73, 76 m (Benicassim), coll. J. A. Díaz; UPSZMC 190966, field#POR798, (MEDITSGSA06N), St. 14 (2020), GOC-73, 96 m (St Carles de la Rápita), coll. J. A. Díaz.

### Outer morphology
Large specimens, up to 25 cm in diameter, 1.5–3 cm in width (Fig. 12A). Flattened, concave or irregular and slightly lobulated. With large holes and depressions that increase its exposed surface. Color in life of a dark tint on the upper side, with some whitish areas on its lower side. Color remains after EtOH preservation, yet the EtOH gets colored in black. Openings up to one mm in diameter, essentially located on the upper side of the body, surrounded by circular areas without pigment. Hard consistency. Hispidation localized. Poorly-delimited cortex, less than 0.5 mm thick.

### Spicules
Dichotriaenes (Fig. 12B). Rhabdome short, straight, and fusiform, 310-732/17-58 μm. Cladome with protoclads projected outwards in a 120° angle with respect to the rhabdome 39-158/14-48 μm. Deuteroclads in a 90° angle, 31-170/8-35 μm. Rarely some of the clads may not be bifurcated.

Plagiotriaenes (Fig. 12C). Rabdome straight, most with a swelling just below the cladome. Scarce to absent in some specimens. Rhabdome 146-757/9-51 μm, cladi: 46-344/8-50 μm.

Oxeas (Fig. 12D). Stout, fusiform, slightly curved, 1,058-2,850/11-62 μm.

Oxyasters (Figs. 12E–12F). with numerous and sharp actines, more or less spined, 11–23 μm.

Amphisanidasters (Figs. 12G–12H). Strongly spined, 8–15 μm long.

### Ecology notes
In the MaC, only found at the summit of the EB, an area with gross sand and both dead and live rhodoliths. Other large sponges were also collected in the same dredge, including *Jaspis* sp., *P. compressa* or *P. monilifera*. Also found at two stations of the fishing grounds in front

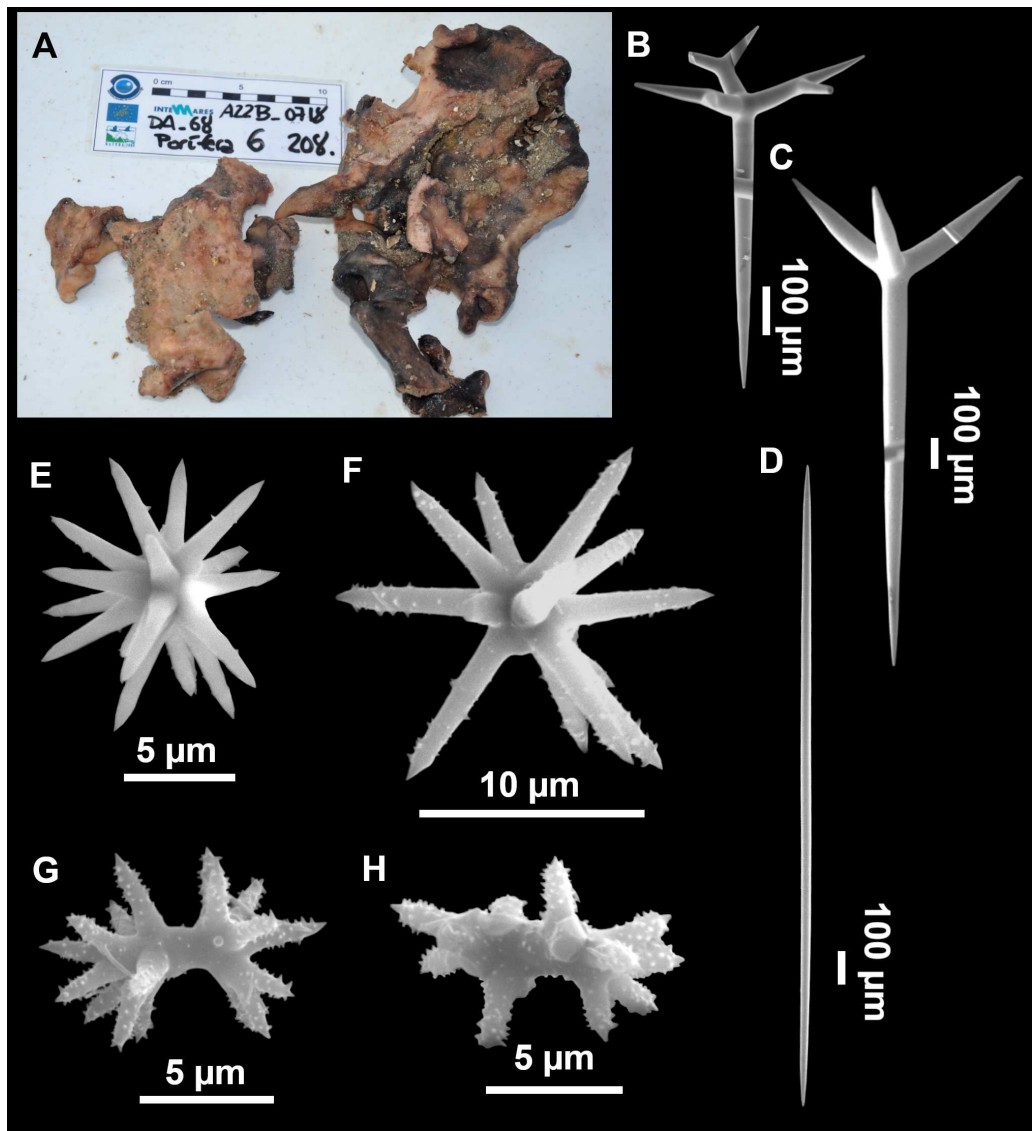

**Figure 12** *Stryphnus ponderosus* (*Bowerbank, 1866*) **specimen field i208_b.** (A) Habitus on deck before fixation. (B) Dichotriaene. (C) Plagiotriaene. (D) Oxeas. (E–F) Oxyasters. (G–H) Amphisanidasters.

of the Ebro delta, at 76 and 95 m depth. Individual i208_b from EB was free of epibionts but specimens from the fishing grounds were almost entirely overgrown by other sponges (*e.g.*, *Desmacella annexa* *Schmidt, 1870*, *Haliclona* cf. *fulva* (*Topsent, 1893*)). The specific association with *D. annexa* has been reported before in the Northeast Atlantic (*Cárdenas & Rapp, 2015*).

### Genetics

Only the miniCOI was obtained for i208_b from the EB seamount (SBP#2689).

## Remarks

This species occurs in the Northeast Atlantic and the Western Mediterranean, from the intertidal to mesophotic depths (0–200 m, this study). For most of the records, only dichotriaenes were found, and no plagiotriaenes reported. Subsequently, a variety of *S. ponderosus* with plagiotriaenes was reported under the name *S. ponderosus var. rudis*, because *Stryphnus rudis Sollas, 1888*, a junior synonym of *S. fortis*, was known to have both dichotriaenes and plagiotriaenes. Latter, *Cárdenas & Rapp (2015)* synonymized *S. ponderosus var rudis* as *S. ponderosus*, also establishing the morphological and ecological distinctions between *S. ponderosus* (shallow temperate North Atlantic and Mediterranean species) and *S. fortis* (deep-sea boreo-arctic species). The presence, absence or abundance ratio between the plagiotriaenes and dichotriaenes seems to have a specific, although highly variable, value: *S. ponderosus* has a prevalence of dichotriaenes while *S. fortis* has a prevalence of plagiotriaenes.

We have found both plagiotriaenes and dichotriaenes in all the specimens. However, plagiotriaenes were abundant in the shallowest specimen (POR778_1, $N = 17$, 76 m), as opposed to very rare in the two deepest specimens (POR798, $N = 6$, 95 m; i208_b, $N = 1$, 135 m). It may suggest that plagiotriaenes are early dichotriaenes stages, and that specimens living in deeper waters having more silica at their disposal, their spicules are more easily fully developed and mature. *Cárdenas & Rapp (2015)* suggested that population differences could explain the different ratios between plagiotriaenes and dichotriaenes found in the different specimens of *S. fortis*. Future laboratory experiments with different silica levels could help understand the relationship between plagiotriaenes and dichotrianees in these species.

We only managed to sequence a Folmer minibarcode (130 bp) from specimen i208_b. The sequence is identical to a sequence of *S. ponderosus* from Northern Ireland (HM592685), a fact that confirms the Mediterrano-Atlantic distribution of the species. It has a 2 bp difference with *S. fortis* from Norway (HM592697) and a 4 bp difference with *S. mucronatus*.

In the Mediterranean, *S. ponderosus* has been widely reported. It is known from the Gulf of Lion (*Vacelet, 1969*), the Alboran Sea (*Maldonado, 1992*), the Catalan Coast (*Uriz, 1981*), the Adriatic Sea (*Babiç, 1922*) and the Aegean Sea (*Voultsiadou, 2005*). In the Balearic Islands, it is only known by an undocumented report at the Cabrera archipelago littoral (*Uriz, Rosell & Martín, 1992*). The present record is the second in the Balearic Islands, and the first on a Mediterranean Seamount.

Family Calthropellidae Lendenfeld, 1907
Genus *Calthropella Sollas, 1888*
Subgenus *Calthropella Sollas, 1888*
*Calthropella (Calthropella) pathologica*
(*Schmidt, 1868*)
(Figs. 13 and 14, Table 6)

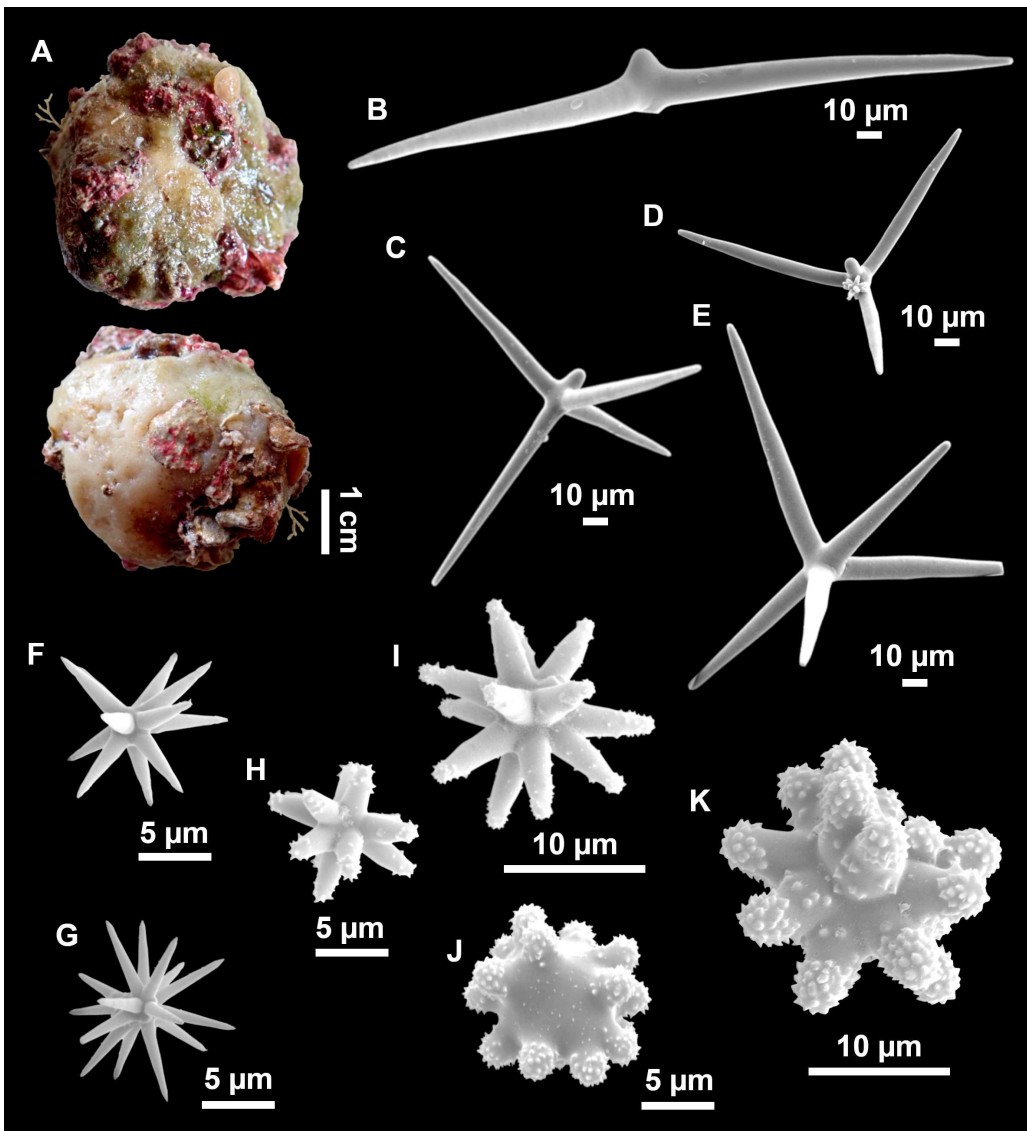

**Figure 13** *Calthropella (Calthropella) pathologica (Schmidt, 1868)*, **specimen i693.** (A) Habitus on deck before fixation. (B–E) Different calthrop modifications. (F–G) Oxyasters. (H–K) Tuberculated strongylasters.

## Material examined

UPSZMC 190806-07, field#i693-i682, St. 43 (INTEMARES0720), 116–118 m, rock dredge, MaC (EB), coll. J. A. Díaz.

## Comparative material

*Calthropella (C.) pathologica*, lectotype, MNHN DT753, Schmidt collection#66, Algeria, 'Exploration Scientifique de l'Algérie', 1842; paralectotype, MNHN DT754, Schmidt collection#87, Algeria, 'Exploration Scientifique de l'Algérie', 1842; PC324, La Ciotat, 3PP cave, 28S: AF062596; MNHN DCL4076, field#ASC6/327-6, Apulian Platform, off Cape

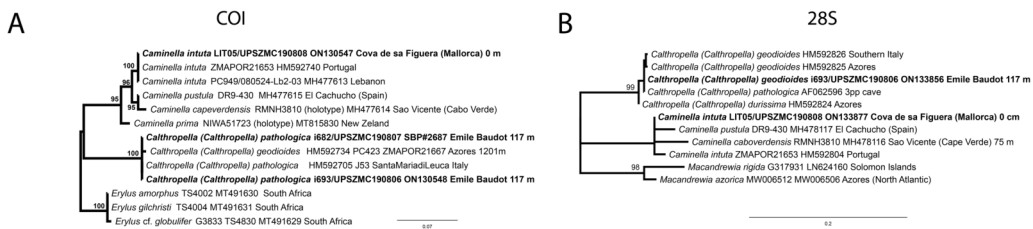

**Figure 14  Detail of the COI (A) and 28S (B) trees for Calthropellidae and *Caminella*.** Specimen codes are written as "field number/museum number" followed by Genbank accession number. The original trees can be seen as Figs. S1–S2.

**Table 6  Spicule measurements of *Calthropella (Calthropella) pathologica* and *Calthropella (Calthropella) geodioides*, given as minimum-mean-maximum for total length/minimum-mean-maximum for total width; all measurements are expressed in μm.** Specimen codes are the field#. Specimens measured in this study are in bold.

| Material | Depth (m) | Oxeas (length/width) | Calthrops I (length/width of actine) | Calthrops II (length/width of actine) | Oxyasters (length) | Spherasters or strongy-lasters (length) |
|---|---|---|---|---|---|---|
| *C. (C.) pathologica* i682 EB | 117 | Always broken (fragments up to 1,652/12) | 54-124-262/6-16-37 4+1 (aborted) and 5 actines | 318-512-879/46-66-94 2, 3, 4+1 (aborted) and 5 actines | 9-11-18 (N = 12) | 6-12-20 |
| *C. (C.) pathologica* i693 EB | 117 | Always broken (fragments up to 2,839/17) | 68-138-278/9-14-26 4+1 (aborted) and 5 actines | 335-452-608/31-46-59 (N = 14) 2, 3, 4+1 (aborted) and 5 actines | 8-12-16 (N = 13) | 8-15-23 |
| *C. (C.) pathologica* lectotype, paralectotype MNHN DT 753 MNHN DT754 Algeria (*Van Soest, Beglinger & de Voogd, 2010*) | – | Always broken (fragments up to 2,000/12) | – | 32-366/5-72 short-shafted triaenes and mesotriaene modifications Curved and stunted cladi. No dichocalthrops | smooth: 9-10-12 lightly spined: 23-25-27 | 9-18-24 |
| *C. (C.) geodioides* holotype, Near Cape St. Vincent (Portugal) (*Sollas, 1888*) | 534 | 736/93 | 3+1 (aborted) actine Dichocalthrop modification present | 785/85 3+1 (aborted) actine | – | Spheraster 25 |
| *C. (C.) geodioides* Terceira (Azores) (*Topsent, 1904*) | 599 and 845 | – | 3+1 (aborted) and 4 actines Dichocalthrop modification present | 2, 3 and 4 actines Dichocalthrop modification present | 12-15 | 20 |
| *C. (C.) geodioides* ZMAPOR 21667 Terceira (Azores) (*Van Soest, Beglinger & de Voogd, 2010*) | 1,201 | Invariably broken, at least 500/5. Absent in some specimens | Dichocalthrops (few, absent in some specimens) Pc: 75-92/12 Dc: 28-31 Rh: 92-120 | 102-351-705/11-52-128 possibly divisible in two categories: 102-180 and 434-705 3 actines | 13-18 | 7-28 |

**Notes.**

Rh, rhabdome; Cl, clad; pc, protoclad; dc, deuteroclad; -, not found/not reported; EB, Emile Baudot.

Santa Maria di Leuca, Southern Italy, 39.56, 18.43, 560–580 m, ROV dive 327-6, Ifremer MEDECO leg1 (ifremer), 17 Oct. 2007, coll. J. Reveillaud, erroneously assigned to *C. (C.) geodioides* (*Carter, 1876*) in *Cárdenas et al. (2011)*, COI: HM592705, 28S: HM592826.

*Calthropella (C.) geodioides*, holotype, NHM 82.7.28.16, Cape St. Vincent, Portugal, 534 m; ZMAPOR 21667, EMEPC/G3-D4-Ma11a, SE of Terceira Island, Azores, 38.4265°N, 26.8206°W, 1201 m, 18 May 2007, COI: HM592734, 28S: HM592825 (*Cárdenas et al., 2011*), SEM images presented in *Van Soest, Beglinger & de Voogd* (*2010*, Fig. 24).

### Outer morphology

Massively irregular. Specimen i682 measures $7 \times 9 \times 4$ cm and specimen i693 measures $5 \times 4 \times 3.5$ cm, both overgrowing coralligenous red algae. Color in life beige with areas of dirty green, more localized in one of the body sides. On a transversal section, the green color persists in the first mm and then fades away turning into a beige choanosome. This first mm corresponds to a well-delimited cortex, which can be distinguished with the naked eye. Stony hard consistency, surface optically smooth, slightly rugose to the touch, overgrown by encrusting sponges (*e.g.*, *Jaspis* sp.); pores inconspicuous.

### Spicules

Calthrops (Figs. 13B–13E), with 2–5 cladi and large variability, including aborted actines, teratogenic modifications and stylote ends. On a wide but continuous size range, overall measuring 54-879/6-94 µm. Potentially divisible in two categories (54-278/6-37 µm and 318-879/31-94 µm) but with intermediate sizes. Mesocalthrop modifications present in both large and small sizes but more common in small ones.

Oxeas (not shown), thin, invariably broken, scarce, up to 2,839/17 µm.

Oxyasters (Figs. 13F–13G), scarce, with many rays and sometimes a few small spines (only detectable by SEM), 9–18 µm.

Strongylasters to spherasters (Figs. 13H–13K) with more or less spiny actines, variations with large centrum/short actines to small centrum/longer actines. Sizes of both kinds overlap so they probably belong to the same category, 6–23 µm.

### Ecology notes

Both specimens were found at the same station; rhodolith bed at the summit of the EB, and both included rhodoliths to their bodies, which probably served as substrate. The station was rich in massive sponges like *P. monilifera*, *Jaspis* sp. or *Spongosorites* spp.

### Genetics

We obtained the Folmer COI of i693 (ON130548) but only the miniCOI from i682 (SBP# 2687). 28S (C1-C2) was obtained from i693 (ON133856).

### Remarks

We assign with hesitation the Balearic material to *C. (C.) pathologica*. As stated by *Van Soest, Beglinger & de Voogd (2010)*, *C. (C.) pathologica* and *C. (C.) geodioides* are morphologically similar, and may be differentiated by (i) the shape of the asters (longer actines in *C. (C.) pathologica*), (ii) occasional calthrops with dichotomous clads in *C. (C.) geodioides* (absent

in *C. (C.) pathologica*) and (iii) mesocalthrops and dimesocalthrops in *C. (C.) pathologica* (absent in *C. (C.) geodioides*).

New spicules preparations were made from the holotype of *C. (C.) geodioides* and thick sections were made of the lecto- and paralectotype of *C. (C.) pathologica*; SEM images of the spicules of the lectotype of *C. (C.) pathologica* were previously presented by *Van Soest, Beglinger & de Voogd* (*2010*, Fig. 26). Based on re-examination of the types, MNHN DCL4076 from Santa Maria di Leuca was re-identified as *C. (C.) pathologica* and not *C. (C.) geodioides* as originally identified by *Cárdenas et al. (2011)*. *C. (C.) pathologica* possesses long thin oxeas that form occasional large bundles, visible on the thick sections of the lectotype. We also found fragments of long oxeas in *C. (C.) geodioides.*

We confirm that asters usually have longer actines in *C. (C.) pathologica* than in *C. (C.) geodioides*. Dichocalthrops have not been found in any of the *C. (C.) pathologica* examined, including the holotype, which is in accordance with the literature. On the other hand, dichocalthrops of all sizes are more or less abundant in *C. (C.) geodioides*, including the type, where they are particularly numerous. Given the extreme variability in calthrop morphology (including cladi size, number and aberrant forms), this character may depend on ecological factors and is potentially misleading so it needs to be further tested in the future. After examining our comparative material, the presence of one or two (mesocalthrop) actines appears to be a solid character: in *C. (C.) pathologica*, small calthrops show 'meso' modifications, both in the form of 4 (fully developed) + 1 underdeveloped actine (Fig. 13C) and in the form of five (fully developed) actines (Fig. 13E) while in *C. (C.) geodioides* they always have 4 (fully developed) + 1 (underdeveloped) actine. Regarding the large calthrops, in *C. (C.) pathologica* those can have 4+1 and five actines while in *C. (C.) geodioides* they always show three, four or 4+1 actine. Analysis of COI/28S sequences revealed 2 bp. differences between the Azores specimen ZMAPOR 21667 (HM592734), and Mediterranean ones (including those from the EB seamount (i682 and i693), and specimen MNHN DCL4076 (HM592705) from Santa Maria di Leuca, Italy). To conclude, both species appear valid for now, based on three morphological differences and COI/28S.

Note: we noticed a typo in the *C. (C.) geodioides* material described by *Van Soest, Beglinger & de Voogd* (*2010*, p. 59 and Fig. 22): it should not be 'ZMAPOR 21666' (EMEPC/G3-D03A-Ma012) which is a *C. (C.) durissima*, IDed and sequenced by *Cárdenas et al. (2011)*. *Van Soest, Beglinger & de Voogd (2010)* meant 'ZMAPOR 21667' (EMEPC/G3-D4-Ma11a). However, the collecting information is correct.

Family Geodiidae Gray, 1867
Subfamily Erylinae *Sollas, 1888*
Genus *Caminella* Lendenfeld, 1894
*Caminella intuta* (*Topsent, 1892*)
(Figs. 14 and 15)

## Material examined

UPSZMC 190808, field#LIT05, "Cova de sa Figuera" (cave), east of Mallorca, free apnea, 0–0.5 m, coll. J. A. Díaz.

## Outer morphology

Hemispherical, 1.5 cm in maximum diameter (Fig. 15A). Ectosome light brown on live specimen, and after fixation in EtOH. Choanosome color not recorded on live specimen, whitish after fixation. Cortex has a stony hard consistency and is slightly rough to the touch, choanosome is hard but compressible. The cortex can be separated from the rest of the body, ∼0.5 mm thick. Surface covered with circular pores, which have a ring-shape in life (open), and a dot-shape after fixation (contracted). Three circular oscula, 1–2 mm in diameter in live specimen, slightly smaller due to contraction after fixation.

## Spicules

Dichotriaenes, robust, with short conical rhabdome, 398–399 ($N = 2$)/65-72 ($N = 3$), long protoclads and short deuteroclads, 60-95/67-150 ($N = 5$) and 248-360/49-68 ($N = 5$), respectively.

   Oxeas, fusiform, scarce, 1313-1860/15-34 ($N = 3$).

   Sterrasters (Figs. 15B–15D), spherical to oval; no clear rosettes but intricate brain-like surface covered with small warts (Fig. 15BC)

   Spiny spherasters to spherules (Figs. 15E–15G), 5-9-12 μm in diameter.

   Oxyasters (Figs. 15H–15I), 4–15 actines, 13-19-27 μm in diameter. Actines are acanthose with robust and triangular spines that are also microspined. Large oxyasters tend to have less actines than smaller ones.

## Ecology and distribution

A single specimen found in the dark area of a littoral cave, firmly attached to a vertical wall, no more than 30 cm deep. Our specimen was living in an area that emerges with high waves, a fact suggesting that the species can survive short periods of air exposure. No epibionts.

## Genetics

Folmer COI (ON130547) and 28S (C1-C2) (ON133877) were obtained. COI was identical to previously sequenced *C. intuta* from Lebanon, and Portugal. The 28S was also identical to specimen ZMAPOR 21653 (Portugal), but differs in 1 bp with specimen SME PL617PC-7 from Cosquer Cave, Marseille, France.

## Taxonomic remarks

The external morphology, spicular set/sizes matches well with a revision of the species, including a redescription of type material (*Cárdenas et al., 2018*). Genetically, the only discrepancy with the published sequences is the 1 bp difference in the 28S sequence when compared to a specimen from Marseille (PL617PC-7). Unfortunately, no COI is available for that same specimen. This is the first record of the species in the Balearic Islands, and the shallowest for the species.

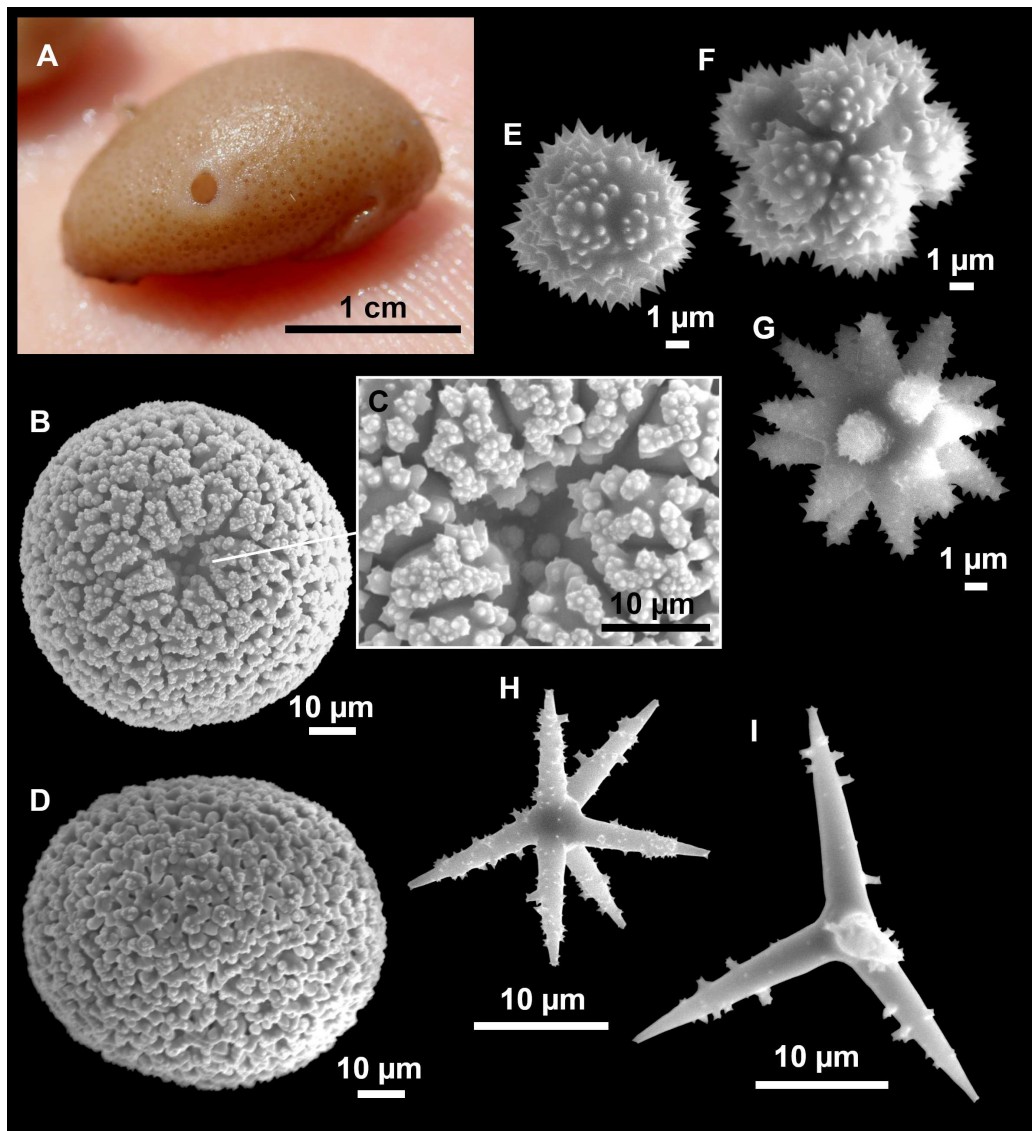

**Figure 15** *Caminella intuta* (*Topsent, 1892*), **specimen LIT05.** (A) Habitus in fresh state, just after collection. (B–D) Sterrasters with (C) surface details. (E–G) Spherules to spherasters. (H–I) Oxyasters.

Genus *Caminus* *Schmidt, 1862*
*Caminus vulcani* *Schmidt, 1862*
(Figs. 10 and 16, Table 7)

## Material examined

UPSZMC 190809, field#i142_C, St. 51 (INTEMARES0718), MaC (EB), beam trawl, 135 m, coll. J. A. Díaz; UPSZMC 190810, field#i254_4, St. 50 (INTEMARES1019), MaC (AM), beam trawl, 102 m, coll. J. A. Díaz; UPSZMC 190811, field#i391_2, St.

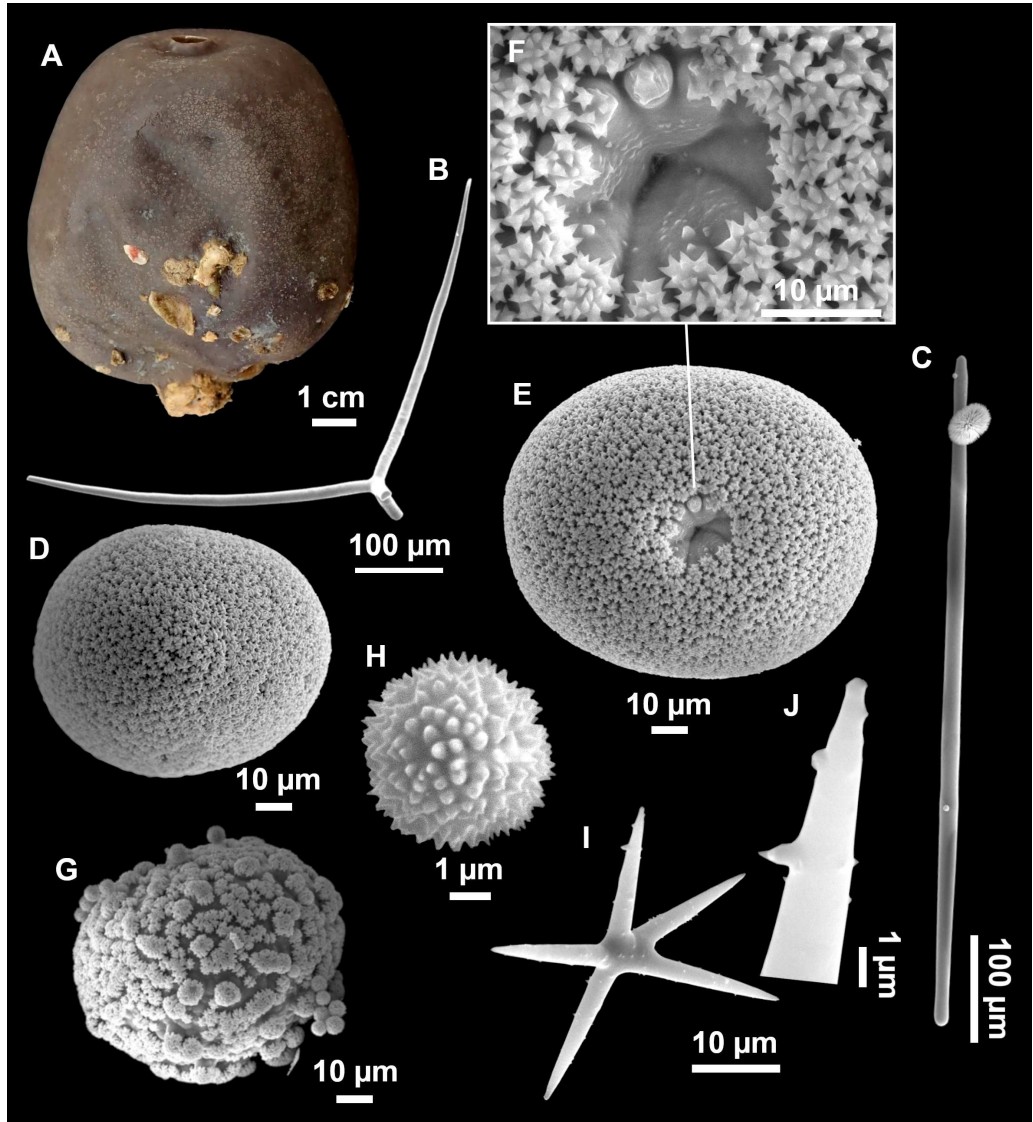

**Figure 16** *Caminus vulcani Schmidt, 1862*, **specimen i254_4.** (A) Habitus after ethanol fixation. (B) Orthotriaene. (C) Strongyle. (D–G) Sterrasters with (F) detail of the rosettes. (H) Spherules. (I) Oxyaster with (J) detail of the spines.

158 (INTEMARES1019), MaC (EB), beam trawl, 146 m, coll. J. A. Díaz; UPSZMC 190812, field#i526, St. 18 (INTEMARES0720), MaC (AM), beam trawl, 114 m, coll. J. A. Díaz.

## Comparative material

*Caminus vulcani*, MNHN DT2288, slide, Banyuls, France, 30–40 m, specimen studied by *Topsent (1894)*; SME, wet specimen, Cassidaigne canyon, off Marseille, France, 100–150 m, trawl, 16 June 1961, specimen studied by *Vacelet (1969)*.

  *Caminus xavierae* **sp. nov.**, ZMAPOR 20422, holotype, Cueva Agua Dulce, Tenerife, Canary Islands, 5–10 m, scuba diving, field#CAN.07.05, 15 Jan 2007, coll. J. R. Xavier.

Díaz et al. (2024), *PeerJ*, DOI 10.7717/peerj.16584

**Table 7** Spicule measurements of *Caminus vulcani* (*Schmidt, 1862*) and *Caminus xavierae* sp. nov., given as minimum-mean-maximum for total length/minimum-mean-maximum for total width; all measurements are expressed in μm. Specimen codes are field#. Specimens measured in this study are in bold.

| Material | Depth (m) | Cortex thickness (mm) | Oxeas (length/width) | Strongyles (length/width) | Orthotriaenes Rhabdome (length/width) Clad (length/width) | Sterrasters (diameter) | Oxyasters (length) | Spherules (length) |
|---|---|---|---|---|---|---|---|---|
| ***C. vulcani* i526 AM** | 114 | – | 774/ 18 (*N* = 1) | 521-700-856/ 8-15-20 | Rh: 606 (*N* = 1)/15-20 (*N* = 2) Cl: 355-378/11-16 (*N* = 2) | 79-96-113 | 35-59-99 (2-6 actines) | 3-5-7 |
| ***C. vulcani* i391_2 EB** | 146 | – | 672-787/ 8-18 (*N* = 4) | 510-697-836/ 14-19-22 (*N* = 17) | Rh: -/19-28 (*N* = 5) Cl: 465-537/16-25 (*N* = 5) | 83-104-133 | 34-47-95 2-7 actines | 3-4-6 |
| ***C. vulcani* i254_4 AM** | 102 | 1.5-2 | 424-562/ 3-9 (*N* = 3) | 511-745-962/ 10-15-20 (*N* = 22) | Rh: 668 (*N* = 1)/15-24 (*N* = 4) Cl: 253-495/12-21 (*N* = 4) | 72-93-112 | 36-50-65 4-8 actines | 3-4-5 |
| ***C. vulcani* Cassidaigne, France** (*Vacelet, 1969*) | 100-150 | – | 378-637-861/ 5-7-10 (*N* = 14) | 449-658-806/ 10-16-21 | Rh: 376-597 (*N* = 3)/15-22 (*N* = 4) Cl: 266-487/11-24 (*N* = 4) | 71-92-106 | 36-50-72 2-6 actines (2 actines = large) | 3-4-5 |
| *C. vulcani* South Gulf of Lion, Banyuls (*Topsent, 1894*) | 30-40 | – | – | 850/ 15-17 | Rh: 480-570/15-17 Cl: 350-380 | 105-115/ 85-88 | 40 (mean) 2-5 actines | 4 |
| *C. vulcani* several specimens including neotype (*Uriz, 2002*) | – | 1.5-2 | – | 850-880/ 15-17 | Rh: 480-572/15-17 Cl: 320-360/- (in chord length, with long and straight clads) | 100-115/ 87-90 | 35-42 | 3-4 |

Peer

**Table 7** (*continued*)

| Material | Depth (m) | Cortex thickness (mm) | Oxeas (length/width) | Strongyles (length/width) | Orthotriaenes Rhabdome (length/width) Clad (length/width) | Sterrasters (diameter) | Oxyasters (length) | Spherules (length) |
|---|---|---|---|---|---|---|---|---|
| ***C. xavierae* sp. nov. Holotype (ZMAPOR 20422) Canary Islands** | 5-10 | 0.5 | broken | 232-405-520/ 4-12-17 | Rh: 326-458/ 5-14-20 (*N* = 9) Cl: 77-225-326/- | 70-81-92 | 13-25-37 (2– 5 actines) | 3-5 |
| ***C. xavierae* sp. nov. paratype UPSZMC 190814 Canary Islands** | 5-10 | 0.5 | 379-450-531/ 7-9-11 | 476-541-639/ 11-15-18 | Rh: broken Cl: 228-241/14-23 (*N* = 2) | 50-73-86 | 25 (*N* = 1) | 2-4-5 |
| *C.* cf. *xavierae* sp. nov. Canary Islands (*Cruz, 2002*) | 5-10 | – | – | 250-440/- | – | 80-120 | 16-24 (3–5 actines) | 3-5 |
| *Caminus carmabi* Bonaire, Caribbean (*Van Soest, Beglinger & de Voogd, 2010*) | 120-137 and 198 | – | – | 600-860-936/ 14-21-25 | Rh: - Cl: 250-650-1,020/18-24-30 | 140-190- 210/125-144- 162 | 51-65-81 (4– 8 actines) | 4-5-7 |

**Notes.**

Rh, rhabdome; Cl, clad; pc, protoclad; dc, deuteroclad; -, not found/not reported; EB, Emile Baudot; AM, Ausias March.

### Outer morphology

Subspherical sponges (Fig. 16A) 3–8 cm in diameter, with an apical rounded oscula (2–4 mm in diameter), with a raised rim. Larger specimens (i254_4, i526) tend to acquire an ellipsoid, constricted shape. Same color in life and after preservation in EtOH: dark grayish ectosome and brownish choanosome. Hard (1 mm thick) but breakable cortex, pulpy choanosome. Smooth surface, with a mosaic visible to the naked eye, consisting of whitish polygonal patterns, which reflect the distribution of the pores; these patterns are even more obvious on dried cortex.

### Spicules

Orthotriaenes (Fig. 16B), few, with clads curved forward, sometimes sinuous, short straight rhabdomes, only slightly longer than the cladi. Malformations like stylote termination or aberrant actines present in both rhabdome and cladome. Rhabdome: 606-668/15-28 $\mu$m, clads: 253-537/11-25 $\mu$m.

Strongyles (Fig. 16C), straight or curved, with round tips, 510-962/10-22 $\mu$m. Immature stages with oxeota ends are also present, 424-787/8-22 $\mu$m.

Sterrasters (Figs. 16D–16G), spherical, with a very pronounced hilum (Figs. 16E and 16F), 72–133 $\mu$m in diameter.

Spherules (Fig. 16H), rugose, 3–6 $\mu$m in diameter.

Oxyasters (Figs. 16I–16J), with small spines, 34–99 $\mu$m in diameter and having 2–8 actines, oxyasters with 2 actines tend to be larger.

### Ecology notes

Only found at the summit of the EB and the AM, inhabiting the mesophotic zone (depth range 102–146 m).

### Genetics

COI (ON130546) and 28S (C1-C2) (ON133892) were obtained from specimen i526. This is the first 28S fragment published for the genus *Caminus*. COI sequence was 8 bp different with *C. xavierae* **sp. nov.** from Tenerife (EU442205), and only 1 bp different with *Caminus carmabi Van Soest, Meesters & Becking, 2014* from the Caribbean (MT815828).

### Taxonomic remarks

See remarks below for *C. xavierae* **sp. nov.**

*Caminus xavierae* **sp. nov.** Díaz & Cárdenas
(Figs. 10 and 17; Table 7)

As *Caminus vulcani*: (*Cárdenas et al., 2011*; *Cárdenas, 2020*) (Figs. 5F and 5G).

### Etymology

Named after sponge biologist Joana R. Xavier for collecting this species in the Canary Islands, and for her continuous efforts and leadership to support deep-sea sponge research.

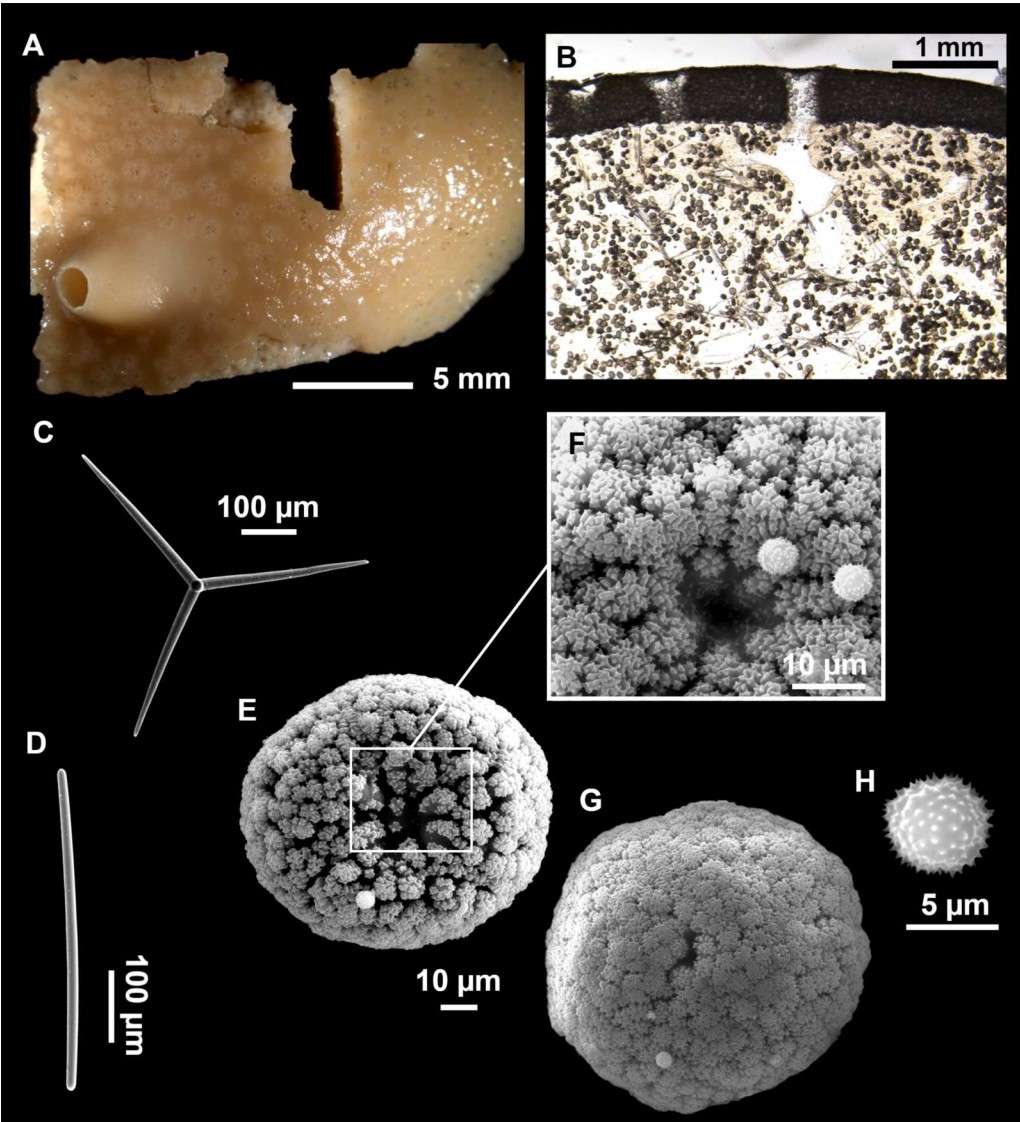

**Figure 17** **Holotype of *Caminus xavierae* sp. nov. (ZMAPOR 20422), Tenerife, Canary Islands.** (A) Habitus after ethanol fixation. (B) Optical microscope image of a thick transversal section; three cribriporal pores visible across the cortex. (C) Orthotriaene. (D) Strongyle. (E–G) Sterrasters with (F) detail of the rosettes with spherules; picture E was already used in *Cárdenas* (*2020*, Fig. 5F). (H) Spherules. Oxyasters not shown. Figure source credit: Paco Cárdenas.

## Material examined

Holotype: ZMAPOR 20422 (wet specimen), UPSZTY 190813 (thick section and spicule slide), field#CAN.07.05, Cueva Agua Dulce, Tenerife, Canary Islands, 5–10 m, scuba diving, 15 Jan. 2007, coll. J. R. Xavier.

Paratype: UPSZTY 190814, spicule slide preparation, same locality as holotype, field# CAN.07.06, 15 Jan. 2007, coll. J. R. Xavier.

## Comparative material

*Caminus vulcani* (this study).

*Caminus carmabi*, HBOI 11-V-00-1-007, specimen code 200005111007, Kaap Sint Marie, South coast, Curacao, 12.180550, -69.083980, 282 m, Johnson Link II-3209, 11 May 2000, id: P. Cárdenas, COI: MT815828.

## Outer morphology

Holotype is 2.5 × 1 cm, massive encrusting with a unique oscule opening with raised margins (2–5 mm high) (Fig. 17A). Pores are distributed on the body of the sponge with the typical *Caminus* star-shaped pattern. Color in EtOH is light yellow to light brown; the raised oscule is lighter. Choanosome is beige.

## Skeleton

Typical *Caminus* skeleton (Fig. 17B) with a distinct cortex (0.5 mm thick) essentially made of sterrasters, and covered with a thin layer of spherules. Below, a subcortical fibrous layer and a few triaenes supporting the cortex with their clades. In the choanosome, bundles of strongyles with no particular orientation, and numerous sterrasters, spherules and oxyasters.

## Spicules (holotype and paratype)

Orthotriaenes (Fig. 17C), not abundant. sometimes ectopic clades on the rhabdome, rhabdome: 326-458/5-20 μm, clads: 77-326/14-23 μm.

Strongyles (Fig. 17D), straight or curved, 232-639/4-18 μm.

Sterrasters (Figs. 17E–17G), spherical to elongated, common unequal length of the actines giving a cauliflower-like aspect, 50–92 μm in diameter.

Spherules (Fig. 17H), 2–5 μm in diameter.

Oxyasters, sometimes looking like strongylasters (with thicker actines), 2–5 actines, many irregular, 13–37 μm in diameter.

## Ecology notes

Until now only found in underwater caves (Agua Dulce and San Juan) in Tenerife, Canary Islands.

## Genetics

COI from the holotype (EU442205) and the paratype (COI unpublished) are identical and differ in 8 bp with the COI of *C. vulcani* and 7 bp with the COI of *C. carmabi.*

## Taxonomic remarks on *Caminus vulcani* and *Caminus xavierae* sp. nov.

*Caminus vulcani* is a well-known species, easily recognizable due to its macroscopical habit and its spicular set. In the Mediterranean, the species is recorded from shallow waters including caves (*Topsent, 1894*, 30–40 m; *Pulitzer-Finali, 1983*, 15 m; *Grenier et al., 2018*) to mesophotic depths (*Vosmaer, 1894*, 150–200 m; *Vacelet, 1969*, 100–150 m; *Maldonado, 1992*, 70–120 m). In general terms our material matches with the revision by *Uriz (2002)* only differing in that MaC specimens have strongyles in a wider size range (510-962/10-22

μm *vs* 850 *vs* 880/15-17 μm) and larger oxyasters (36–95 *vs* 35–42 μm). Besides, type material of *C. vulcani* comes from the Adriatic Sea (*Schmidt, 1862*), so we consider that our material is conspecific with the type because of geographical proximity and morphological similarities. It is reported here for the first-time in the Balearic Islands.

The only published Atlantic records of *C. vulcani* came from shallow caves in Tenerife, Canary Islands (*Cruz, 2002*; *Cárdenas et al., 2010*). Interestingly, COI from those Canary Island specimens differs by 8 bp with COI of our specimens from the MaC seamounts, which clearly indicates that they are two separate species. A new species is proposed for Canary Island specimens, *Caminus xavierae* sp. nov., characterized by three main spicule differences with *C. vulcani*: (i) shorter strongyles (232-639/4-18 μm *vs.* 510-962/8-22 μm in *C. vulcani*), (ii) smaller sterrasters on average (average sizes of 73–80 μm *vs.* average sizes of 92–104 μm) and (iii) shorter oxyasters (13–37 μm *vs.* 36–99 μm) (Table 7). The "cauliflower" morphology of the sterrasters (*vs.* regular subspherical shape in the other *Caminus* species) may be another specific character but SEM examination of more specimens is required to confirm this.

*Caminus xavierae* is the fourth species of *Caminus* in the Atlantic after the Caribbean *C. carmabi*, the Caribbean/Brazilian *Caminus sphaeroconia Sollas, 1886* and the Mediterranean *C. vulcani*. *C. sphaeroconia* differs from all with the absence of oxyasters while the sterrasters (average of 190/144 μm) and oxyasters (average of 65 μm) of *C. carmabi* are much larger than in *C. xavierae*.

Interestingly, the COI sequence of *C. vulcani* (i526) and *C. carmabi* showed they are genetically closer to each other (1 bp. difference) than to *C. xavierae* sp. nov. (7–8 bp). This confirms the morphological similarities observed between them, highlighted in the original description of *C. carmabi*: "*Our material is most similar to Mediterranean Caminus vulcani Schmidt, 1862, but in that species sterrasters are smaller (105-115/85-88) and calthrops have also shorter and thinner cladi*" (*Van Soest, Meesters & Becking, 2014*)". *C. vulcani* and *C. carmabi* are therefore mesophotic sister species. Such small genetic differences in COI on either side of the Atlantic is in accordance with what is observed in other tetractinellid sister-species (Cárdenas P., unpublished data). In the 28S tree, *C. vulcani* groups with *Geodia* and not with the Erylinae (Fig. 10). This unexpected result implies a long branch and no bootstrap support so it may be explained by the short length of our fragment and low polymorphism of the sequence obtained.

Genus *Erylus* Gray, 1867
*Erylus* cf. *deficiens Topsent, 1927*
(Figs. 18–19, Table 8)

## Material examined
UPSZMC 190838, field #LIT10, Calo d'en Rafalino (Cala Morlanda), Mallorca, semi-submerged cave, 0–0.5 m, free apnea, coll. J. A. Díaz.

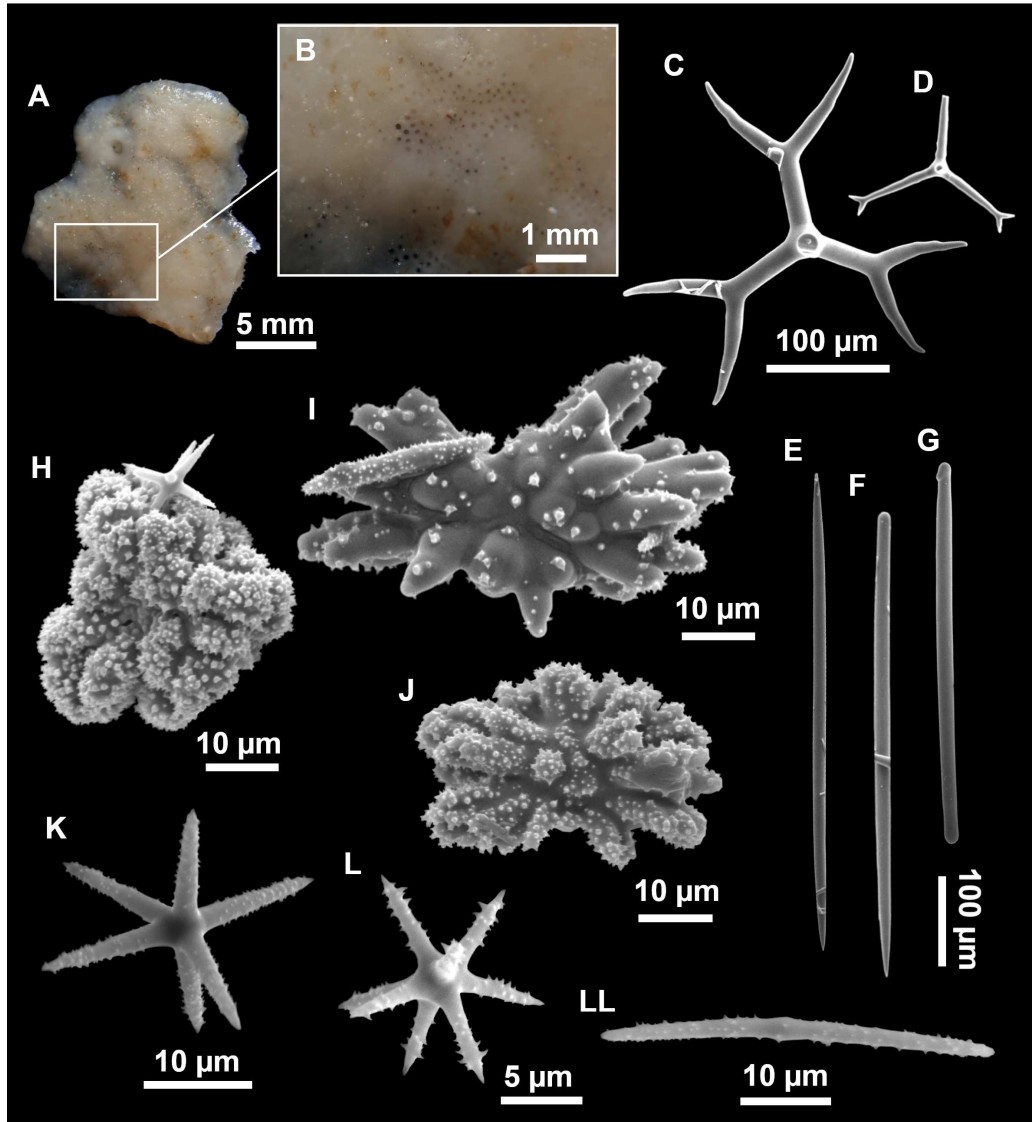

**Figure 18** *Erylus* **cf.** *deficiens* (*Topsent, 1927*)**, specimen LIT10.** (A) Habitus after ethanol fixation with (B) details of the pores. (C) Dichotriaene. (D) Juvenile dichotriaene. (E–G) Oxea, style and strongyle. (H–J) Aspidasters. (K–L) Oxyasters. (LL) Microrhabd.

## Comparative material

*Erylus deficiens*, holotype, MNHN DT1111 (slide), Porto Santo Bay, Madeira, 33°2′N, 16°19′45″W, 100 m, St. 801, 2 July 1897, trawl; ZMAPOR 21693, Gettysburg Peak, Gorringe Seamount, 32 m, 3 June 2006, coll: J. Xavier, field# GOR 06.01, COI: HM592687, 28S: HM592823, specimen identified as *Erylus* sp. in *Xavier & Van Soest (2007)* and *Cárdenas et al. (2011)*; ZMAPOR 20419, Reserva do Garajau, Madeira, 7 m, 17 Feb. 2005, coll: J. Xavier, field#MAD.05.02.31, COI: EU442204, 28S: EU552088.

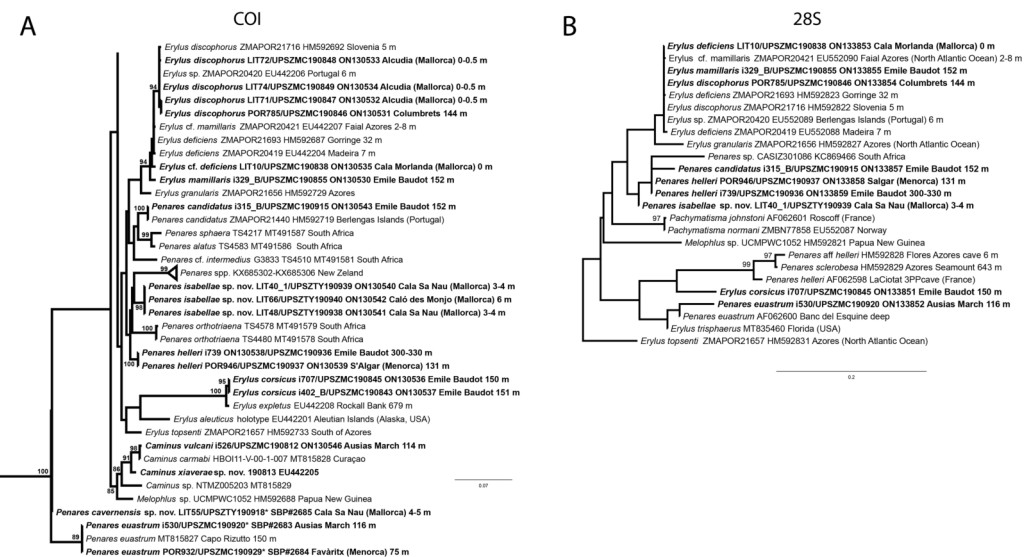

**Figure 19** Detail of the COI (A) and 28S (B) trees of Erylinae (*Erylus, Penares, Pachymatisma, Caminus, Melophlus*). Specimen codes are written as "field number/museum number" followed by Genbank accession number. The original trees can be seen in Figs. S1–S2.

## Outer morphology

Small crust (Fig. 18A), 4–5 cm in maximum diameter, 0.2–0.4 cm in width. Whitish in life, dark brown after EtOH fixation. Skin smooth and hard to the touch, choanosome crumbly. Minute pores, <0.1 mm in diameter (visible to the naked eye), gathered (Fig. 18B). Oscules not observed.

## Spicules

Dichotriaenes (Figs. 18C–18D), uncommon malformations like aborted actines and stylote terminations. Juvenile stages with protoclads much longer than deuteroclads, which are very small, sometimes barely visible. Rhabdome:145–182 ($N = 4$)/8-17-26, the protoclad: 47-67-86/8-14-25 μm and the deuteroclad: 11-79-135/4-11-18 μm.

Orthotriaenes, rare, rhabdome 76 ($N = 1$)/8-11 μm ($N = 4$), cladi 61-133/6-10 μm ($N = 4$).

Oxeas (Fig. 18E), robust, straight to slightly curved, and fusiform, 280-503-647/4-11-17 μm.

Styles to strongyles (Figs. 18F–18G) look like oxea modifications, tend to be shorter and thicker, especially the strongyles. Styles measure 193-520/14-15 μm ($N = 2$), while strongyles measure 248-412-504/11-17-27 μm ($N = 14$).

Aspidasters (Figs. 18H–18J), very scarce, always with an irregular shape due to unequal actine lengths, 31-42-56 μm ($N = 13$).

Oxyasters (Figs. 18K–18L), with 4–9 spined actines, 12-21-36 μm.

Microrhabds (Fig. 18LL), very spiny, with less spines at the center, some being centrotylote, 22-36-55/2-3-6 μm.

**Table 8  Spicule measurements of *Erylus discophorus* (*Schmidt, 1862*), *E. mamillaris* (*Schmidt, 1862*) and *E. deficiens*, given as minimum-mean-maximum for total length/minimum-mean-maximum for total width; all measurements are expressed in μm.** Specimen codes are field#. Specimens measured in this study are in bold.

| Material | Depth (m) | Macroscopical morphology | Oxeas (length/width) | Orthotriaenes Rhabdome (length/width) Clad (length/width) | Dichotriaenes Rhabdome (length/width) Protoclad (length/width) Deuteroclad (length/width) | Aspidasters (diameter) | Spined microrhabds (I) (length/width) | Smooth microrhabds (II) (length/width) | Oxyasters (length) |
|---|---|---|---|---|---|---|---|---|---|
| *E. discophorus* holotype Lesina (Adriatic) (*Sollas, 1888*; *Topsent, 1928*[*]) | – | Irregular, flattened, tuberose mass. Cortex 0.2-0,25 mm thick. Black internally because of pigment-cells in the cortex | 1,060-1,240/35 | – | Rh: 556/52 Pc: 127-143/- Dc: 175-368/- | 84-106/77 15 thick (thin, circular or elliptical) 65 (all circular)[*] | 28/3.5 13-45/2-5[*] | – | 46 3-12 actines |
| *E. discophorus* Adriatic (*Pulitzer-Finalli, 1983*) | 0-30 | Small, encrusting or insinuating. Color in life white, cream, brownish. | 500-1,600/- | – | Rh: 250-750/- Pc: - Dc: - | 40-110 | 12-60 | – | 12-40 |
| **E. discophorus POR785 Columbrets** | 144 | Massive, large, fleshy, rounded body with bulbous processes. Beige color. Dark ring surrounding the oscula | 600-777-978/6-21-29 Styles: 650/25 ($N = 1$) | Rh: 135-464/9-18 Cl: 96-164/7-17 ($N = 3$) | Rb: 217-390-557 ($N = 5$)/13-34-62 Pc: 57-80-95/12-32-52 Dc: 49-162-262/8-27-44 | 64-76-95/45-60-71 (most circular, some subcircular) | 14-22-50/1-3-5 (abundant) | 46-54-68/4-5-6 ($N = 10$) | 10-23-29 (4-11 actines) |
| **E. discophorus LIT71 Coves de na Dana (Cave)** | 0-1 | Encrusting, whitish | 458-624-734/ 5-11-16 Styles: 384-541/10-12 ($N = 3$) Strongyles: 402-466/11-13 ($N = 3$) | – | Rb: -/8-20 Pc: 46-82/7-15 Dc: 10-109/3-12 ($N = 5$) | 40-47-54 ($N = 14$) (most circular, some subcircular) | 19-33-55/1-2-3 (abundant) | 61-70-89/3-3-3 ($N = 6$) | 7-15-30 |
| **E. discophorus LIT72 Coves de na Dana (Cave)** | 0-1 | Encrusting, whitish | 388-480-652/5-9-11 ($N = 15$) Styles: 439/12 ($N = 1$) Strongyles: 183-351/16-17 ($N = 2$) | Rb: -/10 Cld: 105/8 ($N = 1$) | Rb: 123-273 ($N = 3$)/8-21 ($N = 8$) Pc: 48-70-111/8-12-17 ($N = 11$) Dc: 19-57-91/6-10-14 ($N = 11$) | 33-48-71 | 16-26-46/1-2-3 (abundant) | 45-58-76/2-3-5 ($N = 21$) | 6-14-21 |
| **E. discophorus LIT74 Coves de na Dana (Cave)** | 0-1 | Encrusting, whitish | 580-764-956/ 8-14-22 Styles: 647/25 ($N = 1$) Strongyles: 385-796/11-22 ($N = 4$) | – | Rb: 221-360-508/17-21-26 ($N = 5$) Pc: 52-80-100/9-18-31 ($N = 15$) Dc: 46-95-172/8-14-23 ($N = 15$) | 44-73-92 | 17-38-68/2-3-5 (abundant) | 41-63-86/4-4-5 ($N = 9$) | 8-19-30 (5–12 actines) |
| *E. discophorus* Bay of Naples (cave) (*Pulitzer-Finali, 1972*) | 2 | Very small, cushion-shaped, white with brown shades | 480-915/10-17 Styles: 400-710/12-18.5 Strongyles: 355-515/14-20 | – | Rh: 150 pm, Cladome: 400 | 37-57/37-49 | 24-62 | – | 8-24 (5–12 actines) |

Díaz et al. (2024), *PeerJ*, DOI 10.7717/peerj.16584

**Table 8** (*continued*)

| Material | Depth (m) | Macroscopical morphology | Oxeas (length/width) | Orthotriaenes Rhabdome (length/width) Clad (length/width) | Dichotriaenes Rhabdome (length/width) Protoclad (length/width) Deuteroclad (length/width) | Aspidasters (diameter) | Spined microrhabds (I) (length/width) | Smooth microrhabds (II) (length/width) | Oxyasters (length) |
|---|---|---|---|---|---|---|---|---|---|
| *Scutastra cantabrica* paratype NHM 30.1.21.5 Santander Spain | – | Massive | 449-677-938/4-10-14 | – | Rb: 137-205-260 (N = 5)/7-10-13 (N = 11) Pc: 85/- (N = 1) Dc: 52/- (N = 1) | 32-41-50 (ellipsoidal, underdeveloped) | 21-38-70/1.5-4 (most are centrotylote) | – | 18-26-40 |
| *E. mamillaris* holotype Adriatic NHM slide by *Sollas (1888)* Strasbourg slide by *Topsent (1928)*[*] NHM slide by *Uriz (2002)*[**] | – | Massive lobes, each with a single large oscule at the extremity. External color is black | 1,500/32 750-1,500/18-32[**] | – | Rh: 716/44 Pc: 90/36 Dc: 90 Rh: 532-717/25-44 Pc: 90/32-35 Dc: 60-90/-[**] | 77.5-106/43-51.6 (ellipsoidal) 70-80/38-42 (circular to elongated)[*] 62-106/29-52 (ellipsoidal to more elongated)[**] | 23.7/4 11-18/2 13-24/2-4 (occasionally centrotylote)[**] | – | 19 14-28[**] |
| *E. mamillaris* Bay of Naples (cave) (*Pulitzer-Finalli, 1972*) | 1 | Fragment, brown | 610-(700-900)-1,420/16-20-32 | – | Rh: 280-700/20-40 Pc: 70-110 Dc: 25-260 | 70-89-105/35-47-60 20-30 thick | 16-22-29/2-2.5 | – | 13-23 (6-12 actines) |
| *E.* cf. *mamillaris* ZMAPOR 20421 Faial, Azores | 2-8 | Massive, dark color | 465-833-1,126/7-20-30 | Rh: - Cl: 92-225 (N = 3) | Rb: 182-415-612/13-30-45 (N = 21) Pc: 62-90-127/- Dc: 23-131-200/- | 77-85-102/32-50-42 (all elongated) | 11-16-24/<2.5 | – | 11-16-24 |
| *E.* cf. *mamillaris* Faial, Azores (*Boury-Esnault & Lopes, 1985*) | 6-10 | Massive, lobated | spA: 320-642-1,166/6-16-26 spB: 288-506-736/5-10-13 | – | Rb: 253-275-281/4-5-6 Pc: 77-83-96/3-5-6 Dc: 32-67-83/3-4-6 | spA 68-90-94/36-43-55 spB 30-42-50/25-32-38 | spA 16-20-26/3-3-5 spB 16-20-23/3-3-3 | – | spA 13-17-23 spB 8-12-13 |
| *E.* cf. *mamillaris* i329_B EB | 152 | Massive | 429-817-1,037/5-19-28 (N = 5) | – | not measured | 64-85-109/33-67-91 | 15-20-24/1-2-3 (few) | – | 14-29-56 (6-13 actines) |
| *E.* cf. *mamillaris* i179_A EB | 138 | Small massive | 699-843-949/7-17-24 (N = 6) | – | Rb: 335 (N = 1)/27-40-57 (N = 5) Pc: 115-133-151/27-34-52 (N = 5) Dc: 119-139-161/19-27-39 (N = 5) | 69-97-121/43-68-86 | 18-24-31/2-3-4 (few) | – | 17-35-50 (4–13 actines) |
| *E. deficiens* holotype MNHN DT1111 Madeira | 100 | Massive, very large, lobated, large, upper part black | 510-726-969/3-6-10 | – | Rb: 175-229-360 (N = 9)/7-9-15 (N = 14) Pc: 25-52-72/- (N = 7) Dc: 25-54-132/- (N = 7) | 20-31-55 (mostly discoidal, underdeveloped) | 17-37-80/>2.5 (very abundant) | – | 8-15-22 |

Díaz et al. (2024), *PeerJ*, DOI 10.7717/peerj.16584

Peerj

**Table 8** (*continued*)

| Material | Depth (m) | Macroscopical morphology | Oxeas (length/width) | Orthotriaenes Rhabdome (length/width) Clad (length/width) | Dichotriaenes Rhabdome (length/width) Protoclad (length/width) Deuteroclad (length/width) | Aspidasters (diameter) | Spined microrhabds (I) (length/width) | Smooth microrhabds (II) (length/width) | Oxyasters (length) |
|---|---|---|---|---|---|---|---|---|---|
| ***E. deficiens* ZMAPOR 21693 Gorringe Bank** | 32 | Massive, large | 316-513-643/2-6-9 | – | Rb: 84-208-303/3-5-8 (N = 16) Pc: 157/- (N = 1) Dc: 121/- (N = 1) | – | 22-35-52/thin | – | 8-14-30 |
| ***E. deficiens* ZMAPOR 20419 Madeira** | 7 | Massive, black | 400-749-918/ 2-7-10 | Cld: 75-88-125 (N = 7) | Rh: 107-211-275 (N = 4)/4-7-12 (N = 12) Pc: 50-55-62 (N = 5) Dc: 45-54-65 (N = 5) | 27-37-49 (N = 3) (very rare) | 17-32-80/<2.5 (thin, not centrotylote) | – | 9-14-29 |
| ***E. cf. deficiens* LIT10 Caló den Rafelino (Cave)** | 0 | Small encrusting | 280-503-647/ 4-11-17 | Rb: 76 (N = 1)/8-11 (N = 4) Cld: 61-133/6-10 (N = 4) | Rb: 145-182 (N = 4)/8-17-26 Pc: 47-67-86/8-14-25 Dc: 11-79-135/4-11-18 | 31-42-56 (N = 13) | 22-36-55/2-3-6 | – | 12-21-36 |
| *E. cf. deficiens* Port-Cros, France (*Vacelet, 1976*) | 10-42 | Massive, large | 450-850/ 5-10 | – | Rh: 160-320/5-10 Pc: 40-120/5 Dc: 70-110 | 25-30 or 25-30/30-40 (discoidal to ellipsoidal) (very rare or absent) | 15-32 (up to 50) (abundant) | – | 7-17 |

**Notes.**

Rh, rhabdome; Cl, clad; pc, protoclad; dc, deuteroclad; -, not found/not reported; EB, Emile Baudot.

*In cases where measurements originate from various authors, asterisks are employed to indicate the respective source.

### Ecology and distribution

Single specimen found in a littoral, semi-submerged cave in the intertidal zone, periodically exposed to the air. The cave receives freshwater inputs that may increase silicon levels.

### Genetics

Folmer COI (ON130535) and 28S C1-C2 (ON133853) were obtained.

### Taxonomic remarks

See general discussion on *E. discophorus, E. mamillaris* and *E. deficiens* below, after the description of *E. mamillaris*.

*Erylus discophorus* (*Schmidt, 1862*)
(Figs. 19–20, Table 8)

### Material examined

UPSZMC 190846, field#POR785, St. 6 (MEDITS06N20), fishing ground off Columbretes Islands, GOC-73, 144 m, coll. J. A. Díaz.

UPSZMC 190847-49, field#LIT71, field#LIT72 and field#LIT74, Coves de na Dana (Alcudia), Mallorca, semi-submerged cave, scuba diving, 0–1 m, coll. J. A. Díaz and A. Frank.

### Comparative material

*Scutastra cantabrica* Ferrer-Hernández, 1912, paratype, NHM 30.1.21.5, wet specimen, Santander, Spain; *Erylus discophorus*, ZMUC, off São Pedra Bay, São Vincente, Madeira, 40 m, St. 40, originally identified as *S. cantabrica* by *Burton (1956)*, here re-identified.

### Outer morphology

Fishing ground specimen POR785 is massive (Fig. 20A), lobulate, measuring 10 cm in height and 7 cm in diameter. Color beige with dark shades in life, and after fixation in EtOH. Dark shades more present on the upper part and around the oscula. Choanosome beige. Surface smooth. Consistency hard but slightly flexible. Circular oscula, 2–3 mm in diameter, placed at the top of the lobules. Inhalant pores not observed.

Cave specimens are encrusting (Fig. 20B), 0.3–0.5 cm width, spreading 7–8 cm on the vertical walls. Color in life whitish to beige with some brownish areas, probably caused by diatoms. Same color for the ectosome and the choanosome. Color slightly paler after fixation in EtOH. Inner channels visible only in areas where the body was thinner. Surface smooth, but wrinkled after collection due to contraction. Hard consistency. When alive, many small and circular oscula visible, 1–2 mm in diameter, aligned on the top of small ridges (Fig. 20B). Pore groupings visible to the naked eye, in depressed parts of the specimen.

### Spicules

Dichotriaenes, robust, with short and fusiform rhab. Rhabdome: 123-557/8-62 μm, protoclad: 46-111/7-52 μm and deuteroclad: 10-262/3-44 μm.
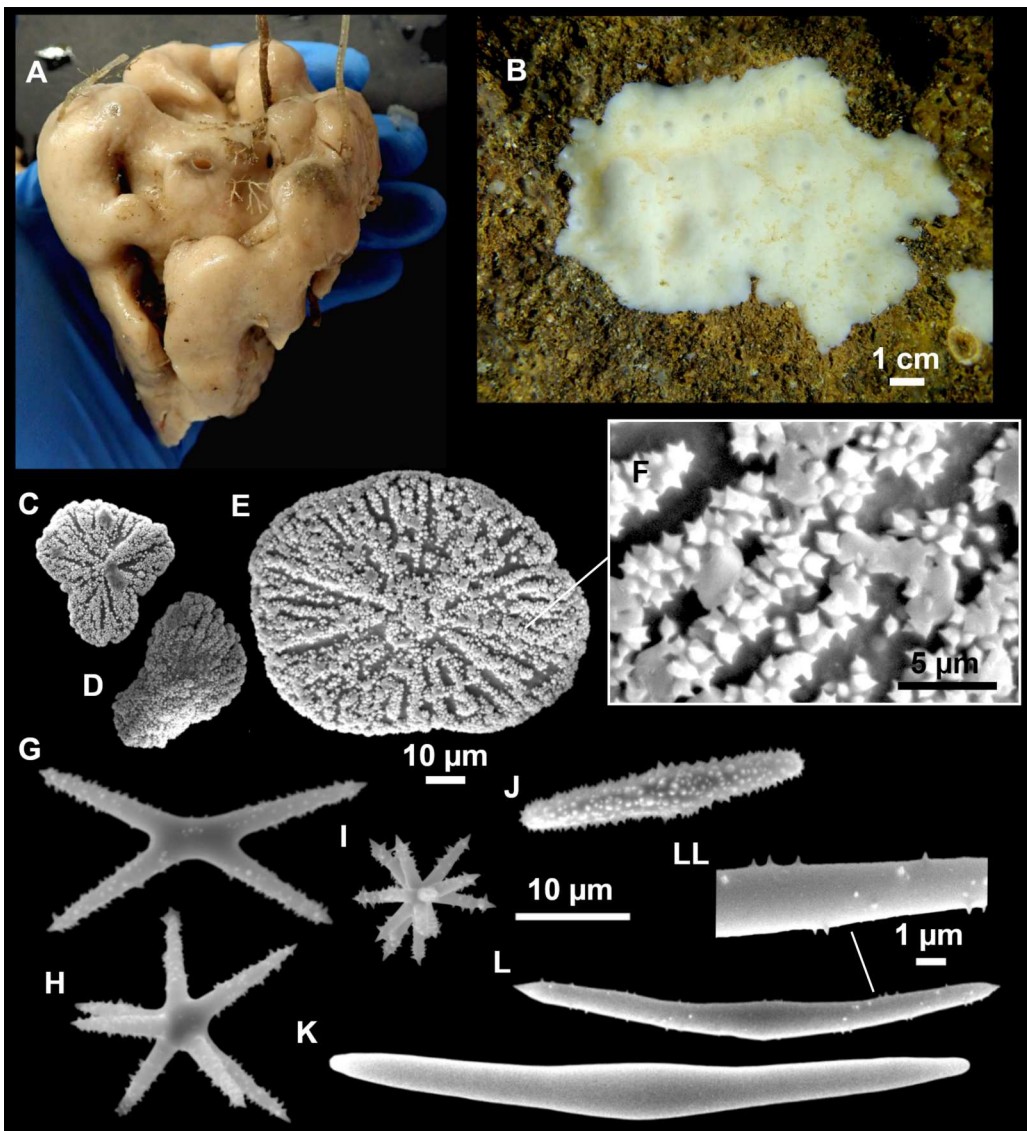

**Figure 20** *Erylus discophorus* (*Schmidt, 1862*). (A) Habitus of POR785 on deck. (B) Cave specimen LIT74 *in situ*. (C–F) Aspidasters of LIT74 with (F) detail of the rosettes. (G–I) Oxyasters from POR785. (J) Microrhabds I from POR785. (K) Smooth microrhabd II from LIT74. (L) Microspined microrhabd II from LIT74 with (LL) detail of the spines.

Orthotriaenes, very scarce, only found in POR785 ($N = 3$) and LIT72 ($N = 1$), rhabdome 135-464/9-18 μm, clad 96-164/7-17 μm.

Oxeas, slightly curved and fusiform, 388-978/5-29 μm, sometimes modified to styles (384-650/10-25 μm) and strongyles (183-796/11-22 μm).

Aspidasters (Figs. 20C–20F), circular to slightly elongated, 33–95 μm (max. diameter).

Oxyasters (Figs. 20G–20I), with 4–12 spined actines, 6–30 μm.

Microrhabds I (Fig. 20J), densely recovered with robust spines, centrotylote, 14-68/1-5 μm.

Microrhabds II (Figs. 20K–20LL) uncommon, smooth to microspined, curved and centrotylote, 41-89/3-6 μm.

### Ecology and distribution

Species found in a fishing ground close to Columbrets (POR785) and in a shallow water cave with freshwater inflow (LIT71, LIT72 and LIT74). Cave specimens were very abundant, and found just below the water surface. The only previous mention of *E. discophorus* in the Balearic Islands was from shallow caves off Cabrera Archipelago (*Uriz, Rosell & Martín, 1992*).

### Genetics

Folmer COI was obtained from all specimens (POR785, ON130531; LIT71, ON130532; LIT72, ON130533; LIT74, ON130534) whereas 28S (C1-C2) was obtained only from POR785 (ON133854).

### Taxonomic remarks

See general discussion after the description of *E.* cf. *mamillaris*.

*Erylus* cf. *mamillaris* (*Schmidt, 1862*)
(Figs. 19 and 21; Table 8)

### Material examined

UPSZMC 190850, field#i142_B, St. 51 (INTEMARES0718), MaC (EB), 128 m, beam trawl, coll. F. Ordines; UPSZMC 190851-52, field#i179_A-179_B, St. 60 (INTEMARES0718), MaC (EB), 138 m, beam trawl, coll. F. Ordines; UPSZMC 190853-56, field#i314, field# i329_A, field#i329_B, field#i329_C, St. 124 (INTEMARES1019), MaC (EB), 152 m, beam trawl, coll. J. A. Díaz.

### Comparative material

*Erylus* cf. *mamillaris*, ZMAPOR 20421, Ponta Furada, Faial Island, Azores, 2–8 m, 6 Sept. 2005, field#FUR05.09.14, coll: J. R. Xavier, COI: EU442207, 28S: EU552090.

### Outer morphology

Massive, ovoid and lobated sponges (Fig. 21A), which often agglomerate foreign sediments, pebbles, worm tubes. The largest specimen (i314) is 9.5 × 5 cm. Single apical oscule on each lobe. Dark brown on its upper side, progressively fading to light brown or whitish at its basal area; choanosome lighter, cream colored. Surface visually smooth but rough to the touch, texture is quite firm, only very slightly compressible. Cortex less than 1 mm thick, clearly distinguishable. Choanosome fleshy, light brown and showing a well-developed aquiferous system.

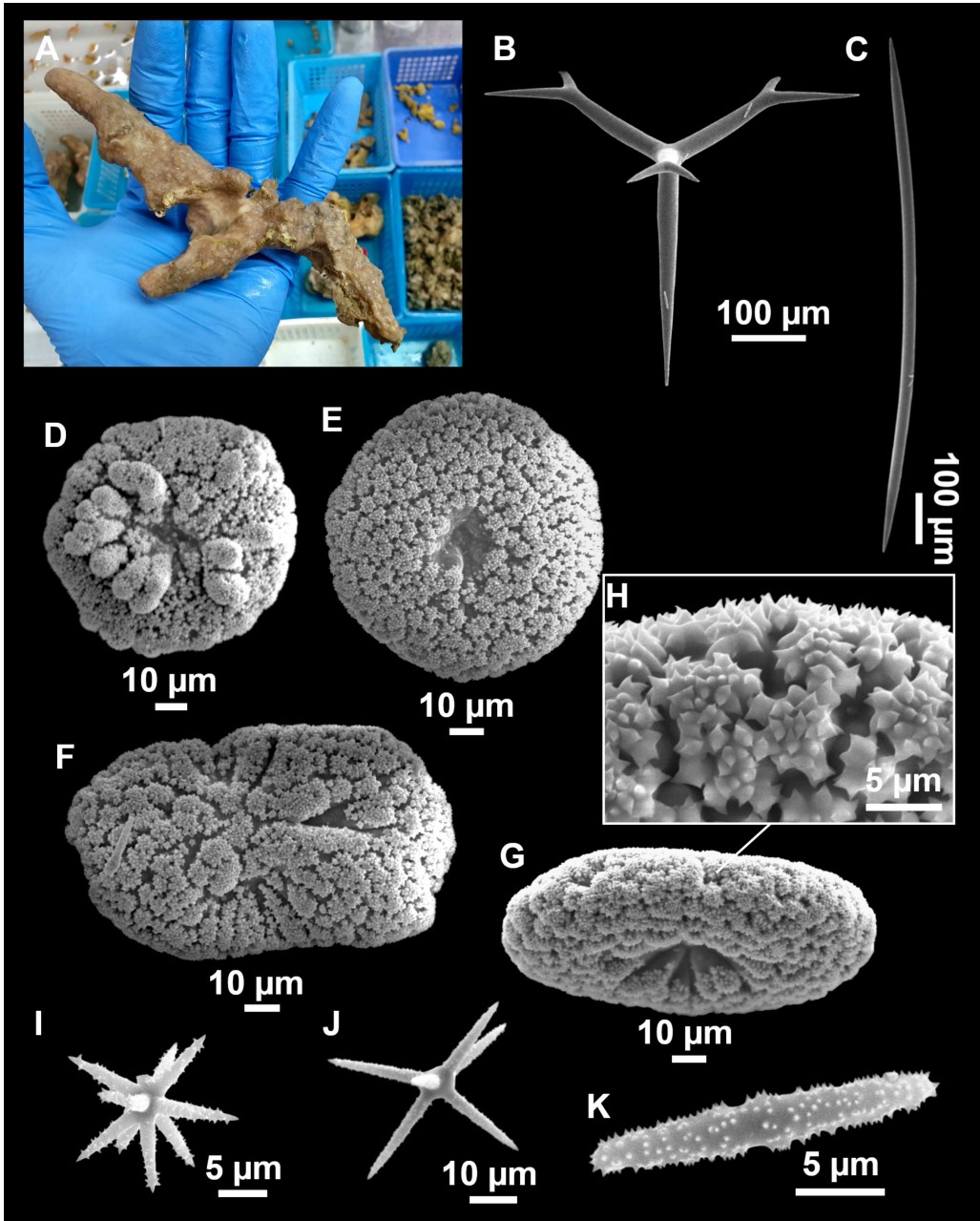

**Figure 21 *Erylus* cf. *mamillaris* (*Schmidt, 1862*).** (A) Habitus of i314 on deck. (B–K) SEM images of i179_1. (B) Dichotriaene. (C) Oxea. (D–H) Aspidasters with (H) detail of rosettes. (I–J) Oxyasters. (K) Microrhabd.

## Spicules

Dichotriaenes (Fig. 21B), scarce, may be significatively thick. Rhabdome: 335 ($N = 1$)/27-
40-57 ($N = 5$) µm, protoclad: 115-133-151/27-34-52 ($N = 5$) µm and the deuteroclad:
119-139-161/19-27-39 ($N = 5$) µm.

Oxeas (Fig. 21C) robust and fusiform, slightly curved 429-1,037/5-28 μm. One single stylote modification was observed in specimen i179_B (not measured).

Aspidasters (Figs. 21D–21H), slightly elongated, 64-121/33-91 μm (length/width).

Oxyasters (Figs. 21I–21J), 4–13 spined actines, 14–56 μm in diameter.

Microrhabds (Fig. 21K), spined and centrotylote, 15-31/1-4 μm.

### Ecology and distribution

Species found at several stations on the EB summit, always on sedimentary bottoms at mesophotic depths, just below the photic zone.

### Genetics

Folmer COI (ON130529) and 28S C1-C2 (ON133884) obtained from specimen i329_B.

### Taxonomic remarks on *Erylus discophorus*, *E. mamillaris* and *E. deficiens*

*E. mamillaris*, *E. discophorus* and *E. deficiens* form a poorly understood complex of Mediterranean and Northeast Atlantic species with similar spicule sets: dichotriaenes, spiny microrhabds, spiny oxyasters and aspidasters (*Sollas, 1888*; *Topsent, 1927*; *Topsent, 1928*; *Cárdenas et al., 2011*; *Cárdenas, 2020*). The conspecificity of specimens assigned to one or the other species have been extensively debated in the literature (*Sollas, 1888*; *Von Lendenfeld, 1894*; *Marenzeller, 1889*; *Topsent, 1901*; *Topsent, 1928*; *Pulitzer-Finali, 1972*). A character proposed to differentiate these three species are aspidasters size, morphology and abundance: larger and more elongated in *E. mamillaris vs.* small and more rounded in *E. discophorus* and extremely scarce to absent in *E. deficiens* (*Topsent, 1928*). We note that this has some consequences on the rosette arrangement of aspidasters in the three species, giving more of a radial pattern in *E. discophorus*, a more regular pattern in *E. mamillaris* and a more irregular/disorganized pattern in *E.* cf. *deficiens*. Also, the size and abundance of microrhabds seem to differ between species, being smaller and less common in *E. mamillaris* and larger and more common in *E. discophorus* (*Sollas, 1888*; *Von Lendenfeld, 1894*) and *E. deficiens* (*Topsent, 1928*; *Vacelet, 1976*; cf. holotype redescription in Table 8). *Von Lendenfeld (1894)* also mentioned microrhabd morphology as a significant character, being more heterogeneous in *E. discophorus* than in *E. mamillaris* (with pointed and rounded variations). In the same work, the presence of differently spined to smooth microrhabds in *E. discophorus* is also mentioned (see Table 8 for details). The Balearic specimens were identified primarily according to aspidaster characters, then the microrhabds were carefully compared (Table 8). Microrhabds are indeed shorter in *E. mamillaris* (15–31 *vs.* 14–68 μm). A second rare category of microrhabds (microrhabds II) was found in *E. discophorus*: larger, smooth to minutely spined. These microrhabds II tend to have pointed tips, close to those drawn by *Von Lendenfeld* (*1894*, Taf III Fig. 41A). This rare spicule type is not singled out in previous descriptions (Table 8) maybe because they are too rare and not easily spotted; for instance, we could not find them in the paratype of *Scutastra cantabrica*. So we are hesitant to consider them as a specific character of *E. discophorus*. However, microrhabds II were absent in all *E. mamillaris* and *E. deficiens* specimens examined, and must be taken into account for a future revision of the types and this species complex.

Regarding spicule size variation in *E. discophorus*, the deeper specimen POR785 had wider oxeas, triaenes and microrhabds and larger oxyasters than the cave specimens (LIT71, LIT72, LIT74). This may be explained by ecophysiological differences, deeper specimens are usually subjected to higher silica concentrations and therefore larger spicules, a phenomenon well described in the Geodiidae (*Cárdenas & Rapp, 2013*). There may be other parameters at play: fishing grounds near Columbrets are a more eutrophic habitat than shallow caves of Mallorca, mainly because of the influence of the river inflows from the Iberian Peninsula.

Macroscopically, *E. deficiens* tends to be described as having a massive lobated external morphology with one single large oscule at the summit of each lobe, and a dark smooth cortex (*Topsent, 1928*). The species was first considered a variety of *E. discophorus* by Topsent, but later erected as a valid species. It was described in Madeira (at 100 m depth), and later collected in the Gorringe Bank (*Xavier & Van Soest, 2007*) and the Ligurian Sea (*Topsent, 1927*; *Vacelet, 1976*). The most remarkable character of *E. deficiens* is the rarity to complete absence of aspidasters. Moreover, *Topsent (1928)* mentions the higher abundance of microrhabds in *E. deficiens* than in *E. discophorus*, to compensate the deficit of aspidasters (the more aspidasters the less microrhabds and vice versa). *Topsent (1928)* also proposed growth habit as a character to separate both species: *E. deficiens* being much larger with lobate processes and *E. discophorus* encrusting and smaller. A spicule slide (MNHN DT 1111) of the holotype of *E. deficiens* from Madeira was compared with specimens morphologically identified as *E. deficiens* from Gorringe Bank (ZMAPOR 21693) (*Xavier & Van Soest, 2007*), Madeira (ZMAPOR 20419) and our cave specimen (LIT10). All specimens were massive except for the encrusting Mallorcan specimen; spicule sizes were all quite similar (Table 8). Microrhabds were similar as in *E. discophorus* but comparatively slightly shorter and thinner.

Type material of *S. cantabrica* from the northern coast of Spain (Santander, Cantabria) is reexamined here for the first time: a new spicule slide was made and spicules measured (Table 8). Shape and size of the spicules conform to those of *E. discophorus*, except for irregular megascleres and the discoid aspidasters which are common but slightly smaller (32-41-50 μm in diameter) and underdeveloped (actines not fused). Such underdeveloped aspidasters and irregular megascleres are actually not uncommon in several *E. discophorus* specimens from Banyuls, France (*Boury-Esnault & Lopes, 1985*) or Portugal (*Cárdenas & Rapp, 2013*, Fig. 12A) and could be linked to low silica concentrations (*Cárdenas & Rapp, 2013*). We therefore confirm the suggestion of *Boury-Esnault & Lopes (1985)* in considering *S. cantabrica* a junior synonym of *E. discophorus*. Another specimen from Madeira identified as *S. cantabrica* by *Burton (1956)* was also re-examined: it has abundant aspidasters (discoidal to elongated, some underdeveloped, 30–55 μm), oxyasters (8–22 μm), mostly straight non-centrotylote spiny microrhabds and regular megascleres: it is therefore re-identified as a typical *E. discophorus*.

*Cárdenas et al. (2011)* had shown that these three *Erylus* species were genetically very close. The present study enriches the sequence sampling to further test the validity of these species. The COI tree (Fig. 19A) is more informative than the 28S tree because we only obtained the more conserved 28S (C1-C2) region which showed no bp differences

between the species (Fig. 19B), so in this case we will only discuss the COI tree. In the COI tree, *E. discophorus* specimens POR785, LIT71, LIT72 and LIT74 strongly group together in a cluster that also includes *Erylus* sp. (EU442206) from Portugal (6 m depth), and *E. discophorus* (HM592692) from Slovenia (5 m). The cluster includes several haplotypes differentiated by 0–2 bp differences. This number of bp differences in COI may indicate the presence of two cryptic species, but also be caused by intraspecific variability. It is far beyond the scope of the present work to elucidate this, and future works with a more extensive sampling shall be conducted. Identical COI sequences were obtained for the encrusting LIT71 and the massive deeper POR785 suggesting that shape is not a diagnostic character. More variable genetic markers would be needed to see if they could be ecotypes. A second paraphyletic group is represented by Atlantic specimens *E.* cf. *mamillaris* (Azores, 7 m) and two *E. deficiens* (Gorringe Bank, 32 m and Madeira, 7 m) with 1 bp. difference. *E.* cf. *mamillaris* has 2 bp differences with both *E. deficiens* sequences. A third group is only represented by the cave Mallorca specimen LIT10 identified as *E.* cf. *deficiens*, which is 6–8 bp different from the other sequences. Since the type locality of *E. deficiens* is Madeira, we can be more or less confident that our specimens sequenced from Gorringe and Madeira are closer to the type locality. This suggests that LIT10 most probably represents a new species from the Mediterranean Sea. However, the difficulty to find clear diagnostic morphological characters and the presence of only one specimen pushes us to delay a new species description. This result also suggests that the character of rare aspidasters may have appeared in different lineages of *Erylus* independently. Finally, the COI sequence of specimen *E.* cf. *mamillaris* i329_B diverges alone, clearly apart from another *E* cf. *mamillaris* from the shallow Azores. This suggests that slightly elongated aspidasters and small microrhabds may not be good specific characters either. All the deep specimens collected from the EB seamount (i179_1, i179_2, i314, i329_A and i329_C) shared the same spicular characters so they must all belong to the same species, which may be conspecific with *E. mamillaris* from the Adriatic Sea. However, comparison with type material would be necessary to address this matter in future works.

Altogether, these phylogenetic results cast doubt on the current spicule characters (aspidaster morphology mainly) used to discriminate these species. To our knowledge, this is the first time such a sponge species complex is revealed in the Mediterranean Sea with many COI haplotypes from populations from different depths and habitats. Additional genetic markers and specimens from these different populations will be necessary to resolve the *Erylus discophorus/mamillaris/deficiens* complex.

*Erylus corsicus* Pulitzer-Finali, 1983
(Figs. 19 and 22; Table 9)

**Material examined**

UPSZMC 190845, field#i707, St. 45 (INTEMARES0720), MaC (EB), 150 m, beam trawl, coll. J. A. Díaz; UPSZMC 190841, field#i389_1, MaC (EB), St. 158 (INTEMARES1019),

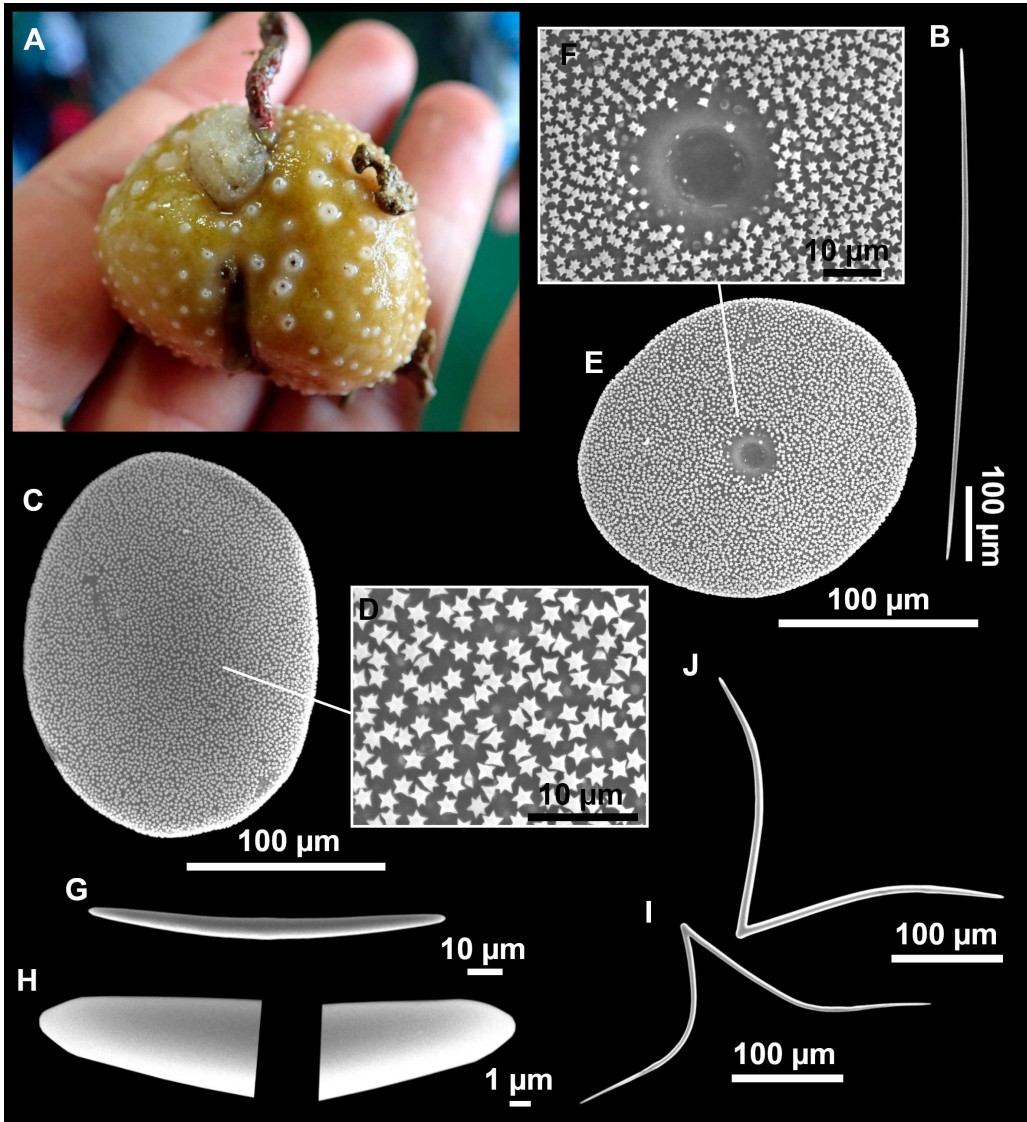

**Figure 22 *Erylus corsicus Pulitzer-Finali, 1983*, specimen i707.** (A) Habitus on deck. (B–J) SEM images. (B) Oxea. (C–F) Aspidasters with (D–F) detail of the rosettes and hilum. (G) Microrhabd with (H) detail of microrhabd tips. (I–J) Toxas.

beam trawl, coll. J. A. Díaz; UPSZMC 190840, field#i356_A, MaC (EB), St. 136 (INTEMARES1019), beam trawl, coll. J. A. Díaz; UPSZMC 190843, field#i402_B, MaC (EB), St. 167 (INTEMARES1019), 151 m, beam trawl, coll. J. A. Díaz.

## Comparative material

*Erylus corsicus*, holotype (slide), MSNG 47157 (NIS.85.14), off Calvi (Corsica), 121–149 m, 14 July 1969.

**Table 9 Spicule measurements of *Erylus expletus*, *E. corsicus* and *E. papulifer*, given as minimum-mean-maximum for total length/minimum-mean-maximum for total width; all measurements are expressed in μm.** Specimen codes are field#. Specimens measured in this study are in bold.

| Material | Depth (m) | Oxeas (length/width) | Triaenes Rhabdome (length/width) Protoclad (length/width) Deuteroclad (length/width) Clad (length/width) | Aspidasters (length/width) | Toxas (length/middle width) | Microrhabds (length/width) |
|---|---|---|---|---|---|---|
| *E. corsicus* holotype MSNG 47157 Corsica | 121-149 | 448-782-1,132/6-11-20 | Rh: 176/6 Cl: 109/5 (N = 1) (orthotriaenes) | 146-185-249/84-127-155 Ratio 1.2-2.0 (many irregular) | 122-195-279/3-6-9 (2 actined, N = 18) 136/3 (3 actined, N = 1) | 39-103-143/2-5-9 |
| *E. corsicus* i707 EB | 147 | 571-943-1,203/9-15-19 | Rh: 654/18 Cl: 323/20 (N = 1) | 172-215-250/139-164-202 Ratio 1.1-1.5 (mostly regular) | 296-425/7-10 (2-actined, N = 7) | 42-91-171/4-6-10 |
| *E. corsicus* i356_A EB | 146 | 534-1,055-1,342/6-17-25 | – | 133-183-230/106-145-181 Ratio 0.9-2.1 (many irregular) | 204-346/6-8 (2-actined, N = 6) (rare) | 35-73-136/3-5-8 |
| *E. corsicus* i402_B EB | 151 | 616-1,152-1,371/6-16-21 | Rh: 340-578/12-31 (N = 3) Cl: 172-387/11-31 (N = 4) One stylote end modification | 144-200-227/120-158-185 Ratio 1.1-1.5 (mostly regular) | 270/8 (1-actined N = 1) (rare) | 44-99-165/2-5-9 |
| *E. papulifer* holotype MSNG 47155 Corsica | 135 | 633-1,034-1,355/13-16-22 | Rh: -/28-43 Pc: 152-238/24-33 Dc: 178-318/19-28 (N = 8) (mostly dichotriaenes) | 150-175-192/114-150-173 Ratio 1.0-1.5 20-30 thick (subspherical) | 117-167/7-9 (1 actined, N = 4) 68-135-201/4-7-11 (2 actined) 96-149/3-7 (3 actined, N = 8) 66/4 (4 actined, N = 1) | 46-60-70/3-4-5 |
| *E. papulifer* paratype MSNG 47156 (Nis19.4c) Corsica | 135 | 1,033-1,235-1,462/12-17-24 (N = 18) | Rh: 541-606 (N = 2)/20-43 (N = 9) Pc: 151-236/11-32 Dc: 126-353/12-28 (N = 9) (mostly dichotriaenes) | 136-178-195/114-139-157 Ratio 1.1-1.5 (oval to lemon shaped) | 67-133-184/3-6-11 (2 actined, N = 20) 112-140/4-7 (3 actined, N = 2) | 52-67-86/2-4-6 |

| Material | Depth (m) | Oxeas (length/width) | Triaenes Rhabdome (length/width) Protoclad (length/width) Deuteroclad (length/width) Clad (length/width) | Aspidasters (length/width) | Toxas (length/middle width) | Microrhabds (length/width) |
|---|---|---|---|---|---|---|
| **E. papulifer paratype MSNG 47156 (Nis19.4b) Corsica** | 135 | 548-1,090-1,451/ 9-15-19 (N = 19) | Rh: -/18-u-31 Pc: 205-221-238/16-21-29 Dc: 91-152-236/12-15-20 (N = 4) (mostly dichotriaenes) Rh: 186/27 Cl: 281/22 (N = 1; ortho-triaene) | 150-172-199/116-135-149 Ratio 1.2-1.4 | 74-103-155/2-5-8 (2 actined, N = 13) 72-99-121/2-3-6 (3 actined, N = 6) | 49-59-74/ 3-4-6 |
| *E. papulifer* Marseille caves, France (*Pouliquen, 1972*) | 2-10 | 350-900/ 10-12 | Rh: (short) Cl: 320 Pc: 100-120 Dc: 130-150 (ortho- and dichotriaenes) | 90-160/- | 120-180/- | 30-80/3-5 |
| *Erylus expletus* MNHN DT837/1326 holotype off São Jorge, Azores (*Topsent, 1904*; *Topsent, 1927*; *Topsent, 1928*) | 1,022 | 700–1,700/18 | Rh: - Cl: 385-525 (orthotriaenes) | 245-295/ 147-205 30-50 thick (lemon shaped) | 1-2 actines size not reported | 80-90/- 180-230/- 2 sizes |
| **Erylus expletus ZMAPOR 18142 Rockall Bank, NE Atlantic** | 679 | 500-1,077-1,536/ 8-19-25 | Rh: 449-479-530/20-22 (N = 4) Cl: 235-398-459 (N = 12) (orthotriaenes) | 290-337-367/190-230-255 25-35 thick (oval to lemon shaped) | 2 actins: 110-224-296/4-7-12 3 actins (N = 4): 150-199-235/4-5-7 | 37-59-88/2-3-5 118-179-230/5-8-10 2 sizes |
| *Erylus* cf. *expletus* Tenerife, Canary Islands on *Ircinia* sp. (*Cruz, 2002*) | *Circalitoral zone* | 530-700/- | Thin orthotriaenes, actines measure 130-480 | 180-300/- | 80-160/- | 30-160/- |

**Notes.**
Rh, rhabdome; Cl, clad; pc, protoclad; dc, deuteroclad; -, not found/not reported; EB, Emile Baudot.

*Erylus papulifer Pulitzer-Finali, 1983*, holotype, MSNG 47155 (NIS.19.4a), wet specimen, off Calvi Corsica, 135 m, 18 July 1975; paratype, MSNG 47156 (NIS.19.4c), same locality as holotype.

*Erylus expletus Topsent, 1927*, holotype, MNHN DT837 and DT1326, two slides, off Säo Jorge, Azores, 1 Aug. 1895, St. 616, 1022 m; ZMAPOR 18142, SE of Rockall Bank, West coast of Ireland, 55°30′13.93″N, 15°47′6.18″W, 679 m, 2 Sept. 2004, coll: R. van Soest, field# M2004/33-05, id. P. Cárdenas, COI: EU442208.

## Outer morphology

Massive globular sponges, 1 to 3.5 cm in diameter (Fig. 22A). Hard, slightly compressible cortex, fleshy choanosome. After fixation in EtOH, the choanosome contracts and tends to get separated from the cortex, which remains resilient. Beige color in life, some individuals have a brownish tinge in localized areas, product of diatom colonization. The brownish stains disappear after EtOH fixation, and then both cortex and choanosome are beige. Small abundant openings, often with a lighter colored ring, scattered all over the surface, 0.1–1 mm in diameter.

## Spicules

Orthotriaenes, very scarce, cladi curved outwards and rhabdome straight. Rhabdome: 340-654/12-31 μm. Cladi: 172-387/11-31 μm.

Oxeas (Fig. 22B), slightly curved and fusiform, 534-1,371/6-25 μm.

Aspidasters (Figs. 22C–22F), discoid to elongated but most are oval, being up to two times longer than wider. Underdeveloped or aberrant forms are present, more or less common depending on the specimen. Rosettes of type 3 (Fig. 22D) *sensu Cárdenas (2020)*; well defined hilum (Fig. 22F). On average, length: 133–250 μm, width: 106–202 μm.

Microrhabds (Figs. 22G–22H), on a wide size range, smooth, slightly curved and faintly centrotylote, 35-171/2-10 μm.

Toxas (Figs. 22I–22J), scarce, most 2-actined, rarely 3-actined. With a central swelling, overall measuring 204-425/6-10 μm.

## Ecology notes

Species found at the summit of the EB, between 146–151 m, on sandy sedimentary bottoms. The species is always covered by a brownish coating produced by diatom aggregations (observation with microscope). The diatoms seem to be favored by the smooth surface that offers the aspidasters on its external side, acting as substrate.

## Genetics

COI was obtained from specimens i707 (ON130536) and i402_B (ON130537) while the 28S (C1-C2) fragment has only been obtained from specimen i707 (ON133851).

## Taxonomic remarks

Thought to be divergent reduced oxyasters (*Topsent, 1928*), toxas are a rare spicule type in *Erylus* species. And yet, three very similar *Erylus* with toxas are recorded from the Mediterranean Sea: *E. corsicus, E. papulifer* and *E. expletus* (this last one is also reported from the Northeast Atlantic). The type material of these three species was compared here for

the first time to test their validity and to compare with our material (Table 9). *E. expletus* is the first one to be described, from the deep waters of the Azores, 1022 meter depth (*Topsent, 1927*; *Topsent, 1928*) and subsequently from Rockall Bank (*Van Soest et al., 2007*), shallow waters of the Canary Islands (*Cruz, 2002*) and underwater caves around Marseille, France (*Pouliquen, 1972*). *E. papulifer* is based on six specimens collected from mesophotic depths in Corsica (*Pulitzer-Finali, 1983*). The type material has been re-examined here (Figs. 23A–23K): they are subglobular sponges, the holotype measures 2.5 cm in diameter (Fig. 23A) while paratypes measure about one cm in diameter. They have a hard but breakable consistency, slightly compressible. Color after formalin fixation is dirty beige with orangish pink (peach) areas. Small abundant openings are scattered all over the surface, 0.1–1 mm in diameter. In terms of external morphology, *E. expletus* (*Topsent, 1928*, Pl. I, Fig. 20), *E. papulifer* (Fig. 23A) and our material (Fig. 22A) are very similar. Actual differences between *E. papulifer* and *E. expletus* were not clearly stated by *Pulitzer-Finali (1983)* except for "*the shape and size of aspidasters*". *E. corsicus* is also described by *Pulitzer-Finali (1983)*, based on a single tiny specimen, which was completely digested to make a single spicule preparation, which was examined here. The *E. corsicus* specimen was collected in the same dredge as several *E. papulifer* specimens. *Pulitzer-Finali (1983)* based *E. corsicus* on having larger and more irregular aspidasters than *E. papulifer*, and a wider length range of microrhabds (48–160 μm *vs.* 50–80 μm in *E. papulifer*).

After careful observation and measurements of spicules (Table 9), we first note that *E. papulifer* is the only species with dichotriaenes while *E. expletus* and *E. corsicus* have very few orthotriaenes. *E. expletus* seems to be discriminated by (i) a majority of aspidasters with a characteristic lemon-shape (*Topsent, 1928*, pl. V, Fig. 10), while *E. corsicus/papulifer* have oval to lemon-shape or irregular aspidasters (Figs. 22C–22E and 23D), (ii) two separate sizes of microrhabds *vs.* only one in *E. corsicus/papulifer* (Figs. 22G and 23K) and (iii) absence of 3-actin toxas (in the holotype), which are present in *E. corsicus/papulifer*, but quite rare (Fig. 23F), which makes this third character difficult to use. Following the two first characters mentioned, our material is not *E. expletus*. No 3-actin toxas were found in our material but since they can be quite rare, we may have missed them. Then our material shares some characters with *E. corsicus* and some with *E. papulifer*. As in *E. corsicus*, our material has rare orthotriaenes, irregular aspidasters (not in all our specimens), and larger aspidasters, toxas and microrhabds. As in *E. papulifer*, some of our specimens have mostly regular aspidasters and have 1-actin toxas. This suggests that the irregular aspidasters may be due to the environment and therefore would not be a reliable specific character. SEM observations of toxas in the holotype of *E. papulifer* revealed spines at the tips of the actines (Figs. 23I–23J), which were not present in our material (Figs. 22I–22J) or in *E. expletus* from Rockall Bank (data not shown). This new character needs further confirmation in order to consider it as diagnostic. All things considered, our specimens were identified as *E. corsicus*, the second report of *E. corsicus* after its original description. However, the validity of *E. corsicus* originally based on a single tiny specimen from the same locality as *E. papulifer* remains dubious and would need to be further tested with additional material, and more importantly genetic sequences from other Mediterranean populations. In the Mediterranean Sea, *E. expletus* has been reported from shallow water caves off Marseille,

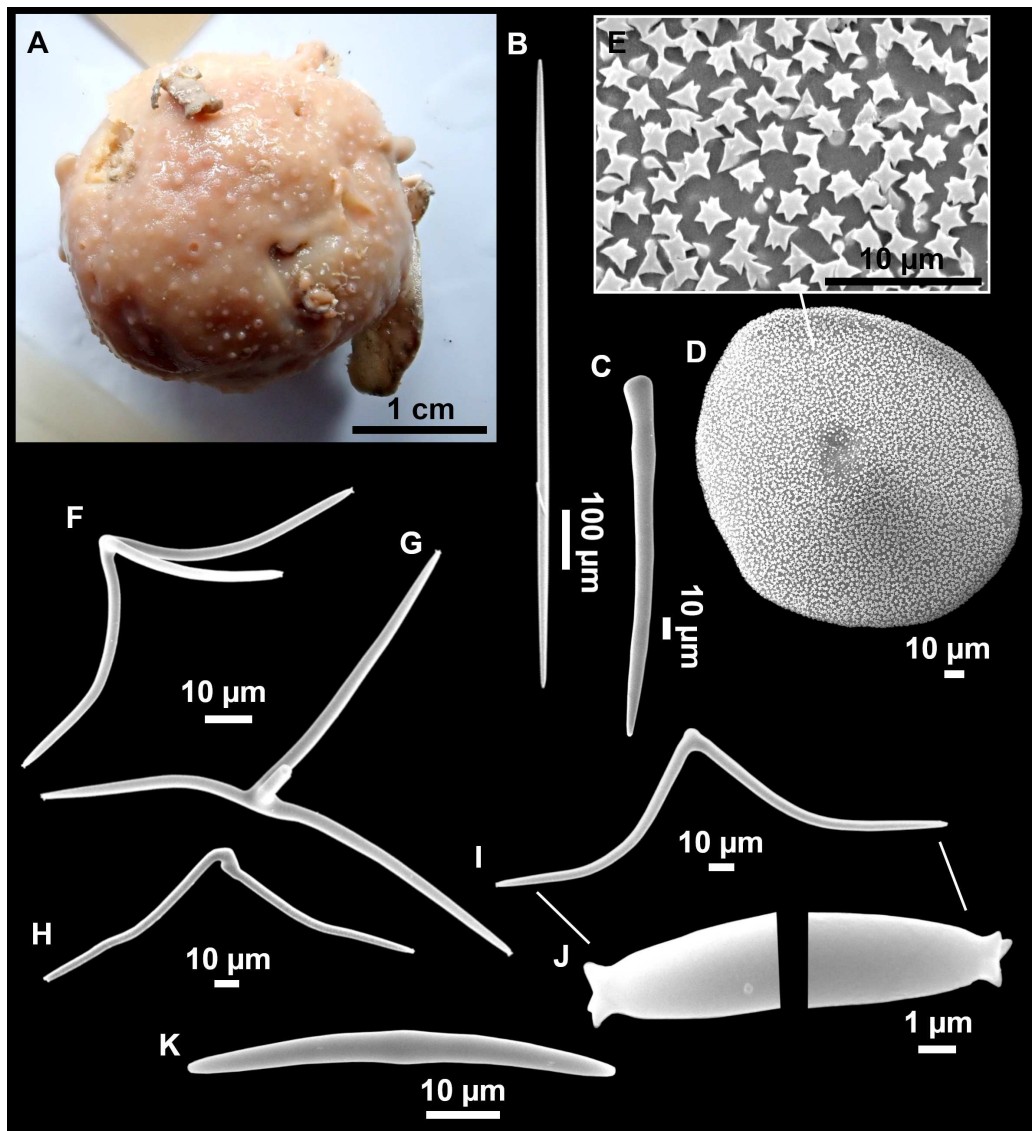

**Figure 23 Holotype of *Erylus papulifer* *Pulitzer-Finali, 1983*, MSNG 47155, Corsica.** (A) Habitus, formalin preservation. (B) Oxea. (C) Style. (D) Aspidaster with (E) detail of the rosettes. (F–J) Oxyasters with (J) detail of the spurs at oxyasters tips. (K) Microrhabd.

France (*Pouliquen, 1972*). Since the Marseille specimens have smaller aspidasters, toxas and oxeas than *E. expletus* and only one size of microrhabds, we propose that they are instead *E. papulifer*, thus restricting *E. expletus* to the North Atlantic. More specimens and sequences of *E. expletus*, *E. papulifer* and *E. corsicus* are now necessary to test the robustness of these spicular characters. The COI tree currently confirms that our *E. corsicus* and *E. expletus* are different but very close species with only a 3 bp difference (Fig. 19A). The phylogenetic position of *E. corsicus/E. expletus* within the Erylinae is still ambiguous and not supported (Fig. 19).

Genus *Penares* Gray, 1867
*Penares euastrum* (*Schmidt, 1868*)
(Figs. 19 and 24; Table 10)

### Material examined

UPSZMC 190921-190922, field#i142_A-i146_4, St. 51 (INTEMARES0718), MaC (EB), 128 m, beam trawl, coll. J. A. Díaz; UPSZMC 190924, field#i524_b, St. 17 (INTEMARES0720), MaC (AM), 113 m, beam trawl, coll. J. A. Díaz; UPSZMC 190926-190927 and UPSZMC 190920, field#i528-529 and field#i530, St. 18 (INTEMARES0720), MaC (AM), 112 m, coll. J. A. Díaz, beam trawl; UPSZMC 190928, field#POR469, St. 219 (MEDITSES052017), Son Bou (South-west of Menorca), 65 m, GOC-73, coll. J. A. Díaz; UPSZMC 190929, field# POR932_1, St. 74 (MEDITSES052020), Favàritx (Northeast off Menorca), 75 m, GOC-73, coll. J. A. Díaz; UPSZMC 190930, field#POR975, St. 78 (MEDITS052020), Ciutadella (West off Menorca), 56 m, GOC-73, coll. J. A. Díaz; UPSZMC 190931, field#POR1141, St. 185 (MEDITS052021), Sa Costera (North of Mallorca), 61 m, GOC-73, coll. J. A. Díaz; UPSZMC 190932, field#POR1253, St. 02 (MEDITS0521_PITIUSSES), South of Formentera, 54 m, GOC-73, coll. J. A. Díaz.

### Comparative material

*Penares euastrum*, holotype, MNHN Schmidt collection#76, wet specimen, La Calle, Algeria, coll: H. de Lacaze-Duthiers.

### Outer morphology

Massive, irregular lobose sponge, reaching about 12 cm in maximum diameter (Fig. 24A). Surface dark brown except at the basal area, where it is whitish. Pores are located on white circular areas throughout the body, giving a characteristic appearance. Circular oscula placed apically. Cortex <1 mm.

### Spicules

Dichotriaenes (Fig. 24B), scarce, whose cladome has longer protoclads than deuteroclads. Rhabdome: 344-387/19-36 , protoclad: 148-206/20-45 , deuteroclad: 46-220/11-36 μm. In the observation of several *P. euastrum* specimens we have observed modified dichotriaenes with one, two and three (orthotriaenes) of its clads non-bifurcated. Also, we observed one rhabdome modified to a rounded end (Fig. 24B).

Oxeas, with variable morphology; some are straight while others are bend or abruptly curved. Tips may be pointed, stepped or rounded, resembling strongyles, measuring 465-1390/4-38 μm.

Aspidasters (Figs. 24C–24E), shape variable between individuals, some showing a predominance of discoid to oval (POR932_1) or only oval (i529, i530, POR469, POR1253), sometimes irregular (Fig. 24E). Stumps in the hilum area found only in POR932_1, having several stumps per aspidaster and showing a bulbous shape, type 3 rosettes *sensu Cárdenas (2020)*, measuring 97-182/74-142 μm (maximum diameter/min diameter).

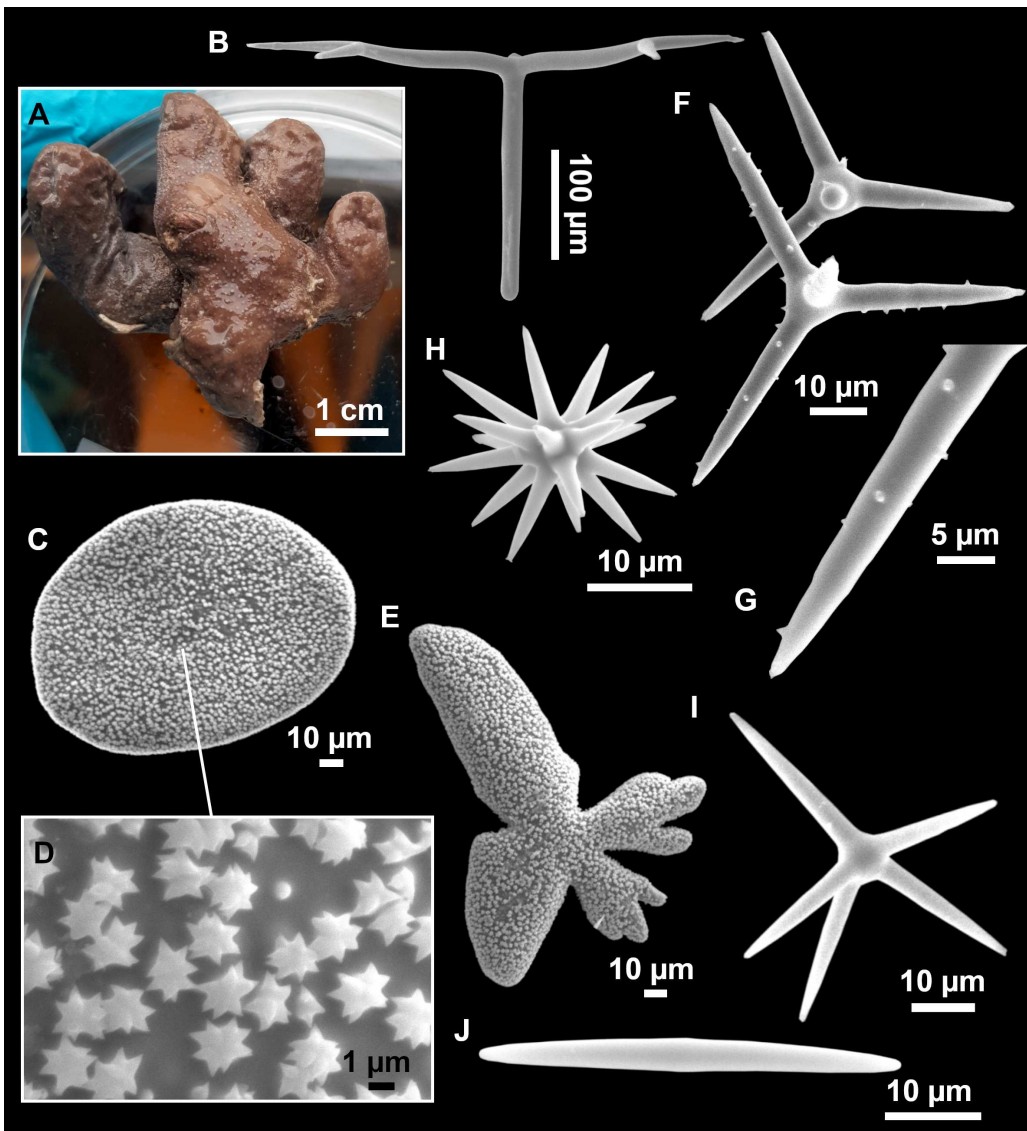

**Figure 24  *Penares euastrum* (*Schmidt, 1868*), specimen i142_A.** (A) Habitus after EtOH fixation. (B) Dichotriaene. (C–E) Aspidasters with (D) detail of the rosettes. (F) Oxyasters I with (G) detail of the spines. (H–I) Oxyasters II. (J) Microrhabd.

Oxyasters I (Fig. 24F), with 2–10 actines, smooth or with few spines, barely visible with an optical microscope (Fig. 24G). There is a relationship between size and number of actines, being in general larger those with less actines. Measuring 25–74 μm in diameter.

Oxyasters II (Figs. 24H–24I), with >10 actines, smooth or with few spines, not visible to the optical microscope, measuring 8–28 μm in diameter.

Microrhabds (Fig. 24J), smooth, centrotylote, with blunt ends 27-74/2-6 μm.

## Ecology and distribution

Common species usually found at mesophotic depths: on fishing grounds of the platform, on *Peyssonnelia* and rhodolith bottoms and at the summit of the AM and the EB seamounts. May have several sponge epibiont species like *Timea* sp. and some poecilosclerids.

## Genetics

The Folmer short miniCOI was obtained from i530 and POR932_1 (SBP#2683 and SPB# 2684), while 28S (C1-C2) (ON133852) was obtained from i530.

## Taxonomic remarks

See remarks below for *Penares cavernensis* **sp. nov.**

*Penares cavernensis* **sp. nov.** Díaz & Cárdenas
(Figs. 19 and 25; Table 10)

## Etymology

*Penares cavernensis*, from the Latin *cavernae* (cave), referring to the habitat where the species was discovered.

## Material examined

Holotype: UPSZTY 190918, field#LIT55, Cova Cala Sa Nau, east of Mallorca, 3–4 m, scuba diving, coll. J. A. Díaz & J. Cabot.

Paratypes: UPSZTY 190917, field#LIT45, Cova Cala Sa Nau, east of Mallorca, 3–4 m, scuba diving, coll. J. A. Díaz; UPSZTY 190919, field#LIT65, Caló des Monjo, west of Mallorca, 6 m, scuba diving, coll. J. A. Díaz & A. Frank.

## Outer morphology

Massive-encrusting, lobated sponges (Figs. 3F and 25A), about 2–5 cm in maximum diameter, and 1–3 cm in height. Surface color can be dark brown to beige, whitish rim around the oscula. Choanosome whitish. Same coloration in life and after fixation in ethanol. Several circular oscula located apically, 1–3 mm in diameter, smaller whitish pores are visible with the naked eye. No hispidation present, surface smooth to the touch. Very thin cortex present, about 0.4 mm. Hard but slightly compressible consistency. The skin wrinkles after EtOH preservation. In the holotype, a large cloaca can be observed below the main apical oscula.

## Spicules

Dichotriaenes (Fig. 25B), scarce, morphology and abundance varies with the individuals. They are very scarce in specimens found in Cala sa Nau (LIT45 and LIT55), with robust rhabdome and cladome. In the specimen from Caló des Monjo (LIT65), dichotriaenes are more abundant but with thinner rhabdome and cladome, also showing teratogenic modifications and aborted actines. Overall measuring: rhabdome 407-515/7-27 μm, protoclad: 86-213/9-22 μm, deuteroclad: 49-139/7-15 μm.

**Table 10  Spicule measurements of *Penares euastrum, Penares cavernensis* sp. nov. and *Penares aspidodiscus*, given as minimum-mean-maximum for total length/minimum-mean-maximum for total width; all measurements are expressed in μm.** Specimen codes are the field#. Specimens measured in this study are in bold.

| Material | Depth (m) | Oxeas (length/width) | Orthotriaenes Rhabdome (length/width) Clad (length/width) | Dichotriaenes Rhabdome (length/width) Protoclade (length/width) Deuteroclade (length/width) | Aspidasters (Maximum diameter/minumum diamater) | Microrhabds (length/width) | Oxyasters I (length) | Oxyasters II (length) |
|---|---|---|---|---|---|---|---|---|
| ***P. euastrum* holotype MNHN#76 La Calle, Algeria** | – | 872-1,043-1,348/ 13-18-21 (*N* = 5) | Rh: 526/29 Cl: 364/21 (*N* = 1) | Rh: -/29-34 Pt: 155-224/21-26 Dt: 89-164/17-20 (*N* = 3) | 100-131-165/ 60-91-107 (oval) | 37-48-56/ 2-3-4 (*N* = 14) (abundant) | 32-49-62 (2-6 actines) | 8-11-16 |
| ***P. euastrum* POR932_1 Menorca** | 75 | 465-895-1,178/ 4-12-17 (*N* = 19) (strongyle modifications) | Always with one or two bifurcated clads | Rh:-/33-36 Pt: 154-180/23-28 Dt: 59-71/14-21 (*N* = 2) | 97-121-142/8 5-103-118 (discoid to oval) (bulbous stumps in the hilum) | 43-55-74/ 2-3-4 (abundant) | 25-49-59 (2-6 actines) | 8-15-28 |
| ***P. euastrum* POR975 Menorca** | 56 | 638-868-1,302/ 6-16-25 (*N* = 12) (strongyle and style modifications) | n.m. | n.m. | 116-138-154/88-102-116 (*N* = 16) (discoid to oval) | 37-46-55/2-3-4 (*N* = 11) (scarce) | 25-53-74 (*N* = 19) (2–6 actines) | 9-13-19 (*N* = 19) |
| ***P. euastrum* POR1253 Formentera** | 53 | 661-1,013-1,271/ 15-24-38 (*N* = 11) | n.m. | n.m. | 118-143-172/74-89-98 (mostly oval) | 27-42-59/2-3-3 (scarce) | 43-60-77 (*N* = 21) (2-5 actines) | 8-13-15 |
| ***P. euastrum* i142_A EB** | 128 | n.m. | n.m. | n.m. | 128-151-182/94-109-142 (*N* = 11) | 39-58-71/3-5-5 (*N* = 13) (abundant) | 29-48-63 (3–9 actines) (*N* = 14) | 15-20-22 (*N* = 7) |
| ***P. euastrum* i524_b AM** | 113 | 587-849-1,172/ 13-19-26 (*N* = 9) | n.m. | n.m. | 111-145-171/77-107-125 (discoid to oval) | 43-54-69/3-4-6 (abundant) | 39-49-57 (4–10 actines) | 12-17-21 |
| ***P. euastrum* i530 AM** | 112 | 756-1,045-1,390/ 15-24-33 | n.m. | Rh: 344-387/19-31 (*N* = 3) Pt: 148-185-206/20-31-45 Dt: 46-138-220/11-23-36 (*N* = 8) | 121-144-168/87-98-109 (oval) | 36-49-67/3-4-5 (abundant) | 34-53-66 (3-7 actines) | 11-17-24 |
| *P. euastrum* Banyuls, France as *E. stellifer* (*Topsent, 1894*) | 25-30 | 1,000/ 20-25 | Rh: - Cl: 250/27 | Rh: - Pt: 220-270/28-30 Dt:50-100/- | 135/ 95 (oval) | 55-65 | actines 23 long (3-5 actines) | actines 5 long |
| *P. euastrum* Ibero-Moroccan Gulf, North Atlantic (*Boury-Esnault, Pansini & Uriz, 1994*) | 1,378 | 910-1,015-1,150/ 16-20-20 (oxeas to styles and strongyles) | – | Rh: short Pt: 160-210-250/30-35-40 Dt: 200-237-260/25-35-41 | 150-160-164/ 110-125-140 (oval to irregular) | 35-52-82 (microxeas) | 35-53-80 (3–7 actines) | 12-14-15 (few) |
| *P. euastrum* Banc East Gettysburg, North Atlantic, as *E. stellifer* (*Lévi & Vacelet, 1958*) | 95 | 950/- | – | Rh: 70/35 Pt:175 Dt:100-160 | 130/ 100 (oval) | 33-40 (microxeas) | 25-35 (3–5 actines) | 5-14 |
| *P. euastrum* Canary Islands, North Atlantic (*Cruz, 2002*) | Infralittoral | 500-850 (strongyles fusiform) | – | Rh: 220-400 Pt: 120-160 Dt: 80-210 | 156/ 60 (oval to irregular) | 24–40 (microstrongyles) | actines 20-40 long (3-5 actines) | 10-16 (microspined) |

**Table 10** (*continued*)

| Material | Depth (m) | Oxeas (length/width) | Orthotriaenes Rhabdome (length/width) Clad (length/width) | Dichotriaenes Rhabdome (length/width) Protoclade (length/width) Deuteroclade (length/width) | Aspidasters (Maximum diameter/minumum diamater) | Microrhabds (length/width) | Oxyasters I (length) | Oxyasters II (length) |
|---|---|---|---|---|---|---|---|---|
| *P. cavernensis* sp. nov. LIT55, holo-type Cova Cala Sa Nau | 3-4 | 766-1,006-1,259/ 7-14-22 (N = 12) | – | Rh: -/27 Pt: 141/22 Dt: 139/15 (N = 1) | 74-102-127/ 62-80-96 (discoid to oval, often with thin stumps ) | 36-45-58/ 2-3-3 (abun-dant) | 32-44-56 (2–5 actines) | 8-10-13 |
| *P. cavernensis* sp. nov. LIT45, paratype Cova Cala Sa Nau | 3-4 | 426-627-853/ 5-8-12 (N = 19) | Rh: 426-548/13-18 (N = 3) Cl: 155-233-291/7-13-17 (N = 5) | Rh:441/20 Pt: 183/15 Dt: 77/10 (N = 1) | 84-113-135/ 66-84-98 (discoid to oval, often with thin stumps) | 35-43-55/ 1-2-3 (abun-dant) | 34-48-68 (2–5 actines) | 7-11-14 |
| *P. cavernensis* sp. nov. LIT65, paratype Cova Cala Sa Nau | 6 | 524-839-1,033/ 6-10-15 (N = 16) | Rh: 289 (N = 1)/9-13 (N = 2) Cl: 109-242/8-9 (N = 2) | Rh: 407-515 (N = 3)/7-13-16 (N = 4) Pt: 86-134-213/9-12-14 (N = 4) Dt:49-66-74/7-10-12 (N = 4) | 67-97-128/ 54-77-94 (discoid to oval, rarely with thin stumps) | 28-36-56/ 1-2-2 (abun-dant) | 32-50-68 (2-4 actines) | 6-9-14 |
| *P. aspidodiscus* holo-type Monaco, Medt. (*Topsent, 1928*) | 123 | usually with blunt ends | yes | yes (often Pt>Dt) | 130/ 110 (discoid, often with thin stumps) | 35–60 (mi-croxeas) | 10–55 (3–5 actines) | <16 |

**Notes.**

Rh, rhabdome; Cl, clad; pc, protoclad; dc, deuteroclad; -, not found/not reported; n.m., not measured; EB, Emile Baudot; AM, Ausias March.

Orthotriaenes (not shown), scarce and mostly broken, with long clads slightly curved inwards, rhabdome: 289-548/9-18 µm, cladi: 109-291/7-17 µm.

Oxeas (Fig. 25C), straight or curved, with slightly stepped tips, 426-1,259/5-22 µm.

Aspidasters (Figs. 25D–25F), discoid to oval (the majority are oval), very thin, several having one or more characteristic stump(s) in the hilum area (Fig. 25E), which are isolated actines; type 3 rosettes *sensu Cárdenas (2020)*, measuring 67-135/54-98 µm (maximum diameter/min diameter). Translucent appearance with an optical microscope due to their extreme thinness.

Oxyasters I (Figs. 25G–25J), with very thin actines, smooth or minutely spined, 2–5 actines, 32–68 µm.

Oxyasters II (Figs. 25K–25LL), many spined actines, (Fig. 25LL), 6–14 µm.

Microrhabds (Fig. 25M), smooth, curved and centrotylote, with blunt ends, 28-58/1-3 µm.

## Ecology and distribution

Found on vertical walls of shallow caves, Mallorca Island, at 3–4 m depth.

## Genetics

The Folmer miniCOI has been obtained from the holotype (SPB#2685).

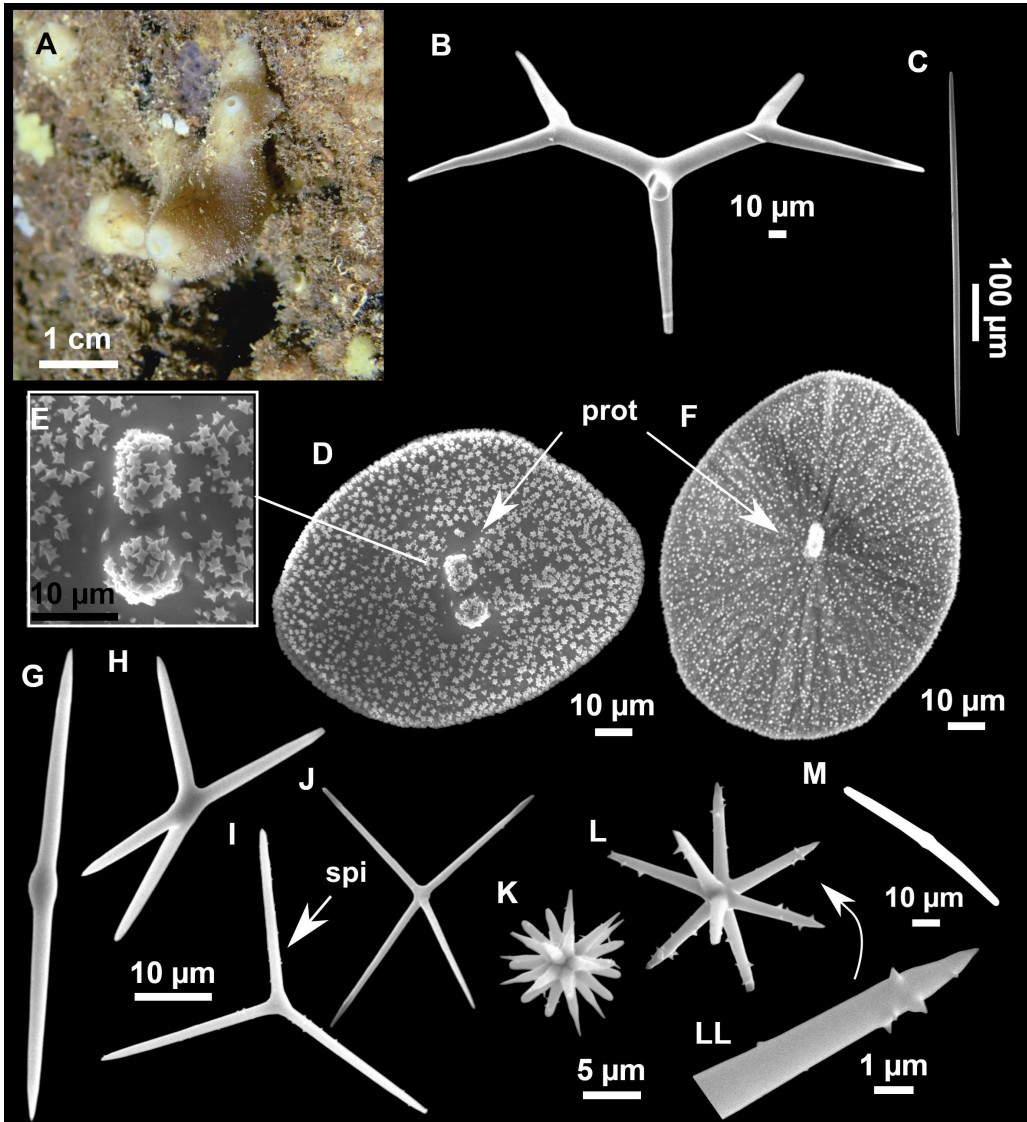

**Figure 25** **Holotype of *Penares cavernensis* sp. nov., UPSZTY 190918 (LIT55).** (A) Habitus *in situ*, at 3–4 m depth, Cova Cala Sa Nau. (B–M) SEM images of the spicules. (B) Dichotriaene. (C) Oxea. (D–F) Aspidaster with (E) detail of the rosettes and the central protuberances. (G–J) Oxyasters I. (K–LL) Oxyasters II with (LL) detail of the spines. (M) Microrhabd. prot, protuberance; spi, spines.

## Taxonomic remarks on *Penares euastrum* and *Penares cavernensis* sp. nov.

*Penares euastrum* is a former *Erylus* species (*i.e.,* Geodiidae with aspidasters) that was moved to the genus *Penares* by *Cárdenas et al. (2010)* based on its close phylogenetic position with *P. helleri*, the type species of the genus *Penares*. The Schmidt holotype of *P. euastrum* from Algeria was here examined, and all the spicules remeasured for the first time (Figs. 26A–26I, Table 10). SEM pictures notably showed that the oxyasters I and II can be spiny to slightly spiny (Figs. 26D–26H) and not always smooth as most of the previous

descriptions suggested. Oxyasters of our *P. euastrum* material (Figs. 24F–24I) were similar, smooth to spiny. *P. euastrum* is a common Mediterranean species, regularly reported from caves to mesophotic depths. We had originally identified our cave specimens from Mallorca as *P. euastrum*, however, while the two miniCOI from our mesophotic specimens (i530 and POR932_1) were a perfect match to the COI of a mesophotic *P. euastrum* from Italy (MT815827), the miniCOI sequence from our cave specimen LIT55 showed a 2 bp difference with the Italian and the Balearic islands mesophotic specimens. This is a significant difference considering how conserved COI is in demosponges (*Schuster et al., 2017*), the geographical closeness of the two habitats and the short size (130 bp) of the miniCOI. In fact, having 2 or more bp differences in a full (about 640 bp) COI sequence is generally considered indicative of different species. External morphology differences are few and subtle but consistent: mesophotic specimens are larger and with a darker coloration while cave specimens are encrusting to massive-encrusting and of a paler color. Although spicules are very similar with overlapping size ranges (Table 10), overall cave specimen spicules are thinner and smaller. Aspidasters of *P. euastrum* are slightly larger and wider (97-182/74-142 μm *vs* 67-135/54-96) with rosettes more densely disposed (Figs. 24D, 25E and 26C). Besides, aspidasters of *P. cavernensis* sp. nov. are more translucid under the light microscopy than those of *P. euastrum*, probably because of having less densely arranged rosettes and being thinner (although we did not manage to get thickness measurements of the aspidasters). Moreover, aspidasters from cave specimens are occasionally discoid (much rarer in mesophotic specimens) and commonly have ectopic actines in the hilum area, which is quite distinct (Figs. 25D–25F). Ectopic actines have been found in all three specimens of *P. cavernensis* sp. nov. examined but only in one of the nine Balearic Islands *P. euastrum* studied (specimen POR932_1), and are absent in the holotype (see Table 10, Figs. 24 and 26). The holotype of *P. euastrum* from Algeria is a large mesophotic specimen (3.5 × 3 cm) (Fig. 26); its spicule measurements are closer to our mesophotic specimens, which are therefore formally identified as *P. euastrum*. *Topsent (1928)* had also distinguished a variety of *P. euastrum* called *Erylus aspidodiscus,* which, based on its morphological similarity with *P. euastrum* should actually be moved to the genus *Penares* as well, as *Penares aspidodiscus* comb. nov. This species was unfortunately based on a single specimen, from mesophotic depths close to Monaco (123 m) with spicule sizes closer to those of the holotype of *P. euastrum*. It was characterized by having only discoid aspidasters, commonly with ectopic actines, just like in *P. cavernensis* sp. nov. Since one of our *P. euastrum* also showed ectopic actins (POR932_1), it seems that this character may be shared by *P. euastrum* and *P. cavernensis* sp. nov., although being much more common in the second. Also, *P. cavernensis* sp. nov. does not have exclusively discoidal aspidasters like in *P. aspidodiscus,* in fact the majority of the aspidasters are oval in *P. cavernensis* sp. nov. *P. aspidodiscus* thus shares some characters from *P. euastrum* and some from *P. cavernensis* sp. nov. and is of uncertain taxonomic status at this point.

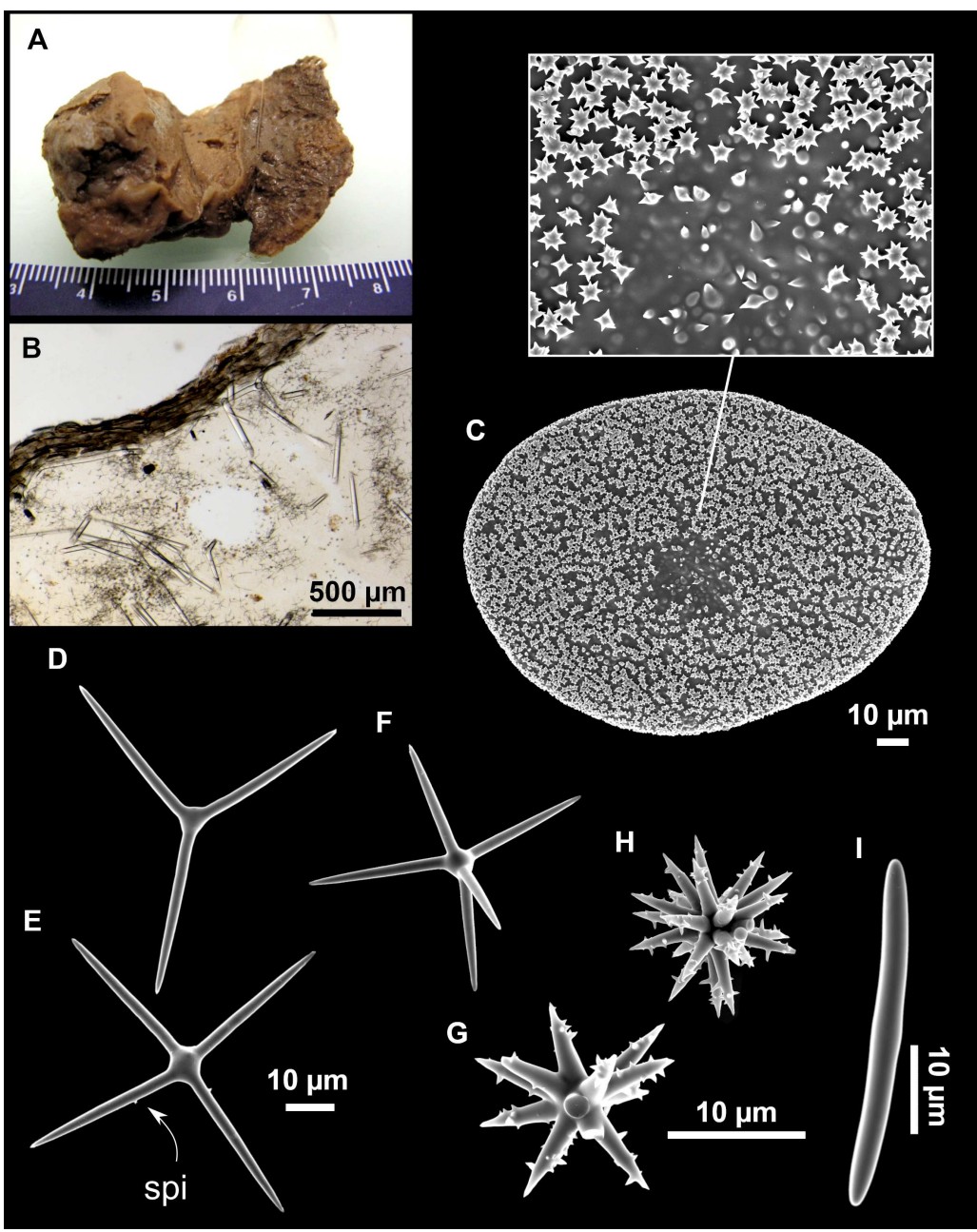

**Figure 26 Holotype of *Penares euastrum*, MNHN Schmidt collection#76, La Calle, Algeria.** (A) Habitus. (B) Optical microscope image of a thick transversal section. (C) Aspidasters with detail of the rosettes and hilum. (D–F) Oxyasters I with arrow (spi) indicating the spines. (G–H) Oxyasters II. (I) Microrhabd.

Regarding 28S, we only manage to get a C1-C2 fragment of mesophotic *P. euastrum* (i530). Interestingly, it showed 3 bp difference with another mesophotic *P. euastrum* from the 'Banc de l'Esquine' (off La Ciotat, France, AF062600), which may indicate that there is even more diversity than suspected in the *P. euastrum* complex.

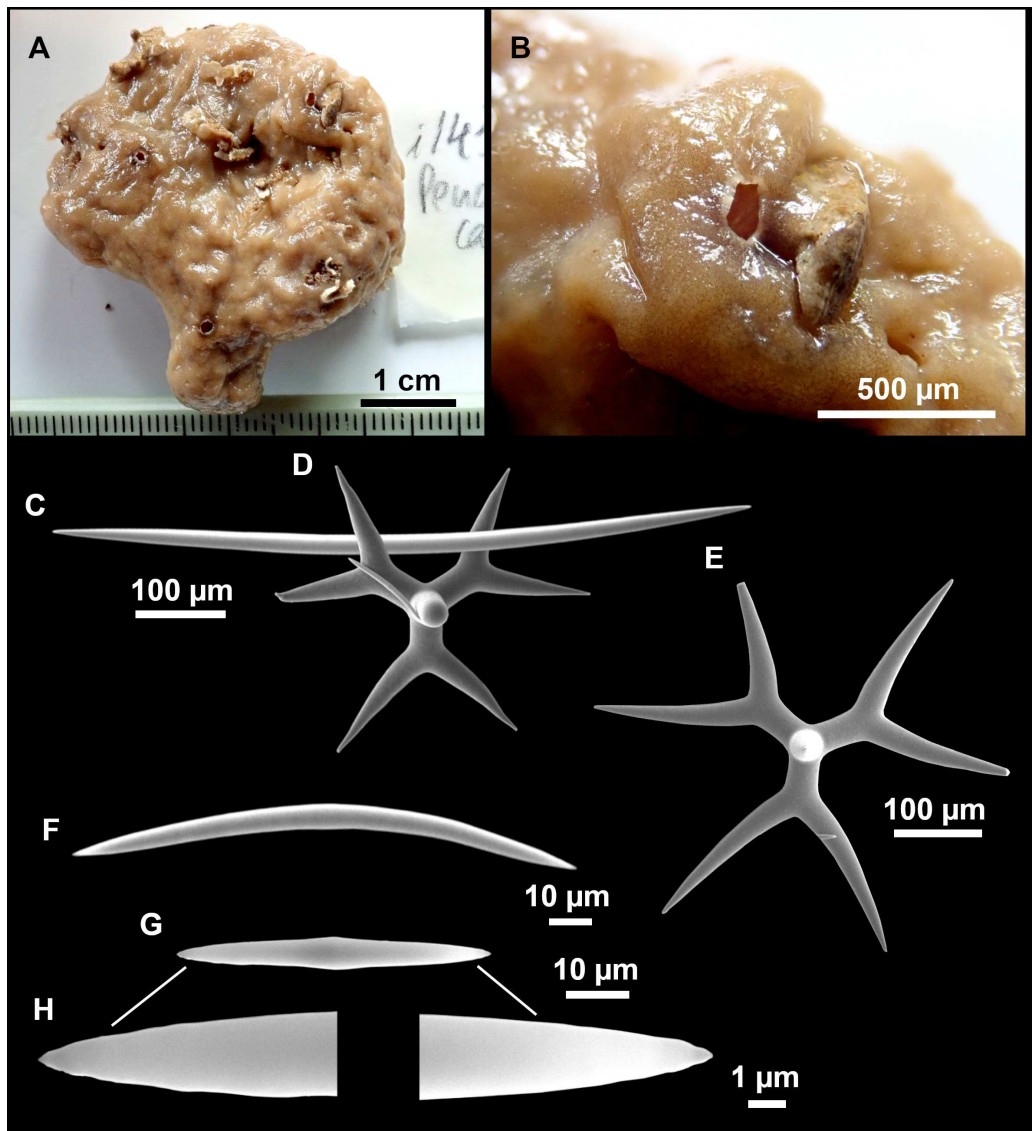

**Figure 27** *Penares candidatus* (*Schmidt, 1868*), **specimen i143_G.** (A) Habitus after EtOH fixation. (B) Detail of the oscula of i143_G. (C) Oxea. (D–E) Dichotriaene. (F) Large microxea. (G) Small microxea with (H) detail of the tips.

*Penares candidatus* (*Schmidt, 1868*)
(Figs. 19 and 27; Table 11)

## Material examined

UPSZMC 190913, field#i143_G, St. 51, INTEMARES0718, MaC (EB), 128 m, beam trawl, coll. F. Ordines; UPSZMC 190914-15, field#i315_A-315_B, St. 124, (INTEMARES1019), MaC (EB), 152 m, beam trawl, coll. J. A. Díaz.

**Table 11  Spicule measurements of *Penares candidatus*, given as minimum-mean-maximum for total length/minimum-mean-maximum for total width; all measurements are expressed in μm.** Specimen codes are the field#. Specimens measured in this study are in bold.

| Material | Depth (m) | Oxeas (length/width) | Dichotriaenes Rhabdome (length/width) Clad (length/width) Protoclade (length/width) Deuteroclade (length/width) | Microrhabds (length/width) |
|---|---|---|---|---|
| **i143_G EB** | 128 | 402-668-974/ 7-14-25 | Rh: 197-229-274/24-31-38 ($N = 3$) Pt: 52-68-97/9-27-39 ($N = 12$) Dt: 37-163-255/6-24-38 ($N = 12$) | 35-109-207/3-7-12 (centrotylote or not) |
| **i315_A EB** | 152 | 464-774-1,156/ 5-16-26 | Rh: 225-310/30-34-39 ($N = 2$) Pt: 59-72-87/17-35-49 ($N = 20$) Dt: 66-191-258/15-29-40 ($N = 20$) | 35-99-216/2-5-9 (centrotylote or not) |
| Holotype MNHN#86 Algeria (*Sollas, 1888*) | – | 816/- | Rh: short Pt: 71 Dt: 177 | 50-250 (centrotylote or not) |
| Cap de Creus & Banyuls Western Mediterranean (*Topsent, 1894*) | 30-40 (Banyuls) 90-100 (Cap de Creus) | 825-1,200/ 23-25 | Rh: 265/30 Pt: 76 Dt: 165 | 30-250 (centrotylote or not) |
| Blanes Western Mediterranean (*Bibiloni, 1981*) | 6 (facies of *Peyssonnelia rubra*) | 450-500/ 10 | Rh: short Pt: 60 Dt: 100 | 50-100/4-5 (not centrotylote) |
| Canary Islands (*Cruz, 2002*) | – | 320-700/- | Rh: 92-160 Pt: 48-92 Dt: 12-96 | 40-60 (not centrotylote) |

**Notes.**

Rh, rhabdome; Cl, clad; pc, protoclad; dc, deuteroclad; -, not found/not reported; EB, Emile Baudot.

**Comparative material**

*Penares candidatus*, holotype, MNHN Schmidt collection#80, wet specimen, Algeria, 'Exploration Scientifique de l'Algérie', 1842.

**Outer morphology**

Massive, lobated sponges (Fig. 27A), up to 5 cm in maximal diameter. Greenish to dirty gold in life, light to dark brown with a faint yellowish tinge after preservation in ethanol. Choanosome beige after ethanol fixation, cavernous. Cortex about 0.2 mm thick. Hard consistency, only slightly compressible. No hispidation, smooth surface and heavily wrinkled. Several circular oscula, 2–3 mm in diameter, located at the apex of the lobules and spread over the body. A characteristic dark rim surrounds the oscula (Fig. 27B). Pores inconspicuous.
### Spicules

Dichotriaenes (Figs. 27D–27E), with a short, straight, and sharp rhabdome, measuring 197-310/24-39 µm ($N = 5$). Cladome with cladus of approximately the same length as rhabdome. Protoclads measuring 52-97/9-49 µm, deuteroclads measuring 37-258/6-40 µm.

Oxeas (Fig. 27C), robust, fusiform, slightly bent, with sharp tips, 402-1,156/5-26 µm.

Microrhabds (Figs. 27F–24H), smooth, curved and with sharp ends (Fig. 27H), measuring 35-216/2-12 µm. Some are very slightly centrotylote, more clear in small spicules.

### Ecology notes

Species found at two stations, both at the EB summit: one at 132 m and the other one at 152 m. Both stations corresponded to sponge grounds, with a great sponge diversity and abundance, including large tetractinellids like *Discodermia polymorpha* (*Pisera & Vacelet, 2011*) or *Erylus* spp., small axinellids and haplosclerids like *Petrosia (Petrosia) ficiformis* (Poiret, 1789) and *Petrosia (Strongylophora) vansoesti* (*Boury-Esnault, Pansini & Uriz, 1994*).

### Genetics

COI (ON130543) and 28S (C1-C2) (ON133857) sequences of specimen i315_B were obtained. The 28S sequence represents the first published sequence of this marker for the species.

### Taxonomic remarks

This is the first report of *P. candidatus* in the Balearic Islands, and the deepest for the species, extending its bathymetric range down to 152 m. The species is also reported for the first time on a seamount. *P. candidatus* is easily recognizable by its external appearance and the absence of oxyasters, a unique feature amongst North Atlantic/Mediterranean *Penares* species. However, there is some heterogeneity in the literature regarding spicule sizes (Table 11). Microrhabds have smaller size ranges in *Bibiloni (1981)* (50–100/4–5 µm), and *Cruz (2002)* (40–60 µm) and are never centrotylote in contrast to the holotype (50–250 µm), specimens from Banyuls, France (30–250 µm) (*Topsent, 1894*) and our material (35–216/2-12 µm). A similar tendency is observed with the oxeas, which are smaller in Bibiloni and Cruz (450-500/10 µm and 320–700 µm, respectively) than in our specimens (825-1,200/23-25 µm and 402-1,156/5-26 µm, respectively) and the rest of the literature (Table 11). These discrepancies may be explained by the depth, and thus silica availability, where individuals were collected, as Bibiloni specimens came from shallower waters (Blanes, 6 m), while possibly the holotype and our specimens were collected at greater depths (>30 m, and up to 152 m). Another explanation is that *P. candidatus* represents a species complex between some shallow and deeper populations, as for *Penares helleri* and *Penares euastrum* (see discussion above and below), but this is currently not supported by COI sequences, which are 100% identical for specimens from 5 m (Berlengas Islands, Portugal, HM592719) and our mesophotic specimens.

*Penares helleri* (*Schmidt, 1864*)
(Figs. 19 and 28; Table 12)

## Material examined

UPSZMC 190933, field#i142_D, St. 51 (INTEMARES0718), MaC (EB), 128 m, beam trawl, coll. F. Ordines; UPSZMC 190934, field#i152, St. 52 (INTEMARES0718), MaC (EB), 107–110 m, rock dredge, coll. F. Ordines; UPSZMC 190935, field#i233, St. 48 (INTEMARES1019), MaC (AM), 123 m, beam trawl, coll. J. A. Díaz; UPSZMC 190936, field#i739, St. 52 (INTEMARES0720), MaC (EB), 300-330 m, beam trawl, coll. J. A. Díaz; UPSZMC 190937, field#POR946, St. 76 (MEDITS2020), s'Algar (east of Menorca), 131 m, GOC-73, coll. J. A. Díaz.

## Comparative material

*Penares* sp.1, PC325, 3PP cave, La Ciotat, France, scuba-diving, 28S: AF062598, originally identified as *P. helleri* in *Chombard, Boury-Esnault & Tillier (1998)*.

   *Penares* sp.2, ZMAPOR 21658, field#FLW.06.48, Gruta Enchareus (cave), Flores, Azores, 12 m, scuba-diving, 11 July 2006, coll: J. R. Xavier, 28S: HM592828, originally identified as *P. helleri* in *Cárdenas et al. (2011)*.

## Outer morphology

Massive ovoid and lobated sponges (Fig. 28A), colors alive are dark brown on its upper side, progressively fading to light brown or whitish in its basal area. Some specimens are entirely whittish (*e.g.*, i739). Colors after ethanol fixation are brown (basal area) to dark brown (apical area). Surface visually smooth but rough to the touch, slightly compressible. Cortex 0.5–0.7 mm thick clearly distinguishable. Choanosome fleshy, light brown after ethanol fixation and showing a well developed aquiferous system. 1–3 small oscules (mm size) irregularly distributed, uniporal pores, much smaller (<1 mm) grouped in some areas. In some specimens the ethanol acquires a dirty gold coloration during the fixation.

## Spicules

Dichotriaenes (Figs. 28B–28C), abundant, rhabdome: 236-293/24-62 $\mu$m, longer deuteroclad than protoclad: protoclad 51-103/18-59 $\mu$m, deuteroclad 99-265/17-50 $\mu$m.
   Oxeas, robust, fusiform and slightly bent, 684-1,641/10-42 $\mu$m.
   Oxyasters (Figs. 28D–28H), microspined (Fig. 28E), with 4–10 actines, 11–56 $\mu$m.
   Microrhabds (Figs. 28I–28K), on a wide but continuous size range, smooth, can have mucronated ends, which are only visible under SEM microscope (Fig. 28K), centrotylote, measuring 19-171/2-8 $\mu$m.

## Ecology and distribution

Species found in sedimentary bottoms from mesophotic to upper bathyal depths (100–300 m). It is common at the fishing grounds of the upper slope of the Mallorca and Menorca shelf and at the summit and slopes of the AM and the EB.

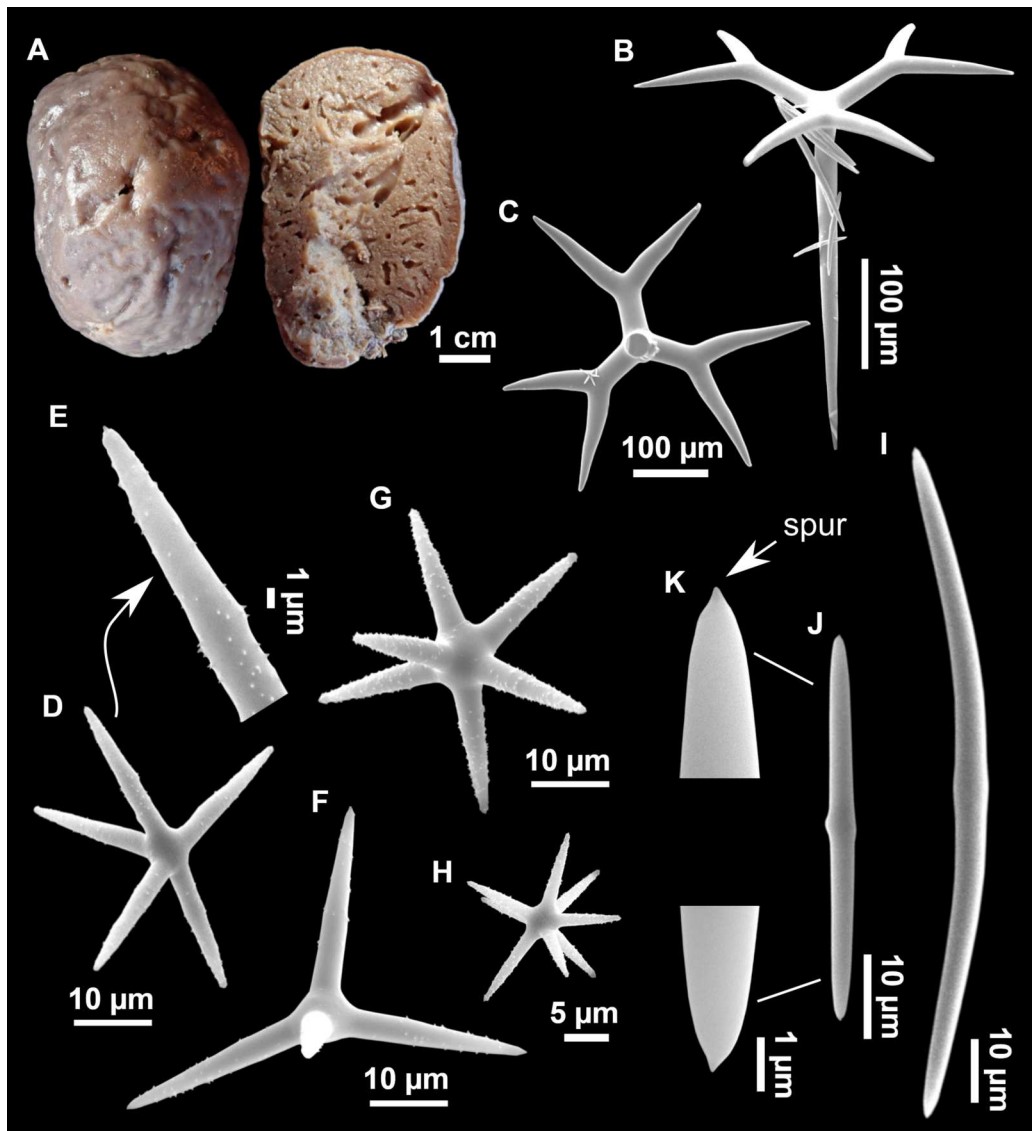

**Figure 28** *Penares helleri* (*Schmidt, 1868*), **specimen POR946.** (A) Habitus after preservation. (B–C) Dichotriaenes. (D–H) Oxyasters with (E) detail of the spines. (I–K) Microrhabds with (K) detail of the spurs at the tips.

## Genetics

The COI (ON130538, ON130539) and 28S (C1-C2) fragments (ON133859, ON133858) have been obtained from specimens i739 (EB) and POR946 (s'Agar). These are the first sequences from mesophotic *P. helleri*.

## Taxonomic remarks

See below taxonomic remarks for *Penares isabellae* **sp. nov.**

**Table 12 Spicule measurements of *Penares isabellae* sp. nov. and *Penares helleri*, given as minimum-mean-maximum for total length/minimum-mean-maximum for total width; all measurements are expressed in μm.** Specimen codes are field#. Specimens measured in this study are in bold.

| Material | Depth (m) | Oxeas (length/width) | Dichotriaenes Rhabdome (length/width) Protoclade (length/width) Deuteroclade (length/width) | Microrhabds (length/width) | Oxyasters (length) |
|---|---|---|---|---|---|
| **P. helleri POR946 Menorca** | 131 | 684-1,099-1,337/10-22-30 | Rh: -/24-44-59 (N = 12) Pt: 54-74-86/23-41-59 (N = 12) Dt: 99-170-222/17-35-50 (N = 12) | 31-82-146/2-4-7 (centrotylote) | 13-28-45 |
| **P. helleri i739 EB** | 300-330 | 755-1,243-1,641/10-26-42 | Rh: 236-293 (N = 3)/25-36-62 Pt: 51-68-103/18-33-52 (N = 17) Dt: 112-179-265/17-28-49 (N = 17) | 29-94-150/3-5-8 (centrotylote) | 12-33-56 |
| **P. helleri i152 EB** | 107-110 | n.m. | n.m. | 36-72-121/2-3-6 (N = 14) (centrotylote) | 15-30-53 |
| **P. helleri i233 AM** | 123 | n.m. | n.m. | 19-98-171/2-4-8 (centrotylote) | 11-27-48 |
| P. helleri holotype, Vis (=Lissa), Croatia (*Sollas, 1,888*) | 64.7 | 1,430/39 | Rh: 400/35 Pt: 60-90 Dt: 190-240 | 32-150/6 (centrotylote) | actin of 20 |
| P. helleri Gibraltar, Atlantic side (*Boury-Esnault, Pansini & Uriz, 1994*) | 521 | 710-1,050-1,550/18-28-40 | Rh: 345-377-410/28-31-35 Pt: 65-79-90/40-47-55 Dt: 175-222-260/40-45-50 | 20-105-150/4-8-10 (centrotylote) | 30-41-50 |
| P. helleri Gulf of Naples (*Pulitzer-Finali, 1972*) | 1–10 m | 1,100/25 or 600-750/- (slightly centrotylote in shallow specimen) | Rh: short Pt: 86-135/ Dt: 54-145/ | 30-190/10, reaching 230 in shallow specimen (rarely centrotylote) | 13-40 |
| P. helleri Ionian Sea (Porto tricase)* Ligurian Sea (Bogliasco)** (*Pulitzer-Finali, 1983*) | 15 and 30* 15** | 1,000-1,500/15-35 | Rh: 200-400/25-40 Cladome: 150-330 | 30-240 (moderately centrotylote) | 13–50 |
| P. helleri var. *subtilis* Gulf of Naples, rocky bottom (# 775:2, #783:2) (*Sarà & Siribelli, 1960*) | 20–25 | 374-510/6-7 | Only one found | 52-80/1-1.7 (#783:2) | – |

**Table 12** (*continued*)

| Material | Depth (m) | Oxeas (length/width) | Dichotriaenes Rhabdome (length/width) Protoclade (length/width) Deuteroclade (length/width) | Microrhabds (length/width) | Oxyasters (length) |
|---|---|---|---|---|---|
| *P. isabellae* sp. nov. holotype, LIT48 Cala sa Nau | 3-5 | 661-802-913/7-13-17 | Rh: 169 ($N=1$)/15-22 ($N=2$)<br>Pt: 38-59/16-19 ($N=2$)<br>Dt: 105-114/12-17 ($N=2$) | 40-74-145/2-3-5 | 13-21-30 (11-18 actines) |
| *P. isabellae* sp. nov. paratype, LIT40_1 Cala sa Nau | 3-5 | 436-638-821/6-8-14 | – | 35-84-168/2-3-6 | 10-17-28 (10-17 actines) |
| *P. isabellae* sp. nov. paratype, LIT66 Caló des Monjo | 3-5 | 353-640-854/6-10-14 | – | 31-98-165/2-3-4 | 9-17-29 (8–17 actines) |

**Notes.**

Rh, rhabdome; Cl, clad; pc, protoclad; dc, deuteroclad; -, not found/not reported; n.m., not measured; EB, Emile Baudot; AM, Ausias March.

*Penares isabellae* **sp. nov.** Díaz & Cárdenas
(Figs. 19 and 29; Table 12)

### Etymology

Named after Isabel Fullana Riera (a.k.a. Bel Fullana), a Mallorcan painter.

### Material examined

Holotype: UPSZTY 190938, field#LIT48, Cova Cala Sa Nau, east of Mallorca, 3–4 m, scuba diving, coll. J. A. Díaz.

Paratypes: UPSZTY 190939, field#LIT40_1, Cova Cala Sa Nau, east of Mallorca, 3–4 m, scuba diving, coll. J. A. Díaz; UPSZTY 190940, field#LIT66, Caló des Monjo, west of Mallorca, 6 m, scuba diving, coll. J. A. Díaz and A. Frank.

### Outer morphology

Massive-encrusting sponges (Figs. 4E and 29A), 8–9 cm in length (sometimes more), 0.5 cm in height. Whitish gray in life and after preservation in ethanol. Surface smooth, slightly rough to the touch. Flexible and compressible. Cortex less than 0.5 mm, clearly distinguishable. Choanosome whitish. Circular oscula 3–5 mm in diameter (Fig. 29B). Minute pores distinguishable with the naked eye when alive, located on depressed areas present all over the surface (Fig. 29C), due to body contraction after fixation, these areas are more difficult to see.

### Spicules

Dichotriaenes (Fig. 29D), very scarce (only found in the holotype), with short rhabdome, cladome may show aberrant cladi. Rhabdome: 169 ($N=1$)/15-22 ($N=2$) µm, protoclad 38-59/16-19 ($N=2$) µm, deuteroclad 105-114/12-17 ($N=2$) µm.

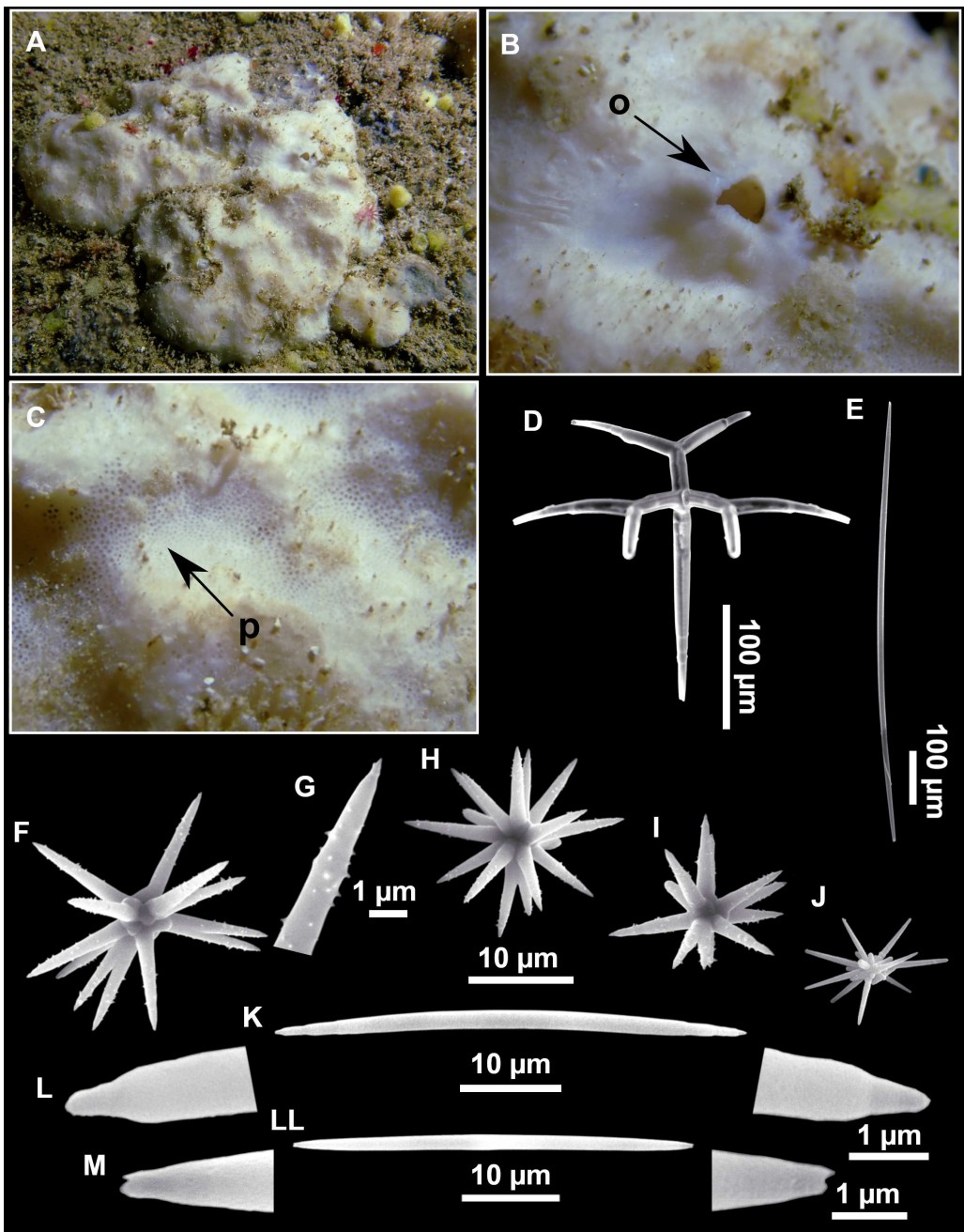

**Figure 29 Holotype of *Penares isabellae* sp. nov., UPSZTY 190938 (LIT48).** (A) Habitus *in situ* at 3–5 m depth, Cala sa Nau. (B) Detail of the oscula. (C) Detail of the pores. (D–M) SEM images of the spicules. (D) Dichotriaene. (E) Oxea. (F–J) Oxyasters. (K–LL) Microrhabds with (L–M) detail of the microrhabd tips. o, oscula; p, pores.

Oxeas (Fig. 29E), slightly bent, tips may be pointed or blunt (*i.e.,* modifications to style and strongyle), 353-913/6-17 µm.

Oxyasters (Figs. 29F–29J), with up to 17–18 microspined actines (Fig. 29FG). Overall measuring 9–30 μm.

Microrhabds (Figs. 29K–29M), very abundant, on a wide but continuous size range, measuring 31-168/2-6 μm, smooth, slightly bent (rarely abruptly bent), with stepped tips (Fig. 29L) that may be mucronated (Fig. 29M).

### Ecology and distribution

This species was discovered in shallow caves, the two exact same localities/habitat as *P. cavernensis* **sp. nov.** (Fig. 3I) (see above).

### Genetics

The Folmer COI (ON130540, ON130541, ON130542) were obtained from holotype LIT48 and paratypes LIT40_1 and LIT66, while a 28S (C1-C2) fragment was obtained from the paratype LIT40_1 (ON133860).

### Taxonomic remarks on *Penares helleri* and *Penares isabellae* sp. nov.

*Penares helleri* is a common Mediterranean species found at both infralittoral caves and the mesophotic zone with one report from the Atlantic side, but close to Gibraltar at 521 m depth, its current deepest record (*Boury-Esnault, Pansini & Uriz, 1994*). Cave specimens are reported to be massive or encrusting while mesophotic ones tend to be reported as massive. Otherwise cave and open-sea specimens share the same spicular set; smooth microrhabds, oxyasters, oxeas and dichotriaenes. However, reported spicular sizes are very heterogeneous (Table 12), with some specimens having longer and/or thicker microrhabds/oxeas. Also, dichotriaenes seems to be very common in some specimens, but very scarce in others. The mentioned differences are usually explained in the literature (*Sarà, 1961*) by differences in habitat conditions, for example, currents may affect the growing morphology, leading to encrusting *versus* massive specimens and depth may influence nutrient availability, affecting the silica content and thus spicule development. Our results reveal that *P. helleri* is a species complex with at least four cryptic but genetically different species, a fact that challenges the use of ecophysiological traits to explain morphological differences.

A first species (POR946, i739, i152 and i233) corresponds to the true *P. helleri,* as macroscopic morphology (massive), spicules (robust and abundant dichotriaenes, thick microxeas and large oxyasters with 4–10 actines) and habitat (mesophotic to upper bathyal) matches the holotype.

A second species was revealed by our molecular markers. The full Folmer COI clearly suggested that our cave specimens (LIT40_1, LIT48 and LIT66) was a different species with a significant 7 bp. difference with the mesophotic *P. helleri*; 28S (C1-C2) shows a 1 bp. difference, knowing that the C1-D1-C2 fragment is quite conserved (we are missing the much more variable D2 part, which is ideal to discriminate species). This whitish encrusting species had thinner microxeas, smaller oxyasters with >10 actines and very scarce dichotriaenes. This species actually resembles *P. helleri* forma *subtilis* Sarà & Sribelli, 1960 found on rocky bottoms in the Gulf of Naples (20–25 m) and in caves. *P. subtilis* is described as whitish encrusting, small, with weak microxeas and with very scarce dichotriaenes, just as in our cave specimens. In fact, in her thesis *Bibiloni (1990)* assigned

cave *P. helleri* from Mallorca to the *subtilis* variety. However, the description of *P. subtilis* is incomplete (the size of oxyasters is notably missing) and it suggests there are two distinct sizes of microrhabds (32–80 µm and 103–213 µm) *vs.* only one in our material (31–165 µm). Furthermore, the type material of *P. helleri* var. *subtilis* (specimens #773:2 and #783:2) could not be revised here, it is unfortunately missing and presumably lost (pers. comm., 'Stazione Zoologica Anton Dohrn', Naples, curator Dr. Andrea Travaglini; 'Museo Civico di Storia Naturale "G. Doria"', Genoa, curator Dr. Maria Tavano; 'Museo zoologico, Naples University', curator Dr. Roberta Improta; 'Museo di zoologia', Bari, Dr. Giovanni Scillitani). Instead of risking to mis-identify this common Mallorcan species, we decided to create *P. isabellae* **sp. nov.**, before someone can sequence several *P. helleri* from the Gaiola area in the Gulf of Naples for comparison.

The fourth species in this complex was originally identified as *P. helleri* in *Chombard, Boury-Esnault & Tillier (1998)* but is here again revealed different thanks to 28S. The full 28S (C1-D2) fragment of a specimen from the 3PP cave (La Ciotat, France, AF062598) has surprising 12 bp. and 13 bp differences with respectively our mesophotic *P. helleri* and *P. isabellae* **sp. nov.** This specimen (PC325) was re-examined here, it is a cave specimen with many dichotriaenes, large oxyasters (13–47 µm) as in *P. helleri* and non-centrotylote shorter microxeas (25–105/2–4 µm), as in *P. isabellae* **sp. nov.** Interestingly, it has common double-bent microxeas, which were never observed in our Balearic specimens. More "*P. helleri*" from the 3PP cave need to be sequenced in order to confirm and understand this potential new species. Meanwhile, this brings additional data to possible cryptic cave faunas. Marine caves are often considered isolated habitats, with cave fauna being poorly connected and having low gene flow, a fact that promotes speciation and high levels of endemism (*Juan et al., 2010*). Patterns of genetic connectivity between cave sponges are understudied, but some works pointed to a high isolation pattern (*Muricy et al., 1996*).

Finally, a fifth species is represented by the specimen ZMAPOR 21658 from another shallow cave, this time on Flores Island, in the Azores. This Flores *P. helleri* is genetically much closer to the 3PP cave specimen than to the Balearic specimens. Indeed, this Flores specimen and the 3PP cave specimen group in a well supported clade, along with a sequence of *Penares sclerobesa* (*Topsent, 1904*), while the Balearic specimen seem to group closer to the *Erylus mamillaris/discophorus/deficiens* complex (Fig. 19). The Flores specimen also has double-bent microrhabds, like those observed in the 3PP cave specimen, which thus may be a good character for future discrimination.

To conclude, the discovery of a polyphyletic *P. helleri* brings new taxonomic issues because if indeed the type of *P. helleri* from the Adriatic Sea is conspecific with our mesophotic specimens, then the phylogenetic relationship *P. helleri/P. euastrum* (based on the position of the 3PP "*P. helleri*" sequence) is no more. This means that the reallocation of *P. euastrum* (and *P. cavernensis* **sp. nov.**) to the genus *Penares* would not be justified. Our current phylogenetic COI and 28S trees now reveal a complex mix of *Erylus* and *Penares* species, with mostly poorly-supported nodes, thus begging for better and additional markers.

Subfamily Geodiinae Gray, 1867
Genus *Geodia* Lamarck, 1815
*Geodia matrix* **sp. nov.** Díaz & Cárdenas
(Figs. 10 and 30; Table 13)

### Etymology

Named *matrix* in analogy with something that harbors other elements, because the species always incorporates all kinds of substrata on its body, and also is a substrate to many other sponge epibionts; also after the 1999 film by the Wachowski sisters.

### Material examined

Holotype: UPSZTY 190881, field#i577_1, St. 21 (INTEMARES0720), MaC (AM), 109 m, beam trawl, coll. J. A. Díaz.

Paratypes: UPSZTY 190876, field#i146_1A, St. 51 (INTEMARES0718), MaC (EB), 128 m beam trawl, coll. F. Ordines; UPSZTY 190879, field#i244_B1, St. 50 (INTEMARES1019), MaC (AM), 102 m, beam trawl, coll. J. A. Díaz; UPSZTY 190880, field#i545, St. 19 (INTEMARES0720), MaC (AM), 111–94 m, rock dredge, coll. J. A. Díaz; UPSZTY 190882, field#i577_2, St. 21 (INTEMARES0720), MaC (AM), 109 m, beam trawl, coll. J. A. Díaz.

### Comparative material

*Geodia canaliculata* Schmidt, 1868, holotype, MNHN-DT750, MNHN Schmidt collection# 61, wet specimen and slide, Algeria, 'Exploration Scientifique de l'Algérie', 1842.

### Outer morphology

Ramose sponge, growing repent and incorporating all kinds of gravels from the substrate. In life, light brown (Fig. 30A), dark brown after ethanol fixation. Choanosome dirty beige after ethanol fixation. The holotype is 7 × 2.5 × .5 cm. Surface smooth to hispid. Hard but breakable consistency. Small uniporal openings barely visible, often at the tips of the lobes, probably oscules, but maybe also pores. Cortex 0.3–0.5 mm thick. The species is usually covered with other sponges, like *Hexadella* sp. (Fig. 30A), haplosclerids, poecilosclerids and an orange encrusting *Timea* sp.

### Spicules

Oxeas (Fig. 30B), slightly curved and fusiform, 566-1,949/7-28 μm.

Plagiotriaenes (Fig. 30C), rhabdome stout and long, straight to slightly bent, with a sharp end. Smaller ones with a marked swelling below the cladome. Clads are usually disposed in a 60–70° angle with the rabdome, they are short and triangular, straight or slightly curved upwards, some aberrant or underdeveloped. Rhabdome: 728-1,623/9-36 μm, cladi: 29-179/7-29 μm.

Sterrasters (Figs. 30D–30F), spherical, smooth rosettes (Fig. 30F), measuring 26–71 μm.

Spheroxyasters (Figs. 30G–30I), smooth, with triangular actines. Young spherasters have few spines (Fig. 30G) while fully developed spheroxyasters have many spines concentrated at the tips of the actines (Figs. 30H–30I), resembling the rosettes of the sterrasters. Measuring 11–33 μm in diameter.

**Table 13 Spicule measurements of *Geodia matrix* sp. nov. and *Geodia canaliculata*, given as minimum-mean-maximum for total length/minimum-mean-maximum for total width; all measurements are expressed in μm.** Balearic specimen codes are the field#. Specimens measured in this study are in bold.

| Material | Depth (m) | Oxeas (length/width) | Anatriaenes Rhabdome (length/width) Clad (length/width) | Plagiotriaenes Rhabdome (length/width) Clad (length/width) | Sterrasters (diameter) | Oxyasters I (length) | Oxyaster II (length) | Spherasters (length) |
|---|---|---|---|---|---|---|---|---|
| ***G. matrix* sp. nov. i244_B1 EB** | 102 | 910-1,377-1,785/ 9-16-24 | – | Rh: 748-1,113-1,615/11-22-32 (*N* = 14) Cl: 41-133-179/8-18-24 (*N* = 17) | 40-49-57 | 36-58-92 2-7 actines | 18-26-32 6-11 actines (*N* = 18) | 12-23-33 |
| ***G. matrix* sp. nov. i146_1A AM** | 128 | 817-1,317-1,705/ 6-15-23 | – | Rh: 728-1,038-1,415/9-22-36 Cl: 29-93-146/7-17-29 | 26-49-63 | 38-55-79 2-6 actines | 14-24-28 6-11 actines (*N* = 7) | 12-18-22 |
| ***G. matrix* sp. nov. Holotype i577_1 AM** | 109 | 867-1,454-1,879/ 12-21-28 (*N* = 17) | – | Rh: 769-1,259-1,606/14-23-30 Cl: 78-121-168/15-21-28 (*N* = 17) | 44-52-60 | 43-65-89 2-6 actines | 17-22-32 6-9 actines (*N* = 19) | 16-20-31 |
| ***G. matrix* sp. nov. i577_2 AM** | 109 | 566-1,405-1,949/ 7-18-28 | – | Rh: 786-1,193-1,623/13-22-31 Cl: 73-97-140/11-18-27 (*N* = 10) | 43-55-71 | 42-64-81 2-7 actines | 17-25-37 7-11 actines (*N* = 14) | 11-20-31 |
| ***G. matrix* sp. nov. i545 EB** | 94-111 | 1,016-1,477-1,713/12-19-23 (*N* = 14) | Rh: 1,350/6 Cl: 15/7 (*N* = 1) | Rh: 761-1,050-1,311/13-18-30 Cl: 71-117-165/11-16-27 (*N* = 9) | 41-53-63 | 50-62-81 2-5 actines (*N* = 18) | 16-23-31 4-9 actines (*N* = 17) | 17-21-26 (*N* = 13) |
| ***G. canaliculata* holotype, Algeria MNHN DT750** | – | n.m. | n.m. | n.m. (malformed cladomes with large axial canals) | 44-50-62 | – | 12-21-30 (*N* = 20) (few) | 20-22-25 (*N* = 5) (very few) |
| *G. canaliculata* La Calle, Algeria (*Topsent, 1901*) | *"Coralligen banks"* | 1,900/ 33 | – | Aberrant cladomes Rhabdome robust, >1,000/30 | 45-60 | – | – | 20-25 |

**Notes.**

Rh, rhabdome; Cl, clad; -, not found/not reported; n.m., not measured; EB, Emile Baudot; AM, Ausias March.

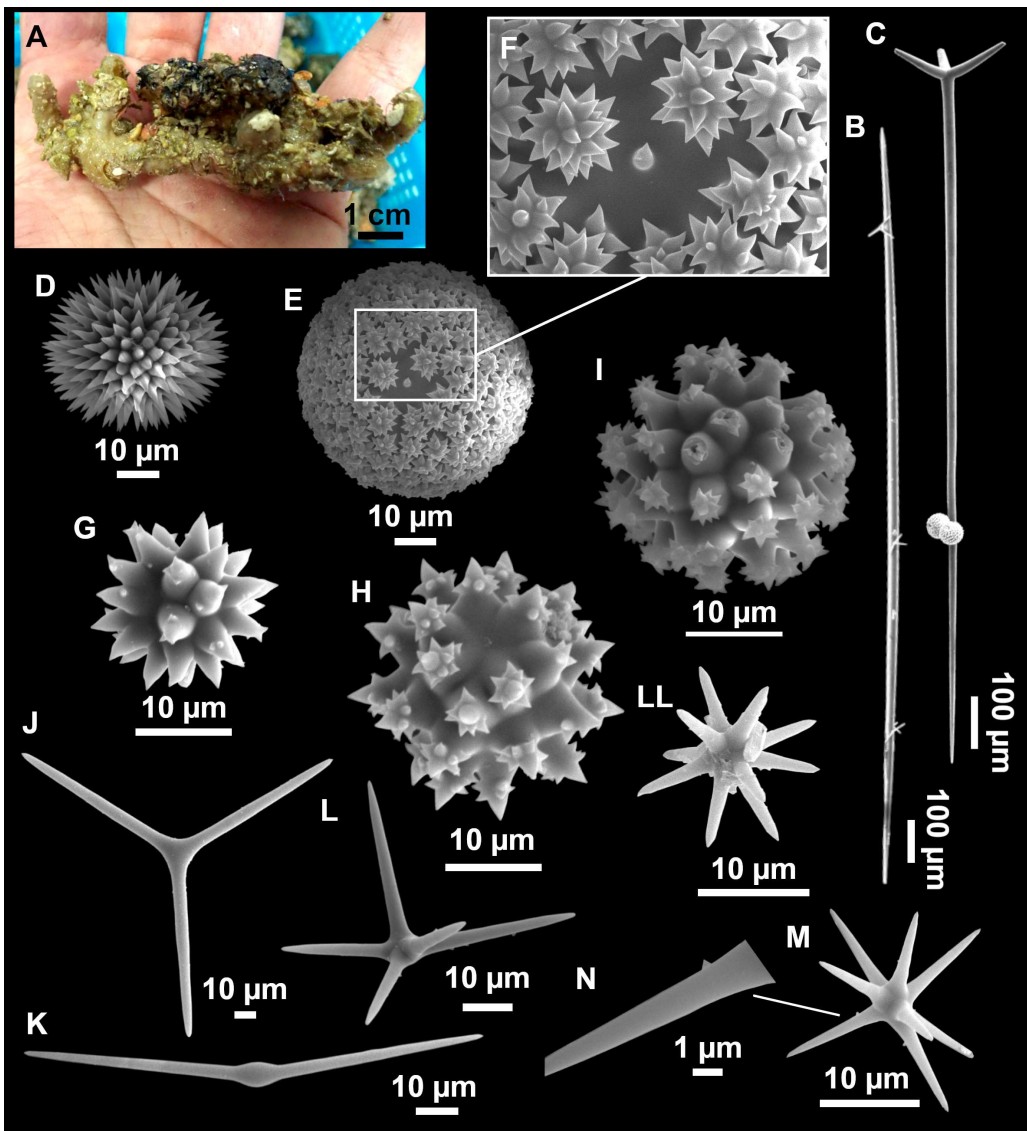

**Figure 30  Holotype of *Geodia matrix* sp. nov., UPSZTY 190881 (i577_1).** (A) Habitus on deck with a *Hexadella* sp. epibiont (dark lilac). (B-N) SEM images of the spicules. (B) Oxea. (C) Plagiotriaene. (D) Immature sterraster. (E) Sterraster with (F) detail of the rosettes. (G–I) Spherasters. (J–L) Oxyasters I. (LL–N) Oxyasters II with (N) detail of the spines.

Oxyasters I (Figs. 30J–30L), 2–7 actines. The less actines they have, the larger they are. Actines are essentially smooth with a few occasional small spines. Measuring 36–92 μm in diameter.

Oxyasters II (Figs. 30LL–30N), uncommon, with 4–11 slightly microspined actines (Fig. 30N), 16–37 μm in diameter.

## Ecology and distribution

Always found associated with rhodolith beds at the summit of the AM and the EB seamounts, where it can be very abundant and significantly contribute to the overall biomass in several kg per 100m2 (*Díaz et al., 2024*). Due to its relatively large size and high abundance, the species may play a role as habitat builder, providing shelter to smaller associated fauna. It is always overgrown by other sponges, mostly *Hexadella* sp., *Timea* sp. and several haplosclerids.

## Genetics

COI (ON130523, ON130524, ON130525) and 28S (C1-C2) (ON133885, ON133886, ON133887) were obtained from i146_1A, i244_B1, i577_1 (holotype) and i577_2.

## Taxonomic remarks

*Geodia matrix* **sp. nov.** appears to have uniporal oscules/pores, which is only found in a few temperate Atlanto-Mediterranean species, previously grouped in the genus *Isops*: *Geodia geodina* (*Schmidt, 1868*), *G. canaliculata*, *Geodia globus Schmidt, 1870* and *Geodia pachydermata Sollas, 1886*. Two more species should be considered for comparison, since their opening morphologies are unknown: *Geodia echinastrella Topsent, 1904* and *Geodia spherastrella Topsent, 1904*.

Geodia geodina has smaller spheroxyasters (10–18 μm), which do not develop spines at their tips (cf. below). *G. globus*, is a poorly described species only reported once from Portugal but a re-description of the type suggests it has strongylasters (9 μm), and no spherasters (*Burton, 1946*). *G. pachydermata* has a very typical external warty appearance, and much larger sterrasters (>200 μm). *G. echinastrella* from the Azores, resembles *G. matrix* **sp. nov.** in having similar size spherical sterrasters (47–50 μm) and similar sized smooth oxyasters (22–26 μm), however, it has smaller spherasters (15–18 μm) and is lacking the large oxyaster category. *G. spherastrella* also from the Azores has larger and ellipsoidal sterrasters (90–110 μm), smaller spherasters (14–16 μm) and only one size of spiny oxyasters (25 μm).

Of all known Atlanto-Mediterranean species, *G. matrix* **sp. nov.** is closest to *G. canaliculata,* a poorly-known species reported twice from the coast of Algeria more than 100 years ago (*Schmidt, 1868*; *Topsent, 1901*). The holotype (Fig. 31) was revised by *Topsent (1938)* but re-examined here and thick sections were made (Figs. 31D–31G): it is a massive sponge with uniporal openings (Topsent also could not distinguish oscules from pores), which explains its placement in the former genus *Isops* by *Topsent (1901)*. It can be added to the detailed description given by *Topsent (1938)* that the holotype is overgrown by an encrusting poecilosclerid with hymedesmoid skeleton (reddish color, Figs. 31D–31E) and the cortex is 0.25–0.3 mm thick (measured on thick sections). On our sections, oxyasters are essentially found just below the cortex (moderate abundance) while spheroxyasters (at different stages of development) are found throughout the choanosome (low abundance, Fig. 31F), with rare presence in the cortex. More surprisingly, slightly flexuous microtoxas were found, often slightly centotylote (160–220 μm long; Fig. 31G); they look like flattened flexuous microtoxas such as the ones from the encrusting sponge

*D. annexa,* a common epibiont in the Atlanto-Mediterranean Sea region. They clearly appeared on the thick sections we made, fairly abundant throughout the choanosome, and with no particular orientation. Such spicules have never been observed before in *Geodia* species so we consider them to be probably foreign, but we also note that there are no other spicule contamination in the choanosome than these flexuous microtoxas (*i.e.,* no other typical *D. annexa* spicules). As a result, despite some shared characters with *G. canaliculata* (irregular lobose/ramose shape with numerous foreign bodies and epibionts, uniporal oscules/pores, sterraster size, characteristic spherasters), *G. matrix* **sp. nov.** has two significant differences: (i) well-developed regular plagiotriaenes (*vs.* aborted cladomes), and (ii) large and small oxyasters (*vs.* only small oxyasters).

Attention should be paid to the fact that *G. matrix* **sp. nov.** tends to be overgrown by *Timea* sp. This species has spheroxyasters of a similar shape and size (6-19-30) as those of *G. matrix* **sp. nov.** and so they mixed in our first spicule preparations. The subtle difference is that spheroxyasters of *G. matrix* **sp. nov.** have a larger centrum and heavily spined actines, especially at their tips. *Timea* sp. has a characteristic orange color in life but it turns brownish after ethanol fixation, so its presence is hard to notice once specimens are fixed. After carefully digesting separately clean cortex and choanosome, we observed that *Geodia* spheroxyasters are much more abundant in the cortex, suggesting that spheroxyasters could be ectocortical or located just below the cortex. However, we could not confirm this as in the thick sections the cortex was always covered by the *Timea* sp. We further note that spheroxyasters are overall much more abundant in *G. matrix* **sp. nov.** than in the holotype of *G. canaliculata*; this characteristic should be confirmed with new specimens of *G. canaliculata*. According to their similar spicules, *G. canaliculata* and *G. matrix* **sp. nov.** are undoubtedly phylogenetically close species. It is also clear that the typical spheroxyasters in both species are homologous. There has been some confusion regarding these spheroxyasters in *G. canaliculata* (*Topsent, 1901*; *Topsent, 1938*), which SEM observations of *G. matrix* **sp. nov.** have helped us to understand. *Schmidt (1868)* and *Topsent (1901)* mention the presence of smaller sterrasters, with less actines than the regular ones, later thought to be a different category of spicule (*Topsent, 1938*). These are actually fully-developed spheroxyasters with spiny tips, as observed in *G. matrix* **sp. nov.** (Fig. 30I) or in other species but with a smaller size (*G. pachydermata, G. spherastrella*). The confusion arises because these spheroxyasters are almost as large as the sterrasters and sometimes mixed with them in the cortex.

COI and 28S tree suggest that *G. matrix* **sp. nov.** is related to species *Geodia parva* Hansen, 1885, *Geodia phlegraei* (Sollas, 1880), *G. geodina*, *Rhabdastrella intermedia* Wiedenmayer, 1989 and *Rhabdastrella* sp.1 South Africa. (Fig. 10). Those species do not belong to the three main *Geodia* clades (*Cydonium*[P], *Depressiogeodia*[P] and *Geodia*[P])(*Cárdenas et al., 2011*), and may represent a fourth group in *Geodia* based on

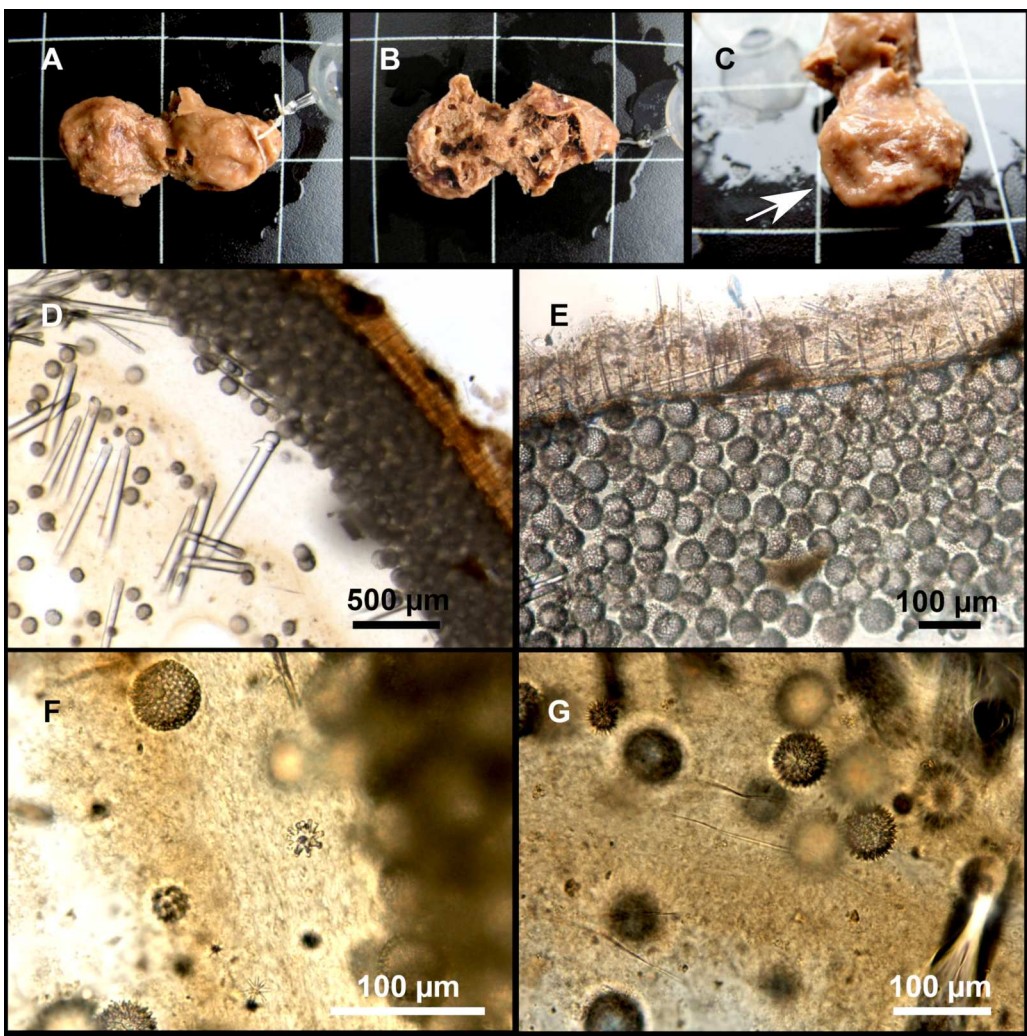

**Figure 31  Holotype of *Geodia canaliculata Schmidt, 1862*, MNHN DT750, Algeria.** (A–C) Habitus, notably showing the uniporal openings in C (arrow). (D–G) Optical microscope images of thick sections. (D) Transversal section of the cortex with an underdeveloped triaene. (E) Detail of the cortex and the poecilosclerid epibiont. (F) Detail of the choanosome with spheroxyasters. (G) Detail of the choanosome with foreign toxas. Scale grid of A–C: 1 cm.

shared smooth oxyasters and presence of spheroxyasters, despite no bootstrap support for the clade at the moment.

*Geodia geodina* (*Schmidt, 1868*)
(Figs. 10, 32, 33 and 34; Table 14)

### Synonym

*Stelletta geodina Schmidt, 1868*
*Cydonium geodina* (*Schmidt, 1868*)
*Sidonops geodina* (*Schmidt, 1868*)
*Synops anceps Vosmaer, 1894* (new synonym)
*Isops anceps* (*Vosmaer, 1894*) (new synonym)
*Geodia anceps* (*Vosmaer, 1894*) (new synonym)

Not *Geodia anceps* in *Sitjà et al. (2019)* from the Gulf of Cadiz: renamed in the present paper as *Geodia phlegraeioides* **sp. nov.**

## Material examined

UPSZMC 190862-190863, field#i140_A and field#i140_B, St. 51 (INTEMARES0718), MaC (EB), 135 m, beam trawl, coll. F. Ordines; UPSZMC 190868, field#i391_3, St. 158 (INTEMARES1019), MaC (EB), 146 m, beam trawl, coll. J. A. Díaz; UPSZMC 190870-190871, field#i575 and field#i576, St. 21 (INTEMARES0720), MaC (AM), 109 m, beam trawl, coll. J. A. Díaz; UPSZMC 190872, field#i708, St. 45 (INTEMARES0720), MaC (EB), 150 m, beam trawl, coll. J. A. Díaz.

## Comparative material

*Geodia geodina,* lectotype (designated here), MNHN Schmidt collection#91 (large specimen), paralectotype (designated here), MNHN Schmidt collection#93 (small specimen), both specimens in the same jar registered under MNHN DT752, wet specimens, Algeria, 'Exploration Scientifique de l'Algérie', 1842; MNHN DCL728, spicule slide, East Gettysburg, Gorringe Bank, St. 149, trawl, 95 m (*Lévi & Vacelet, 1958*); MNHN (unregistered), field#JC46, Jean Charcot Madeira 1966, SW of Deserta Islands, 32°21′30″N, 16°30′18″W, 100–130 m, wet specimen, 18 July 1966, id: P. Cárdenas.

*Geodia anceps*, syntype, NHM 1955.3.24.1 (=RMNH POR0655), wet specimen, label from the 'Rijksmuseum-Leiden' saying 'coll. no. 655, fragment of type sp.', Vosmaer personal number N557, between Capri and Naples, 150–200 m, 12 Feb. 1891.

## Outer morphology

Massive, globular (Fig. 32A), to ramose or lobated. Larger specimens up to 12 cm in maximum diameter. Grayish ocher in life, dark brown after ethanol fixation. Always paler on the lower side of the body (protected from the light). Surface smooth to hispid, smooth to the touch. Hard, only slightly compressible consistency. Cortex less than 0.5 mm thick, clearly distinguishable. Choanosome fleshy, whittish. Uniporal oscules are grouped on the top surface of specimens or at the top of lobes, always contracted on deck and after ethanol fixation. Uniporal pores gathered in depressed areas, visible to the naked eye in some specimens.

**Table 14 Spicule measurements of *Geodia geodina*, *Geodia phlegraeioides* sp. nov. and related species *Geodia phlegraei* and *Geodia parva*, given as minimum-mean-maximum for total length/minimum-mean-maximum for total width; all measurements are expressed in μm.** Balearic specimen codes are the field#. Specimens measured in this study are in bold.

| Material | Depth (m) | Cortex thickness (mm) | Oxeas (length/width) | Anatriaenes Rhabdome (length/width) Clad (length/width) | Orthotriaenes Rhabdome (length/width) OPD ortho/ proto/ deuteroclads Clad (length/width) | Sterrasters (diameter) | Oxyasters I (diameter) | Oxyasters II (diameter) | Spherasters (diameter) |
|---|---|---|---|---|---|---|---|---|---|
| *G. parva* holotype ZMBN 100 (*Cárdenas & Rapp, 2013*) | – | – | 773-1,194-1,625/ 14-21-34 (*N* = 6) | – | Rh: 360-697-1,000/ 20-26-33 (*N* = 8) OPD: 102-161-232 (*N* = 7)/56/44 | 75-85-93 | 19-42-64 | | 13-16-21 |
| *G. phlegraei* holotype NHM 1910.1.1.840 (*Cárdenas & Rapp, 2013*) | 330 | 0.64 | 1,825-3,293-448/ 20-41-60 | Rh: >3,760/8-19-25 (*N* = 6) Cl: 48-72-130 (*N* = 6) | Rh: 586-2,129-3,640(*N* = 10)/ 12-50-72 OPD: 80-416-660/ 220-250(*N* = 2)/ 100-250 (*N* = 2) | 82-93-102/76-86-95 | 17-24-41 | | 12-17-24 |
| **G. anceps syntype NHM 1955.3.24.1 Capri-Naples** | 150-200 | – | 900-1,907-2,580/ 8-28-42 | Rh: 1,689 (*N* = 1)/ 7-8-9 (*N* = 3) Cl: 20-30-37 | Rh: 830-1,426-1,920/18-39-62 OPD: 130-262-390 160 (*N* = 1)/130 (*N* = 1) | 50-61-70 | 30-51-75 (3-8 actines) | 12-17-25 (>8 actines) | 10-13-15 (*N* = 10) (sometime spiny at tips) |
| **G. geodina lectotype MNHN DT752 Algeria** | – | 0.15-0.2 | 1,536/20 (*N* = 1) | | Rh: >1,050/8-17-25 (*N* = 12) OPD: 88-199-320/-/- | 38-42-45 | 35-53-65 (3-8 actines) (very abundant) | 10-17-27 | 10-15-18 |
| **G. geodina paralectotype MNHN DT752 Algeria** | – | 0.15-0.2 | broken | – | Rh: broken/10-17-25 OPD: 100-197-326/-/- | 35-40-45 | 20-40-62 (>3 actines) (very abundant) | 8-14-18 | 8-14-20 |
| *G. geodina* MNHN syntypes (*Topsent, 1938*) | – | 0.5 | 2,200-2,400/ 20-25 | – | Rh: 1,250/ 16-18 OPD: 200-300/-/- | 40-45 | 30 (actine) (4-8 actines) | 15-17 | – |
| **G. geodina i140_b EB** | 128 | – | 1,494-1,856 –2,534/6-19-31 (*N* = 12) | Rh: 1,421-1,845-2,020/10-14-17 (*N* = 10) Cl: 21-28-37/8-10-13 (*N* = 10) | Rh: 1,316/27 (*N* = 1) Cl: 206-294-343/ 20-27-33 (*N* = 12) | 41-51-59 | 38-61-86 (2-8 actines) | 11-21-27 (8-12 actines) | 10-14-18 |
| **G. geodina i708 AM** | 130 | – | 748-1,918-2,614/8-25-36 | Rh: 2,006-2,687 (*N* = 2)/9-10 (*N* = 3) Cl: 3-37/8-10 (*N* = 3) | Rh: 797-1,650-1,947/ 16-39-54 (*N* = 11) Cl: 112-279-372/ 13-35-50 (*N* = 10) | 46-54-64 | 41-53-77 (3-8 actines) | 14-25-38 (6-15 actines, largest one has 6 actines) | 11-14-18 (uncommon) |
| **G. geodina i575 AM** | 110 | – | 1,208-1,668-2,176/14-24-35 | Rh: 1,180 (*N* = 1)/5-17 (*N* = 2) Cl: 22-97/5-16 (*N* = 2) | Rh: 419-1,085-1,395/ 11-24-32 (*N* = 9) Cl: 71-284-470/ 11-26-36 (*N* = 31) | 46-57-66 | 40-54-74 (3-6 actines) | 14-23-30 (8-19 actines) | 12-14-15 (uncommon, *N* = 11) |
| **G. geodina i576 AM** | 110 | – | 1,357-1,742-2,145/11-23-36 (*N* = 20) | Rh: 1,597 (*N* = 1)/8-14 (*N* = 10) Cl: 20-36/9-14 (*N* = 10) | Rh: 523-1,092/ 13-22 (*N* = 2) Cl: 99-246-340/ 10-23-30 (*N* = 17) | 47-52-64 | 38-51-61 (4-9 actines, never 2) | 15-25-36 (~7-14 actines) | 13-14-17 (*N* = 11) |
| **G. geodina i140_a EB** | 128 | – | 1,185-1,740 –2,500/16-24-38 | Rh: 1,428/19 (*N* = 1) Cl: 39/16 (*N* = 1) | Rh: all broken Cl: 175- 270-330/ 20-28-32 (*N* = 10) | 44-56-68 | 36-61-88 (6-10 actines) | 16-22-35 (*N* = 19) (6-13 actines) | 8-14-22 |

| Material | Depth (m) | Cortex thickness (mm) | Oxeas (length/width) | Anatriaenes Rhabdome (length/width) Clad (length/width) | Orthotriaenes Rhabdome (length/width) OPD ortho/ proto/ deuteroclads Clad (length/width) | Sterrasters (diameter) | Oxyasters I (diameter) | Oxyasters II (diameter) | Spherasters (diameter) |
|---|---|---|---|---|---|---|---|---|---|
| *G. anceps* Alboran Sea (*Maldonado, 1992*) | 70-120 | 0.5-0.7 | 1,085-3,000/15-30 600-1,812/ 2-4 (flexuous; in 1 specimen) | Rh: 200-961/ 4-10 Cl: 15-25 | Rh: 433-1,700/ 15-26 OPD: 80-273/ 140-160/113-170 | 44-68 | 30-46 (actine length) (2-5 actines) | 17-27 (>6 actines) | 12-25 |
| **G. geodina MNHN DCL728 Gorringe Bank**, (*Lévi & Vacelet, 1958*) | 95 | 0.5 | 640-1,468-2,150/10-18-25 | Rh: 1,254-1,636-1,971 (*N* = 5)/4-7-12 Cl: 13-29-50 | Rh: 1,020-1,561 (*N* = 2)/ 13-18-22 (*N* = 10) OPD: 133-258-367 | 42-48-50 | 43-54-67 (3-5 actines) (very abundant) | 12-20-37 | 10-13-16 |
| *G. phlegraeioides* holotype, MNCN-P224-11 Gulf of Cadiz (*Sitja et al., 2019*) **(this study)*** | 895 | 0.5 | 2,122-3,406/ 16-42 | Rh: Up to 1,500/ 6-8 Cl: - | Rh: 375-2,770/ 14-70 OPD: 97-580/ 122-378/ 121-338 | 77-91 **75-81-87*** | 31-50 (2-5 actines) (rare) -* | 18-30 (6-8 actines) **12-19-30*** | 13-29 (sometime spiny) |
| **G. phlegraeioides paratype, UPSZTY 190887 (DR15-972) Le Danois Bank** | 650 | 0.5 | broken | – | Rh: broken/ 28-61 (*N* = 2) OPD: 93-255-459 (*N* = 5)/ 86-180-247 (*N* = 11)/ 194-269-351 (*N* = 10) | 87-94-99 | | 19-36-57 | 14-21-32 |
| **G. phlegraeioides DR15-869c Le Danois Bank** | 650 | – | broken | – | Rh: - OPD: 281/ 140/249-254 | 41-78-92 | | 12-24-61 | 9-14-17 |
| **G. phlegraeioides DR15-862c Le Danois Bank** | 650 | – | 145-975-2,995 (*N* = 12)/ 15-20 (*N* = 2) | – | Rh: 1,012-1,134 (*N* = 2)/ 37-45 (*N* = 2) OPD: 131-273-508 (*N* = 4)/ 28-131-307 (*N* = 5)/53-217-580 (*N* = 5) | 76-83-91 | 42 (*N* = 1) | 16-19-25 (*N* = 20) | 10-16-20 |
| **G. phlegraeioides DR15-882 Le Danois Bank** | 650 | – | 136-1,161-2,892 (*N* = 22)/- | – | Rh: 955-1,130-1,324 (*N* = 3)/ 37 (*N* = 1) OPD: 90-307-575 (*N* = 9)/ 92-172-220 (*N* = 15)/ 93-201-315 (*N* = 15) | 58-83-96 | | 14-36-94 | 12-18-25 |
| **G. phlegraeioides COLETA#5803 Banc Princesse Alice, Azores** | 432 | – | >3,328/ up to 45 | Rh: >3,840/ 10-19-25 (*N* = 7) one ana(meso)triaene Cl: 35-58-87 (*N* = 7) | Rh: 1,740-2,333-2,670 (*N* = 16)/ 35-42-50 OPD: 327-336-449/-/- | 60-69-87 | | 10-15-25 | 8-16-25 (large centrum) |

**Table 14** (*continued*)

| Material | Depth (m) | Cortex thickness (mm) | Oxeas (length/width) | Anatriaenes Rhabdome (length/width) Clad (length/width) | Orthotriaenes Rhabdome (length/width) OPD ortho/ proto/ deuteroclads Clad (length/width) | Sterrasters (diameter) | Oxyasters I (diameter) | Oxyasters II (diameter) | Spherasters (diameter) |
|---|---|---|---|---|---|---|---|---|---|
| *G. phlegraeioides* **COLETA#6243 Banco Voador, Azores** | 418 | 0.5-0.6 | broken | – | Rh: broken/ 20-46-50 (*N* = 11) OPD: 225 (*N* = 1)/ 184-249-357 (*N* = 12)/ 153-252-357 (*N* = 15) | 67-78-85 | | 13-22-40 | 12-15-18 (large centrum) |
| *G. phlegraeioides* Italy (*Longo, Mastrototaro & Corriero, 2005*) | 738-809 | 0.65 | 1,200-1,592-2,000/20-26-40 | Rh: 1,100-1,900-2,500/ 4-8-10 Cl: -/45-70 | Rh: 540-1,120/- Cl: 90-174-260/- n.m./n.m. | 88-99-106 | 40-56-68 (4-6 actins) | 34-48-66 (many actins) | 10-14-16 |
| *G. echinastrella* holotype, Azores (*Topsent, 1904*) | 318 | "thick" | n.m. | – | orthotriaenes | 47-50 | | 22-26 | 15-18 |

**Notes.**

Rh, rhabdome; OPD, ortho/proto/deuteroclads; -, not found/not reported; n.m., not measured; EB, Emile Baudot; AM, Ausias March.

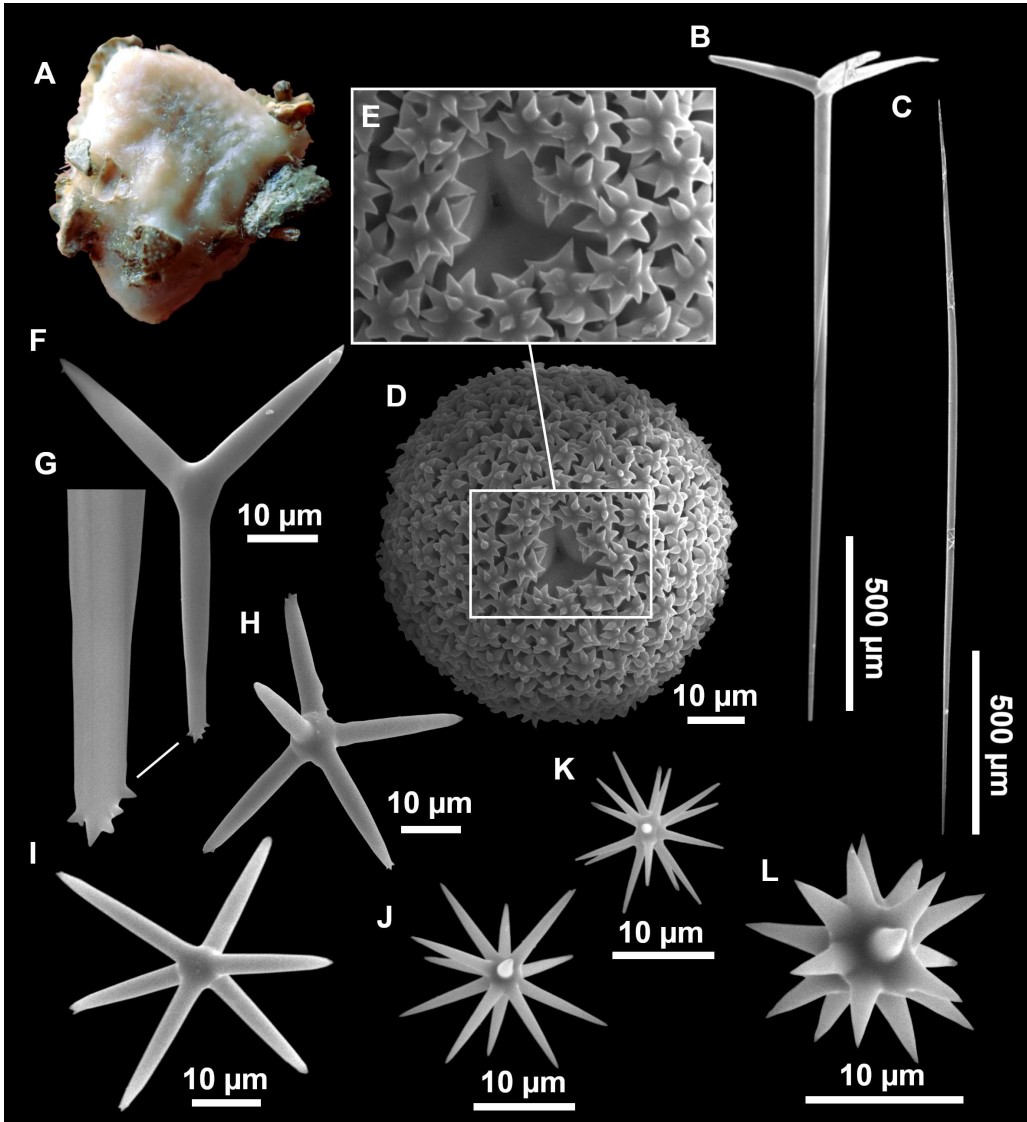

**Figure 32** *Geodia geodina* (*Schmidt, 1868*)**, specimen i708.** (A) Habitus after EtOH fixation. (B) Orthotriaene. (C) Oxea. (D) Sterraster with (E) detail of the rosettes. (F–I) Oxyasters I with (G) detail of the spines. (J–K) Oxyasters II. (L) Spheraster.

## Spicules

Orthotriaenes (Fig. 32B), rhabdome straight and fusiform; the smaller ones have a triangular swelling at the joint with the cladome. Clads are disposed in a 100° angle with the rhabdome, some being slightly tortuous or having its tips curved inwards. A single cladome modification in the form of dichotriaene was observed. Rhabdome measuring 419-1947/11-54 μm, cladi measuring 71-470/11-50 μm.

Oxeas (Fig. 32C), slightly curved and fusiform, measuring 748-2,614/6-38 μm.

Anatriaenes, uncommon, with straight, fusiform rhabdome, 1,180-2,687/5-19 μm. Clads with short cladi, evenly curved, measuring 3-97/5-16 μm.

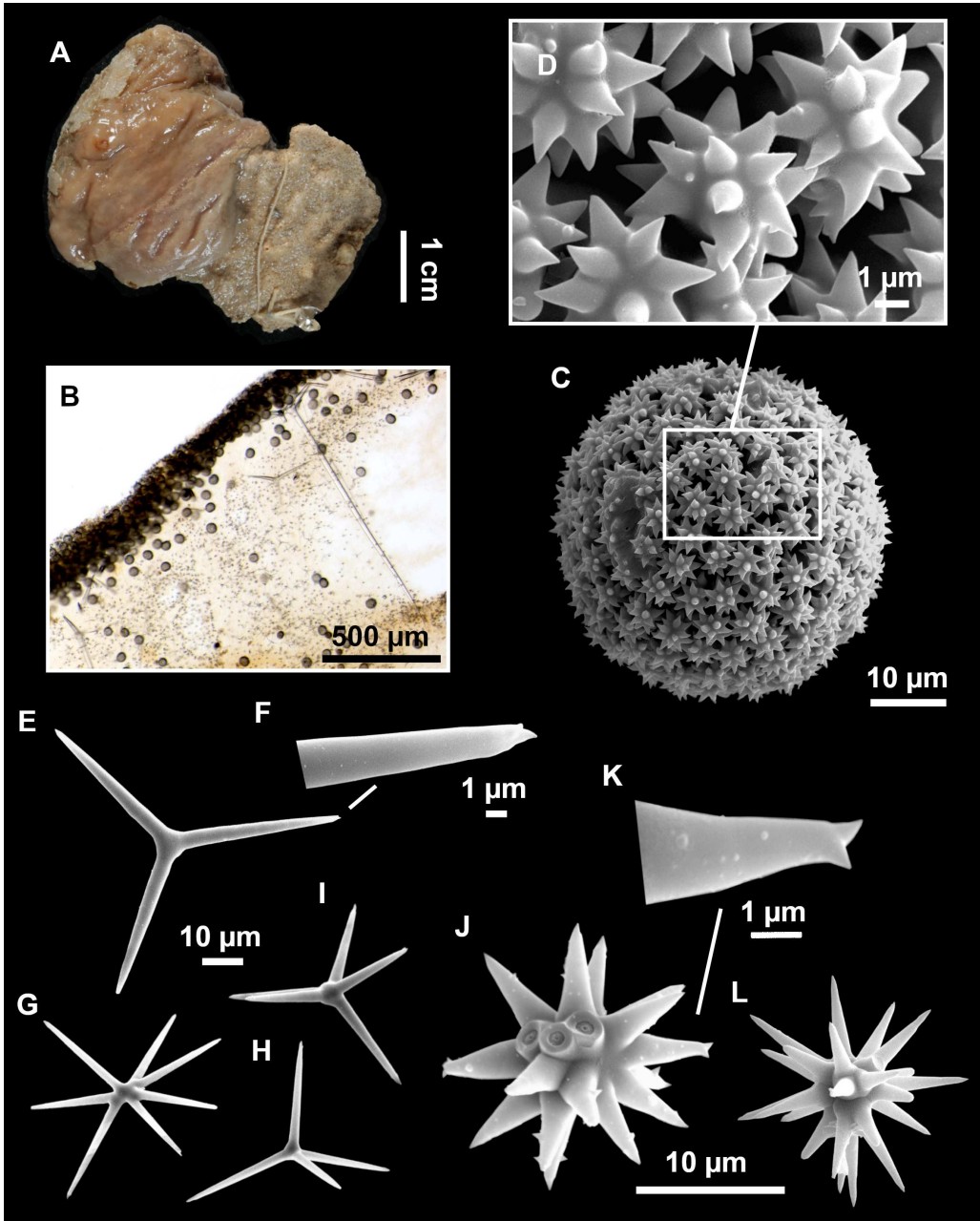

**Figure 33 Lectotype of *Geodia geodina* (*Schmidt, 1868*), MNHN Schmidt collection#91 Algeria.** (A) Habitus. (B) Optical microscope image of a thick transversal section. (C) Sterraster with (D) detail of the smooth rosettes. (E–I) Oxyasters I with (F) detail of the spines. (J) Spheraster with (K) detail of the spines. (L) Oxyaster II.

Sterrasters (Fig. 32D), spherical, with smooth rosettes having 4–12 conical rays (Fig. 32E), measuring 41–68 μm.

Oxyasters I (Figs. 32F–32I), large, smooth actines, usually with a few spines at its tips (Fig. 32G), 36–88 μm (2–10 actines).

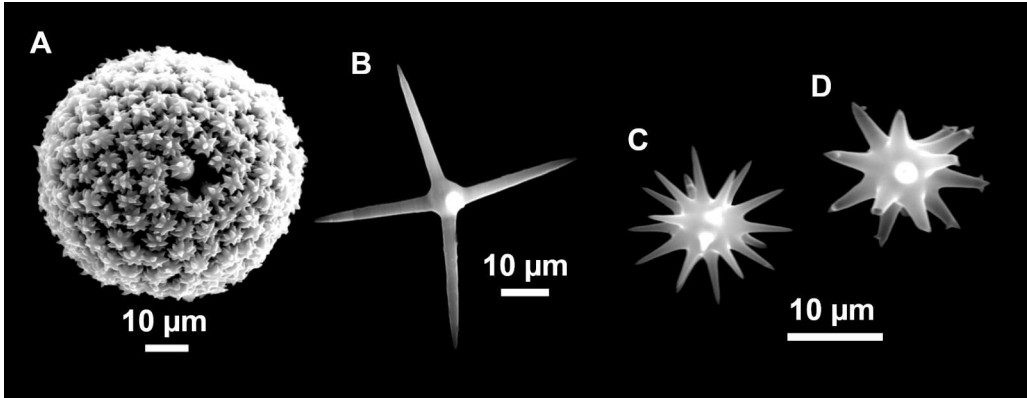

**Figure 34 Syntype of *Geodia anceps* (*Vosmaer, 1894*), NHM 1955.3.24.1, between Capri and Naples, Italy.** (A) Sterraster. (B) Oxyaster I. (C) Oxyaster II. (D) Spherasters.

Oxyasters II (Figs. 32J–32K), smaller than oxyasters I, smooth actines. Measuring 14–38 µm (∼6–19 actines).

Spherasters (Fig. 32L), with triangular actines that can be smooth or microspined at its tips, measuring 8–22 µm.

## Ecology and distribution

Circalittoral species found at the summit of the AM and the EB, although reaching greater depths at the EB. It can be very abundant in some stations, suggesting that it is a habitat-forming species due to its large size and sometimes intricate body shape. It is used as a substrate by epibionts, especially other sponges, like *Hexadella* sp., *Timea* sp., as well as several haplosclerids like *Haliclona poecillastroides* (*Vacelet, 1969*).

## Genetics

COI (ON130519, ON130520, ON130521, ON130522) has been obtained for i575, i708, i576 and i140_A, while 28S (C1-C2) (ON133879, ON133880) were obtained from i575 and i708.

## Taxonomic remarks

See taxonomic remarks on *G. geodina* and *G. phlegraeioides* **sp. nov.** below.

*Geodia phlegraeioides* **sp. nov.** Díaz & Cárdenas
(Table 14)

## Etymology

Named '*phlegraeioides*' to highlight its phylogenetic and morphological closeness with *Geodia phlegraei* (Sollas, 1880) from boreal waters.

## Type material

Holotype, MNCN/1.01/1026 (wet specimen), UPSZTY 190886 (thick sections and spicule slide), Almazán mud volcano, Gulf of Cadiz, 36°3′17.39″N, 7°19′43.20″W–36°3′36.6″N, 7°19′13.2″W (INDEMARES-CHICA), beam trawl, 894–896 m, 4 March 2011, coll: C. Farias, originally identified as *G. anceps* in *Sitjà et al. (2019)*.

Paratype, UPSZTY 190887, field#DR15-972, Le Danois Bank, Cantabrian Sea, station DR-15, 650 m, 44°6′20.64″N, 5°9′16.2″W (SponGES0617), rock dredge, 23 June 2017, coll: P. Rios.

## Other non-type material examined

*Geodia phlegraeioides* **sp. nov.**, CPORCANT, DR10-490 and -500, Le Danois Bank, Cantabrian Sea, station DR-10, 44°6′4.8″N, 4°38′18″W, 541 m, 17 June 2017, rock dredge, Expedition SponGES0617, coll: P. Rios; CPORCANT DR15-869c, -862c, -882, same station as paratype; COLETA#5803 ( =PC566, spicule slide), Banc Princesse Alice, Azores, 37°49′19.2″N, 20°27′43.2″W, 432 m, 28 Feb. 2011, subglobular specimen, bycatch from long line demersal fishery TB/137/MBO/2011; COLETA#6243 (=PC567, spicule slide), Banco Voador, Azores, 37°28′58.8″N, 30°50′34.8″W, 418 m, 24 June 2010, fragment of a specimen, Coral Fish D33-V10, Palangre de fundo.

## Comparative material

*Geodia phlegraei*, holotype, NHM 1910.1.1.840, Korsfjord, SW Bergen, Norway, 1878, 60°9′60″N, 5°10′0″E, 330 m, coll: Rev. A. M. Norman.

*Geodia parva*, holotype, ZMBN 100, spicule slide, unknown station, Norwegian North Sea Exp. 1876–78.

## Outer morphology and skeleton

Massive, subspherical. External color whitish to light brown, alive and in ethanol; choanosome slightly more tanned. Surface is smooth to hispid. Uniporal oscules (up to ~1 mm in diameter) and minute uniporal pores (~0.2 mm in diameter). Cortex is ~0.5 mm thick. Typical geodiid skeleton with ectocortical spheroxyasters and choanosomal oxyasters. The holotype is the largest specimen we have seen so far, 6.5 × 5 × 2.5 cm; for a detailed description of the holotype including its spicules, see *Sitjà et al. (2019)*.

## Spicules (Table 14)

Ortho- and dichotriaenes, robust, straight rhabdome 375-2770/11-70 μm, cladi slightly forward oriented, clads of orthotriaenes: 90–580 μm, protoclads: 28-378, deuteroclads: 53–580 μm.

Oxeas, curved and fusiform, 136-3,406/16-45 μm.

Sterrasters, spherical, with smooth rosettes, 41–99 μm.

Oxyasters, smooth, 11–93 μm.

Spheroxyasters, smooth with a few microspines at tips essentially, 9–32 μm. Specimens from the Azores tend to have a larger centrum.

### Ecological notes

Some specimens were found growing on other sponges: DR15-869c was found on *C. (C.) geodioides*, the holotype was growing on a *Pachastrella* sp. The holotype was budding (*Sitjà et al., 2019*).

### Genetics

COI were obtained from the holotype (OR045844), COLETA#5803 (OR045842), COLETA#6243 (OR045843) and DR15-869c (OR045845).

### Taxonomic remarks on *G. geodina* and *G. phlegraeioides* sp. nov.

Our material from the EB and AM seamounts matches with *G. geodina*, a species described from Algeria (*Schmidt, 1868*) and subsequently reported in the Gulf of Naples (*Pulitzer-Finali, 1972*) and the Gorringe Bank (*Lévi & Vacelet, 1958*). We have compared our material with the type material from Schmidt (lectotype (#91) and paralectotype (#93), here designated), for which new thick sections and SEM was done (Fig. 33), and the spicules re-measured (Table 14). The only differences were in the smaller sizes of sterrasters (35–45 μm *vs* 41–68 μm) and of the orthotriaenes in the type material (Table 14). In this process, a close morphological similarity with *G. anceps,* a better-known Mediterranean *Geodia,* was also noticed. A syntype of *G. anceps* (NHM 1955.3.24.1) was examined, with new spicule and SEM preparations (Fig. 34). The type materials of *G. geodina* and *G. anceps* shared the same external morphology, spicule set and morphologies, with similar size ranges (Table 14). Again, the only noticeable difference was the smaller size of the sterrasters (35–45 μm *vs.* 50–70 μm) and of the orthotriaenes in the types of *G. geodina*, which may be a result of different depths or habitats, as it is known that sterraster and triaene size can be influenced by these parameters (*Cárdenas & Rapp, 2013*). All previous records of *G. anceps,* from the Gulf of Naples (*Vosmaer, 1894*; *Pulitzer-Finali, 1972*) or the Alboran Sea (*Maldonado, 1992*), as well as our material, come from mesophotic depths (70–200 m). Unfortunately, we have no locality/depth data for the types of *G. geodina* so it is impossible to test this hypothesis at the moment. Also, anatriaenes with small cladomes were occasionally found in the syntype of *G. anceps* (this study), in previous reports (*Vosmaer, 1894*; *Pulitzer-Finali, 1972*; *Maldonado, 1992*), in a NHM Schmidt type slide of *G. geodina* (*Burton, 1946*) as well as in our material from the Balearic Islands; however, no anatriaenes were found in our preparations of the *G. geodina* types. Not finding anatriaenes is not surprising since they can be rare and are often localized to certain parts of the sponge in some *Geodia,* so they can be easily overlooked. To conclude, we propose that *G. anceps* becomes a junior synonym of *G. geodina.*

One single *G. anceps* report is from the Atlantic: from the Gulf of Cadiz, 895 m depth (*Sitjà et al., 2019*). It was stated at the time that sterrasters were larger than in Mediterranean specimens (*Sitjà et al., 2019*). Besides that, we also noted other unusual features in this specimen, such as a mix of orthotriaenes and dichotriaenes, and very rare oxyasters I. Examination of additional Northeast Atlantic specimens originally identified as *G. anceps* from Le Danois Bank (Cantabrian Sea) at 650 m and two specimens collected South of the Azores (418–432 m) revealed these exact same characters: (i) larger sterrasters with an

average size of 68–94 µm (*vs.* 40–61 µm in *G. geodina*), (ii) common dichotriaenes mixed with orthotriaenes, (iii) usually only one category of oxyasters with a continuum of sizes from 11 to 93 (*vs.* two separate sizes in *G. geodina*) and (iv) slightly larger spheroxyasters with an average size of 14–21 µm (*vs.* 13–15 µm in *G. geodina*). Although *Sitjà et al. (2019)* report two oxyaster categories, only one category was found during our re-examination of the specimen. Likewise, only one large oxyaster was found in specimen DR15-862c from Le Danois Bank, suggesting that a second larger category of oxyasters may occasionally be produced in *Geodia phlegraeioides* **sp. nov.** but they are very rare or integrated in a continuum with the first category. On the other hand, large oxyasters are always very common in *G. geodina,* and in a clear separate category. These morphological differences were supported genetically: COI Folmer sequences of these Atlantic specimens had a highly significant 18 bp difference with *G. anceps.* We propose the name *Geodia phlegraeioides* **sp. nov.** for this North Atlantic species resembling *G. anceps* and found deeper, down to the upper bathyal zone (418–895 m). The specimen described by *Sitjà et al. (2019)* is designated as the holotype, while one specimen from Le Danois Bank is designated as a paratype.

One deep report of *G. anceps* at 738–809 m, from the deep-sea coral reef off Cape Santa Maria di Leuca, Italy (*Longo, Mastrototaro & Corriero, 2005*) is possibly *G. phlegraeioides* **sp. nov.**: it has larger sterrasters (88–106 µm) and the common dichotriaenes. However, two categories of oxyasters are reported with similar size ranges, which is atypical and needs to be revised. If this was indeed *G. phlegraeioides* **sp. nov.**, it would suggest that this North Atlantic species can also be found in the upper bathyal Mediterranean Sea, below the *G. geodina* zone in the mesophotic area. Likewise, *G. geodina* would also be found in the mesophotic zone in the Atlantic, above the deeper *G. phlegraeioides* **sp. nov.** In fact, a *G. geodina* spicule slide (MNHN-DCL 728) from the Gorringe Bank in the Northeast Atlantic, 95 m depth (*Lévi & Vacelet, 1958*) was re-measured and it fits clearly better the description of *G. geodina* than that of *G. phlegraeioides* **sp. nov.** (Table 14). This was independently confirmed by the examination of a MNHN large *Geodia* specimen (~12 cm long) from Deserta Islands (close to Madeira) from 100–130 m, that we also identified as *G. geodina*.

With respect to our phylogenetic trees (Fig. 10), both *G. geodina* and *G. phlegraeioides* **sp. nov.** group with *G. matrix* **sp. nov.**, former species of *Isops* (*G. phlegraei, G. parva*), which all share as previously said smooth oxyasters, spheroxyasters, and external morphology. In the COI tree, *G. phlegraeioides* **sp. nov.** was the sister species to the clade *G. phlegraei* +*G. parva*; there were respectively 5 and 6 bp differences between them. *G. phlegraei* is a boreal species found from 40 to 3000 m depth, while *G. parva* is its arctic counterpart (*Cárdenas & Rapp, 2013*), so *G. phlegraeioides* **sp. nov.** is the temperate version of *G. phlegraei,* thus explaining our choice for the species name. *G. phlegraeioides* **sp. nov.** seems to have a more irregular shape than *G. phlegraei* (which is subglobular when young and then bowl shaped when larger), with fewer oscules, less visible pores, it can have lots of dichotriaenes (*G. phlegraei* only has orthotriaenes), its sterrasters are round (*vs.* usually oval in *G. phlegraei*). *G. geodina* is also quite close to *G. phlegraei* but again, the shape is more irregular with smaller, less abundant oscules and invisible pores. *G. geodina* has smaller

spherical sterrasters (*vs.* oval in *G. phlegraei*) with two sizes of oxyasters (*vs.* one size in *G. phlegraei*), and the large ones have fewer actines than in *G. phlegraei*.

*Geodia bibilonae* **sp. nov.** Díaz & Cárdenas
(Figs. 10 and 35; Table 15)

### Etymology
Named after Dr. Maria Antònia Bibiloni, who initiated the studies of sponges from the Balearic Islands, from 1982 to 1993.

### Material examined
Holotype: UPSZTY 190857, field#i715_1, St. 45 (INTEMARES0720), MaC (EB), beam trawl, 150 m.

Paratypes: UPSZTY 190860-190861, field#i674-field#i675, St. 42 (INTEMARES0720), MaC (EB), 143–139 m, beam trawl, coll. J. A. Díaz; UPSZTY 190858-190859, field#i780-field#i781, St. 54 (INTEMARES0720), MaC (EB), 124–210 m, rock dredge, coll. J. A. Díaz.

### Outer morphology
Found in two different morphologies: globose when living in sedimentary bottoms, with a lot of gravel incorporated on its body (i715_1; Fig. 35A), encrusting, slightly hemispherical, and free of foreign materia when growing on rocks (i674, i675, i780, i781; Fig. 35B). Relatively small, ∼2 cm in diameter. Whitish beige on deck and after ethanol fixation. Choanosome fleshy, pale beige after ethanol fixation. Surface hispid. Hard consistency, incompressible. Small cribriporal openings (about 0.2 mm) present all over the body, visible to the naked eye on deck. After preservation, they are also visible but not so patent. Cortex more than 0.5 mm thick. Typical geodiid skeleton with ectocortical spherasters, endocortical sterrasters and choanosomal oxyasters.

### Spicules
Oxeas (Fig. 35C), thin and long, slightly curved, 1,058-2,765/12-32 µm. A second category of smaller oxeas were found in small numbers ($N = 5$) and only in i675 (Table 15). Those spicules are probably contamination, as its shape was similar to the isoactinal oxeas (oxeas II) of *Craniella* cf. *cranium*.

Orthotriaenes (Fig. 35D), with long, fusiform rhabd and slightly curved clads. The clads may end tipping downward. Sometimes a clad may be bifurcated. Juvenile stages show a bulbous swelling at the uppermost part of the rhabd. Rhabdome length: 594-2,224/14-59 µm, cladi 78-306/12-49 µm.

Protriaenes (not shown), only found in specimen i675. With long, straight or slightly curved rhabdome and pointed clads. Rhabdome: 1,472-2,389-3,446/9-11-14 µm ($N = 10$), cladome: 69-102-157/8-11-13 µm ($N = 16$).

Anatriaenes (not shown), scarce, rhabdome: 2,567 ($N = 1$)/2-10 µm, cladome: 16-86/2-14 µm.

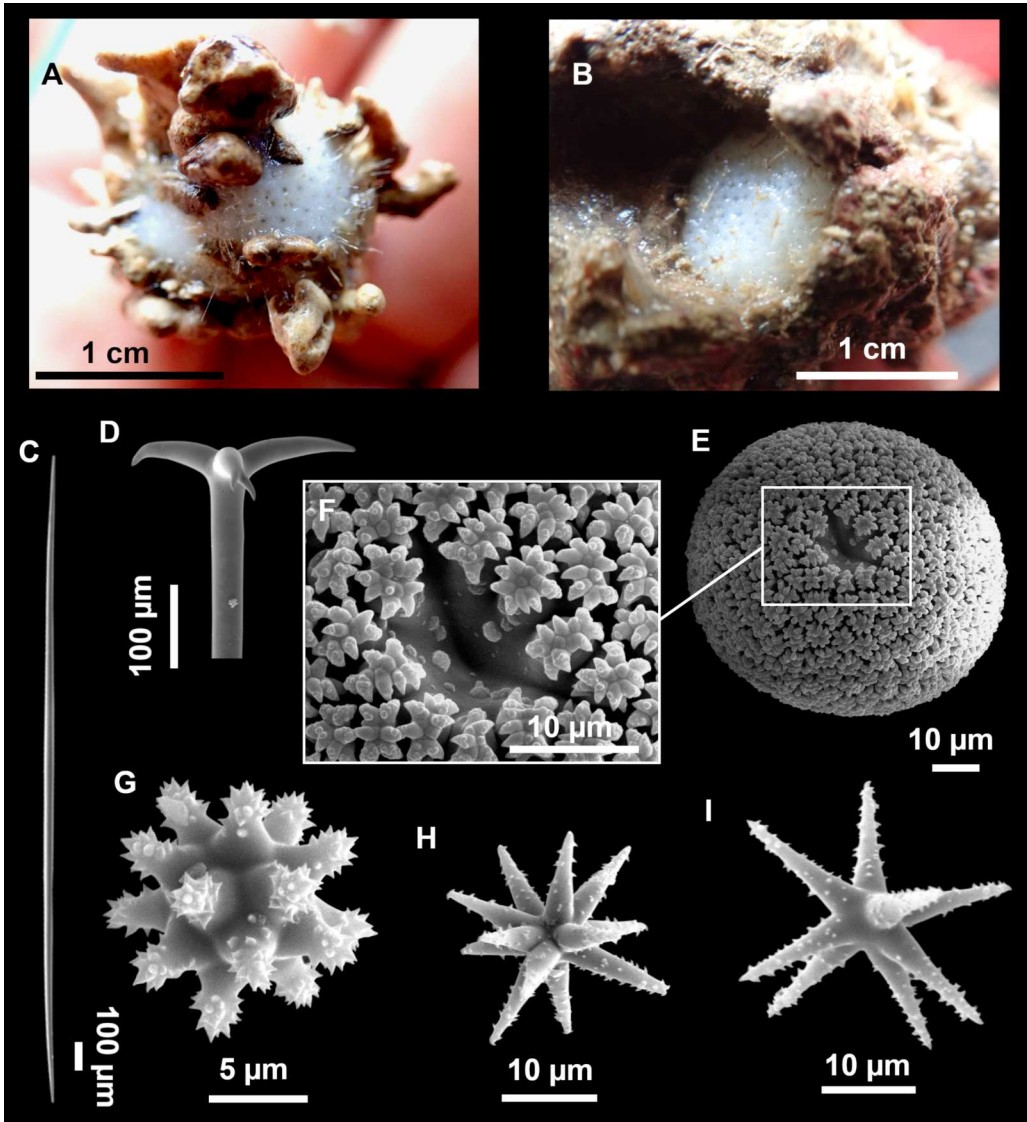

**Figure 35** *Geodia bibilonae* **sp. nov.** (A) Habitus of the holotype UPSZTY 190857 (i715_1) on deck. (B) Habitus of the paratype i780 on deck. (C–H) SEM images of the holotype spicules. (C) Oxea. (D) Ortho-triaene. (E) Sterraster with (E1) detail of the warty rosettes and hilum. (F) Spheraster. (G–H) Oxyasters.

Sterrasters (Fig. 35E), rounded, with warty rosettes (Fig. 35F), 39–69 µm in diameter. Strongylasters-spherasters (Fig. 35G), with spines on the tips of the actines, 9–18 µm. Oxyasters (Figs. 35H–35I), normally with 6–12 (sometimes up to 16), spined actines, 21–48 µm.

## Ecological notes

Species found in the mesophotic to aphotic zone, in both sedimentary and rocky bottoms. In the sedimentary bottoms the species is collected as a rounded mass with many agglutinated sediments while when found on hard substrata it is encrusting, hidden in crevices of rocks.

**Table 15   Spicule measurements of *Geodia bibilonae* sp. nov, *G. microsphaera* sp. nov. group and the related species *G. cydonium* (Linnaeus, 1767), given as minimum-mean-maximum for total length/minimum-mean-maximum for total width; all measurements are expressed in μm. Balearic specimen codes are the field#. Specimens measured in this study are in bold.**

| Material | Depth (m) | Oxeas (length/width) | Anatriaenes Rhabdome (length/width) Clad (length/width) | Protriaenes Rhabdome (length/width) Clad (length/width) | Orthotriaenes Rhabdome (length/width) Clad (length/width) | Sterrasters (diameter) | Oxyasters (diameter) | spherasters (diameter) |
|---|---|---|---|---|---|---|---|---|
| ***G. bibilonae* sp. nov. holotype, i715_1 EB** | 150 | I. 1,058-1,968-2,765/ 12-22-32 | Rh: 2,567/ 8 (N = 1) Cl: 32-65/ 9-14 (N = 3) | – | Rh: 731-1,360-2,243/ 24-38-59 Cl: 88-169-246/ 14-30-49 | 39-51-63 | 21-29-42 (6-16 actines) | 9-12-15 |
| ***G. bibilonae* sp. nov. paratype, i675 EB** | 141 | I. 1,478-1,970-2,213/ 14-20-24 (N = 18) II. 247-388/ 4-5 (N = 5) | Rh: broken Cl: 16-86/ 2-10 (N = 5) | Rh: 1,472-2,389-3,446/9-11-14 (N = 10) Cl: 69-102-157/8-11-13 (N = 16) | Rh: 594-1,524-1,885/ 14-35-47 Cl: 78-209-306/ 12-31-46 (N = 16) | 46-60-69 | 21-37-48 | 9-13-18 |
| ***G. microsphaera* sp. nov. holotype, i589_2 AM** | 109 | 1,057-1,696/ 10-16 (N = 7) | – | Rh: 1,682-2,451/8-9 Cl: 66-116/8-9 (N = 3) | Rh: 557-1,121 (N = 7)/ 9-20-25 (N = 9) Cl: 57-185-254/ 8-16-25 (N = 9) | 31-38-44 | 25-33-49 (N = 11) | 9-13-20 |
| ***G. microsphaera* sp. nov. paratype, i589_3 AM** | 109 | 844-1,164-1,493/ 10-14-16 (N = 19) | – | Rh: 2,948-3,504/10-10 Cl: 72-116/9-10 (N = 2) | Rh: 311-1,137-1,558/ 5-18-27 Cl: 34-193-294/ 4-15-25(N = 15) | 30-38-44 | 16-32-41 (N = 10, 4-8 actines) | 10-15-21 (N = 22) |
| ***G. microsphaera* sp. nov. paratype, i589_8 AM** | 109 | 827-1,242-1,694/ 9-16-23 (N = 22) | – | Rh: 1,394-3,438 (N = 9)/ 7-10-13 (N = 18) Cl: 44-93-118/5-8-11 (N = 17) | Rh: 583-985-1,278/ 9-22-29 Cl: 56-179-332/ 7-18-27 (N = 22) | 38-44-51 | 26-40 (5-7 actines, N = 6) | 5-13-23 |
| *G. cydonium* Cataluna, Spain Specimen 146 (*Uriz, 1981*) | 30-35 | I. 2,000-2,500/25 II. 600-700/10-12 | Rh: 3,000 Cl: 50 | Rh: 3,000 Cl: 60 | Rh: 1,200-1,300/ 20-25 Cl: 150-200/- | 50 | 15 | Chiasters: 15 Spherasters: 18-20 |

**Notes.**
Rh, rhabdome; Cl, clad; -, not found/not reported; EB, Emile Baudot; AM, Ausias March.

With that in mind, the spherical morphology is likely an artifact, as a consequence of the body contraction when collected, given that the small sediments that act as substrate in these bottoms are not heavy enough to avoid the body contraction. This, however, must be corroborated through direct observation of living specimens.

### Genetics

Folmer COI (ON130526, ON130527 and ON130528) and 28S (C1-C2) (ON133882, ON133883, ON133881) sequences were obtained from i780, i674 and i715_1 (holotype). Two haplotypes were found for COI and 28S, which differed in 1 bp for both markers.

### Taxonomic remarks

See taxonomic remarks on *G. bibilonae* **sp. nov.** and *G. microsphaera* **sp. nov.** below.

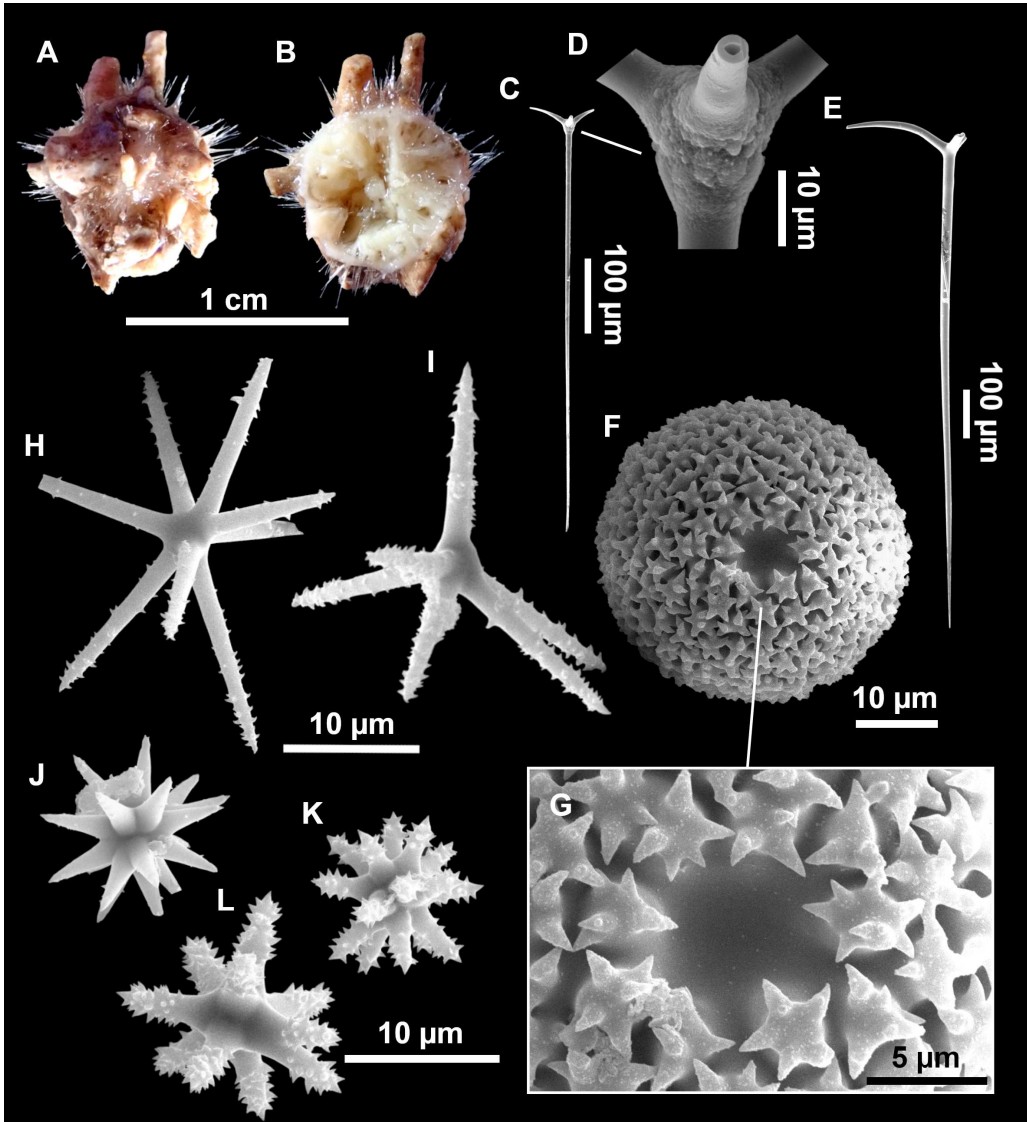

**Figure 36 Holotype of *Geodia microsphaera* sp. nov., UPSZTY 190883 (i589_2).** (A–B) Habitus after ethanol fixation. (C–L) SEM images of the spicules. (C) Juvenile orthotriaene with (D) subterminal swelling. (E) Orthotriaene. (F) Sterraster with (G) detail of the warty rosettes and hilum. (H–I) Oxyasters. (J–L) Spherasters.

*Geodia microsphaera* **sp. nov.** Díaz & Cárdenas
(Figs. 10 and 36; Table 15)

## Etymology

The name '*microsphaera*' refers to the small size of its sterrasters.

## Material examined

Holotype: UPSZTY 190883, field#i589_2, St. 21 (INTEMARES0720), MaC (AM), 109 m, beam trawl, coll. J. A. Díaz.

Paratypes: UPSZTY 190884-190885, field#i589_3-i589_8, same station and collector as the holotype.

## Comparative material

*Geodia cydonium,* ZMBN 85220, field#HC1, Hidden Cleft Cave, Brixham, Devon, England, collected above water at low tide, 1 Sept. 2008, colls. F. Crouch and C. Proctor, id: P. Cárdenas, COI: HM592715, 28S: HM592814.

## Outer morphology

Small (0.8 cm in diameter), globose sponge almost entirely covered with gravels (Figs. 34A–34B). Color on deck not recorded, of whitish surface and choanosome after fixation in ethanol. Surface fairly hispid, hard consistency. Cortex patent, 0.5 cm thick. Openings not visible. Typical geodiid skeleton with ectocortical oxyspherasters, endocortical sterrasters and choanosomal oxyasters. Choanosome fleshy.

## Spicules

Orthotriaenes (Figs. 36C–36E), fusiform rhabd and curved cladi. In juvenile forms, swellings present at the uppermost part of the rhabd (Fig. 34D). Rhabdome length: 311-1,558/5-29 µm, cladi: 34-332/4-27 µm.

Protriaenes (not shown), rare in i589_2 and i589_3 but common in i589_8. Long and thin, rhabdome straight, slightly curved or bent, cladome straight. Rhabdome: 1,394-3,504/7-13 µm, cladi: 44-118/5-13 µm.

Oxeas (not shown), fusiform, usually straight, some slightly bent or curved, 827-1696/9-23 µm.

Sterrasters (Fig. 36F), small and spherical, with warty rosettes (Fig. 36G), 31–51 µm.

Oxyasters (Figs. 36H–36I), with 4–10 long and spined actines, overall measuring 16–49 µm

Strongylasters (Figs. 36J–36L), with the spines concentrated at its tips, overall measuring 5–23 µm.

## Ecological notes

All specimens were found at a single station: the summit of the AM, in the mesophotic zone, composed of detrital bottoms with gross sand and gravels. Due to its small size and the amount of sediment that it agglutinates, the species may have been neglected from many other stations with similar features. In the field, it was almost indistinguishable from *G. bibilonae* **sp. nov.** and *S. dichoclada.*

## Genetics

COI (ON130529) and 28S (C1-C2) (ON133884) sequences were obtained from the holotype (i589_2).

## Taxonomic remarks on *Geodia bibilonae* sp. nov. and *Geodia microsphaera* sp. nov.

Both new species have very similar external morphology, spicule sets and yet clearly different COI/28S sequences; they were also found on different seamounts. *Geodia bibilonae* **sp. nov.** probably has cribriporal oscules and pores since only cribriporal openings could be found and we then assumed that some of those were inhalant and others exhalant. It can be therefore compared with North Atlantic/Mediterranean species with indistinctive cribriporal oscules/pores, some of which were formerly grouped in the genus *Cydonium* Fleming, 1828, now a synonym of *Geodia*. These species are the shallow/mesophotic species *Geodia cydonium*, *Geodia conchilega* Schmidt, 1862, *Geodia tuberosa* Schmidt, 1862 and the deep-sea North Atlantic *Geodia macandrewii* and *Geodia nodastrella* Carter, 1876. To this we need to add the poorly described species *Geodia pergamentacea* Schmidt, 1870 from Portugal. We can easily discard *G. macandrewii* and *G. nodastrella*, which are massive sponges found in the North Atlantic at deeper depths (Cárdenas & Rapp, 2013; Cárdenas & Rapp, 2015) and have for the former much larger sterrasters and for the later very characteristic ectocortical spherasters; both have also been sequenced and have different COI and 28S (Fig. 10). The shallow *G. conchilega* has a characteristic thick cortex and fairly large oval sterrasters, quite different from those in our new species; its COI/28S sequences are also different from those of our new species. As for the two poorly known Schmidt species, *G. pergamentacea* and *G. tuberosa*, for which no illustrations are published, we can rely on redescriptions (Sollas, 1888; Burton, 1946) from fragments and/or slides of type material. Although their succinct and incomplete descriptions make it challenging to identify these species, they are clearly different from our new species with much larger sterrasters (spherical 90 μm for *G. tuberosa* and oval 60–80 μm for *G. pergamentacea*). Burton (1946) further suggests that *G. pergamentacea* is a synonym of *G. conchilega* but this remains to be confirmed.

We are left with *G. cydonium*, a species complex (Cárdenas et al., 2011). Both *G. bibilonae* **sp. nov.** and *G. microsphaera* **sp. nov.** seem to be close to *G. cydonium* in terms of spicule set (small spherical sterrasters with warty rosettes, spiny oxyasters, spiny spherasters to strongylasters variations) and without molecular markers we would have been tempted to consider them conspecific with *G. cydonium*. However, although COI/28S trees (Fig. 10) confirmed that our new species belong to the 'Cydonium' clade, they also suggested that they are different from the *G. cydonium* sequenced so far. Our new species group closer to *G. cydonium* (ZMBN 85220) from Devon, England, which has similar spicules with similar sizes (sterraster size 52–60 μm, closer to those of *G. bibilonae*); however, this *G. cydonium* comes from a different habitat, intertidal shallow cave, sometimes emerged at very low tide. *G. microsphaera* **sp. nov.** even shares the exact same 28S (C1-C2) with the English *G. cydonium*, but a different COI with a 6 pb difference; this is not so surprising since the 28S C1-C2 fragment is more conserved, it is missing the more variable D2 fragment. *G. bibilonae* **sp. nov.** COI is even more different with a 17–18 bp difference with the English *G. cydonium*. So, despite similar spicule sets *G. bibilonae* **sp. nov.** and *G. microsphaera* **sp. nov.** are genetically different and live in different habitats. As for the Mediterranean *G. cydonium*, reports all show two sizes of oxeas, *vs.* only one in our new species. We hypothesize that

several cryptic species are hiding under the overused name *G. cydonium*; *G. bibilonae* **sp. nov.** and *G. microsphaera* **sp. nov.** are the first ones to be formally identified, while others await description (Cárdenas P., unpublished results) along with a necessary revision of *G. cydonium*, a task beyond the scope of this study.

*Geodia bibilonae* **sp. nov.** and *G. microsphaera* **sp. nov.** differ in that *G. bibilonae* **sp. nov.** has orthotriaenes with longer and thicker rhabdomes (594-2224/14-59 μm *versus* 311-1558/5-29 μm), thicker cladomes (12–49 μm *versus* 4–27 μm), longer oxeas (1058–2765 μm *versus* 827–1696 μm) and larger sterrasters on average (average 51–60 μm *versus* 38–44 μm). Genetically, *G. bibilonae* **sp. nov.** and *G. microsphaera* **sp. nov.** differ by 13–14 bp for the COI and 3–4 bp for the 28S (C1-C2) fragment. For *G. bibilonae,* two COI haplotypes with a 1 bp difference were detected (i780/i715_1 *versus* i674), and two 28S haplotypes with a 1 bp difference were detected (this time i780/#i674 *versus* i715_1). This reminds us what was reported for *Stelletta dichoclada* except that here there is variation within the same seamount and that both markers do not distinguish two clear populations.

Family Pachastrellidae Carter, 1875
Genus *Characella Sollas, 1886*
*Characella pachastrelloides* (*Carter, 1876*)
(Figs. 6 and 37; Table 16)

## Material examined

UPSZMC 190815, field#i527, St. 18 (INTEMARES0720), MaC (AM), 116 m, Beam trawl, coll. J. A. Díaz.

## Comparative material

*Spongosorites maximus Uriz, 1983*, CEAB.POR.BIO.89, holotype, Fora de les Garotes, off Blanes, Catalan coast, Spain, trawl fishing ground, 150–250 m (Fig. 37B).

## Outer morphology

Massive sponge, 9 × 6 × 5 cm attached to a rhodolith at its base (Fig. 37A). Alive, dark olive in the upper area, beige at the base. After ethanol fixation, surface and choanosome pale beige. Surface smooth with only faint hispidation in the groves. Hard but slightly spongy consistency. Diffuse cortex, less than 0.5 mm thick. Choanosome fleshy, not cavernous. Openings not visible.

## Spicules

Orthotriaenes (Figs. 37C and 37I), most are aberrant, in the form of orthomonoaenes and orthodiaenes, some with aborted cladome, others with ectopic actines on the rhabdome. Rhabdome length: 187-320-416/8-16-25 μm, cladi: 133-228-334/10-15-21 μm ($N = 7$).

Dichotriaenes (not shown), only two found, as modified orthotriaenes. Rhabdome length: 16-332/13-13 μm. Protoclad: 116-134/12-14 μm. Deuteroclad: 101-110/11-14 μm ($N = 2$).

Díaz et al. (2024), *PeerJ*, DOI 10.7717/peerj.16584

**Table 16** **Spicule measurements of *Characella pachastrelloides* and *Characella tripodaria*, given as minimum-mean-maximum for total length/minimum-mean-maximum for total width; all measurements are expressed in μm.** Balearic specimen codes are field#. Specimens measured in this study are in bold.

| Material | Depth (m) | Oxea (length/width) | Anatriaene Rhabdome (length/width) Clad (length/width) | Triaene Rhabdome (length/width) Clad (length/width) Protoclade (length/width) Deuteroclade (length/width) | Microxea I (length/width) | Microxea II (length/width) | Amphiaster (length) |
|---|---|---|---|---|---|---|---|
| *C. pachastrelloides* holotype, Portugal (*Sollas, 1888*; *Maldonado & Uriz, 1996*) | 683 | 3,660-4,620/ 84-100 | Rh:3,660-6,640/21 Cl: 100-170/- | Rh: 850/70 Cl: 490/- (orthotriaenes + pseudocalthrops) | 245/6 | 47/9 | 13 |
| ***C. pachastrelloides*** **i527 AM** | 116 | 681-1,088-143/8-21-35 | – | Rh: 187-320-416/8-16-25 ($N = 7$) Cl: 133-228-334/10-15-21 ($N = 7$) Pr: 116-134/12-14 Dt: 101-110/11-14 ($N = 2$) (orthotriaenes mainly) | 93-176-254/ 1-3-4 | 19-30-44/ 1-2-3 | 13-26-44 ($N = 8$) |
| ***Spongosorites*** ***maximus*** **holotype CEAB.POR.BIO.89 Catalan Coast** | 150-250 | 1,004-1,696-23/74/10-29-61 | – | Rh: 311-495-716/36-56-78 ($N = 5$) Cl: 205-361-555/34-51-78 ($N = 6$) (orthotriaenes) | 145-177-225/ 3-4-5 | 33-48-59/ 2-3-3 ($N = 19$) | 11-15-27 |
| *C.tripodaria* holotype, NHM:68:3:2:36, Algeria (*Maldonado & Uriz, 1996*; *Cárdenas & Rapp, 2012*) | – | 1,000-1,600/ 10-40 | Rh:-/10 Cl: 25/- | Rh: 180-400/15-22 Cl: 180-400 (plagiotriaene pseudocalthrops) | 115-180/ 2-3 | 35-45/2-3 | 10-18-30 ($N = 15$) |

Díaz et al. (2024), *PeerJ*, DOI 10.7717/peerj.16584

**Table 16** (*continued*)

| Material | Depth (m) | Oxea (length/width) | Anatriaene Rhabdome (length/width) Clad (length/width) | Triaene Rhabdome (length/width) Clad (length/width) Protoclade (length/width) Deuteroclade (length/width) | Microxea I (length/width) | Microxea II (length/width) | Amphiaster (length) |
|---|---|---|---|---|---|---|---|
| *C. tripodaria* i153_1B EB | 107-110 | 689-1,456-2,148/ 15-29-49 | Rh: 723-1,006-1,767/7-13-18 Cl: 27-66-108/7-12-18 (N = 18) | Rh: 207-298-434/ 16-23-30 (N = 17) Cl: 221-315-426/15-22-26 (plagiotriaene pseudocalthrops) | 86-126-152/ 2-3-5 | 25-35-53/ 1-2-4 | 8-14-27 |
| *C. tripodaria* i777 EB | 102-105 | 697-1,369-1,909/7-22-41 | – | – | 86-125-160/ 2-3-4 | 22-33-40/ 1-2-3 (N = 6) | 14-17 (N = 2) |

**Notes.**

Rh, rhabdome; Cl, clad; -, not found/not reported; EB, Emile Baudot; AM, Ausias March.

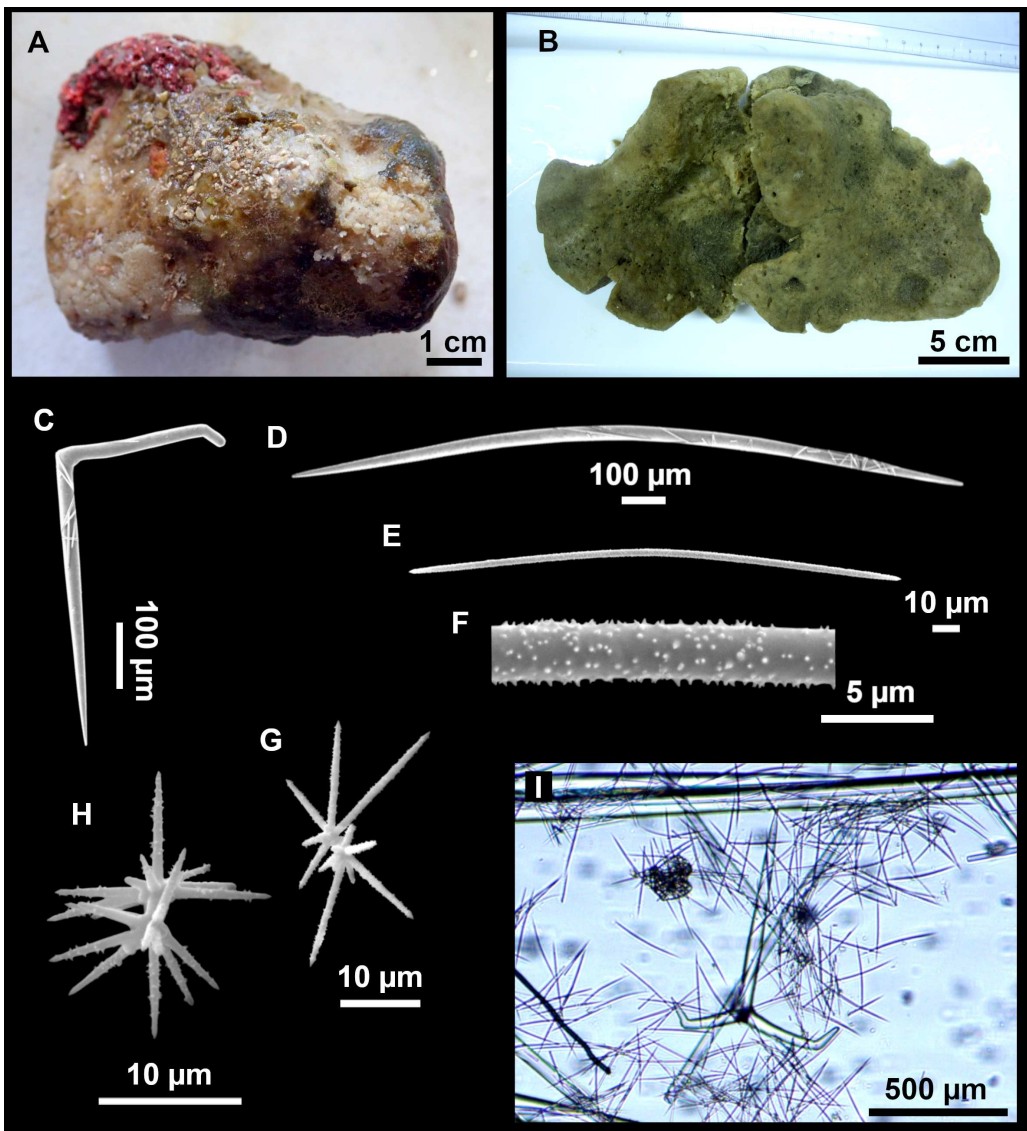

**Figure 37** *Characella pachastrelloides* (*Carter, 1876*). (A) Habitus of specimen i527 on deck. (B) Habitus of *Spongosorites maximus Uriz, 1983*, (CEAB.POR.BIO.89) holotype. (C–H) SEM images of spicules from i527 (C) Orthomonoaene. (D) Oxea. (E) Microxea I with (F) detail of the spines. (G–H) Amphiasters. (I) Optical microscope image of an orthotriaene and microxeas.

Oxeas (Fig. 37D), very abundant, centrocurved, 681-1,088-1,437/8-21-35 μm.

Microxeas I (Figs. 37E–37F), very abundant, thin and centrocurved, microspined (Fig. 37F), with sharp ends, 93-176-254/1-3-4 μm.

Microxeas II (not shown), rare, spiny, some centrotylote, 19-30-44/1-2-3 μm.

Amphiasters (Figs. 37G–37H), rare, with long spined actines, aborted actines are common, shafts are clear or with aborted actines, length: 13-26-44 μm (N = 8).

### Ecology and distribution
Species only found once at the top of the AM, on a rhodolith bed.

### Genetics
COI (ON130551) and 28S C1-C2 (ON133873) sequenced.

### Taxonomic remarks
See below discussion of *C. tripodaria*.

*Characacella tripodaria* (*Schmidt, 1868*)
(Figs. 6 and 38; Table 16)

### Material examined
UPSZMC 190816, field#i153_1B, St. 52 (INTEMARES0718), MaC (EB), rock dredge, 110–107 m, coll. F. Ordines; UPSZMC 190818, field#i777, St. 53, MaC (EB), rock dredge, 108–102 m, coll. J. A. Díaz.

### Comparative material
*Characacella tripodaria*, MNHN DT756, holotype, Schmidt collection#107, Algeria, 'Exploration Scientifique de l'Algérie', 1842.

### Outer morphology
Massive irregular (i153_1B) or massive elongated (i777), up to four cm in diameter. In life, pale beige, gray after ethanol fixation, same color on the surface and in the choanosome. Surface mostly hispid with some smooth areas. Hard consistency. Diffuse cortex, less than 0.5 mm thick. Openings not visible.

### Spicules
Plagiotriaene pseudocalthrops (Figs. 38B–38D), only found in i153_1B, the three clades often curved making them look like plagiotriaene. Sometimes triactinals, with an aborted actine. Rhabdome length: 207-298-434/16-23-30 µm. Cladi: 221-315-426/15-22-26 µm (N = 17).

Anatriaenes (Fig. 38E), only in i153_1B, scarce, with a fusiform rhabdome, the cladome may have clads projecting straight from the center or drawing a soft curvature. In some cases, there are aborted clads. Rhabdome length: 723-1,006-1,767/7-13-18 µm, cladome: 27-66-108/7-12-18 µm (N = 18).

Oxeas (Fig. 38D), slightly curved, 689-2,148/7-49 µm.

Microxeas I (Figs. 38B–38E), thin and slightly curved in the center, with sharp ends, minutely spined, 86-160/2-5 µm.

Microxeas II (Fig. 38H), rare (i153_1B) to very rare (i777), thin, some are straight while others are abruptly curved in the center, with some stylote and strongylote modifications, 22-53/1-4 µm.

Amphiasters (Figs. 38F–38G), rare (i153_1B) to very rare (i777), some are normal while others have malformations and aborted actines, 8–27 µm.

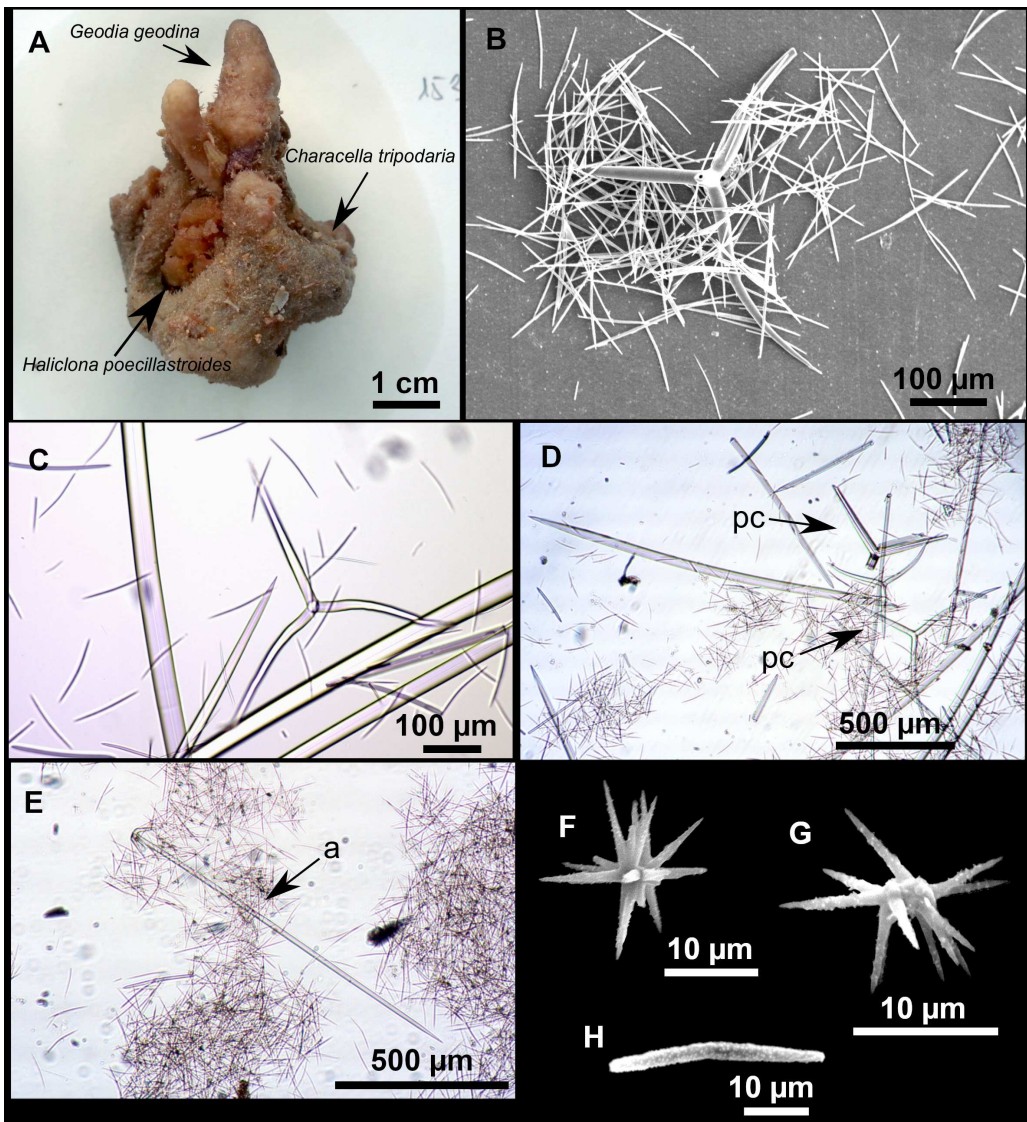

**Figure 38** *Characella tripodaria* (*Schmidt, 1868*). (A) Habitus of i153_1B after ethanol fixation, overgrown with *Geodia geodina* and *Haliclona poecillastroides*. (B–H) Spicule images of i153_1B. (B) SEM image showing a plagiotriaene pseudocalthrop and microxeas. (C–D) Several plagiotriaene pseudocalthrops (pc), oxeas and microxeas. (E) Anatriaene (a) and microxeas I. (F–G) Amphiasters. (H) Microxeas II.

## Ecology and distribution

Both specimens were found at the summit of the EB, together with other massive sponges like *Spongosorites* spp., *Pachastrella monilifera* and several axinellids.

## Genetics

COI was obtained from i777 (ON130552) while only the short miniCOI was obtained from i153_1B (SBP#2690). The 28S (C1-C2) fragment was obtained from both i777 and i153_1B (ON133871 and ON133872).

## Taxonomic remarks on *C. pachastrelloides* and *C. tripodaria*

There are two *Characella* species documented in the Mediterranean Sea and the nearby North Atlantic, *C. pachastrelloides* and *C. tripodaria,* both sharing an almost identical set of spicules. The taxonomic history, characters and distribution of the two species have previously been discussed (*Maldonado & Uriz, 1996*; *Cárdenas & Rapp, 2012*). After revision of the holotype of *C. tripodaria, Cárdenas & Rapp (2012)* concluded that no clear differences could be made between the two species, even though the amphiasters seemed to have more actines on the shaft in *C. tripodaria*. Our COI and 28S sequences both suggested a partition of i527 *versus* i153_1B and i777. Specimen i527 from AM Seamount is a perfect COI match with boreal *C. pachastrelloides* (haplotype 1) from Norway (HM592672), Scotland (HM592749) and the Globan Spur (Celtic Sea) (MK085975). The COI of i153_1B and i777 (haplotype 3) has a 2 bp. difference with the boreal sequences and 1 bp. difference with two fairly deep *C. pachastrelloides* from the Gulf of Cadiz (HM592713) and southern Portugal (HM592709) (haplotype 2), both close to the type locality of *C. pachastrelloides.* So we now have three COI haplotypes in this complex. We have looked for morphological differences between i527 *versus* i153_1B/i777: i527 is large, with a smooth surface while i153_1B/i777 specimens are smaller and very hispid but such shape/surface variations have already been observed by *Topsent (1904)* in the Azores within *C. pachastrelloides,* so they do not seem to be diagnostic. The triaenes however look different: i153_1B has pseudocalthrops often with three curved forward-oriented clades, that we decided to call plagiotriaene pseudocalthrops (Figs. 38B–38D), while i527 has rather irregular orthotriaenes with most of the time straight clades. The taxonomic value of triaene morphological variations in *C. pachastrelloides* has already been raised, but always failed to reveal a phylogenetic pattern (*Topsent, 1904*; *Maldonado & Uriz, 1996*; *Cárdenas & Rapp, 2012*). Looking back on previous descriptions it does seem however that triaenes may reflect different populations or species. We noticed that (i) the Norwegian, Scottish and Globan Spur specimens (hap 1, mesophotic to upper bathyal) all had (i) short-shafted orthotriaenes (sometimes looking like pseudocalthrops) mixed with dichotriaenes, with clads usually straight, except when irregular; no anatriaenes found; amphiasters were moderately abundant, usually with a "clean" shaft (*i.e.,* no extra actines there). North Atlantic Iberian specimens (hap 2, upper bathyal), the closest to the type locality of *C. pachastrelloides* in terms of geography and depth, had anatriaenes and the same type of triaenes as hap1 but usually larger/thicker (*Cárdenas & Rapp, 2012*, Table 2); amphiasters are moderately abundant as well, usually with a "clean" shaft. Finally, our specimens i153_1B/i777 from EB (hap 3, mesophotic zone) have anatriaenes and pseudocalthrops, usually with curved clades, sometimes malformed/aborted, no dichotriaenes; amphiasters are somewhat in lower numbers, with additional actines on the shaft. This description fits the holotype of *C. tripodaria* we examined, and in which we found triaenes with curved clades similar to that drawn by *Schmidt (1868)*, pl. III, Fig. 10A), no dichotriaenes, and similar amphiasters (*Cárdenas & Rapp, 2012*, Fig. 2J). The triaenes of i527 from AM look more like those in hap1 and hap2, short-shafted orthotriaenes mainly, but they are on average smaller and more irregular than in its North Atlantic counterparts. We concluded that in the Balearic Islands we had *C. pachastrelloides* (i527) and *C. tripodaria* (i153_1B and

i777) at the same depth, but so far on different seamounts. Following this, we consider for now hap1 and hap2 (1 bp. difference) to be haplotypes of two distinct populations of *C. pachastrelloides*. Therefore, hap3 with 1–2 bp difference with *C. pachastrelloides* is the first COI sequence of *C. tripodaria*.

Besides, microxea I lengths are identical between *C. tripodaria* specimens (86–152 μm and 86–160 μm, respectively) and much smaller than those of *C. pachastrelloides* (93–254 μm). This pattern in microxea I sizes is consistent with the sizes reported for other *C. pachastrelloides* and *C. tripodaria*: for *C. pachastrelloides*, it matches with the holotype (246 μm) and the Norwegian specimen ZMBN 80248 (80–259 μm; *Cárdenas & Rapp, 2012*). It should be noted that COI and 28S are available for ZMBN 80248, being identical to i527. Conversely, the microxea I sizes for the holotype of *C. tripodaria* are also in the same size range (115–180 μm) than those of the Balearic Islands. Also, as already discussed in the literature, amphiaster morphology seems to be different between *C. pachastrelloides* and *C. tripodaria*. We found that those spicules are rare and may have aborted actines in both species. However, amphiasters of *C. tripodaria* tend to have a more elevated number of actines, which are also shorter, and a shorter shaft, a character already also noted by *Maldonado & Uriz (1996)* and *Cárdenas & Rapp (2012)*. Notwithstanding the mentioned, amphiaster morphology is, by today, a weak character to discern between *Characella* species, as it is probably influenced by ecophysiological factors and/or intraspecific variability. Further works shall study a larger number of individuals and compare its amphiasters and its genetic sequences to clarify its systematic value.

In the course of another project, the holotype of *Spongosorites maximus Uriz, 1983* had been examined (Fig. 37B) with the making of new spicule preparations (Table 16). Unexpectedly, orthotriaenes and amphiasters were found, which, combined with two sizes of microxeas, suggested that it was in fact a *C. pachastrelloides*. Therefore, *S. maximus* becomes a junior synonym of *C. pachastrelloides*, further confirming the presence of this species in the Mediterranean Sea. This large specimen (25 × 16 cm) had been collected on the fishing grounds off the Catalan coast (Fora de les Garotes) at mesophotic depths (150–250 m). The third record of a Mediterranean *C. pachastrelloides* is from south Malta at 607 m depth (*Calcinai et al., 2013*), the description of orthotriaenes, abundant amphiasters with few actines suggest that this is a correct identification. For *C. tripodaria*, it has been previously reported from Algeria and the Alboran Sea (*Maldonado & Uriz, 1996*) so this is the third report in the Mediterranean Sea. Both species are reported for the first time off Balearic Islands.

Anatriaenes were not found in *C. pachastrelloides* i527 nor *C. tripodaria* i777 but they were relatively common in *C. tripodaria* i153_1B, which further confirms their presence in this species. Since anatriaenes were previously formally reported from *C. pachastrelloides*, including the holotype (*Maldonado & Uriz, 1996*; *Cárdenas & Rapp, 2012*), they are not a good character to discern between *C. pachastrelloides* and *C. tripodaria*.

Smooth oxeas II have previously been reported with some doubt in a specimen from Norway (*Cárdenas & Rapp, 2012*). Here, similar smooth smaller oxeas can be found in i153_1B/i777 but the most plausible explanation is that they are contamination by haplosclerid sponges (especially *H. poecillastroides*) overgrowing all the studied specimens.

They have the same oxeas, with the same length/width as those found. Similarly, the holotype of *C. tripodaria* is overgrown by an haplosclerid, and possesses such oxeas (*Topsent, 1938*). The report of oxeas II in *Characella luna Dias, Santos & Pinheiro, 2019* from Brazil therefore needs to be considered with caution and confirmed.

Genus *Nethea Sollas, 1888*
*Nethea amygdaloides* (*Carter, 1876*)
(Figs. 6 and 39; Table 17)

## Material examined

UPSZMC 190889, field#POR7(15), St. 181 (MEDITS052016), south west of Cabrera Archipelago, 142 m, GOC-73, coll. P. Ferriol; UPSZMC 190888, field#POR347_b, St. 194 (MEDITS052017), Port d'es Canonge (North of Mallorca), 148 m, GOC-73, coll. J. A. Díaz; UPSZMC 190890, field#i215_b, St. 3 (INTEMARES1019), MaC (SO), 293–255 m, rock dredge, coll. J. A. Díaz.

## Comparative material

*Nethea amygdaloides*, holotype, NHM Norman Coll. slides 1910.1.1.1683-1685, near Cap St. Vincente, Portugal, 534 m, Porcupine Expedition, St. 24, 1870; ZMAPOR 21223, Gulf of Cadiz, 35°18′46.8″N, 6°13′44.4″W, 428 m, 2007, field#CAD07-01, coll: Loïs Maignien, id: P. Cárdenas, 28S: HM592773; MNHN DCL4077, Apulian Platform, off Cape Santa Maria di Leuca, southern Italy, 39°33′54.78″N, 18°26′12.39″E, 562 m, sampling#PBT1(1), ROV dive 327-6, MEDECO leg1 (Ifremer), 17 Oct. 2007, coll: Julie Reveillaud, id: P. Cárdenas, 28S: HM592772; sampling#GBT1-1, Apulian platform, Atlantis mound, off Cape Santa Maria di Leuca, southern Italy, 39°36′43.84″N, 18°30′28.28″E, 648 m, ROV dive 328-7, MEDECO leg1 (Ifremer), 17 Oct. 2007, coll: Julie Reveillaud, id: P. Cárdenas; PC479, SME PL.ACH.P1, Canyon du Planier, off Marseille, France, 43°0′6.08″N, 5°0′12.49″E, 332 m, ROV dive B4-PL-ACH-P01-20091112, 12 Nov. 2009, MedSeaCan campaign, RV *Minibex*, coll: J. Vacelet, id: P. Cárdenas; PC1280, SME TetractCYE8-D2b, Canyon St. Florent, Corsica, 42°45′56.88″N, 9°13′30.72″E, 208 m, 6 Oct. 2018, H-ROV *Ariane* dive 110-08, Cylice-Eco (CYE) campaign, RV *L'Europe*, coll: P. Chevaldonné, id: P. Cárdenas.

## Outer morphology

Massive, or encrusting (i215_b); the largest specimen, POR7(15) (Fig. 39A) measures four cm in diameter and one cm in height. The color of POR7(15) was whitish on deck and whitish gray after preservation in alcohol. In Por347_b, the color after preservation is dark gray. The skin of both individuals is slightly rough to the touch with only localized hispidation visible to the naked eye. Consistency stony hard, leaving a mark when pressed. Single oscula observed in both specimens, measuring 2 mm in POR7(15) and 1 mm in POR347_b. Pores inconspicuous. Specimen POR7(15) had a dark garnet *H. poecillastroides* as epibiont (Fig. 39A).

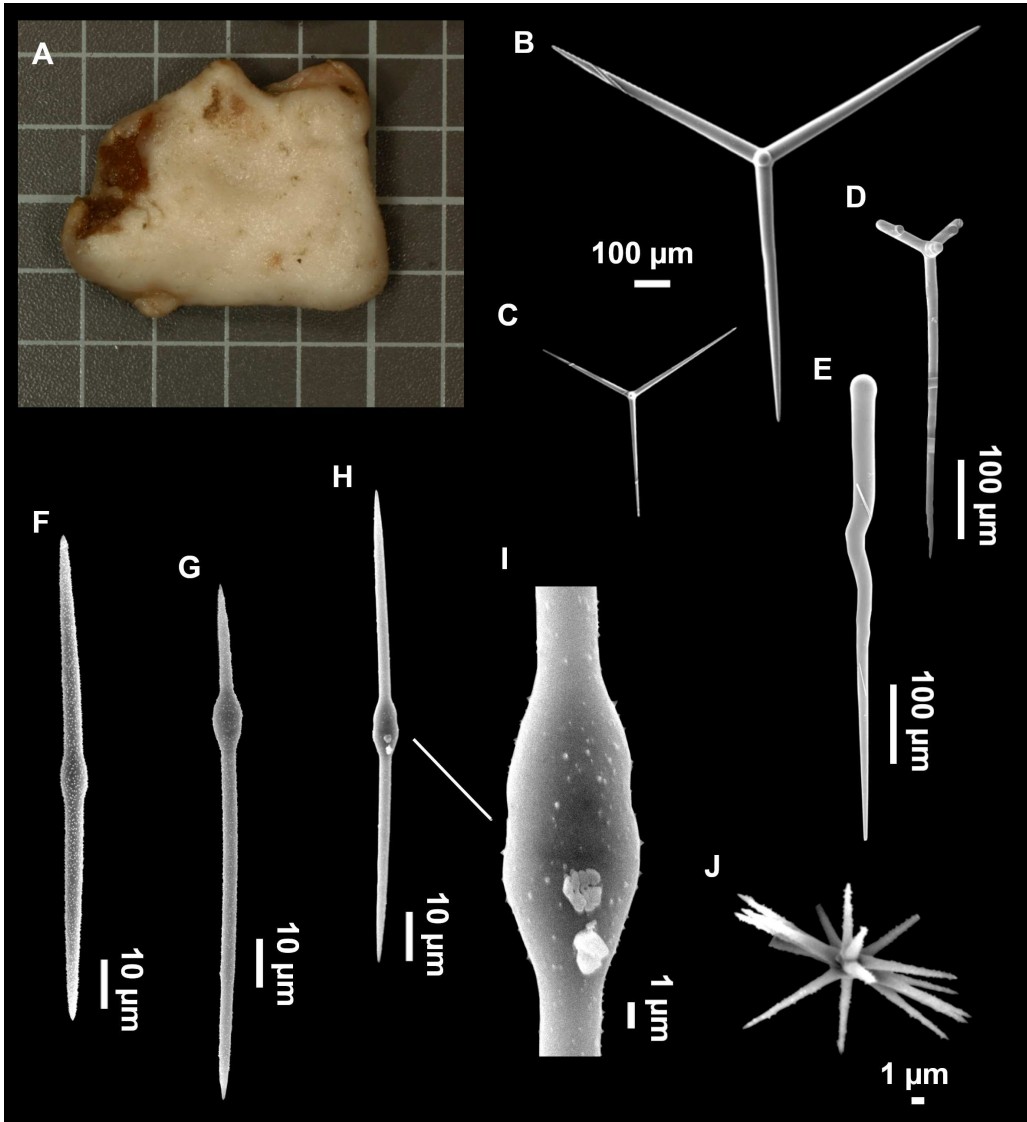

**Figure 39 *Nethea amygdaloides* (*Carter, 1876*).** (A) Habitus of specimen POR7(15) on deck. (B–I) SEM images of specimen POR347_B spicules. (B–C) Regular calthrops. (D–E) Underdeveloped calthrops. (F–H) Microxeas with (I) detail of a central swelling and spines. (J) Amphiaster.

## Spicules

Cathrops (Figs. 39B–39E). Regular ones are mostly tetractinals (Figs. 39B–39C) with actines disposed in equiangular disposition on a same plane and with one of the actines reduced to a stump, making the spicule look like a triactin. Regular actines are mostly straight, occasionally slightly curved; distinct swellings occasionally occur on the actines in Por7(15); actine ends are progressively sharpened or stepped. In Por347_b and Por7(15), malformed or underdeveloped actines are quite common; as well as reduced triactines to one (Fig. 39E) or two actines. Actines overall measuring 114-910/5-48 µm.

**Table 17  Spicule measurements of *Nethea amygdaloides,* given as minimum-mean-maximum for total length/minimum-mean-maximum for total width; all measurements are expressed in μm.** Balearic specimen codes are the field#. Specimens measured in this study are in bold.

| Material | Depth (m) | Oxea (length/width) | Calthrops (length/width of actine) | Microxea (length/width) | Amphiaster (length) |
|---|---|---|---|---|---|
| **Holotype NHM 1910.1.1.1683-1685 Cap St. Vincent, Portugal** | 534 | 796-1,076-1,673/ 10-15-20 | 184-539-755/ 18-47-67 | 35-94-155/ 2-5-8 | 17-21-25 |
| **Por347_B North of Mallorca** | 148 | 682-857-1,008/ 9-13-18 | 250-496-829/ 9-22-34 | 53-106-178/ 2-4-6 | 8-15-24 |
| **Por7(15) Cabrera archipelago** | 142 | 615-1,123-1,863/ 6-13-24 | 114-449-910/ 5-25-48 | 70-131-181/ 3-6-10 | 10-16-31 |
| **i215_b SO** | 274 | 747-1,055-1,340/ 7-12-17 (N = 12) | 235-441-670/ 13-28-38 | 67-123-174/ 2-4-5 | 13-20-29 |
| **MNHN DCL4077 Southern Italy** | 562 | 974-1,660/ 10-22 (N = 5) | 181-594-871/ 11-39-59 | 55-104-169/ 2-3-6 | 13-25-54 |
| **GBT 1(1) Southern Italy** | 648 | 1,354-2,036/ 15-27 (N = 6) | 256-673-1,048/ 12-34-51 | 80-133-192/ 2-4-6 | 12-16-27 (N = 12) |
| **ZMAPOR 21223 Gulf of Cadiz, Atlantic** | 428 | 835-2,068/ 12-24 (N = 7) | 190-734-996/ 10-37-50 | 63-109-157/ 1-4-6 | 12-15-19 (N = 15) |
| **PC479 Marseille, France** | 330 | 914-1,770/ 15-28 | 215-734-1,007/ 13-34-47 | 72-126-171/ 4-5-7 | 9-18-35 |
| **PC1280 Canyon St. Florent Corsica** | 208 | 527-1,267-1,776/ 8-19-29 | 173-608-871/ 7-25-41 | 74-133-172/ 2-4-7 | 9-16-24 (N = 14) |
| NIS.70.1, PF.263 Corsica (*Pulitzer-Finali, 1983*) | 140 and 200 | 1,200-1,600/ 11-14 (rare) | 650/37 (some reduced to di-actines) | 65-140/ 1.5-5.5 | Abundant (NIS.70.1), rare (PF.263) |

**Notes.**
   SO,  Ses Olives.

Oxeas (not shown), long, thin, slightly curved and with sharp tips. Measuring 615-1,863/6-24 μm.

Microxeas (Figs. 39F–39I), mostly bent, but some are straight; in i215_b they are quite curvy with sometimes double bends. With a marked (Fig. 39F) or subtle (Figs. 39G–39H and 39I) microspination. In Por347_b, many microxeas have a distinct swelling in the middle or the upper part; in Por7(15) and i215_b such swellings are very rare. Length: 53–181 μm, width: 2–10 μm.

Amphiasters (Fig. 39J), moderately abundant, 14–17 actines, both regular and underdeveloped forms are found. Clear shaft, sometimes reduced/absent shaft making some amphiasters look like euasters. Length: 8–31 μm.

## Ecology and distribution

Uncommon, found on upper slopes of the Mallorca shelf and SO seamount.

## Genetics

The COI of specimens POR347_B, POR7(15) (ON130544 and ON130545), and the miniCOI fragment of i215_b (SBP#2686) have been obtained. The 28S (C1-C2) fragment was obtained from i215_b (ON133878).

## Taxonomic remarks

The specimens are assigned to the genus *Nethea* on the basis of the triactinal calthrops with three actines disposed on a single plane and a fourth actine absent or reduced to a stump. When they resurrected *Nethea*, *Cárdenas et al. (2011)* suggested to include all species with triactial calthrops and amphiasters: *Nethea nana* (Carter, 1880) from the Indian Ocean, *Nethea capitolii* (*Mothes et al., 2007*) from Brasil and *N. amygdaloides*. The case of *Characella connectens* (*Schmidt, 1870*) from Florida is unclear since *Maldonado (2002)* observed in the type material triactinal calthrops but also two sizes of microxeas instead of one, as in *Characella* species: a revision of the type material is necessary.

*Nethea amygdaloides* is occasionally reported from mesophotic to bathyal depths (103–2,165 m) of the northeast Atlantic (*Topsent, 1892*; *Topsent, 1904*; *Topsent, 1928*; *Sitjà et al., 2019*; *Ríos et al., 2022*) and shallow to mesophotic depths (25–200 m) of the Western Mediterranean Sea (*Topsent, 1895*; *Pulitzer-Finali, 1983*). Comparative material from the upper bathyal depths in the Apulian platform (off Cape Santa Maria di Leuca, southern Italy) also shows that this species can be found quite deep also in the Mediterranean. The species was previously reported on a monticule at the southeast of the EB (*Maldonado et al., 2015*; Table 1) but no description was given. Here, the species is reported for the second time in the Balearic Islands region, and fully described/barcoded. The Folmer primers together do not seem to work for this species (Cárdenas P., unpublished data) so we tried to sequence COI in two parts (miniCOI + part 2) which worked, thus giving us the first COI sequence for this species. This is an opportunity to test the phylogenetic position of this species which is very uncertain with 28S (C1-D2) (*Cárdenas et al., 2011*; Fig. 6B). Unfortunately, our COI tree (Fig. 6A) also gives an uncertain position for *N. amygdaloides*, diverging between the Vulcanellidae and lithistid families, alone on a poorly-supported branch. The two 28S (C1-D2) sequences previously obtained by *Cárdenas et al. (2011)* were quite different: specimen MNHN DCL4077 (HM592772) from Italy and ZMAPOR 21223 (HM592773) from the Gulf of Cadiz had a significant 19 bp difference plus a deletion of 4 bp, a result difficult to explain at the time. Our shorter 28S (C1-C2) fragment (ON133878) is a 100% match to the sequence from Italy confirming this difference. Such a high genetic difference between Northeast Atlantic and Mediterranean specimens suggests two clear different species, and a species complex for *N. amygdaloides*.

In order to explore this hypothesis, we decided to measure spicules from the type, which has never been revised since its original description (*Carter, 1876*). *Maldonado (2002)* mentions a type slide (NHM 00047; I.1.2) but does not give any measurements or drawings. Here, three slides were examined (NHM 1910.1.1.1683-1685); although not explicitly labeled as type slides, these slides have the label '*Porcupine Exped. Pachastrella amygdaloides Cr*'. Since the holotype is the only specimen collected during the Porcupine Expedition, we can be sure that these spicule slides are from the holotype. These slides also

have purple hand-written numbers '43' (1684), '44' (1683) and '45' (1685), which seem older, and may explain why *Maldonado (2002)* examined a slide with number '47'. On slides 1,683 and 1,685, the genus has been changed to *Poecillastra* with hand-written ink, maybe after its genus reallocation (*Sollas, 1888*). In addition to the holotype slides, spicules from other specimens were measured: *N. amygdaloides* off southern Italy (MNHN DCL4077, GBT1(1)), Marseille (PC479), Corsica (PC1280) and the Gulf of Cadiz (ZMAPOR 21223) (Table 17). The spicule sizes of the three specimens collected from the Balearic Islands match well with those of the holotype (Table 17): minor differences were found in the size of the calthrops (with thicker actines in the holotype) and the microxeas (slightly shorter in the holotype). Overall, no significant size differences could be identified between the two Northeast Atlantic and the seven Mediterranean specimens. Spicule morphologies were then compared. The triactines can have straight or slightly bent actines: in the holotype the majority of the actines are bent, while actines in Mediterranean species seem to be more straight, but more Atlantic specimen are required to confirm this difference. Besides, the northeast Atlantic specimen ZMAPOR 21223 also had triactines with fairly straight actines, as well as irregular/aborted actines, unlike the holotype. Irregular triactines were especially common in the most shallow specimens from the Balearic Islands (Por347_B and POR7(15)) where reductions to one-two actines were found, as well as in GBT1(1) from Italy. We also noticed that swellings along the actines were quite common in the Mediterranean specimens (i215b, GBT1(1), PC479, PC1280) albeit not always present (POR347_B, POR7(15)); these were not present on the actines of the northeast Atlantic specimens (type and ZMAPOR 21223). We also compared microxea morphology but no differences could be found: centrotylote microxeas are generally uncommon, but specimen Por347_B had plenty (Figs. 39F–39I), maybe because its microxeas were thinner, which revealed more swellings. Finally, we cannot exclude differences in external morphology between the Atlantic and Mediterranean populations but unfortunately Atlantic specimens are few and either poorly described or only a fragment (*e.g.*, ZMAPOR 21223). To conclude, although 28S suggests *N. amygdaloides* to be a species complex, we refrain from any taxonomical action before more specimens can be revised (*e.g.*, Topsent's numerous specimens from the Azores) and sequenced.

A second size of oxeas, resembling renierid spicules, is quite common in our samples. As for *C. pachastrelloides* and *C. tripodaria* (see above), those spicules are probably foreign, because the species shares habitat with haplosclerids with spicules of a similar size and morphology. In fact, a specimen of *H. poecillastroides* (#CFM-IEOMA-6393) reported by *Díaz et al. (2020)* was collected along with POR347_b, with a similar spicule size. Those spicules were not present in the holotype and in specimens from Corsica (PC1280), Marseille (PC479) and Italy (MNHN DCL4077).

Genus *Pachastrella* *Schmidt, 1868*
*Pachastrella monilifera* *Schmidt, 1868*
(Fig. 40; Table 18)

## Material examined

UPSZMC 190891-190892, field#i139_A-i139_B, MaC (EB), St. 51 (INTEMARES0718), 128 m, beam trawl; UPSZMC 190893-190894 field#i153_3 and field#i157, MaC (EB), St. 52 (INTEMARES0718), 109 m, rock dredge, coll .F. Ordines; UPSZMC 190897, field# i352_4, MaC (EB), St. 136 (INTEMARES1019), 146 m, beam trawl, coll. J. A. Díaz; UPSZMC 190898, field#i650, MaC (AM), St. 34 (INTEMARES0720), 105–111 m, rock dredge, coll. J. A. Díaz; UPSZMC 190899-190900, field#i687 and field#i688, MaC (EB), St. 43 (INTEMARES0720), 117 m, rock dredge, coll J. A. Díaz; UPSZMC 190901, field# i771, MaC (EB), St. 53 (INTEMARES0720), 104 m, rock dredge, coll J. A. Díaz; UPSZMC 190904, field#i827_2, MaC (EB), St. 25 (INTEMARES0820), 100 m, ROV, coll J. A. Díaz.

## Comparative material

*Pachastrella monilifera*, MNHN-DT-410, holotype, Schmidt collection#65, Algeria, 'Exploration Scientifique de l'Algérie', 1842.

## Outer morphology

Massive irregular or cup-shaped, 2–15 cm in diameter. Coralligenous algae (alive and dead), used as substrate and incorporated in the body (Fig. 40A). Several sponge epibionts, including *H. poecillastroides*, *Jaspis* sp., *P. compressa* and *Craniella* cf. *cranium* (Figs. 40B–40C). No oscula observed. In some specimens, several minute orifices are placed in a central depression (Fig. 40B). Same color on deck and after ethanol fixation; whitish gray. Stony hard consistency. Surface smooth with localized hispidation areas. Diffuse cortex ca. 500 μm thick.

## Spicules

Calthrops, large size range (Figs. 40D–40E). Three of its actines are curved, corresponding to the cladome, and a fourth one straight, slightly longer, which is the rhabdome. Differentiation between cladome and rhabdome is obvious in larger spicules, but unclear in medium/small ones. For this reason clads and rhabdome measurements were merged. On the largest calthrops, actines may be rarely dichotomous or aberrant (Fig. 40D). Actine length: 23-779/4-112 μm.

Oxeas, long and thin, found mostly broken. However, in specimen i153_3A several unbroken ones were measured, length: 972-2,326-3,907/10-17-24 μm ($N = 11$). It cannot be excluded that longer oxeas may be present.

Amphiasters (Figs. 40F–40G), scarce, microspined rays radiating from two distal axes, sometimes from the shaft, 7–19 μm.

Microstrongyles I (=microrhabds, Fig. 40H), spiny, sometimes centrotylote, elongated to spherical, 8-21/2-6 μm.

Microstrongyles II (=microrhabdose streptasters), very scarce, spiny, thin, elongated and curved, rarely centrotylote, 17-56/1-2 μm.

## Ecology and distribution

Found at the summit of the EB and the AM, on coralligenous algae bottoms (100–146 m) that serve as substrate for growth. It was also found growing on *S. mucronatus* (i827_1).

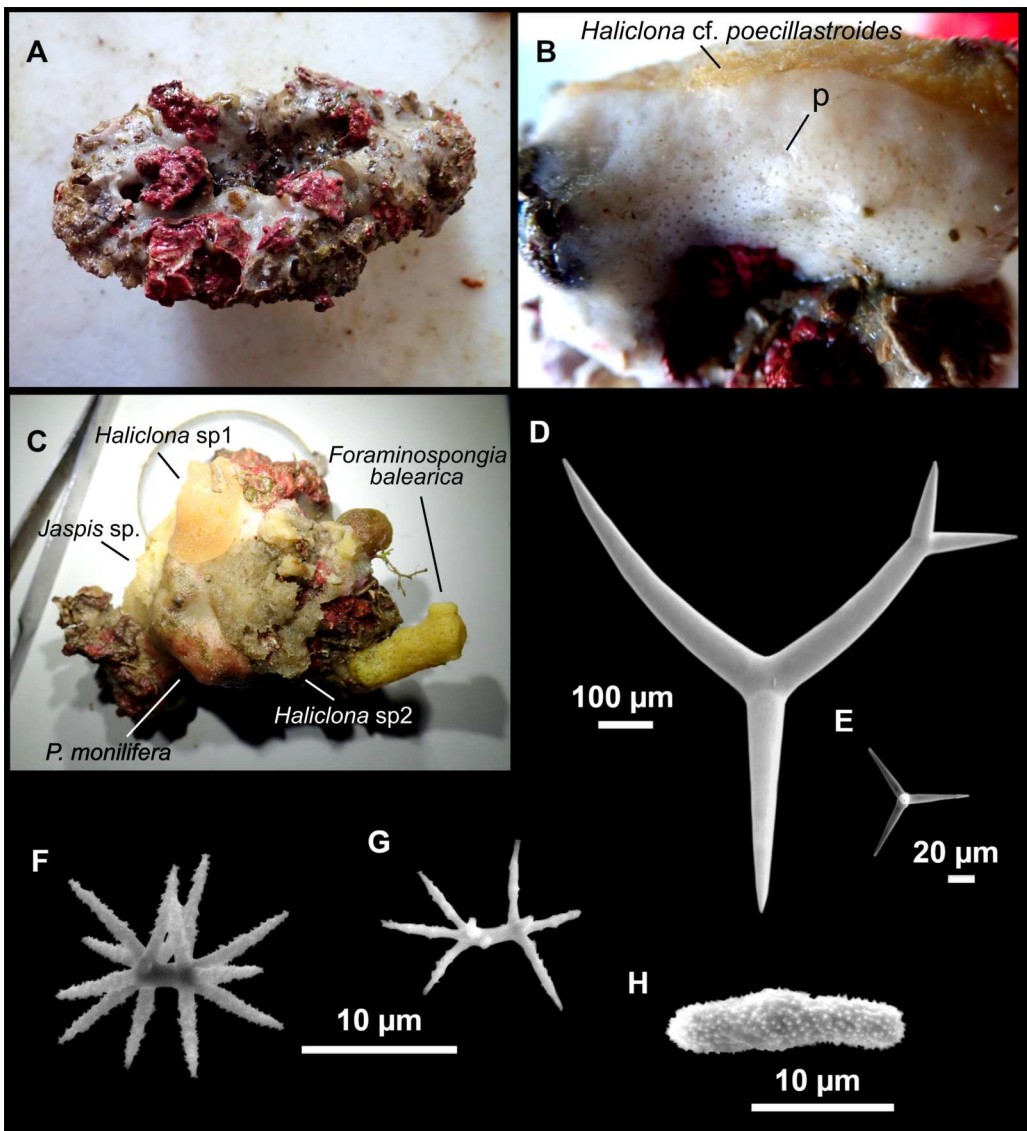

**Figure 40** *Pachastrella monilifera Schmidt, 1868.* (A) Specimen i688 on deck. (B) Specimen i687 on deck, showing the pores (p) and overgrowth by *Haliclona* cf. *poecillastroides.* (C) Specimen i824_1 on deck overgrown by several demosponges. (D–H) SEM images of spicules from i139_A. (D–E) Calthrops. (F–G) Amphiasters. (G) Microrhabd I.

This species is often overgrown with sponge epibionts such as *Jaspis* sp., *H. peocillastroides*, *Vulcanella aberrans* (*Maldonado & Uriz, 1996*) or *C.* cf. *cranium.*

### Genetics
Folmer COI was sequenced from i688 (ON130559) but only the second part of COI was sequenced for i771 (ON130560); 28S (C1-C2) was obtained only from i688 (ON133874).

### Taxonomic remarks
See below discussion of *P. ovisternata.*

*Pachastrella ovisternata* Lendenfeld, 1894
(Figs. 2B and 41, Table 18)

## Material examined

UPSZMC 190905, field#i219_A, MaC (SO), St. 8 (INTEMARES1019), 240 m, rock dredge, coll. J. A. Díaz; UPSZMC 190906-190907, field#i278_B-i278_D, MaC (AM), St. 58 (INTEMARES1019), 139 m, beam trawl, coll. J. A. Díaz; UPSZMC 190908, field#i394_1, MaC (EB), St. 165 (INTEMARES1019), rock dredge, 312 m, coll. J. A. Díaz; UPSZMC 190909, field#i628, MaC (AM), St. 30 (INTEMARES0720), 204 m, rock dredge, coll. J. A. Díaz; UPSZMC 190902, field#i808, small mount west AM, St. 11 (INTEMARES0820), 263 m, ROV, coll. J. A. Díaz; UPSZMC 190911, field#i818_1, small mount east off EB, St. 20 (INTEMARES0820), ROV, 725 m, coll. J. A. Díaz; UPSZMC 190912, field#i820_1, MaC (EB), St. 21 (INTEMARES0820), 425–733 m, ROV, coll. J. A. Díaz

## Outer morphology

Massive, irregular, up to 40 cm in diameter (i808, Fig. 41A), often with several pockets which may develop in deep cavities, these pockets and cavities are covered with minute openings (Figs. 2B and 41B); the nature of these openings in the cavities is uncertain. Surface and choanosome whitish in life and after ethanol fixation; strongly hispid surface; hard consistency; diffuse cortex ca. 0.5 cm thick.

## Spicules

Calthrops (Figs. 41C–41D) in a wide size range, with the rhabdome slightly longer and straighter than the clads. In the larger ones, curvature of the clads is more pronounced; rarely, having only three actines, showing teratogenic modifications or having their actines bifurcated. Actine measurements: 44-1,102/7-142 μm.

Meso/dichotriaenes (Figs. 41E–41G), common, rhabdome and epirhabdome are straight and fusiform, the former slightly longer than the latter. Epirhabdome may be absent or poorly developed. Measuring: rhabdome 47-129/5-16 μm, epirhabdome 27-108/4-15 μm. Protoclads are disposed in a 120° angle with the rhabdome, while deuteroclads are in a 90° angle with the rhabdome. Teratogenic modifications are frequent. Protoclads measuring 13-66/3-17 μm, deuteroclads 4-140/4-14 μm.

Oxeas (not shown), long and fusiform, slightly curved, mostly broken, measuring 980-5,931/7-50 ($N = 15$) μm.

Amphiasters (Fig. 41H), abundant, with well developed microspined actines, measuring 7–22 μm.

Microstrongyles I (=microrhabds) (Fig. 41I), spiny, very abundant, centrotylote or not, elongated straight or slightly curved to spherical, measuring 8-21/1-7 μm.

Microstrongyles II (=microrhabdose streptasters), very scarce, spiny, thin, elongated and curved, 20-47/1-2 μm.

Díaz et al. (2024), *PeerJ*, DOI 10.7717/peerj.16584

**Table 18  Spicule measurements of *Pachastrella monilifera* and *Pachastrella ovisternata*,** given as minimum-mean-maximum for total length/minimum-mean-maximum for total width; all measurements are expressed in μm. Balearic specimen codes are the field#.

| Material | Depth (m) | Oxeas (length/width) | Calthrops (length/width of actine) | Mesodichotriaenes Rhabdome (length/width) Epirhabdome (length/width) Protoclade (length/width) Deuteroclade (length/width) | Microrhabds (length/width) | Amphiasters (length) |
|---|---|---|---|---|---|---|
| *P. monilifera* i139_A EB | 128 | broken | 43-183-522/ 7-26-77 | – | I. 12-16-18/3-4-6 II. 23-33-56/1-1-2 | 9-13-19 |
| *P. monilifera* i153_3A EB | 109 | 972-2,326- 3,907/ 10-17-24 (*N* = 11) | 23-168-779/ 4-25-106 | – | I. 12-17-21/2-4-6 II. 17-27-32/1-1-2 (*N* = 5) | 8-11-16 |
| *P. monilifera* i688 EB | 117 | 3,129/24 (*N* = 1) | 107-300-724/ 8-37-112 | – | I. 8-13-18/2-3-5 II. - | 7-9-12 (*N* = 9) |
| *P. ovisternata* i219_A SO | 240 | n.m. | 44-242-1,036/ 7-34-142 | Rh: 52-67/6-8 (*N* = 3) EpiR: 40-55/7-9 (*N* = 3) Pr: 18-32-44/4-8-12 (*N* = 14) Dt: 25-51-82/4-7-10 (*N* = 14) | I. 9-14-17/4-5-7 II. 23-29-35/1-2-2 (*N* = 5) | 8-12-22 |
| *P. ovisternata* 278_B AM | 139 | 2,691/ 22 (*N* = 1) | 59-301-1,102/ 8-38-140 | Rh: 49-82-129/5-9-16 (*N* = 7) EpiR: 27-59-108/4-8- 15 (*N* = 7) Pr: 16-24-30/4-9-17 (*N* = 13) Dt: 4-62-99/4-8-15 (*N* = 13) | I. 9-15-21/1-3-4 II. 29 − 42/1 − 2 (*N* = 2) | 7-12-16 |
| *P. ovisternata* i278_D AM | 139 | 980-2,244- 5,931/ 7-18-35 (*N* = 10) | 49-292-726/ 7-36-99 | Rh: 50-65-79/5-7-10 (*N* = 7) EpiR: 31-46-60/4-6-8 (*N* = 6) Pr: 13-28-48/3-8-15 (*N* = 11) Dt: 26-51-140/4-7-14 (*N* = 11) | I. 10-15-21/2-3-4 II. 20-29-41/1-1-2 (*N* = 11) | 8-14-19 |

**Table 18** (*continued*)

| Material | Depth (m) | Oxeas (length/width) | Calthrops (length/width of actine) | Mesodichotriaenes Rhabdome (length/width) Epirhabdome (length/width) Protoclade (length/width) Deuteroclade (length/width) | Microrhabds (length/width) | Amphiasters (length) |
|---|---|---|---|---|---|---|
| *P. ovisternata* i628 AM | 240 | 3,222/ 29 (*N* = 1) | 62-340-860/ 7-49-140 | Rh: 79 (*N* = 1)/8-9-12 (*N* = 5) EpiR: broken Pr: 26-32-43/5-7-9 (*N* = 5) Dt: 23-39-59/5-6-7 (*N* = 5) | I. 11-16-18/3-5-6 II. 36-47/1-2 (*N* = 2) | 8-14-19 (*N* = 20) |
| *P. ovisternata* i808 AM | 263 | 3,345-3,587/ 31-50 (*N* = 3) | 87-294-793/ 11-39-115 | Rh: 47-58-68 (*N* = 8)/8-12-15 (*N* = 12) EpiR: 39-56-76/8-11-13 (*N* = 6) Pr: 25-39-66/8-11-15 (*N* = 12) Dt: 28-49-73/5-9-12 (*N* = 5) | I. 8-13-18/3-5-6 II. - | 10-13-20 (*N* = 20) |

**Notes.**

Rh, rhabdome; Cl, clad; -, not found/not reported; EB, Emile Baudot; AM, Ausias March; SO, Ses Olives.

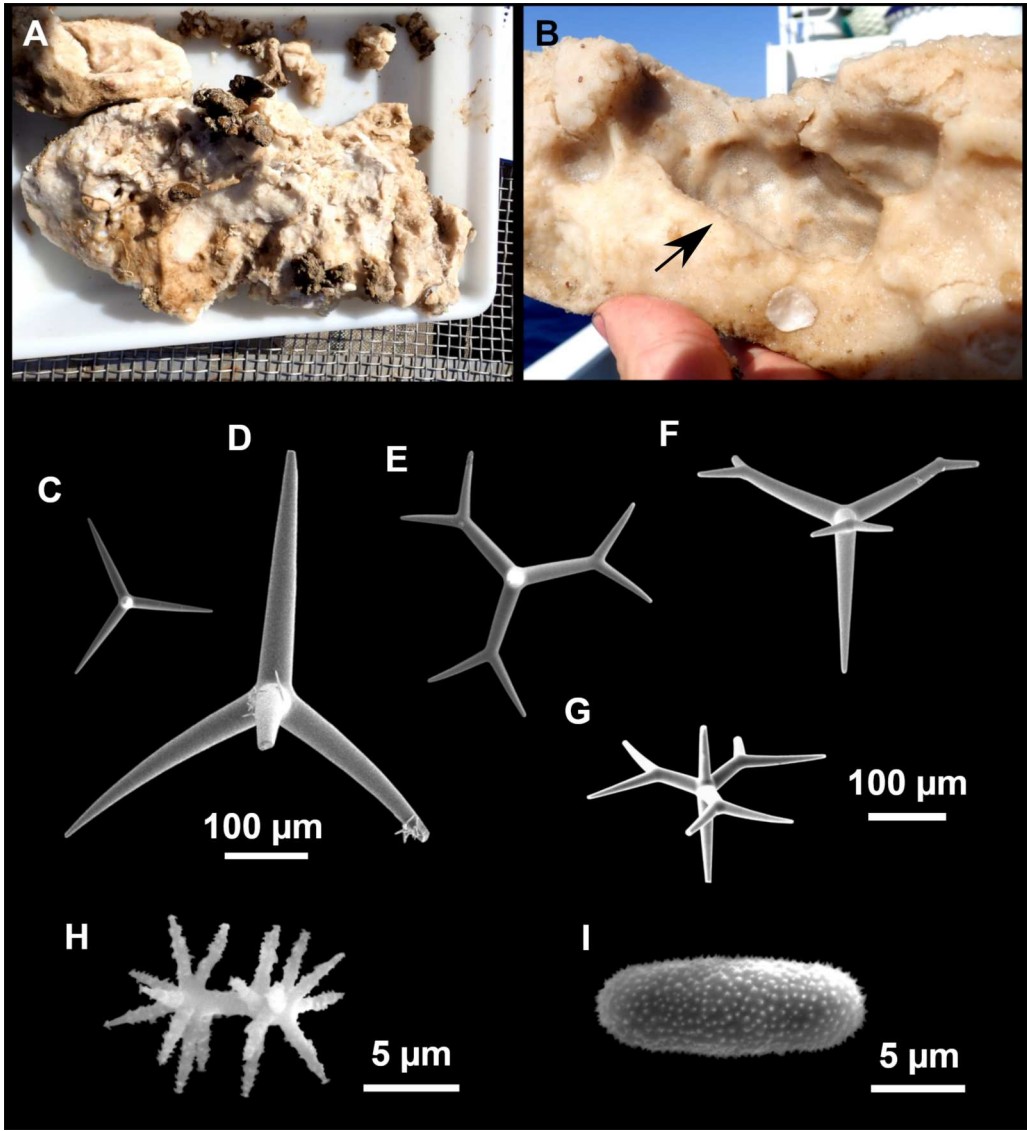

**Figure 41** *Pachastrella ovisternata* **Lendenfeld, 1984.** (A) Habitus of i808 on deck. (B) Habitus of i628 on deck, showing the minute openings placed in large depressions (arrow). (C–I) SEM images of spicules from i808. (C–D) Calthrops. (E–F) Dichotriaenes. (G) Mesodichotriaene. (H) Amphiaster. (I) Microstrongyle I.

## Ecology and distribution

Essentially found in the MaC, below the photic zone (139–733 m). Due to its large size, the species provides habitat to many other invertebrates: sea urchins, brittle stars, shrimps and other small crustaceans, sponges: an unidentified Calcarea, *Haliclona* spp., *Jaspis* sp., *C.* cf. *cranium*, *P. compressa* or *Vulcanella gracilis* (see below).

## Genetics

COI from i808 (ON130558) was obtained, while 28S (C1-C2) was obtained for i808 and i820_1 (ON133876 and ON133875).

## Taxonomic remarks on *Pachastrella monilifera* and *Pachastrella ovisternata*

To discriminate both species, three spicule characters were particularly used: *P. ovisternata* has i) large calthrops with straight actines with irregular endings (*vs.* curved regular actines in *P. monilifera*), (ii) meso/dichotriaenes (absent in *P. monilifera*) and (iii) streptasters as a mix of amphiasters (the majority) along with metasters with less actines (absent in *P. monilifera*). Our specimens of *P. monilifera* also had less abundant amphiasters as well as slightly thinner microstrongyles I (Table 18). Macroscopically, *P. monilifera* tended to be smaller and much less hispid than *P. ovisternata*. In the MaC both species seem to have overlapping by different bathymetric distributions: *P. monilifera* was always collected in the photic zone, associated to coralligenous algae bottoms while *P. ovisternata* appears below the photic zone and reaches bathyal depths, down to 725 m. This is only the second record of *P. ovisternata* in the Mediterranean Sea after its finding off Malta (*Sitjà, 2020*) at 239 m and 752 m. The mesophotic distribution of *P. monilifera* agrees with most of the previous Mediterranean records (*Topsent, 1934*; *Pulitzer-Finali, 1972*) while it also has been commonly reported from upper bathyal depths in the North Atlantic (*Cárdenas & Rapp, 2012*), down to 2,165 m off Essaouira (=Mogador), Morocco (*Topsent, 1928*).

However, we have failed to separate both species genetically: COI and short fragments of 28S (C1-C2) were identical for our specimens of *P. monilifera* and *P. ovisternata*. It is known that in some groups COI shows no variation between sister species, even when apparently strong morphological synapomorphies are present; see the case of *Heteroxya corticata/H. beauforti* (Morrow et al., 2019) or *Thenea muricata/T. valdiviae* (*Cárdenas & Rapp, 2012*). Also, we are missing the 28S D2 fragment, which is the most variable. Our COI tree (Fig. S2) also suggests that *P. ovisternata* is polyphyletic. A revision of the type material is necessary now to decide which *P. ovisternata* is the right one. In any case, this suggests that there is probably another undescribed *Pachastrella* in the Atlanto-Mediterranean region, and that more specimens need to be sequenced to untangle the matter. However, there may be a second explanation: the discrepancy in spicular set, spicule sizes and abundances may just be a depth-related artifact caused by differences in nutrient content or nutrient availability. Silica concentration is known to directly affect the spiculogenesis in demosponges, and proved to be the cause of the presence/absence of isochelae and desmas in *Crambe crambe* (*Maldonado et al., 1999*). Depth directly affects the silica concentration of the water mass through several processes, but mostly because at the photic zone it is disputed by diatoms. This could explain why the deep species *P. ovisternata* develops mesodichotriaenes and more amphiasters and why "*P. ovisternata*" morphotype only appears below the photic zone while "*P. monilifera*" *morphotype* is always shallower. Future studies should address this question using molecular markers with greater resolution, like microsatellites or SNPs.

Hastate oxeas are sometimes reported in *Pachastrella* (*Maldonado, 2002*; *Cárdenas & Rapp, 2012*). While some authors question their origin and point them as foreign, others

consider them as proper (*Maldonado, 2002*). We have found both hastate and regular oxeas in most of *Pachastrella* spp. specimens. However, usually those specimens also harbor several haliclonid as epibionts which could act as the source of the oxeas. In order to discern whether they are proper or exogenous, we measured the oxeas found in *Pachastrella* spp. choanosomes and compared them to those of *Haliclona* spp. growing on the same individuals. Hastate oxeas measured 100-195-310/3-6-12 ($n = 19$) µm in the choanosome of i687 (*P. monilifera*) and 114-236-314/3-8-14 µm in i687_1 (*Haliclona* cf. *poecillastroides* epibiont); 199-316/8-13 µm ($n = 2$) in the choanosome of i628 (*P. ovisternata*) and 281-312-349/7-13-18 µm in i628_1 (*H. poecillastroides* epibiont); 263-279-294/7-11-15 µm ($n = 4$) in i808 and 202-281-334/11-13-16 µm in i808_1 (*H. poecillastroides* epibiont). Besides, in the choanosome of *P. monilifera* specimen i824_1 smaller haliclonid spicules were found, 148-158-172/4-6-8 µm ($n = 4$), *versus* 127-157-181/2-5-7 µm for the oxeas from i824_2, a whitish *Haliclona* sp. epibiont found on that same specimen. The coincidence in sizes and the prevalence of the association between *Pachastrella* and *Haliclona* spp. (mostly *H. poecillastroides*) seems to indicate that both regular and hastate oxeas are foreign, in opposite view to *Maldonado (2002)*. Having those spicules in the choanosome does not mean that they are proper of *Pachastrella*, yet other clearly non-pachastrellid spicules (like sterrasters, clionid microscleres, bubarid vermicular oxeas and *Jaspis* oxyasters and microxeas) have been found in the choanosome of our *Pachastrella* spp. These species act as substrate for many sponges, such as *H. poecillastroides*, *Jaspis* sp., *Craniella* cf. *cranium*, *etc*. Foreign spicules which end up in the choanosome are perhaps filtered by the aquiferous system of the host and then incorporated in its own body, or possibly actively stolen from epibionts. It could also be the case that *Pachastrella* kills the epibionts by overgrowing them and then uses its skeleton as a substrate. Epibiont interactions are widespread in deep-sea sponge communities, and may occur in other massive demosponges, a fact that could explain the presence of hastate oxeas in species like *N. amygdaloides* or *C. tripodaria* as well, and their recurrence in the tetractinellid literature.

*Genus Discodermia* Du Bocage, 1869
*Discodermia polymorpha* *Pisera & Vacelet, 2011*
(Fig. 42, Table 19)

**Material examined**
UPSZMC 190824-190825, field#i141_1-i141_2, MaC (EB), St. 51 (2018), 128 m, beam trawl, coll. F. Ordines; UPSZMC 190829, field#i277_1, MaC (AM), St. 58 (INTEMARES1019), 139 m, beam trawl, coll. J. A. Díaz; UPSZMC 190833-190834, field# i320 and field#i321, MaC (EB), St. 124 (INTEMARES1019), 152 m, beam trawl, coll. J. A. Díaz; UPSZMC 190837, field#i606, MaC (AM), St. 26 (INTEMARES0720), 130 m, beam trawl, coll. J. A. Díaz.

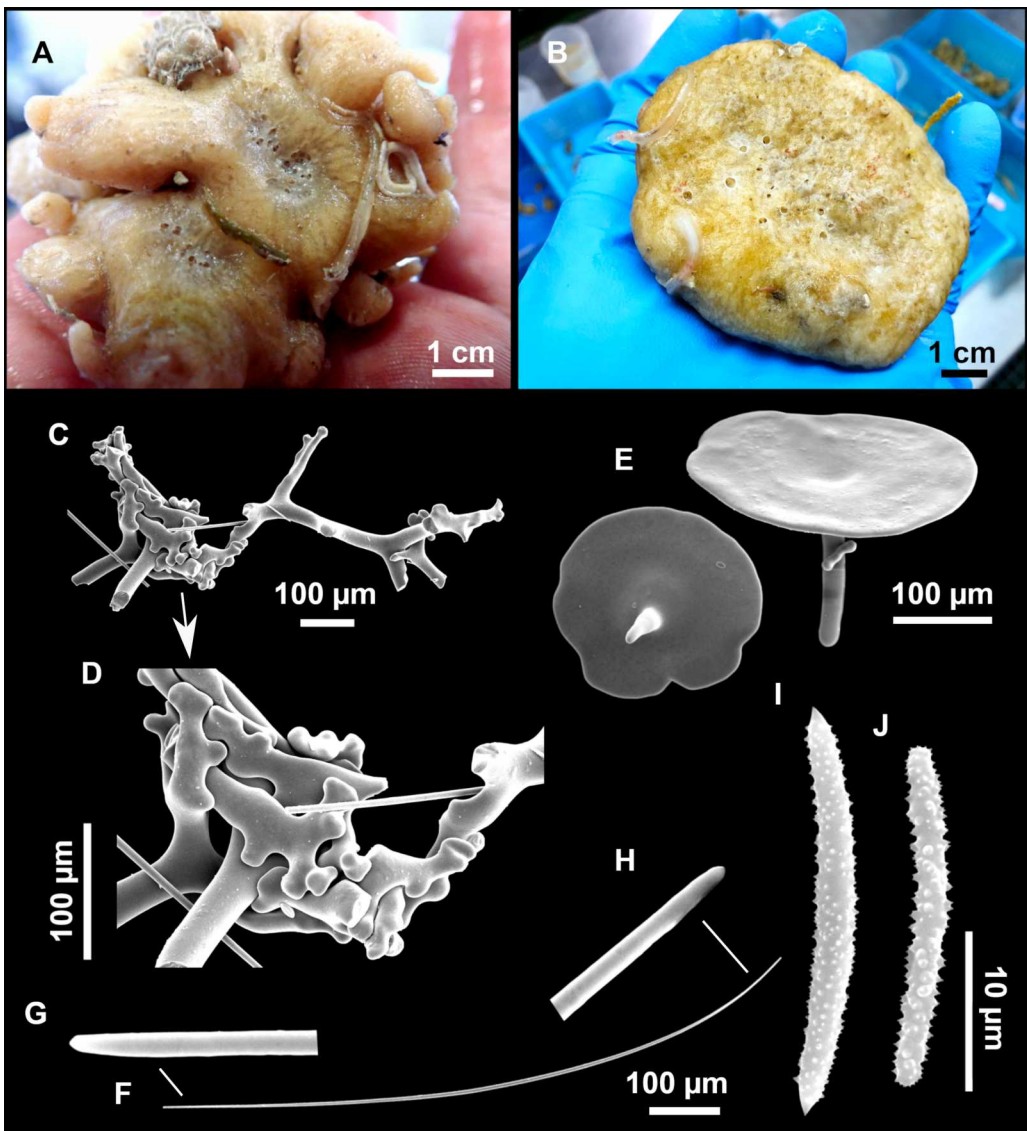

**Figure 42** *Discodermia polymorpha Pisera & Vacelet, 2011*. (A) Habitus of i606 on deck. (B) Habitus of i320 on deck. (C) Tetraclone desmas with (D) detail of the zygosis. (E) Discotriaenes. (F) Diactine with detail of the tips (G–H). (I) Acanthomicroxea. (J) Acanthomicrorhabd.

## Outer morphology

Extremely variable in shape, some have a slightly ramose tendency, some show bulbous processes and protuberances (Fig. 42A), others are spherical or cup-shaped (Fig. 42B). Sizes about 4–9 cm in diameter. Consistency hard and crumbly; surface smooth. Subdermal canals commonly visible by transparency on the surface (Fig. 42A). Beige with brownish shades on deck (Figs. 42A–42B), whitish after ethanol fixation. Uniporal to cribiporal oscules grouped on the top surface (Figs. 42A–42B), about 1–4 mm in diameter, with a patent thin membrane.
**Table 19 Spicule measurements of *Discodermia polymorpha* and related species, given as minimum-mean-maximum for total length/minimum-mean-maximum for total width; all measurements are expressed in μm.** Balearic specimen codes are the field#. Specimens measured in this study are in bold.

| Species | Depth (m) | Diactines (length/width) | Discotriaenas Rhabdome (length/width) Cladome (diameter) | Desmas (length/width) | Acanthomicrorhabds (length/width) | Acanthomicroxeas (length/width) |
|---|---|---|---|---|---|---|
| *D. polymorpha* i141_1 **EB** | 128 | 724-929-1,152/3-5-8 | Rh: 63-93-125/18-22-28 Cl: 151-261-319 | 264-422-651/28-35-46 | 13-21-31/1-2-3 | 20-40-51/2-2-3 |
| *D. polymorpha* i141_2 **EB** | 128 | 673-905-1,126/3-5-8 | Rh: 71-96-110/16-25-48 Cl: 207-247-295 | 255-393-582/38-42-54 | 15-22-27/2-2-3 | 23-38-48/2-2- 4 |
| *D. polymorpha* i321 **EB** | 152 | 683-1,023-1,367/3-6-8 | Rh: 85-97-105/8-14-20 Cl: 155-262-347 | 235-377-520/13-29-55 | 13-20-28/2-2-3 | 43-51-65/2-2-3 |
| *D. polymorpha* i606 **AM** | 130 | 508-945-1,446/5-7-10 | Rh: 63-115-150($N=5$)/11-19-26 Cl: 157-267-358 | 295-405-508/34-40-46 ($N=9$) | 16-25-48/1-2-3 | 48-54-61/1-3-4 |
| *D. polymorpha* Marseille (caves) (*Pouliquen, 1972*) | 4-25 | -/5-8 | Rh: 30-50/- Cl: 200-300 | -/40 | 10-25/2 | 40-60/3 |
| *D. polymorpha* Western Mediterranean and Aegan Sea (*Pisera & Vacelet, 2011*) | Littoral caves and deep sea (360 m) | – | Rh: 60-65/- Cl: 174-366/- | 370-718 μm in diameter | 13-44-37/2-5-4 | 25-68/2-4 |
| *D. polydiscus* Canary Islands (caves) (*Cruz, 2002*) | – | 1,200-1,500/- | Rh: - Cl: 160-440 | -/30-60 | 12-20 | 36-60 |
| *D. polydiscus* holotype Saint Vincent, Caribbean (*Bowerbank, 1869*) | – | Present | – | – | – | – |
| *D. ramifera* holotype, Azores (*Topsent, 1892*) | 318 | Present | Cl: 300 Cl: - | Desmas rays full of tubercles in the extremities | 20-25 | 40-45 |

**Notes.**

-, not found/not reported.

EB, Emile Baudot; AM, Ausias March.

## Skeleton

Cortex formed by a layer of discotriaenes, with their rhabdomes pointed inwards and the discs disposed tightly packed. The discs lay on a layer composed by a dense aggregation

of microrhabds on a collagenous membrane. Choanosome cavernous, fleshy, with tight fascicles of diactines disposed perpendicular to the cortex and a network of desmas. Many incorporated exogenous particles. Microxeas embedded in the walls of the choanosomal chambers.

### Spicules

Tetraclone desmas (Figs. 42C–42D), with smooth clones, zygomes tuberculated, 235-651/13-55 μm.

Discotriaenes (Fig. 42E), concave disc, highly polymorphic: from circular to elliptical, with regular or irregular margins or with several lobules, diameter of 151-358 μm. Rhabdome is short, triangular, sometimes with a swelling below the disc and with a slightly blunt tip, measuring 63-150/8-48 μm.

Diactines (Fig. 42F), thin, widely curved, with variable tips (rounded, tylote, subtylote or blunt, Figs. 42G–42H), measuring 508-1,446/3-10 μm.

Acanthomicroxeas (Fig. 42I), straight to slightly curved, triangular, with sharp tips, 20-65/1-4 μm.

Acanthomicrorhabds (Fig. 42J), curved, with round ends, entirely covered with microspines, measuring 13-48/1-3 μm.

### Genetics

COI obtained for i606 and i321 (ON130549 and ON130550); 28S (C1-C2) obtained for i606, i321 and i320 (ON133891, ON133890 and ON133889).

### Ecology and distribution

Mesophotic species, always collected on the upper slopes of the AM and the EB, just below the photic zone, between 128–152 m depth. Occasional sponge epibionts are a red *Timea* sp., *Hexadella* sp. and *Jaspis* sp. In addition, agglomerations of diatoms attached to the discotriaenes were common, and are probably giving the characteristic brownish shades to their skin.

### Taxonomic remarks

*Discodermia polydiscus* (Bowerbank, 1869), the type species of *Discodermia,* was described from Saint Vincent, in the Caribbean, as "*small, unequally developed, cup-shaped sponge*" with tetraclone desmas, discotriaenes, microrhabds and microxeas. Diactinal spicules were not originally described but found later when revising the holotype (*Pisera & Lévi, 2002*). After its description, the species was reported in Portugal (*Du Bocage, 1869 (1970)*, Canary Islands (*Cruz, 2002*) and the Mediterranean Sea (*Vacelet, 1969*; *Pouliquen, 1969*; *Pouliquen, 1972*; *Voultsiadou, 2005*, Table 19). The assignment of the northeast Atlantic and Mediterranean specimens to *D. polydiscus* was never satisfactorily argued, especially considering the distance between the type locality (Caribbean) and the Mediterranean Sea. Finally, *Pisera & Vacelet (2011)* described a new species, *D. polymorpha,* to include all the previous *D. polydiscus* Mediterranean records, from shallow caves to mesophotic depths in the Aegean Sea (210–360 m). The identity of the remaining northeast Atlantic records of *D. polydiscus* (*Du Bocage (1869 (1970))*; *Cruz, 2002*) are probably inaccurate but

require proper revision of this material. Because *D. polymorpha* had extremely variable macroscopic and spicular characters, *Pisera & Vacelet (2011)* cannot exclude that it could represent a species complex. Our sequences match previous *D. polymorpha* sequences from its type locality, the 3PP cave, in La Ciotat, France (*Chombard, Boury-Esnault & Tillier, 1998*; *Cárdenas et al., 2011*). Our 28S (C1-C2) is 100% identical, while our COI has a 1 bp difference with cave specimens from the type locality. Interestingly, all cave specimens sequenced so far from Medes Islands (Spain), around Marseille (France) and Dugi Otok (Croatia) share the holotype COI (*Pisera & Vacelet, 2011*) which would suggest that western Mediterranean cave populations are somewhat genetically separated from the Balearic mesophotic populations. Several morphological traits observed in our material may support this possibility. *D. polymorpha* from caves were described as small, 1–2 cm in diameter while our specimens are much larger, reaching eight cm in diameter (i320). Also, *D. polymorpha* overall shape was described as "*nearly spherical to irregular masses with protuberances*" (*Pisera & Vacelet, 2011*). This character was actually used to differentiate *D. polymorpha* from the North Atlantic *Discodermia ramifera Topsent, 1892* and *D. polydiscus* which are ramose and cup-shaped to irregular, respectively. However, the morphologies of our mesophotic specimens cover all these shapes: spherical or subspherical (i141_1, i141_3), ramose (i277, i321), and cup-shaped (i320, Fig. 42B). Also, *D. ramifera* and *D. polydiscus* have simple oscules (*i.e.,* uniporal) on elevations, very much like in specimens of *D. polymorpha* from caves that we observed on underwater pictures (courtesy of P. Chevaldonné), looking like white warts. The oscula morphology of our specimens is more diverse and complex than this, with uniporal to cribriporal oscules, usually placed on depressed or flat areas located apically (Fig. 42A). One reason for that may be that we examined live specimens; once fixed in ethanol, many of these oscule groups contract and become less visible to invisible. However, the clear difference between the warty uniporal oscules in live cave specimens *versus* our oscule complexes remains and may be due to difference in depth, habitat differences such as water flow/currents. Similar oscule differences were also observed between cave *Caminella* and mesophotic ones, where oscule walls disappeared in deeper specimens (*Cárdenas et al., 2018*): reduced water flow in caves may stimulate the formation of raised oscules.

Another significant difference of our specimens is the presence of diactines in relatively high abundance, packed together on choanosomal tracks, a spicule only previously reported by *Pouliquen (1972)* but not mentioned by *Pisera & Vacelet (2011)*. Again, the abundance of this spicule may be linked to the mesophotic depths of our specimens. Interestingly, according to *Carvalho et al. (2020)*, diactines are always present in the Northeast Atlantic *D. ramifera,* found at 98–673 m in the Azores (*Topsent, 1892*) the Gulf of Cadiz (*Sitjà et al., 2019*) and the Great Meteor seamount (*Carvalho et al., 2020*). The similarity of spicular set, spicule morphometrics, macroscopic shape and deep-sea habitat of *D. ramifera* and *D. polymorpha* indicate that both are phylogenetically closely related. Indeed, specimens of *D. ramifera* from the Azores (COI: MW000696; 28S: MW006540-6541) are clearly different genetically but sister to *D. polymorpha* (COI: 4–5 bp. difference; 28S (C1-D2): 13–14 bp difference), and are actually 1 bp. closer (with COI) to our mesophotic specimens than to

the cave ones, which suggests that the shallow cave populations appeared after the deep ones.

Family Theneidae Gray, 1872
Genus *Thenea* Gray, 1867
*Thenea muricata* (Bowerbank, 1858)

## Material examined

UPSZMC 190967, field#i232_1, St. 36 (INTEMARES1019), MaC (SO), beam trawl, 619 m.

## Outer morphology

Spherical to hemispherical sponges with basal root-like projections. Sizes are variable between areas: 0.3–1.5 cm in maximum diameter at the seamounts off MaC and 2–5 cm in maximum diameter in some fishing grounds, especially in some stations found north off MeC. Hispid surface, slightly compressible. Colour whitish on deck and after preservation. Apical, circular oscula, from <0.1 mm to about 0.4 cm in the larger ones. Equatorial sieved poral area, barely distinguishable in smaller specimens. The ectosome is visible to the naked eye, measuring about 0.1 mm in thickness.

## Genetics

COI and 28S (C1-C2) were obtained from i232_1 (ON130569 and ON133888).

## Ecology

The species is widely present in the detrital mud stations of both the MaC and the fishing grounds of the Mallorca and Menorca shelf. Large aggregations of the species are annually found in a fishing ground North of Mallorca (MeC), constituted by individuals that reach five cm in diameter. However, most of the specimens collected in the MaC seamounts and the adjacent bottoms were much smaller, barely reaching one cm in diameter. This may be caused by nutrient differences between both areas, and be related to the fact that the MeC is narrower and shallower and more influenced by the strong storms from the north. The species is usually epyphyted by *Epizoanthus* sp., a relationship more common in larger specimens.

## Taxonomic remarks

The species is well known and documented in the literature. The spicular complement agrees with those provided previously (*Cárdenas & Rapp, 2012*). COI and 28S (C1-C2) are a 100% match with already-existing sequences for this species.

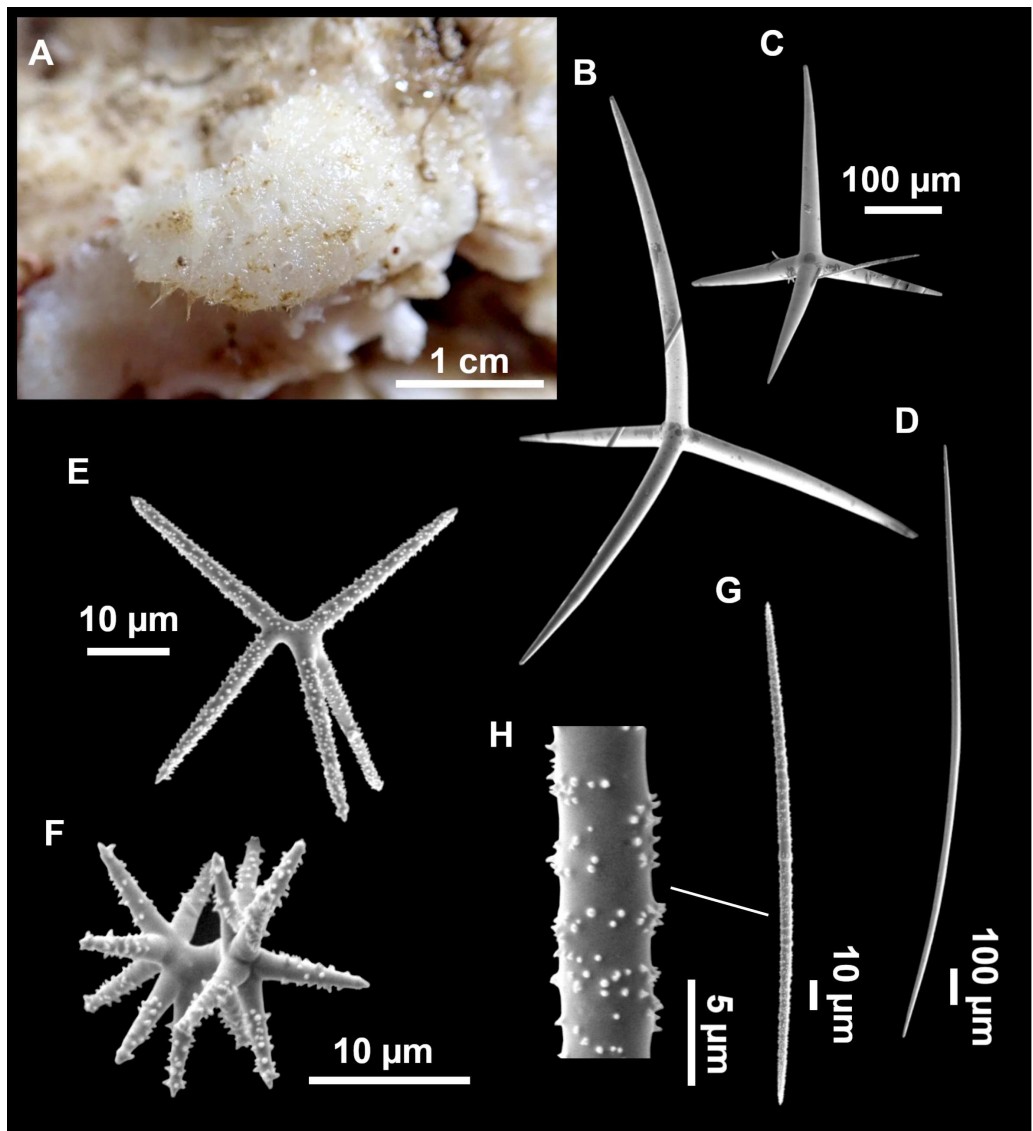

**Figure 43   *Poecillastra compressa* (*Bowerbank, 1866*), specimen i808_9.** (A) Habitus on deck. (B–C) Pseudocalthrops. (D) Oxea. (E) Plesiaster. (F) Metaster. (G) Microxea with (H) details of the spines.

Family Vulcanellidae *Cárdenas et al., 2011*
 Genus *Poecillastra Sollas, 1888*
*Poecillastra compressa* (*Bowerbank, 1866*)
(Fig. 43)

## Material examined

UPSZMC 190941-190942, field#i808_9-#i809, small mount west of AM (MaC), St. 11 (INTEMARES0820), 263 m, ROV, coll. J. A. Díaz.

## Ecology and distribution

Very abundant species, at seamounts it was found associated with rhodolith beds at the EB and AM summits (98–150 m), and in greater depths living on hard and gravel bottoms (down to 511 m). On trawl fishing grounds, it was found on the shelf break and upper slopes, on a more restricted depth range (104–257 m).

## Genetics

COI was obtained from i808_9 and i809 (ON130553 and ON130554), as well as 28S (C1-C2) for i809 (ON133870).

## Taxonomic remarks

Spicule size/morphology perfectly agree with previous descriptions (*Cárdenas & Rapp, 2012*). COI and 28S (C1-C2) are 100% match with already-existing sequences for this species. This deep-sea species is common throughout the Eastern Atlantic and Mediterranean Sea (*Cárdenas & Rapp, 2012*; *Cárdenas & Rapp, 2015*). It can present different external colors, from white, grayish, yellow to orange (*Cárdenas & Rapp, 2012*). Samples UPSZMC 190941-190942 were of the white morphotype but other examined specimens were also orange, yellowish or grayish. *Cárdenas & Rapp (2012)* suggested that color was a morphological variety not related to light irradiance because of bicolor specimens and colored specimens at depths deeper than 100 m. However, in the Balearic islands, orange to yellowish specimens are always found at mesophotic depths, 100–150 m, while non-colored specimens (white and grayish), are present in both mesophotic and aphotic zones. This seems to indicate that colored specimens are conditioned by light irradiance at the bottom while non-colored specimens are more widespread. In other sponge species like *Suberites domuncula* (Olivi, 1792), blue, orange and yellowish colorations are given by carotenoids acquired from bacteria and microalgae (*Cariello & Zanetti, 1981*; *Maia et al., 2021*), which may be lost or not produced when inhabiting deeper waters (deeper *S. domuncula* specimens are grayish, personal observation). In fact, in marine invertebrates, orange and yellow colors that cannot be produced intrinsically are usually linked to carotenoids accumulated in the body and acquired from the medium through feeding on photosynthetic microorganisms. In some cases, colors are lost when inhabiting shaded or cryptic spaces or deeper waters (*Bandaranayake, 2006*). In the case of *P. compressa*, the loss of coloration when the sponge lives in aphotic habitats may be caused by the lack of light or photosynthetic microorganisms in the surrounding waters. However, this does not explain why there are white and bicoloured individuals in the mesophotic zone. Perhaps, those specimens are placed in cryptic areas, hidden from the sunlight, or perhaps they do not have the time to acquire the pigments. In any case, those questions should be addressed in future works exploring the pigment contents of *P. compressa*.

Genus *Vulcanella Sollas, 1886*
*Vulcanella aberrans* (*Maldonado & Uriz, 1996*)
(Fig. 44, Table 20)

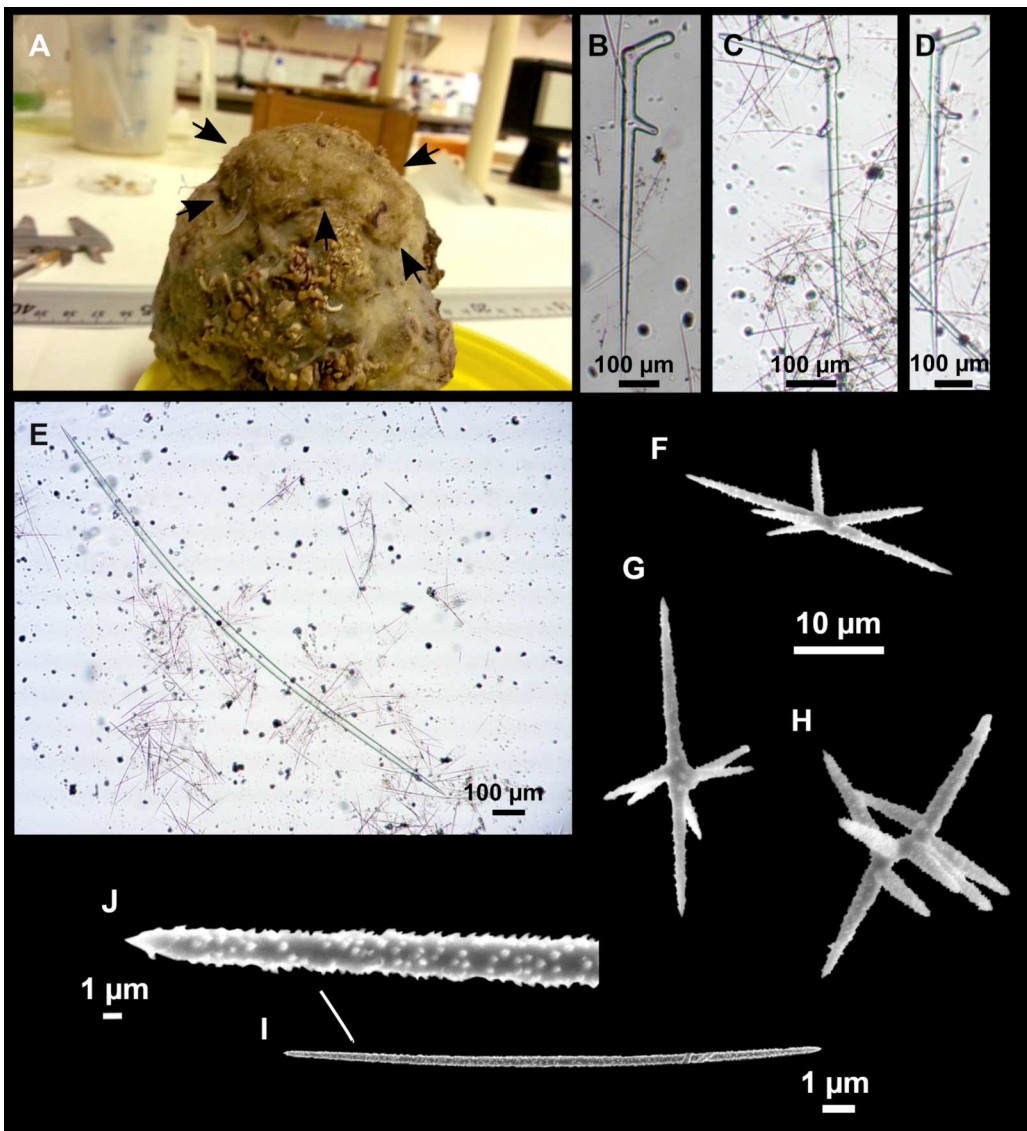

**Figure 44  *Vulcanella aberrans* (*Maldonado & Uriz, 1996*).** (A) Habitus of i139_B1 after ethanol fixation (arrows) growing on a *P. monilifera* specimen (i139_B). (B–D) Irregular plagiotriaenes. (E) Oxeas I and microxeas. (F–H) Metasters to plesiasters. (I) Microxea with (J) detail of the spines.

## Material examined

UPSZMC 190971, field#i139_B1, MaC (EB), St. 51 (INTEMARES0718), 128 m, beam trawl, coll. F. Ordines.

## Comparative material

*Vulcanella aberrans,* paratype, CEAB.BIO.POR.021B, slope of Alboran Island, 70–120 m.

## Outer morphology

Small massive encrusting specimen, subdiscoid (∼4 cm in diameter), with a few foreign pebbles, growing on *P. monilifera* (i139_B). External color is light brown in ethanol, same

**Table 20  Spicule measurements of *Vulcanella aberrans, Vulcanella gracilis* and *Vulcanella* cf. *gracilis* given as minimum-mean-maximum in μm.** Balearic specimen codes are the field#.

| Material | Depth (m) | Oxeas (length/width) | Pseudo-calthrops /calthrops Rhabdome (length/width) Clad (length/width) | Microxeas I (length/width) | Microxeas II (length/width) | Metaster/ plesiasters (length) | Spirasters (length) |
|---|---|---|---|---|---|---|---|
| *Vulcanella aberrans* paratype **CEAB-BIO POR021B** Alboran Island | 70-120 | 602-2,059-2,553/13-33-47 Styles: 674-1,409-1,892/19-28-38 ($N = 4$) | Rh: 596-798/14-24 ($N = 4$) Cl: 110-390/15-25 ($N = 4$) (rare) | 53-165-361/ 1-3-8 | – | 8-17-28 | 10-15-23 |
| *Vulcanella aberrans* **i139_B1 EB** | 128 | I. 1,115-1,580-2,185/11-21-34 (styles, same size) II. 951-1,537/6-10 ($N = 4$) | Rh: 478-591/21-23 ($N = 3$) Cl: 85-275/20-28 ($N = 4$) (rare) | 83-196-371/ 1-3-5 | – | 13-27-43 | 14-18-20 ($N = 4$) scarce |
| *Vulcanella gracilis* **MNHN DCL4082** Southern Italy | 560-580 | between 1,500 and >2,000/16-39-64 | Rh: 481-589-767/38-52-62 ($N = 11$) Cl: 229-336-410/36-51-64 ($N = 11$) | 112-209-326/ 6-9-13 | 80-120-169/ 2-3-6 | 28 ($N = 1$) | 11-19-26 |
| *Vulcanella gracilis* **i303_B AM** | 231-303 | I. 4,550-7,406/9-21 ($N = 3$) II. 1,736-2,663/32-76 ($N = 3$) | Rh: 563-798-1,111/60-70-77 ($N = 5$) Cl: 354-551-661/50-66-72 ($N = 5$) | 155-254-324/ 6-10-12 ($N = 11$) | 94-116-151/ 2-2-3 ($N = 11$) | 12-18-24 ($N = 18$) | 14-16-18 ($N = 4$) |
| *Vulcanella gracilis* **i818_2 EB** | 725 | I. long and thin, broken II. 911-1,884-2,795/29-60-107 ($N = 5$) | Rh: 254-507-717/36-54-70 ($N = 4$) Cl: 181-454-655/24-54-85 ($N = 15$) | 194-262-341/ 9-14-19 | 77-109-171/ 2-4-8 ($N = 20$) | 13-19-25 ($N = 10$) | 10-13-16 ($N = 3$) |
| *Vulcanella* cf. *gracilis* **i279_A AM** | 139 | I. long and thin, broken II. 1,149-1,712-2,565/12-25-53 ($N = 16$) | 114-369-523/13-32-45 | 105-222-351/ 3-6-11 | – | – | – |
| *Vulcanella* cf. *gracilis* **i279_B AM** | 139 | n.m. | n.m. | n.m. | – | 13-20-29 ($N = 8$) | – |

**Notes.**
-, not found/not reported; n.m., not measured; EB, Emile Baudot; AM, Ausias March.

color as *P. monilifera*; internal color is the same. Surface is slightly hispid. Compressible. No oscula or pores observed.

## Spicules

Plagiotriaenes (Figs. 44B–44D), scarce, with malformations such as aborted, missing clads and stumps, affecting both cladome and the rhabdome. Rhabdome measures 478-549-591/21-22-23 μm ($N = 3$), cladi measures 85-191-275/20-24-28 μm ($N = 4$).

Oxeas I (Fig. 44E), abundant, robust and fusiform, slightly or markedly curved, occasionally double bent. Some modified to styles, 1,115-1,580-2,185/11-21-34 µm.

Oxeas II, rare, thin and slender, sometimes centrotylote, smooth, 950-1,269-1,537/6-8-10 ($N = 4$) µm.

Metasters to plesiasters (Figs. 44F–44H), abundant, microspined, short axis and long, robust actines, some of which may be aborted, 13-27-43 µm.

Spirasters, uncommon, may be immature stages of metasters 14-18-20 µm ($N = 5$).

Microxeas (Fig. 44I), thin, slightly curved, some centrotylote, finely microspined (Fig. 41J), sometimes in larger ones a subtle ring-pattern of spines can be seen at the central part. On a wide but continuous size range, 83-196-371/1-3-5 µm.

## Ecology and distribution

Only a single specimen was collected, growing in epibiosis on a large *P. monilifera* specimen, at the mesophotic zone off EB.

## Genetics

No sequences were obtained.

## Taxonomic remarks

The material is assigned to *V. aberrans* essentially on the basis of the presence of malformed plagiotriaenes, of the same size and morphology as those described in the original description, and also similar to those found in the paratype CEAB-BIO POR021, from the closeby Alboran Sea. *Maldonado & Uriz (1996)* describe two categories of microxeas with very close sizes (150–315/3–7 and 65–140/2–2.5) with the largest size being sometimes centrotylote and with spines distributed in a ringed pattern. After measuring a large number of microxeas in our specimen, we concluded that they belonged to one category: size was continuous, with similar morphology, except for larger ones where a weak spiny ringed pattern was sometimes observed. However, this was not always the case, and some large microxeas were identical to small ones. Considering that, the sizes of the microxeas from the paratype and specimen i139_B1 are very similar (53-165-361/1-3-8 µm *versus* 83-196-371/1-3-5 µm). This is the first report of the species in the Balearic Islands after its description in the Alboran sea, slightly extending its distribution in the Western Mediterranean. A potential different population whose status remains to assess, occurs in Norway (*Cárdenas & Rapp, 2012*).

*Vulcanella gracilis* (*Sollas, 1888*)
(Fig. 45, Table 20)

## Material examined

UPSZMC 190974, field#i303_B, MaC (AM), St. 103 (INTEMARES1019), 231–302 m, rock dredge, coll. J. A. Díaz; UPSZMC 190977, field#i818_2, small mount east of EB, St. 20 (INTEMARES0820), 725 m, ROV, coll. J. A. Díaz.

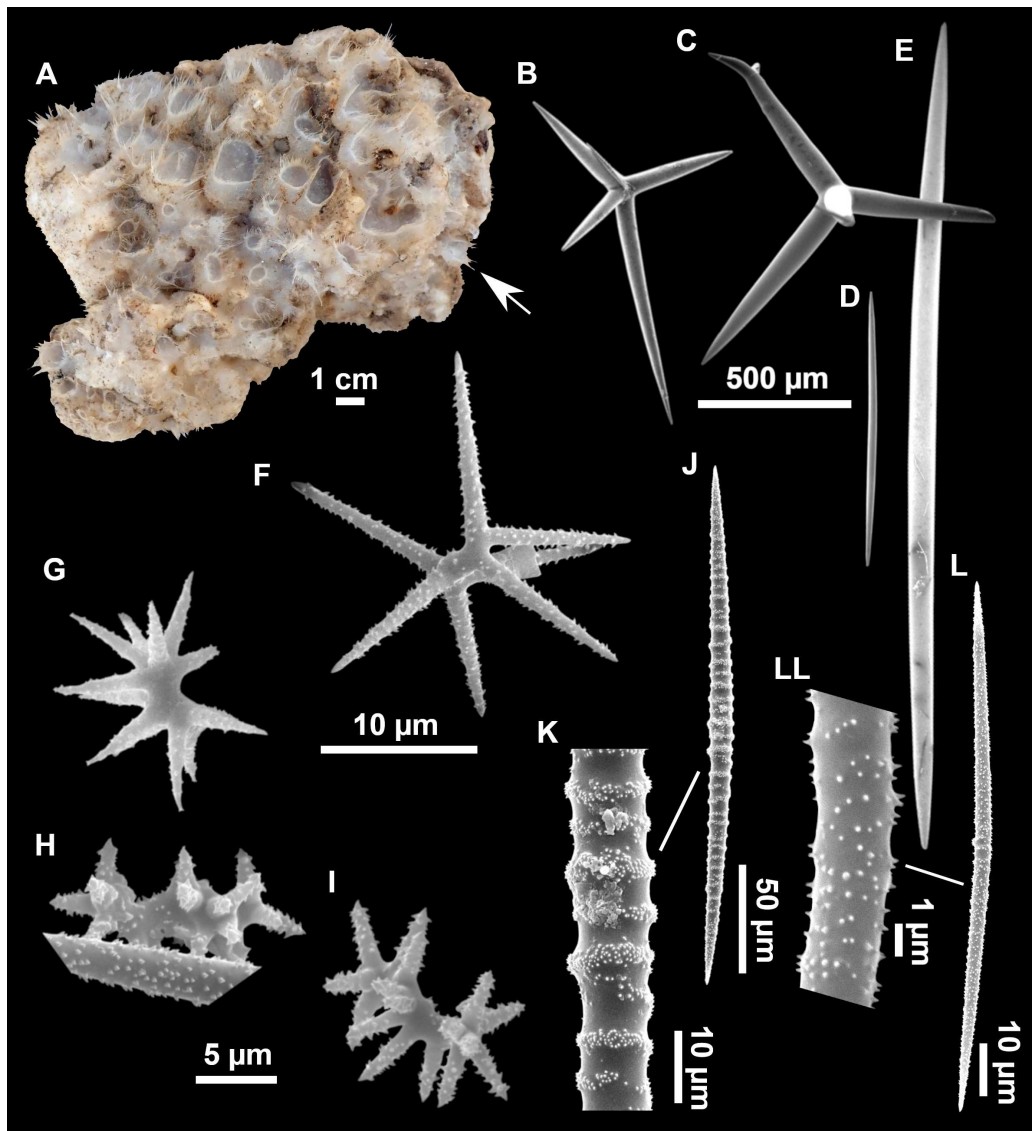

**Figure 45** *Vulcanella gracilis* (*Sollas, 1888*). (A) Habitus of the association between a large *P. ovisternata* specimen (i818_1) acting as substrate for several *V. gracilis* epibionts (arrow), including the specimen i818_2. (B–C) Pseudocalthrops. (D–E) Oxeas. (F) Metaster. (G–I) Metasters to spirasters. (J) Microxea I with (K) detail of the ringed microspination. (L) Microxea II with (LL) detail of the microspination.

## Comparative material

*Vulcanella gracilis,* MNHN DCL4082, Apulian Platform, off Cape Santa Maria di Leuca, southern Italy, 39°33′36″N, 18°25′48″E, 560–580 m, ROV dive 327-6, field#ASC-9/327-6, MEDECO leg1 (Ifremer), 17 Oct. 2007, coll: J. Reveillaud, id: P. Cárdenas, COI: HM592704, 28S: HM592760.

### Outer morphology

Ovoid, massive or encrusting sponges (Fig. 45A), slightly compressible, hispid, up to two cm in diameter. Pale gray in life and after ethanol fixation. Small specimens have a single oscular basket while larger ones can have several (Fig. 45A). Oscular baskets are composed of an atrial sieve with a characteristic smooth thin membrane with openings 0.5–1 mm wide, surrounded by long thin oxeas. i818_2 (Fig. 45A) represents multiple small *V. gracilis* growing on a *P. ovisternata* (i818_1).

### Spicules

Plagiotriaene pseudocalthrops (Figs. 45B–45C) with a very short rhabdome. Some malformations (additional/aborted actines and stylote terminations) may be present. The rhabd measures 254-1111/36-77 µm ($N = 9$) while clads measure 181-655/24-85 µm ($N = 20$).

Oxeas I, around the atrial sieve, very long and thin, smooth, most broken when digested, measuring 4550-7406/9-21 µm ($N = 3$).

Oxeas II (Figs. 45D–45E), fusiform, smooth, thick, measuring 911-1,795/29-107 ($N = 8$).

Metasters (Fig. 45F), spiny, with a short, thin axis and long actines, measuring 12–25 µm.

Spirasters (Figs. 45G–45I), uncommon, with short spined actines and an axis that can be thick or thin, 10–18 µm.

Microxeas I (Fig. 45J), with a very patent and regular spiny annulation clearly visible with the optical microscope (Fig. 45K), straight to gently curved. Overall measuring 155-341/6-19 µm.

Microxeas II (Fig. 45L), microspined but not annulated, (Fig. 45LL) gently curved, 77-171/2-8 µm.

### Ecology and distribution

Very abundant species in the summits of the AM and the EB Seamounts, where it grows on epibiosis with *Hexadella* sp. individuals. It has also been collected growing on rocks on the slopes of the seamounts EB and AM.

### Genetics

Folmer COI and the 28S (C1-C2) fragment were obtained from i818_2 (ON130555 and ON133869).

### Taxonomic remarks

*Vulcanella gracilis* is easily recognizable due to the possession of characteristic strongly tuberculated microxeas, together with a second smaller category of spiny, non annulated, microxeas. Spicule size/morphology (Table 20) agree with previous descriptions from the Mediterranean Sea and measurements from comparative material from Italy (MNHN DCL4082). This is the first record of the species at the Balearic Islands. COI and 28S (C1-C2) are 100% match with already-existing sequences for this species off Cape St. Maria di Leuca, Italy (560–580 m) and off Tangers, Morocco (529 m). However, sequences from the somewhat remote type locality (Cape Verde Islands) and a careful comparison
with the type is warranted in the future. Indeed, the type material (*Sollas, 1888*) did not seem to have rare spirasters as in our specimens (and comparative material), and the microxeas II were smooth and suddenly bent *vs.* spiny and gently bent in our material (and comparative material). Previous sequencing of Cape Verde (*Cárdenas et al., 2018*) or Canary Island specimens (*Caminus xavierae* **sp. nov.**, this study) have shown that the sponge faunas there may be different.

*Vulcanella* cf. *gracilis* (*Sollas, 1888*)
(Fig. 46, Table 20)

### Material examined
UPSZMC 190972-190973, field#i279_A-i279_B, MaC (AM), St. 58 (INTEMARES1019), 139 m, beam trawl, coll. J. A. Díaz.

### Outer morphology
Small hispid basket-like sponges, 0.5–0.7 cm long. Whitish alive and in ethanol discolored to deep purple by neighboring *Hexadella* sp.

### Spicules
Plagiotriaene pseudocalthrops (Fig. 46), abundant, small (Fig. 46B), straight to slightly distorted, many dichotomized and/or with some degree of malformation, 114-369-523/13-32-45 µm.

Oxeas I, atrial oxeas, long and thin, all broken.

Oxeas II (Fig. 46C), fusiform, thick, slightly curved, measuring 1,149-1,712-2,565/12-25-53 µm ($N = 16$); two oxeas thin and centrotylote, may represent immature stages of Oxea I, 670-1,300/7-8 µm ($N = 2$).

Microxeas I (Fig. 46D), slightly curved, centrotylote, with an irregular annulation more patent at the central part of the spicule, measuring 105-222-351/3-6-11 µm.

Streptasters (Figs. 46E–46F), only found in 279_B, rare, spiraster to metaster morphology, 13-20-29 µm ($N = 8$).

### Ecology and distribution
Found as epibiont of *Hexadella* sp., at the upper slope of the AM.

### Genetics
No sequences obtained.

### Taxonomic remarks
The material is assigned to *Vulcanella* cf. *gracilis* because it has similarities but also several spicule differences with *V. gracilis.* The macroscopic morphology is identical, having a clear fenestration surrounded by a ridge of long oxeas. Like *V. gracilis,* it has pseudocalthrop plagiotriaenes, however, (i) they are more abundant, (ii) their rhabdome is proportionally shorter, compared to the cladi, than in *V. gracilis*, and (iii) clads are shorter and thinner,

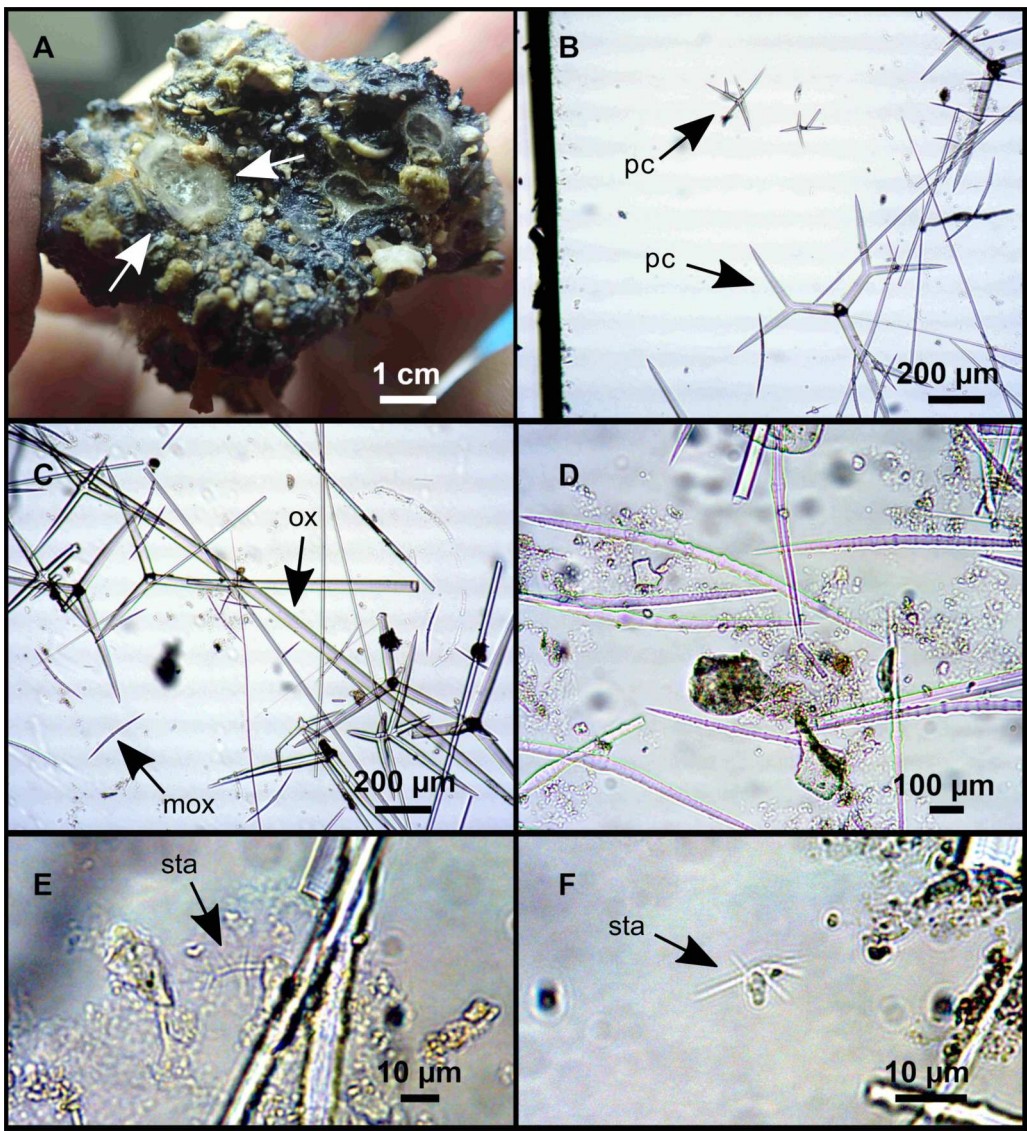

**Figure 46** ***Vulcanella* cf. *gracilis* (*Sollas, 1888*).** (A) Habitus of i279_A on deck (arrows) growing on *Hexadella* sp. (B–F) Optical microscope images of i279_A (B–D) and i279_B (E–F). (B) Pseudocalthrops (pc). (C) Microxeas I (mox), oxeas (ox), and pseudocalthrops. (D) Microxeas I. (E–F) Streptasters (sta).

and (iv) commonly bifurcated, a modification also observed in *V. gracilis,* but less common (*Uriz, 1981*; *Pulitzer-Finali, 1983*; *Boury-Esnault, Pansini & Uriz, 1994*). On the other hand, oxeas are shorter and thinner than in *V. gracilis.* Regarding microscleres, *V.* cf. *gracilis* is missing the microxeas II found in *V. gracilis.* Also, the microxeas sometimes show an irregular annulation pattern, which contrasts with the regular tuberculated annulation of *V. gracilis.* Finally, streptasters are very scarce, only found in i279_B but not in i279_A, while these are relatively common in *V. gracilis.* To conclude, these two samples from the same station might represent a new species or atypical shallower specimens of *V. gracilis.* Genetic markers and additional specimens are necessary to test our hypotheses.

Family Thrombidae *Sollas, 1888*
Genus *Thrombus Sollas, 1886*
*Thrombus abyssi* (Carter, 1873)
(Fig. 47)

## Material examined

UPSZMC 190968-190969, field#i391_5_1 and field#i391_6, MaC (EB), St. 158 (INTEMARES1019), 146 m, beam trawl, coll. J. A. Díaz; UPSZMC 190970, field#i470, MaC (SO), St. 8 (INTEMARES0720), 244–251 m, rock dredge, coll. J. A. Díaz.

## Outer morphology

Small encrusting sponges, spreading 1–3 cm on rocks and other sponges. Beige in life (Fig. 47A), and beige to grayish beige (Fig. 47B) after ethanol fixation. Smooth surface, hard but slightly compressible. Cortex not visible to the naked eye. Pores and oscules inconspicuous.

## Spicules

Acanthotrichotriaenes (Figs. 47C–47F), several morphologies probably representing different developmental stages. Small ones (Fig. 47C), microspined, with a smooth appearance under the light microscope; clads sometimes not bifurcated, and sometimes without an epirhabdome. Large ones appear in two morphologies, with overlapping sizes, mostly trichotomous but also dichotomous or unbifurcated. Often, one morphology (Figs. 47D–47E) is robust with strong and large spines and short epirhabdome, while a second morphology (Fig. 47F), is more slender, has less and smaller spines and a longer epirhabdome. Overall measuring: rhabdome 25-45/2-9 μm, epirhabdome 8-34/4-9 μm, protocladi 2-9/2-8 μm, deuterocladi 5-15/2-7 μm.

Amphiasters (Fig. 47G), common, rays curved and directed inwards, at the end of both axes. Axis is straight and may have some isolated spines. Overall measuring 4–6 μm.

## Ecology and distribution

The species has been found as epibiont of tetractinellid sponges *G. geodina* (i391_6_1) and *V. gracilis* (i391_5_1), and also growing on a dead oyster *Neopycnodonte* sp. shell (i470).

## Genetics

Folmer COI and 28S C1-D2 fragments were obtained from i470 (ON130561 and ON133868).

## Taxonomic remarks

Easily recognizable species due to the presence of trichotriaenes with both rhabdome and epirhabdome together with characteristic "amphiasters". Its sister species *Thrombus niger Topsent, 1904* differs with *T. abyssi* by its black color and the absence of epirhabdome in its trichotriaenes. We have however found trichotriaenes with and without epirhabdome in *T. abyssi* but the former are more abundant. Interestingly, all trichotriaenes observed under SEM had epirhabdomes, but in many cases it was broken, which may suggest an artifact when these spicules are observed under light microscope. The small size of the spicules

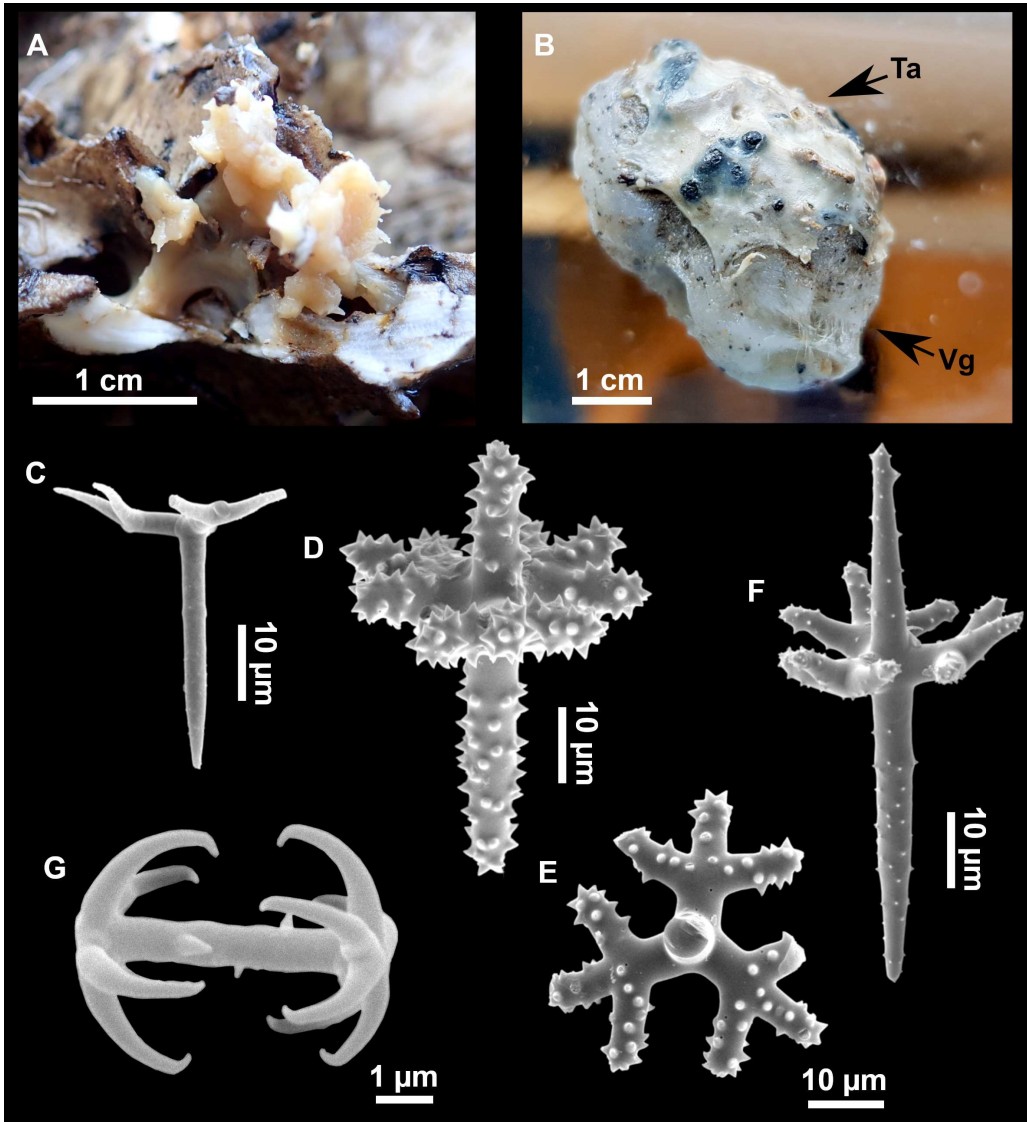

**Figure 47 *Thrombus abyssi* Carter, 1873.** (A) Habitus of i470 on deck. (B) Habitus of i391_5_1 (Ta) after ethanol fixation, growing on a *Vulcanella gracilis* (Vc) specimen i391_5. (C–G) SEM images of i470. (C) Juvenile acanthotrichotriaene. (D–F) Acanthotrichotriaenes. (G) Amphiaster.

and the epirhabdome itself could make it difficult to distinguish this break, and thus the observer may assume that the epirhabdome is not present. The unique morphology of the "amphiasters" indicates that they are not homologous to the amphiasters found in the Astrophorina (*e.g.*, Pachastrellidae, Vulcanellidae, Theneidae). This is the first time COI is sequenced for this species. As for 28S, two haplotypes were previously sequenced from Rockall Bank, 751–784 m (HM592755) and Mingulay Reef in Scotland, 151–159 m (HM592756), with a 1 bp difference; our 28S sequence is the same haplotype as that of the specimen from Rockall Bank.

Suborder Spirophorina Bergquist & Hogg, 1969
Family Tetillidae *Sollas, 1886*
Genus *Craniella Schmidt, 1870*
*Craniella* cf. *cranium* (Müller, 1776)
(Fig. 48)

## Material examined

UPSZMC 190910 (spicule preparation), field#i151_1B and UPSZMC 190873 (spicule preparation), field#i153_4_1, MaC (EB), St. 52 (INTEMARES0718), 109 m, rock dredge, coll. F. Ordines; UPSZMC 190819, field#i172_1A, MaC (EB), St. 60 (INTEMARES0718), 138 m, beam trawl, coll. F. Ordines; UPSZMC 190820, field#i339_5, St. 124 (INTEMARES1019), MaC (EB), 152 m, beam trawl; UPSZMC 190821, field#i409_1_1, MaC (EB), St. 167 (INTEMARES1019), 151 m, beam trawl, coll. J. A. Díaz; UPSZMC 190822, field#i416_D, EB, St. 177 (INTEMARES1019), 155 m, beam trawl, coll. J. A. Díaz; UPSZMC 190823, field#i826_7_1(G), MaC (EB), St. 24 (INTEMARES0820), 150–134 m, ROV, coll. J. A. Díaz.

## Outer morphology

Small, 0.3–1 cm in diameter (Fig. 48A), circular to hemispherical, encrusting or slightly insinuating sponges (Fig. 48B). Pale beige in life and after ethanol fixation. Very hispid, hard and compressible consistency. Choanosome fleshy, thin cortex, less than 0.1 mm width. Pores and oscules inconspicuous.

## Spicules

Anatriaenes (Fig. 48C), with long, thin and flexuous rhabdomes, measuring 744-1,364-2,436/4-7-15 μm ($N = 14$). Cladomes with short pointy clads, usually with a small spine at its top. (Fig. 48D), 13-21-51/5-8-15 μm.

Protriaenes (Fig. 48E), abundant, with long rhabdome, straight at most of its length but ending in a flexuous tip, measuring 777-900-1,045/6-6-7 μm. Cladome with 2–3 clads, often of an unequal length, straight or slightly curved inwards at the tips (Fig. 48F), 25-63-135/4-6-10 μm.

Oxea I (Fig. 48G), anisoactinal (anisoxeas), straight and fusiform, 427-821-1,307/5-11-27 μm.

Oxea II, isoactinal, fusiform, slightly curved, 177-281-470/5-10-23 μm.

Microxea (Fig. 48H), curved, measuring 80-173-243/1-5-5 μm ($N = 18$)

Sigmaspires (Fig. 48I), very abundant, spiny, most C-shaped but also S-shaped, fairly small and measuring 7-8-10 μm in chord.

## Ecology and distribution

Found at the summit and upper slope of the EB, on gravel bottoms. The sponge has been found growing on other sponges like *Spongosorites* sp. (i826_7_1) (Fig. 48A), *Phakellia robusta Bowerbank, 1866* (i409_1_1) (Fig. 48B), *Foraminospongia balearica* Díaz, Ramírez-Amaro & Ordines, 2021 (i172-1A), *P. monilifera* (i151_1B and i153_4_1) or directly on the substrate (i339_5). Interestingly, when growing on *P. robusta* it erodes its tissue,

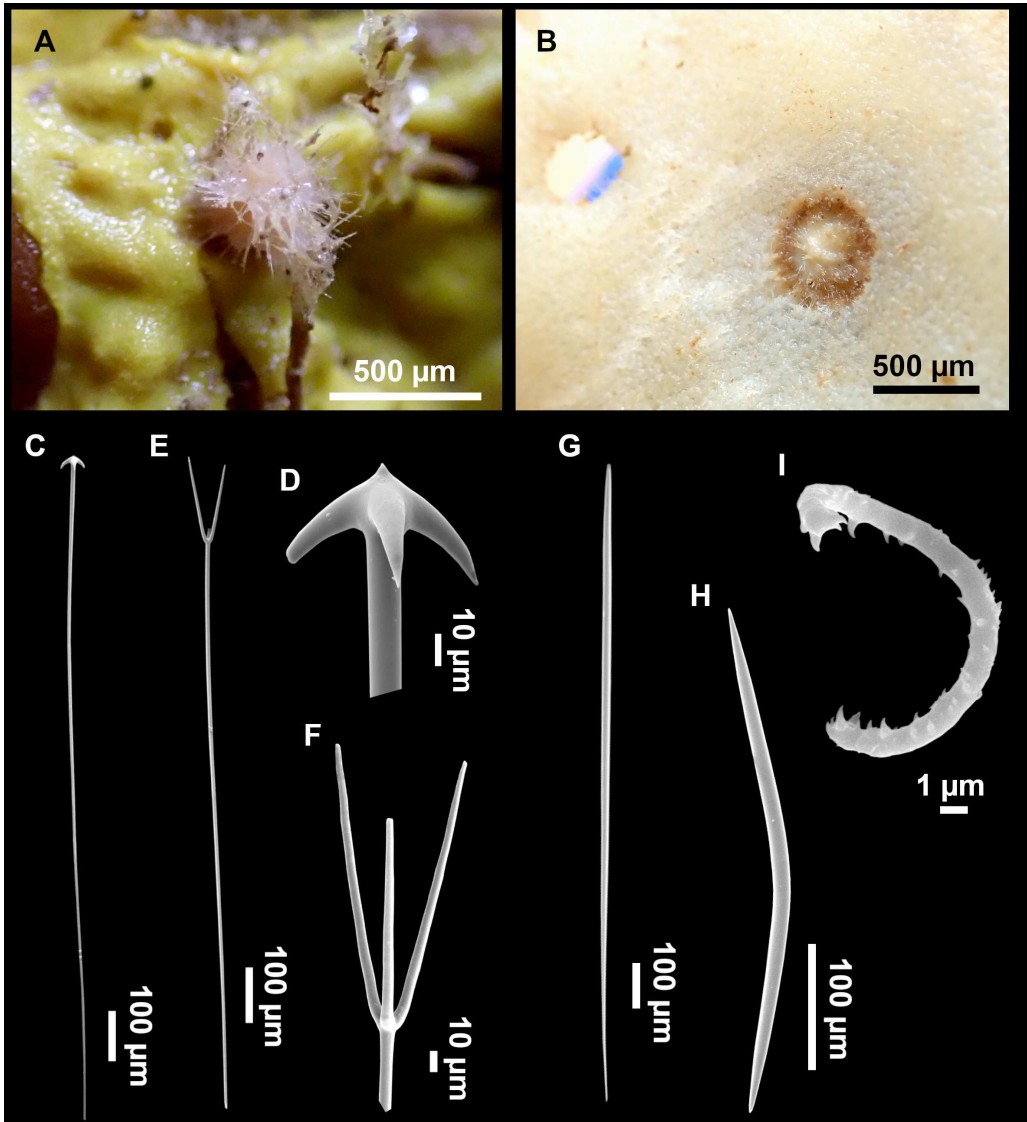

**Figure 48** *Craniella* cf. *cranium* (Müller, 1776). (A) Habitus of i826_7_1(G) growing on *Spongosorites* sp., picture taken on deck. (B) Habitus of i409_1_1 after ethanol fixation, growing on/in *Phakellia robusta* and causing necrosis to the host. The hole on the left may be the remain of another of these interactions. (C–I) SEM images of i827_7_1(G). (C) Anatriaene with (D) detail of the cladome. (E) Protriaene with (F) detail of the cladome. (G) Oxea I (anisoxea). (H) Oxea II. (I) Sigmaspire.

which at the end is cut off, leaving a characteristic hole (Fig. 48B). In ROV records of the MaC seamounts, perforated *P. robusta* specimens are commonly seen. It is unknown if this process is a defense mechanism of the host (potentially mediated through secondary metabolites) or an indirect effect of the epibiosis (like tissue decay due to pore obstruction).
### Genetics

The COI Folmer-Erpenbeck fragments were obtained for i172-1A and i416_D (OR045914 and OR045913), as well as the 28S (C1-C2) fragments of the same specimens (ON133849 and ON133850).

### Taxonomic remarks

A comprehensive revision of *Craniella cranium* is greatly needed, it has been widely reported from the Mediterranean (*e.g.*, *Vacelet, 1969*; *Pulitzer-Finali, 1983*), including the MeC (*Santín et al., 2018*). It is also recorded from the eastern, central and western Atlantic, including both north and south America coasts, and on a broad range of depths and habitats (caves, mesophotic zone, deep sea; see WPD for an overview). Most of the records have not been accompanied by proper descriptions and/or type redescriptions, nor sequencing. Here, we prefer to identify our material *Craniella* cf. *cranium*, until a future revision of the species is done.

## DISCUSSION

### Hidden diversity in mesophotic west Mediterranean waters

The Mediterranean Sea is considered one of the best known marine areas of the world (*Coll et al., 2010*). Regarding sponges, this study challenges this assumption. In a small area such as the Balearic Islands, we are reporting the discovery of six new tetractinellid species: *Stelletta mortarium* **sp. nov.**, *Penares cavernensis* **sp. nov.**, *Penares isabellae* **sp. nov.**, *Geodia bibilonae* **sp. nov.**, *Geodia microsphaera* **sp. nov.** and *Geodia matrix* **sp. nov.** Also, *Stelletta dichoclada* and *Erylus corsicus* are reported here for the second time since their description 40 years ago. *Pachastrella ovisternata*, *Vulcanella aberrans* and *Characella pachastrelloides* are reported only for the second time in the Mediterranean. Others are reported for the first time in the Balearic Islands region: *C. pachastrelloides*, *C. tripodaria*, *Stelletta mediterranea*, *Caminus vulcani*, *Caminella intuta*, *G. geodina*, *Erylus* cf. *deficiens*, *Erylus mamillaris*, *Discodermia polymorpha*, *Thrombus abyssi* and *Vulcanella gracilis*. With this work the number of Mediterranean tetractinellid species is raised from 83 to 89, while in the Balearic Islands, their number more than doubles, from 16 to 39, thus becoming one of the regions with the highest tetractinellid diversity of the Mediterranean (*de Voogd et al., 2023*). The only species that we have not found but that are reported in the Balearic Islands are *Dercitus (Stoeba) plicatus* (*Schmidt, 1868*) and *Geodia cydonium*, both reported by *Bibiloni (1990)* and the lithistids *Neophrissospongia nolitangere* and *Leiodermatium pfeifferae* reported from the MaC and MeC, respectively (*Santín et al., 2018*; *Maldonado et al., 2015*). *Dercitus (S.) plicatus* is an encrusting to excavating sponge, maybe missed due to its cryptic habit. Regarding *G. cydonium*, it is interesting that *Bibiloni (1990)* mentioned the presence of two types of morphologies, one large/massive and one small/encrusting. We suspect that small encrusting individuals belong in fact to *G. bibilonae* **sp. nov.**, but we did not have access to this material for comparison.

Interestingly, in the MaC seamounts we have found 29 of the 34 species here reported, a number that contrasts with the seven species previously found in the MeC (*Santín et al., 2018*). This may be explained by the higher habitat heterogeneity of the MaC seamounts,

but also could be an artifact caused by differences in sampling methodologies, use of molecular markers, lack of specialists and sampling intensity.

Also, *Uriz (1981)* reported 10 species of tetractinellids in the northeastern Iberian Peninsula, three of which (*Stelletta dorsigera*, *Stelletta grubii* and *Stelletta hispida*) were not reported in the Balearic Islands. The shallower distribution of these species may explain why these were not found in the present study. The high oligotrophy of the Balearic Islands may also be a contributing factor. Indeed, nutrient scarcity can have a negative effect on eutrophic species by limiting its physiological demands. Also, oligotrophy causes water transparency which in turn promotes a shift in the taxonomic composition of the benthos, promoting the development of photosynthetic algae and seagrass communities that compete with filter-feeders. In fact, shallow bottoms of the Balearic Islands are dominated by vast meadows of *Posidonia oceanica* (L.) Delille, 1813 and also brown and green algae like *Cystoseira* spp., *Halimeda* or *Caulerpa*, among others. At mesophotic depths, red algae are dominant from 30–40 m to 130–140 m. However, the deepest red algae may not be as efficient in competing for space as the heterotrophic or mixotrophic organisms. The low nutrient content and the intense competition for space in shallow Balearic Islands waters may explain why most of the tetractinellids have been found associated with deep red algae beds, and why some shallow water species are not found.

*Sitja & Maldonado (2014)* listed 26 tetractinellid species in the Alboran Island and surrounding abyssal plains. The upper shelf of the Alboran Island is a Specially Protected Areas of Mediterranean Importance (SPAMI), extensively studied and considered one of the richest spots of the Alboran Sea (*Rueda et al., 2021*), with a surface area comparable to the MaC seamounts. The fact that we found three more tetractinellid species than in the emblematic area of the Alboran Island is a strong argument towards the inclusion of these seamounts in the Natura 2000 framework.

Some of the new species described here are large and massive(*G. matrix* **sp. nov.** and *S. mortarium* **sp. nov.**), and have large population biomasses, which could signify potential sponge ground habitats in these areas (*Díaz et al., 2024*). Those species, together with other large tetractinellids reported (*P. monilifera*, *P. ovisternata*, *C.(C.) pathologica*, *S. fortis* and *S. mucronatus*) are habitat builders that provide three dimensional structure, shelter, and settlement substrate for other organisms like small crustaceans, ophiuroidea, worms and other sponges. Recently, *Díaz, Ramírez-Amaro & Ordines (2021)* described the agelasid genus *Foraminospongia* *Díaz, Ramirez & Ordines, 2021* at the MaC Seamounts. Its type species, *Foraminospongia balearica* Díaz, Ramírez-Amaro & Ordines, 2021, shares habitat with *G. matrix* **sp. nov.**, *G. geodina* and *S. mortarium* **sp. nov.**, and like those it is an abundant, large and characteristic habitat builder. The fact that those large key species have only recently been described indicates that more large species may be waiting to be discovered. This highlights how poorly explored Mediterranean deep-sea habitats are, especially seamounts, and the importance of future seamount research and conservation.

Sampling was done on a broad bathymetric scale (0–725 m) and heterogeneous habitats: from littoral caves, to different sedimentary bottoms (with mud, gravels, soft and coralline red algae), located at trawling grounds and seamounts, and also on the rocky slopes and summits of the seamounts SO, AM and EB. Of the 34 species recorded here, only

five were not recorded from the seamounts (*C. intuta*, *P. isabellae* **sp. nov.**, *P. cavernensis* **sp. nov.**, *E.* cf. *deficiens* and *E. discophorus*), which shows the importance of the MaC seamounts in terms of species richness. On the contrary, on the trawl fishing grounds off Mallorca, Menorca and Ibiza-Formentera, only six species were found (*Poecillastra compressa*, *Nethea amygdaloides*, *P. euastrum*, *P. helleri*, *S. mucronatus*, *Thenea muricata*), all of them also present in the MaC seamounts. MaC has been proposed to be part of the Natura 2000 network due to its highly rich ecosystems and presence of vulnerable marine habitats (*Ordines et al., 2019*; *Díaz, Ramírez-Amaro & Ordines, 2021*; *Massutí et al., 2022*). The present study strengthens this proposal.

## Species complexes revealed by molecular markers

The combined use of morphology and molecular markers has allowed us to detect several species complexes: *Geodia bibilonae* **sp. nov.** and its sister species *Geodia microsphaera* **sp. nov.**, both related to the Northeast Atlantic *G. cydonium*. Their almost identical macroscopic morphology, added to their small sizes and similar spicular set (only distinguishable by differences in spicule size and morphology) makes their distinction and determination by means of morphology alone challenging. Another species complex concerns the species *G. geodina* (previously called *G. anceps* for most records), that we now split in two species: *Geodia phlegraeioides* **sp. nov.** in the upper bathyal Atlantic (phylogenetically closer to *G. phlegraei/parva*) and *G. geodina* in the mesophotic Mediterranean and the mesophotic North Atlantic. This was discovered thanks to the barcoding and careful spicule comparison of several specimens of Mediterranean and Atlantic representatives. Likewise with the species *C. vulcani* and *Caminus xavierae* **sp. nov.**, the first being now restricted to the Mediterranean and the second to the Canary Islands for the moment. Those cases are direct evidence of how the Atlanto-Mediterranean barriers, as well as water masses, have strongly affected sponge speciation. Two major genetic barriers are present between the north Atlantic and the Western Mediterranean: the Strait of Gibraltar, which separates the North Atlantic from the sea of Alboran, and the Almeria Oran front, that separates Alboran from the western Mediterranean. These barriers are known to affect a vast number of marine organisms, including sponges (*Patarnello, Volckaert & Castilho, 2007*; *Riesgo et al., 2019*). Besides, a minor genetic barrier is also known at the Ibiza Channel, between the island of Ibiza and Valencia (*García-Merchán et al., 2012*). In light of that, many sponge species previously thought to have an Atlanto-Mediterranean distribution may in fact represent species complexes, with separate North Atlantic and Mediterranean representatives. The case of *G. geodina* also illustrates that mesophotic Mediterranean species can be found in the Atlantic at similar depths, highlighting the importance of water masses as barriers, already suggested for deep-sea sponge distribution (*Roberts et al., 2021*; *Steffen et al., 2022*).

## Shallow-water caves connections with the deep sea

It is usually assumed that the shallow-water cave fauna is tightly connected to the deep-sea fauna, with some species common to both habitats: the fish *Conger conger* (Linnaeus, 1758), the hexactinellid sponge *Oopsacas minuta* *Topsent, 1927* and the carnivorous sponge

*Lycopodina hypogea* (*Vacelet & Boury-Esnault, 1996*; *Bakran-Petricioli et al., 2007*) are only a few examples. This happens in ecosystems that may be separated by hundreds of km, and is explained by the similarities between both habitats: lack of light (and thus lack of photosynthetic communities), scarcity of nutrients and low hydrodynamic energy (*Vacelet, Boury-Esnault & Harmelin, 1994*). However, our results nuance this assumption: we have shown that several species that were previously thought to inhabit both shallow water caves and deep sea habitats can sometimes have diverged enough to reach species status. This is the case of sister species *P. helleri* (deep sea)/*P. isabellae* **sp. nov.** (caves), and *P. euastrum* (deep sea)/*P. cavernensis* **sp. nov** (caves), both traditionally considered a single species. Also, the COI sequence of deep sea *D. polymorpha* showed 1 bp. difference with cave specimens off Marseille, a genetic differentiation not sufficient alone to justify describing a new species but that suggest some kind of population differentiation.

Sponges are main contributions to the overall diversity and biomass elements in both deep sea and cave habitats (*Gerovasileiou & Voultsiadou, 2012*; *Maldonado et al., 2017*). Mesophotic and cave species may be more susceptible to undergo speciation driven by oceanographic barriers, because of the added effects of habitat discontinuities (habitat patchiness or limited ecological threshold). Different speciation scenarios may have taken place: ancient species with broad bathymetric distribution inhabiting both deep sea and littoral caves, becoming isolated and later differentiated in two different species. Also, throughout new environment colonization (deep-sea species colonizing caves, as suggested in the case of *D. polymorpha*, or vice versa) and posterior reproductive isolation. In some cases the dispersal potential may have been high enough to maintain the gene flow between both ecosystems, in which case a single species remained. This may be the case of the mentioned *D. polymorpha*, but also of *Erylus discophorus*. For the latter, we have found 0–2 bp differences between specimen POR785, found on a fishing ground, and Mallorca shallow water cave specimens LIT71, LIT72 and LIT74. Further works with higher resolution markers are necessary to clarify these questions. Cave and deep-sea sponges would probably represent a good model to study speciation in low-dispersal invertebrates.

Finally, it important to highlight the importance of using molecular markers when studying species inhabiting this kind of ecosystems, especially if they are found in both shallow caves and deeper waters. In those cases, molecular markers represent an independent dataset to test the relevance of possible morphological discrepancies, which may be related to environmental parameters, such as different silica concentration, and/or genetics. In the course of this study, we have always found larger spicules on deeper waters, in both cases: when a single species inhabits caves and the deep-sea (like *D. polymorpha* or *E. discophorus*), and in cases when there are two sister species, one from the deep-sea and one inhabiting the caves (like *P. helleri*/*P. isabellae* **sp. nov** or *P. euastrum*/*P. cavernensis* **sp. nov.**). Translocation experiments or laboratory experiments under different silica concentrations could eventually show if these size characters are fixed in the genetics of the species/populations or simply part of their spicule plasticity range.

## ACKNOWLEDGEMENTS

The authors wish to thank the captain and crew of R/Vs *Ángeles Alvariño*, *Sarmiento de Gamboa* and *Miguel Oliver*, as well as the participants in the INTEMARES surveys and the MEDITS 2016–2021 surveys. Special thanks to Ferran Hierro (University of the Balearic Islands) for the technical assistance at the scanning electron microscope, to Sergio Ramírez-Amaro for his guidance in the laboratory, the members of the GEOMAR group and to Maria Teresa Farriols and Elena Marco-Herrero for the logistic work during the development of the INTEMARES surveys. Thank you to Pilar Ríos, Alejandra Calvo-Día, Ana Colaço and Filipe Porteiro for sharing specimens of *G. phlegraeioides* and helping in measuring their spicules. Many thanks to Adrzej Pisera and Francisca Carvalho for their taxonomic advice on *D. polymorpha*. Special thanks to Bel Fullana, Aida Frank, Joan Cabot, Xavier Busquets and Guillem Mateu for the logistic support during the diving surveys. Thanks to Pere Ferriol for the collection of *N. amygdaloides* POR7(15). Thank you to Pierre Chevaldonné for sharing several comparative specimens of *N. amygdaloides*. Thanks to Elena Gerasimova (University of Bergen) for producing some of the thick sections. Thanks to David Rees for sequencing the COI of *G. phlegraeioides* **sp. nov.** DR15_869c. Thank you to Maria Tavano (Museo Civico di Storia Naturale "G. Doria") for sending the type material of *Stelletta dichoclada*, *Stelletta defensa, Erylus corsicus* and *Erylus papulifer*. Thank you to Monika Myrdal (Museum of Evolution, Uppsala) for helping with the specimen collection deposition and registration.

### Funding

The project LIFE IP INTEMARES is coordinated by the Biodiversity Foundation of the Spanish Ministry for the Ecological Transition and the Demographic Challenge and receives financial support from the European Union's LIFE program (LIFE15 IPE ES 012). MEDITS surveys are co-funded by the European Union through the European Maritime and Fisheries Fund (EMFF), within the National Program of collection, management and use of data in the fisheries sector and support for scientific advice regarding the Common Fisheries Policy. Julio A. Díaz was supported by a predoctoral contract co-funded by the Regional Government of the Balearic Islands and the European Social Fund (FPI/2178/2018) and received a mobility grant (MOB_008_2019) by the Conselleria de Fons Europeus, Universitat i Cultura del Govern de les Illes Balears that allowed Julio A. Díaz to carry out a stay at the Uppsala University. Paco Cárdenas was supported by the SponBIODIV project, a 2021-2022 BiodivProtect joint call for research proposals, under the Biodiversa+ Partnership co-funded by the European Commission and the Swedish funding organization FORMAS (project#2022-01709). This publication was made possible with the support of the project "Improvement of the scientific and technical knowledge for the sustainability of demersal fisheries in the western Mediterranean" (SosMed) funded by Next Generation European funds (Recovery, Transformation and Resilience Plan), for the publication fees, with an agreement between the Spanish Ministry of Agriculture, Fisheries

and Food and CSIC by means of the Spanish Institute of Oceanography. There was no additional external funding received for this study. The funders had no role in study design, data collection and analysis, decision to publish, or preparation of the manuscript.

## Grant Disclosures

The following grant information was disclosed by the authors:
Biodiversity Foundation of the Spanish Ministry for the Ecological Transition and the Demographic Challenge and receives financial support from the European Union's LIFE program (LIFE15 IPE ES 012).
European Union through the European Maritime and Fisheries Fund (EMFF).
National Program of collection, management and use of data in the fisheries sector and support for scientific advice regarding the Common Fisheries Policy.
Regional Government of the Balearic Islands and the European Social Fund: FPI/2178/2018.
Conselleria de Fons Europeus, Universitat i Cultura del Govern de les Illes Balears: (MOB_008_2019).
SponBIODIV project, a 2021-2022 BiodivProtect joint call for research proposals.
Biodiversa+ Partnership co-funded by the European Commission and the Swedish funding organization FORMAS: project#2022-01709.
Next Generation European funds (Recovery, Transformation and Resilience Plan).
Spanish Ministry of Agriculture, Fisheries and Food.
CSIC.

## Competing Interests

The authors declare there are no competing interests.

## Author Contributions

- Julio A. Díaz conceived and designed the experiments, performed the experiments, analyzed the data, prepared figures and/or tables, authored or reviewed drafts of the article, and approved the final draft.
- Francesc Ordines conceived and designed the experiments, authored or reviewed drafts of the article, and approved the final draft.
- Enric Massutí conceived and designed the experiments, authored or reviewed drafts of the article, and approved the final draft.
- Paco Cárdenas conceived and designed the experiments, performed the experiments, analyzed the data, prepared tables, authored or reviewed drafts of the article, and approved the final draft.

## DNA Deposition

The following information was supplied regarding the deposition of DNA sequences:
The sequences, GenBank and Sponge Barcoding project accession numbers are available in the Supplemental Files.

## Data Availability

The COI and 28S standard barcoding DNA sequences are available in the Supplemental Files.

### New Species Registration

The following information was supplied regarding the registration of a newly described species:

Publication LSID

urn:lsid:zoobank.org:pub:A88AE49E-B422-4F9A-A5E0-BB6C6B8FC185

*Caminus xavierae*

urn:lsid:zoobank.org:act:06736269-F059-41F0-A661-D470B24ED66B

*Geodia bibilonae*

urn:lsid:zoobank.org:act:25E34452-ECCB-4750-BF9A-02A737C63B4B

*Geodia matrix*

urn:lsid:zoobank.org:act:7C6012CC-A8B8-40E1-9A0A-851EE2AB03FA

*Geodia microsphaera*

urn:lsid:zoobank.org:act:5E204613-8367-45FD-ADEC-C0D321C97FBB

*Geodia phlegraeioides*

urn:lsid:zoobank.org:act:F0D8FC4D-F4AE-43FA-A7BD-281A41CB7F64

*Penares cavernensis*

urn:lsid:zoobank.org:act:06A77BBF-D954-4A35-8FF4-B5B8C8404E55

*Penares isabellae*

urn:lsid:zoobank.org:act:D21CF35B-8933-4D8A-B68B-C7DF90D9E0B4

*Stelletta mortarium*

urn:lsid:zoobank.org:act:F54E7CED-910D-44A0-A7E0-8DFBE512340B.

### Supplemental Information

Supplemental information for this article can be found online at http://dx.doi.org/10.7717/peerj.16584#supplemental-information.

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
