# Peer review of "From caves to seamounts: the hidden diversity of tetractinellid sponges from the Balearic Islands, with the description of eight new species"

_PeerJ, doi:10.7717/peerj.16584_

## Round 0.1 · original submission · Major Revisions

I have review reports from three experts. I particularly would like the authors to address the point of Material Transfer Agreements (MTA) and also provide necessary reference numbers related to MTA.

Reviewer 1 ·

Basic reporting

Figure 1: Scale, coordinates, bathymetry are a bit small/low quality, and in 1A, the black dots are unclear what they represent.
In line 289, the holotype of Stelleta dichoclada refers to figure 4, but figure 4 shows the specimens from the study and the holotype is actually shown in Figure 5.
In line 292, the images are switched between the brackets, it should be (Fig. 4A-B1 and Fig. 5A).
In line 296, for the whole spicule section, the images are actually referring to the figure 5.
In line 297, I suggest referring to Figure 5C when mentioning the plagiotriaene, despite only there being an image from the holotype.
In line 310, Figure 4D-G refers to anatriaene and some oxyasters, I am thinking this is a mistake and is actually referring to Figure 5D-G from the holotype, which refers to all oxyasters. If this is the case, why point the reader to the holotype oxyasters and not to the ones found in the Balearic specimens? Also, the oxyasters in the holotype seem to have a wider and rounder center when compared to the Balearic specimens. Is this seen in most/all oxyasters?
In line 312, same as above, the Figure 4H shows an oxyaster, whereas in Figure 5H shows a raphid.
In line 459, the measurement ranges of most spicules do not match with those found in Table 4.
In Table 4, the dichotriaene protoclad width of specimen i401_2 have the order switched from larger to smaller.
In line 472, an image showing this spicule type, if possible, would be beneficial.
In line 701, the calthrop range according to the table 6 is 54-879/6-94 µm
In line 723, there is a misspelling C. (C.) geodioides.
In table 6, some mean values are not underlined.
In line 770, an image showing a dichotriaene, if possible, would be beneficial.
In line 823, according to Table 7, the width of strongyles actually ranges from 8-22 µm
In line 828, according to Table 7, the diameter of oxyasters actually ranges from 34-99 µm.
In line 869 the orthotriaene rhabdome measurements are separated with an “x”, rather than the “/” used in the previous spicule descriptions.
In Table 7, orthotriaene clad width measurements are missing in the holotype of Caminus xavierae sp. nov.
In Table 7, orthotriaene clad mean values are missing for the paratype of Caminus xavierae sp. nov. In case the number of clads measured is low, the number of measured spicules is missing in the table.
In line 941, grammatical error in “the deureroclads is then very small”.
In table 12, the legend should be “Spicule measurements of Penares isabellae sp. nov. and Penares helleri”
In line 1493, the images referring to spicules need to be rechecked as they are all incorrect.
In figure 35, the legend does not correspond to the orthotriaene.
In figure 36, the legend points to an oxea in 36C, but the image has no oxea.
In line 2047, no mean values are shown to the orthotriaene measurements.
In line 2224, in other spicules not shown in image, have “(not shown)”. I suggest, for consistency, to add “(not shown)” after oxeas. There is also a full stop missing at the end of the second phrase.
In figure 41, Figure 41B refers to a calthrop, and SEM pictures are (B-I), Calthrops (B-C), dichotriaenes (D-F).
In line 2374 Image 41B refers to a calthrops.
In line 2377, the figure legends need to be updated, same as above.
In line 2469, there is no list of comparative material, I suppose the Pouliquen 1972 and the Pisera & Vacelet, 2011’s material were used as comparative material.
In line 2521, there is an error in the phrase “The identify of the remaining north-east Atlantic records…”
In line 2654, there is an error when referring to metaster with aborted actine “(Fig X)”.
In line 2831, no mean values are presented for spicule measurements.
In line 2917, there is a misspelling of the word biomasses.

Experimental design

In line 343, it is also a good point to mention the slight difference between the rounder and wider center in the holotype’s oxyaster, from Corsica and the specimens from this study.
In line 381, it seems that only the spicule measurements of the i757 EB specimen are presented. Why are the remeasurements of the holotype not presented as the previous species description?
In line 1050, the morphology of the aspidasters in the plates presented for the three species are also seem a bit different. Smaller and a bit amorphous and with less abundance of rosettes in E. cf. deficiens, besides size, the rosette arrangement between E. discophorus and E. cf. mamillaris seem different as well. Are these characters seen in other aspidasters not presented in the images?
In line 1107, it is very interesting that the COI tree is more elucidative than the 28S tree, when the COI is a supposedly a slower evolving gene than the 28S. I am assuming that 28S sequences are the same for the three species, but it should be specified.
In line 1209 I think it is important to address that, according to Table 9 the triaenes seen in E. papulifer are mostly dichotriaenes, as opposed to the orthotriaenes seen in the other two species.
Line 1351 is it considered that the slight spicule size variations could be a result of the depth difference between the shallow cave and mesophotic specimens? This phenomenon is mentioned in line 1072, when discussing E. discophorus. Considering the molecular differences despite the close geographical locations, I agree with the uncertain taxonomic status of P. cavernicola, and that it is a probable separate species to P. euastrum, however, the slight morphological differences between them might not be fully convincing to some authors.
In line 2425, agreed with the need of revision and it is unfortunate that the 28S D2 fragment was not possible to sequence. The presence/absence of mesodichotriaenes seem the most promising for the separation of both species.
In line 2573, we also get specimens barely reaching 1cm in diameter from the Gulf of Cádiz and Southern Portugal and also have Epizoanthus growth. It would be beneficial, if possible to have an outer morphology description of these specimens (Oscule, pores etc.).
In line 2857, I think it is needed to mention somewhere in the discussion the Silica concentrations in the water as a potential factor in the morphological differences between specimens/species, since it has been mentioned in numerous taxonomic remarks when discussing spicular differences.

Validity of the findings

no comment

Additional comments

no comment

·

Basic reporting

The paper by Diaz et al on the Tetractinellida of the Balearic Islands is an excellent work on an important taxon of sponges, collected during several years at diverse depths, including bathyal depths and diverse caves. The descriptions are accurate, with excellent images, including numerous Scanning Electron Microscope of unusual quality. The taxonomic decisions rely not only on accurate morphological descriptions but also on molecular data. In my opinion, it is a taxonomic work on sponges of exceptional quality. I strongly recommend its publication in PeerJ.
Of course the MS is rather long, and some minor mistakes or misspellings could be found. Here are some remarks, or some questions.
General remark for italic use: In the text, the references are in italic (eg line 36: and 23 families (de Voogd et al., 2023). In the References list, line 3017, the review name is not in italic. This differs from the general rules. Is that special rule for PeerJ, or author’s mistake?
Line 59: lithistids are not presented before.
86: those facts or these facts?
157: all caves are said to be created by sea erosion, not karstic. This appears surprising, even for Cova de Cala Sa nau, 76 m in lenght?
Fig. 4, raphides are 4J on Fig.4, but 4H on line 312. Line 310, oxyasters are (Fig. 4D-G), but in fact (F-I) on Fig. 4. Possible other mistake in the illustration.
352: and, not and
354: A SEM plate, not An
416: (Wells, Wells & Gray, 1960): not in italics
505: Schmidt, not Schimdt
595: Vacelet, 1961comma missing
625-627: In Fig. 12, (D-E) Oxeas. (F-G) Oxyasters. (H-I) Amphisanidasters. But in the text, only 12D for Oxeas, 12I for amphisanidasters and 12G for Oxyasters
645: distinctions, not disctinctions.
1494, 1497: Fig. 29 B and C refer to Dichotriaenes or Oxeas in the text, but to morphology and pores in Fig. 29. Check the correspondence between text, Fig. 29 and Legend of Fig. 29.
1586: For Geodia matrix, film by the Wachowski sisters, reference?
1615: 30L-K, not k
1655-1661: toxa-like, toxas. From Fig. 31 G, I would rather think that these spicules are slightly flexuous microtoxas rather than toxas.
1738: size of the rhadome?
1865: of a MNHN, not an
1964-1965: differences between text, legend of Fig. 36 and Fig. 36
1978: could be found in G. microsphaera
2812: Spirophorina, not Sphirophorina
2848-2852: The interactions with P. robusta figured on Fig. 48B need to be explained again in the legend of the Fig.
2865: Discussion. In typography, this heading is similar to the headings in the text for species descriptions, when it is more general. But this is more relevant of presentation than of science.
2964: separate, not separate
2972: it would be good to underline that these are only a few examples
3017: References. In the reference list, the references are not in italic? With an exception for PeerJ (line 3123, and possibly other).
Legend Fig. 3: why some Latin names in bold and other not?
Table 1, sampling stations: In several Stations, the coordinates column Start are noted E instead of N.

Experimental design

No comment

Validity of the findings

No comment

Additional comments

No comment

Reviewer 3 ·

Basic reporting

All criteria passed. It is a clear article that is well-structured and professional. It's an impressive piece of work. I would however recommend that a fluent English-speaking editor reviews the text, before re-submission.

Experimental design

The methods and design is good and descriptions are very clear.

My main concern is related to where the material has been deposited and to the Nagoya protocol:
Page 9 lines 173-177: It is stated that all material is deposited at Museum of Evolution, Uppsala University (Uppsala, Sweden) with Two exceptions: the holotype of Geodia phlegraeioides
sp. nov. was deposited at the MNCN in Madrid (Spain), and the holotype of Caminus xavierae
sp. nov. was deposited at the MNHNC-UP in Porto (Portugal).

Questions: why is not all material and in particular all holotypes not stored in the country where the material was collected, namely Spain? Is this in agreement with regulations in Spain, with permission from the relevant Spanish authorities? Is there an agreement about this transfer of material? Were any permits required for the collection? Have any sub-samples been provided to a natural history museum in Spain?
With respect to the Nagoya Protocol, what agreements have been made and is there a formal Material Transfer Agreement? Where have the DNA extractions of all the material, and in particular of the holotypes, been stored? Are they openly accessible?

Validity of the findings

Clear and comprehensive taxonomic review and descriptions. It is very impressive! The plates are of very good quality and will provide an important reference for future work on sponge biodiversity in the Balearic islands, and the Mediterranean in general.

What I am missing are images of the skeletal architecture, so the images of the slides with the choanosome. This is an important feature for the description and identification. This is given only in a few plates. I understand that this may not be possible for very small or thinly encrusting specimens, but there are ample specimens where the field image shows large sponges that would allow ample tissue for a slide. I suspect that the authors have made slides, so would urge them to include the images in the plates.

The phylogentic trees are good, but there are too many. If I understand correctly, there are basicly two trees: COI and 28S. Subsequently small portions are zoomed out as separate figures. This is unnecessary, you can refer to the two major trees. Please consolidate your trees and make a selection.

---

## Round 0.2 · accepted · Accept

The authors have addressed the points raised by the reviewers.